# Entropy Dissipation for Degenerate Stochastic Differential Equations via Sub-Riemannian Density Manifold

**DOI:** 10.3390/e25050786

**Published:** 2023-05-11

**Authors:** Qi Feng, Wuchen Li

**Affiliations:** 1Department of Mathematics, University of Michigan, Ann Arbor, MI 48109, USA; 2Department of Mathematics, University of South Carolina, Columbia, SC 29208, USA; wuchen@mailbox.sc.edu

**Keywords:** degenerate drift–diffusion process, Lyapunov methods, auxiliary Fisher information, sub-Riemannian density manifold, generalized Bochner’s formula, 53C17, 60D05, 58B20

## Abstract

We studied the dynamical behaviors of degenerate stochastic differential equations (SDEs). We selected an auxiliary Fisher information functional as the Lyapunov functional. Using generalized Fisher information, we conducted the Lyapunov exponential convergence analysis of degenerate SDEs. We derived the convergence rate condition by generalized Gamma calculus. Examples of the generalized Bochner’s formula are provided in the Heisenberg group, displacement group, and Martinet sub-Riemannian structure. We show that the generalized Bochner’s formula follows a generalized second-order calculus of Kullback–Leibler divergence in density space embedded with a sub-Riemannian-type optimal transport metric.

## 1. Introduction

Consider the following Stratonovich stochastic differential equation:(1)$$dX_{t}=b\left(X_{t}\right)dt+\sqrt{2}a\left(X_{t}\right)\circ dB_{t},$$
where $\left(B_{t}^{1},B_{t}^{2},\cdots ,B_{t}^{n}\right)$ is an *n*-dimensional Brownian motion in $R^{n}$, $a\in R^{n+m}\to R^{(n+m)\times n}$ is a matrix-valued function, and $b:R^{n+m}\to R^{n+m}$ is a drift vector field. The convergence analysis of SDE (Equation 1) to its invariant distribution lies in the intersection of differential geometry, analysis, the Lie group (subgroup in quantum mechanics), and probability. The convergence analysis also has broad applications in designing fast algorithms in artificial intelligence (AI) and Bayesian sampling/optimization problems. One key question arises: How fast does the probability density function of SDE (Equation 1) converge to its invariant distribution?

The Gamma calculus, also named Bakry–Émery iterative calculus [1], provides analytical approaches to derive the convergence rate for SDE (Equation 1). This lower bound is known as the Ricci curvature lower bound. However, classical studies are limited to the non-degenerate diffusion coefficient matrix *a*. The classical Gamma calculus is no longer valid when *a* is a degenerate matrix function; see the generalization of Bakry–Émery calculus in [2].

This paper presents a Lyapunov convergence analysis for the degenerate diffusion process. We selected a class of *z*-Fisher information as the Lyapunov functional, where *z* is a matrix function different from matrix *a*. We derived a generalized Gamma calculus by the dissipation of the Lyapunov functional along the diffusion process. We then derived the generalized Bochner’s formula and obtained the exponential convergence condition. Several concrete examples are presented: gradient-drift–diffusions on the Heisenberg group, the displacement group, and the Martinet sub-Riemannian structure. Our approach extends the classical optimal transport geometry, in particular the second-order calculus of the relative entropy in the density manifold studied in [3,4,5,6].

The generalized Gamma calculus was first introduced by Baudoin–Garofalo [2] for sub-Riemannian manifolds. Related results were studied later in [7,8,9,10,11,12,13,14,15]. The commutative property of the iteration of $\Gamma_{1}$ and $\Gamma_{1}^{z}$ (Hypothesis 1.2 in [2]) was crucial in the previous works. Our algebraic Condition 1 does not have this requirement. We can remove this commutative condition in the weak sense. Thus, our results go beyond the step two-bracket-generating condition. We present algebraic conditions for the existence of the generalized Bochner’s formula.

On the other hand, optimal transport on the sub-Riemannian manifold was studied by [16,17,18,19]. An optimal transport metric on a sub-Riemannian manifold was proposed in [18,19]. In this case, the density manifold still forms an infinite-dimensional Riemannian manifold. The Monge–Ampère equation in sub-Riemannian settings was studied in [17]. Our approach is different. We introduced the sub-Riemannian density manifold (SDM) and studied its second-order geometric calculations of relative entropies in the SDM. Using those, we propose a new Gamma z calculus for degenerate stochastic differential equations and established the generalized curvature dimension-type bound. Besides, Refs. [20,21] used the analytical property of optimal transport to formulate the Ricci curvature lower bound in general metric space. Different from [20,21,22], we focused on the geometric calculations in the density manifold introduced by the *z* direction. Following the second-order geometric calculations in the density manifold, we formulated the new Gamma calculus and the corresponding Ricci curvature tensor for the sub-Riemannian manifold. Besides, our derivation also relates to the entropy methods [23,24]. Using entropy methods, Refs. [25,26] derived the convergence rate for degenerate drift–diffusion processes with constant diffusion coefficients *a*. Compared to previous works, we applied the entropy method with Gamma calculus and geometric calculations in the density manifold. It derives a generalized Gamma calculus from the dissipation of auxiliary Fisher information. Several concrete examples of convergence conditions are derived in the Lie-group-induced drift–diffusion processes.

We organize the paper as follows. We introduce the main result in Section 2. It is an explicit convergence rate condition for the density of degenerate SDEs in the $L^{1}$ distance. In Section 3, we provide three examples of the proposed convergence analysis, including gradient-drift–diffusions on the Heisenberg group, the displacement group, and the Martinet sub-Riemannian structure. In Section 4, we present the Lyapunov analysis in the sub-Riemannian density manifold. The generalized Gamma calculus and the proof of the generalized Bochner’s formula is presented in Section 5. Some further discussions for other functional inequalities are presented in Section 6.

## 2. Main Results

In this section, we present this paper’s setting and main results.

### 2.1. Setting

Consider a Stratonovich SDE:(2)$$\begin{array}{c} dX_{t}=b\left(X_{t}\right)dt+\sqrt{2}a\left(X_{t}\right)\circ dB_{t}, \end{array}$$
where $\left(B_{t}^{1},B_{t}^{2},\cdots ,B_{t}^{n}\right)$ is an *n*-dimensional Brownian motion in $R^{n}$, $a:R^{n+m}\to R^{(n+m)\times n}$ is a matrix-valued function, and $b:R^{n+m}\to R^{n+m}$ is a vector field. We refer to [27] (Section 3.13) for the definition of the Stratonovich SDE. According to [28] (Appendix A.7), the SDE (Equation 2) can also be written as the following Itô SDE:(3)$$dX_{l,t}={\tilde{b}}_{l}\left(X_{t}\right)dt+\sum_{i=1}^{n}\sqrt{2}a_{li}\left(X_{t}\right)dB_{t}^{i},\;\mathrm{for}\;l=1,\cdots ,n+m,$$
where
(4)$${\tilde{b}}_{l}=b_{l}+\left(\sum_{i=1}^{n}\nabla_{a_{i}}a_{i}\right){}_{l},\;\mathrm{for}\;l=1,\cdots ,n+m.$$

We denote $\left\{a_{1},\cdots ,a_{n}\right\}$ as the column vectors of matrix *a*, and $\sum_{i=1}^{n}\nabla_{a_{i}}a_{i}$$\in R^{n+m}$ represents
(5)$$\left(\sum_{i=1}^{n}\nabla_{a_{i}}a_{i}\right){}_{l}=\sum_{i=1}^{n}\sum_{k=1}^{n+m}a_{ki}\frac{\partial a_{li}}{\partial x_{k}},\;\mathrm{for}\;l=1,\cdots ,n+m.$$

We denote $a^{T}$ as the transpose of matrix *a* and denote $\left\{a_{1}^{T},\cdots ,a_{n}^{T}\right\}$ as the row vectors of matrix $a^{T}$. In particular, we have $a_{\hat{i}i}=a_{i\hat{i}}^{T}$, for i=1,⋯,n and $\hat{i}=1,\cdots ,n+m$. With some abuse of notation, we also denote $a_{i}^{T}$ as the vector fields corresponding to the row vectors $a_{i}^{T}$, for i=1,⋯,n. We assumed that $\left\{a_{1}^{T}\left(x\right),a_{2}^{T}\left(x\right),\cdots ,a_{n}^{T}\left(x\right)\right\}$ satisfies the strong Hormander condition (or bracket-generating condition):$$\mathrm{Span}\left\{a_{1}^{T}\left(x\right),\cdots ,a_{n}^{T}\left(x\right),\;\left[a_{i_{1}}^{T},\cdots ,\left[a_{i_{k-1}}^{T},a_{i_{k}}^{T}\right]\cdots \right]\left(x\right),\;1\leq i_{1},\cdots ,i_{k}\leq n,k\geq 2\right\}=R^{n+m},$$
where [·,·] represents the Lie bracket between two vector fields. The strong Hörmander condition means that the Lie algebra generated by the vector fields $\left\{a_{1}^{T}\left(x\right),\cdots ,a_{n}^{T}\left(x\right)\right\}$ is of full rank at every point $x\in R^{n+m}$ (see, e.g., [29] (Section 7.4)). This condition ensures the existence of a smooth probability density function of SDE (Equation 2); see the original proofs in [30,31]. For the simplicity of presentation, we assumed the probability density function is strictly positive. Indeed, the positivity of the density follows from the Hörmander condition [32]; for the more technical conditions to show the positivity by using Malliavin calculus, we refer to [33,34] (Theorem 1.4 with *H* = 1/2). Denote $X_{t}∼\rho \left(t,x\right)$, where ρ=ρ(t,x) is the probability density function of SDE (Equation 2). The density function ρ satisfies the Fokker–Planck equation of SDE (Equation 2):(6)$$\partial_{t}\rho \left(t,x\right)=-\nabla_{x}\cdot \left(\rho \left(t,x\right)\tilde{b}\left(x\right)\right)+\sum_{i=1}^{n+m}\sum_{j=1}^{n+m}\frac{\partial^{2}}{\partial x_{i}\partial x_{j}}\left({\left(a\left(x\right)a{\left(x\right)}^{T}\right)}_{ij}\rho \left(t,x\right)\right),$$
with a smooth initial condition:$$\rho_{0}\left(x\right)=\rho \left(0,x\right),\;\int_{R^{n+m}}\rho_{0}\left(x\right)dx=1,\;\rho_{0}\left(x\right)>0.$$

In this paper, we assumed that SDE (Equation 2) has a unique invariant symmetric measure μ, where dμ=π(x)dx with $\pi \in C^{\infty }\left(R^{n+m}\right)$. Here, π solves the equilibrium of Fokker–Planck Equation (Equation 6):$$-\nabla_{x}\cdot \left(\pi \left(x\right)\tilde{b}\left(x\right)\right)+\sum_{i=1}^{n+m}\sum_{j=1}^{n+m}\frac{\partial^{2}}{\partial x_{i}\partial x_{j}}\left({\left(a\left(x\right)a{\left(x\right)}^{T}\right)}_{ij}\pi \left(x\right)\right)=0.$$

We studied a particular class of the vector field *b* for a given invariant distribution π.

**Assumption (Gradient flow formulation):** Suppose that *b*, *a*, and π satisfy the relation:(7)$$b=a⊗\nabla a+aa^{T}\nabla \log\pi ,$$
where $a⊗\nabla a\in R^{n+m}$ represents, for $\hat{k}=1,\cdots ,n+m$,
(8)$$\begin{array}{ccc} {\left(a⊗\nabla a\right)}_{\hat{k}} & = & \sum_{k=1}^{n}\sum_{k^{\prime }=1}^{n+m}a_{\hat{k}k}\frac{\partial }{\partial x_{k^{\prime }}}a_{k^{\prime }k}. \end{array}$$

In the Itô formulation, $\tilde{b}$, *a*, and π satisfy
$${\tilde{b}}_{l}={\left(aa^{T}\nabla \log\pi \right)}_{l}+{\left(a⊗\nabla a\right)}_{l}+{\left(\sum_{i=1}^{n}\nabla_{a_{i}}a_{i}\right)}_{l},$$
for l=1,⋯,n+m. In this case, we can reformulate Equation (Equation 6) as
(9)$$\partial_{t}\rho \left(t,x\right)=\nabla \cdot \left(\rho \left(t,x\right)a\left(x\right)a{\left(x\right)}^{T}\nabla \log\frac{\rho (t,x)}{\pi (x)}\right).$$

We leave the derivation of Formula (Equation 9) in Appendix A. If ρ(t,x)=π(x), then $\log\frac{\rho (t,x)}{\pi (x)}=0$, and π is an invariant density function for SDE (Equation 2). In Section 4, we demonstrate that Fokker–Planck Equation (Equation 6), or its equivalent Formulation (Equation 9), forms a “horizontal” gradient flow in the sub-Riemannian density manifold. We designed a Lyapunov functional to study the convergence behavior of this “horizontal” gradient flow (Equation 9).

**Remark** **1.**
*Formula (Equation 9) can be written as*

$$\partial_{t}\left(\log\rho \left(t,x\right)-\log\pi \left(x\right)\right)\rho \left(t,x\right)=\nabla \cdot \left(\rho \left(t,x\right)a\left(x\right)a{\left(x\right)}^{T}\nabla \log\frac{\rho (t,x)}{\pi (x)}\right).$$


*It has a weak formulation that*

$$\int_{R^{n+m}}\left(\partial_{t}\log\frac{\rho (t,x)}{\pi (x)},\phi \left(x\right)\right)\rho \left(t,x\right)dx=-\int_{R^{n+m}}\left(\nabla \phi \left(x\right),a\left(x\right)a{\left(x\right)}^{T}\nabla \log\frac{\rho (t,x)}{\pi (x)}\right)\rho \left(t,x\right)dx,$$

*where $\phi \in C^{\infty }\left(R^{n+m}\right)$ is a smooth test function.*


**Remark** **2**(Non-gradient flow drift)**.**
*In fact, the proposed method is not limited to the gradient flow assumption of the drift vector field b in (Equation 7). See the details in [35].*

### 2.2. Main Result

We now briefly sketch the main results. Denote a sub-elliptic operator $L:C^{\infty }\left(R^{n+m}\right)\to C^{\infty }\left(R^{n+m}\right)$ as follows:$$Lf=\nabla \cdot \left(aa^{T}\nabla f\right)-{\langle a⊗\nabla a,\nabla f\rangle }_{R^{n+m}}+{\langle b,\nabla f\rangle }_{R^{n+m}},$$
where $f\in C^{\infty }\left(R^{n+m}\right)$.

**Definition** **1**(Generalized Gamma z calculus)**.**
*Consider a smooth matrix function $z:R^{n+m}\to R^{(n+m)\times m}$. Denote Gamma one bilinear forms $\Gamma_{1},\Gamma_{1}^{z}:C^{\infty }\left(R^{n+m}\right)\times C^{\infty }\left(R^{n+m}\right)\to C^{\infty }\left(R^{n+m}\right)$ as*
$$\Gamma_{1}\left(f,g\right)={\langle a^{T}\nabla f,a^{T}\nabla g\rangle }_{R^{n}},\;\Gamma_{1}^{z}\left(f,g\right)={\langle z^{T}\nabla f,z^{T}\nabla g\rangle }_{R^{m}}.$$
*Define Gamma two bilinear forms $\Gamma_{2},\Gamma_{2}^{z,\pi }:C^{\infty }\left(R^{n+m}\right)\times C^{\infty }\left(R^{n+m}\right)\to C^{\infty }\left(R^{n+m}\right)$ as*

$$\Gamma_{2}\left(f,g\right)=\frac{1}{2}\left[L\Gamma_{1}\left(f,g\right)-\Gamma_{1}\left(Lf,g\right)-\Gamma_{1}\left(f,Lg\right)\right],$$

*and*

(10)
$$\begin{array}{ccc} \Gamma_{2}^{z,\pi }\left(f,g\right) & = & \;\frac{1}{2}\left[L\Gamma_{1}^{z}\left(f,g\right)-\Gamma_{1}^{z}\left(Lf,g\right)-\Gamma_{1}^{z}\left(f,Lg\right)\right] \end{array}$$


(11)
$$\begin{array}{ccc} & & +{\mathrm{div}}_{z}^{\pi }\left(\Gamma_{1,\nabla (aa^{T})}\left(f,g\right)\right)-{\mathrm{div}}_{a}^{\pi }\left(\Gamma_{1,\nabla (zz^{T})}\left(f,g\right)\right). \end{array}$$


*Here, ${\mathrm{div}}_{a}^{\pi }$, ${\mathrm{div}}_{z}^{\pi }$ are divergence operators defined by*

$${\mathrm{div}}_{a}^{\pi }\left(F\right)=\frac{1}{\pi }\nabla \cdot \left(\pi \;aa^{T}F\right),\;{\mathrm{div}}_{z}^{\pi }\left(F\right)=\frac{1}{\pi }\nabla \cdot \left(\pi \;zz^{T}F\right),$$

*for any smooth vector field $F\in R^{n+m}$, and $\Gamma_{\nabla (aa^{T})}$, $\Gamma_{\nabla (zz^{T})}$ are vector Gamma one bilinear forms defined as*

$$\begin{array}{ccc} \Gamma_{1,\nabla (aa^{T)}}\left(f,g\right) & = & \langle \nabla f,\nabla \left(aa^{T}\right)\nabla g\rangle =\left(\langle \nabla f,\frac{\partial }{\partial x_{\hat{k}}}\left(aa^{T}\right)\nabla g\rangle \right){}_{\hat{k}=1}^{n+m},\\ \Gamma_{1,\nabla (zz^{T)}}\left(f,g\right) & = & \langle \nabla f,\nabla \left(aa^{T}\right)\nabla g\rangle =\left(\langle \nabla f,\frac{\partial }{\partial x_{\hat{k}}}\left(zz^{T}\right)\nabla g\rangle \right){}_{\hat{k}=1}^{n+m}, \end{array}$$

*with*

$$\begin{array}{ccc} {\mathrm{div}}_{z}^{\pi }\left(\Gamma_{\nabla (aa^{T})}f,g\right) & = & \frac{\nabla \cdot (zz^{T}\pi \langle \nabla f,\nabla \left(aa^{T}\right)\nabla g\rangle )}{\pi },\\ {\mathrm{div}}_{a}^{\pi }\left(\Gamma_{\nabla (zz^{T})}f,g\right) & = & \frac{\nabla \cdot (aa^{T}\pi \langle \nabla f,\nabla \left(zz^{T}\right)\nabla g\rangle )}{\pi }. \end{array}$$



We next demonstrate that the summation of $\Gamma_{2}$ and $\Gamma_{2}^{z,\pi }$ can induce the following decomposition and bilinear forms. They are natural extensions of the classical Bakry–Émery calculus in the Riemannian manifold, i.e., non-degenerate matrix function *a*.

**Notation** **1.**
*For matrix function $a:R^{n+m}\to R^{(n+m)\times n}$, we define matrix Q as*

(12)
$$\begin{array}{c} Q=\left(\begin{array}{ccc} a_{11}^{T}a_{11}^{T} & \cdots & a_{1(n+m)}^{T}a_{1(n+m)}^{T}\\ \cdots & a_{i\hat{i}}^{T}a_{k\hat{k}}^{T} & \cdots \\ a_{n1}^{T}a_{n1}^{T} & \cdots & a_{n(n+m)}^{T}a_{n(n+m)}^{T} \end{array}\right)\in R^{n^{2}\times {\left(n+m\right)}^{2}}, \end{array}$$

*with $Q_{ik\hat{i}\hat{k}}=a_{i\hat{i}}^{T}a_{k\hat{k}}^{T}$. More precisely, for each row (respectively, column) of Q, the row (respectively column) indices of $Q_{ik\hat{i}\hat{k}}$ follow $\sum_{i=1}^{n}\sum_{k=1}^{n}$ (respectively, $\sum_{\hat{i}=1}^{n+1}\sum_{\hat{k}=1}^{n+m}$). For matrix function $z:R^{n+m}\to R^{(n+m)\times m}$, we define matrix P as*

(13)
$$\begin{array}{c} P=\left(\begin{array}{ccc} z_{11}^{T}a_{11}^{T} & \cdots & z_{1(n+m)}^{T}a_{1(n+m)}^{T}\\ \cdots & z_{i\hat{i}}^{T}a_{k\hat{k}}^{T} & \cdots \\ z_{m\hat{1}}^{T}a_{n\hat{1}}^{T} & \cdots & z_{m(n+m)}^{T}a_{n(n+m)}^{T} \end{array}\right)\in R^{\left(nm\right)\times {\left(n+m\right)}^{2}}, \end{array}$$

*with $P_{ik\hat{i}\hat{k}}=z_{k\hat{k}}^{T}a_{i\hat{i}}^{T}$. For smooth function $f\in C^{\infty }\left(R^{n+m}\right)$, for any $\hat{i},\hat{k},\hat{j}=1,\cdots ,n+m$ and i,k=1,⋯,n (or 1,⋯,m), we define vector $C\in R^{{\left(n+m\right)}^{2}\times 1}$ with components*

(14)
$$\begin{array}{c} C_{\hat{i}\hat{k}}=\left[\sum_{i,k=1}^{n}\sum_{i^{\prime }=1}^{n+m}\left({\langle a_{i\hat{i}}^{T}a_{ii^{\prime }}^{T}\left(\frac{\partial a_{k\hat{k}}^{T}}{\partial x_{i^{\prime }}}\right),{\left(a^{T}\nabla \right)}_{k}f\rangle }_{R^{n}}-{\langle a_{ki^{\prime }}^{T}a_{i\hat{k}}^{T}\frac{\partial a_{i\hat{i}}^{T}}{\partial x_{i^{\prime }}},{\left(a^{T}\nabla \right)}_{k}f\rangle }_{R^{n}}\right)\right], \end{array}$$

*where we denote ${\left(a^{T}\nabla \right)}_{k}f=\sum_{k^{\prime }=1}^{n+m}a_{kk^{\prime }}^{T}\frac{\partial f}{\partial x_{k^{\prime }}}$. We define vector $D\in R^{n^{2}\times 1}$ with components*

(15)
$$\begin{array}{c} D_{ik}=\sum_{\hat{i},\hat{k}=1}^{n+m}a_{i\hat{i}}^{T}\frac{\partial a_{k\hat{k}}^{T}}{\partial x_{\hat{i}}}\frac{\partial f}{\partial x_{\hat{k}}},\;\mathrm{and}\;D^{T}D=\sum_{i,k}D_{ik}D_{ik}. \end{array}$$


*We define vector $F\in R^{{\left(n+m\right)}^{2}\times 1}$ with components*

(16)
$$\begin{array}{c} F_{\hat{i}\hat{k}}=\left[\sum_{i=1}^{n}\sum_{k=1}^{m}\sum_{i^{\prime }=1}^{n+m}\left({\langle a_{i\hat{i}}^{T}a_{ii^{\prime }}^{T}\left(\frac{\partial z_{k\hat{k}}^{T}}{\partial x_{i^{\prime }}}\right),{\left(z^{T}\nabla \right)}_{k}f\rangle }_{R^{m}}-{\langle z_{ki^{\prime }}^{T}a_{i\hat{k}}^{T}\frac{\partial a_{i\hat{i}}^{T}}{\partial x_{i^{\prime }}},{\left(z^{T}\nabla \right)}_{k}f\rangle }_{R^{m}}\right)\right]. \end{array}$$


*We define vector $E\in R^{(n\times m)\times 1}$ with components*

(17)
$$\begin{array}{c} E_{ik}=\sum_{\hat{i},\hat{k}=1}^{n+m}a_{i\hat{i}}^{T}\frac{\partial z_{k\hat{k}}^{T}}{\partial x_{\hat{i}}}\frac{\partial f}{\partial x_{\hat{k}}},\;\mathrm{and}\;E^{T}E=\sum_{i,k}E_{ik}E_{ik}. \end{array}$$


*We define vector $G\in R^{{\left(n+m\right)}^{2}\times 1}$ with components*

(18)
$$\begin{array}{ccc} G_{\hat{i}\hat{j}} & = & \sum_{i=1}^{n}\sum_{j=1}^{m}\sum_{j^{\prime },\hat{j},i^{\prime },\hat{i}=1}^{n+m}\left[\left(z_{j\hat{j}}^{T}z_{jj^{\prime }}^{T}\frac{\partial }{\partial x_{j^{\prime }}}a_{i\hat{i}}^{T}a_{ii^{\prime }}^{T}\frac{\partial f}{\partial x_{i^{\prime }}}+z_{j\hat{j}}^{T}z_{jj^{\prime }}^{T}\frac{\partial }{\partial x_{j^{\prime }}}a_{ii^{\prime }}^{T}\frac{\partial f}{\partial x_{i^{\prime }}}a_{i\hat{i}}^{T}\right)\right.\\ & & \left.\;\;\;\;\;\;-\left(a_{i\hat{i}}^{T}a_{ii^{\prime }}^{T}\frac{\partial }{\partial x_{i^{\prime }}}z_{j\hat{j}}^{T}z_{jj^{\prime }}^{T}\frac{\partial f}{\partial x_{j^{\prime }}}+a_{i\hat{i}}^{T}a_{ii^{\prime }}^{T}\frac{\partial }{\partial x_{i^{\prime }}}z_{jj^{\prime }}^{T}\frac{\partial f}{\partial x_{j^{\prime }}}z_{j\hat{j}}^{T}\right)\right]. \end{array}$$


*We define X as the vectorization of the Hessian matrix of function f:*

(19)
$$\begin{array}{c} X^{T}=\left(\begin{array}{ccccc} \frac{\partial^{2}f}{\partial x_{1}\partial x_{1}} & \cdots & \frac{\partial^{2}f}{\partial x_{\hat{i}}\partial x_{\hat{k}}} & \cdots & \frac{\partial^{2}f}{\partial x_{n+m}\partial x_{n+m}} \end{array}\right)\in R^{1\times {\left(n+m\right)}^{2}}. \end{array}$$



**Assumption** **1.**
*Assume that there exists vectors $\Lambda_{1},\Lambda_{2}\in R^{{\left(n+m\right)}^{2}\times 1}$ such that*

$$\begin{array}{ccc} {\left(Q^{T}Q\Lambda_{1}+P^{T}P\Lambda_{2}\right)}^{T}X & = & {\left(F+C+G+Q^{T}D+P^{T}E\right)}^{T}X. \end{array}$$



**Definition** **2**(Hessian matrix)**.**
*For smooth function $f\in C^{\infty }\left(R^{n+m}\right)$, define a matrix function $R:R^{n+m}\to R^{\left(n+m)\times (n+m\right)}$ as*
$$\begin{array}{c} R\left(\nabla f,\nabla f\right)=-\Lambda_{1}^{T}Q^{T}Q\Lambda_{1}-\Lambda_{2}^{T}P^{T}P\Lambda_{2}+D^{T}D+E^{T}E+\left(R_{ab}+R_{zb}+R_{\pi }\right)\left(\nabla f,\nabla f\right), \end{array}$$*where we define the following bilinear forms:*
$$\begin{array}{ccc} R_{ab}\left(\nabla f,\nabla f\right) & = & R_{a}\left(\nabla f,\nabla f\right)-\sum_{i=1}^{n}\sum_{\hat{i},\hat{k}=1}^{n+m}{\langle \left(a_{i\hat{i}}^{T}\frac{\partial b_{\hat{k}}}{\partial x_{\hat{i}}}\frac{\partial f}{\partial x_{\hat{k}}}-b_{\hat{k}}\frac{\partial a_{i\hat{i}}^{T}}{\partial x_{\hat{k}}}\frac{\partial f}{\partial x_{\hat{i}}}\right),{\left(a^{T}\nabla f\right)}_{i}\rangle }_{R^{n}},\\ R_{a}\left(\nabla f,\nabla f\right) & = & \sum_{i,k=1}^{n}\sum_{i^{\prime },\hat{i},\hat{k}=1}^{n+m}{\langle a_{ii^{\prime }}^{T}\left(\frac{\partial a_{i\hat{i}}^{T}}{\partial x_{i^{\prime }}}\frac{\partial a_{k\hat{k}}^{T}}{\partial x_{\hat{i}}}\frac{\partial f}{\partial x_{\hat{k}}}\right),{\left(a^{T}\nabla \right)}_{k}f\rangle }_{R^{n}}\\ & & +\sum_{i,k=1}^{n}\sum_{i^{\prime },\hat{i},\hat{k}=1}^{n+m}{\langle a_{ii^{\prime }}^{T}a_{i\hat{i}}^{T}\left(\frac{\partial }{\partial x_{i^{\prime }}}\frac{\partial a_{k\hat{k}}^{T}}{\partial x_{\hat{i}}}\right)\left(\frac{\partial f}{\partial x_{\hat{k}}}\right),{\left(a^{T}\nabla \right)}_{k}f\rangle }_{R^{n}}\\ & & -\sum_{i,k=1}^{n}\sum_{i^{\prime },\hat{i},\hat{k}=1}^{n+m}\langle a_{k\hat{k}}^{T}\frac{\partial a_{ii^{\prime }}^{T}}{\partial x_{\hat{k}}}\frac{\partial a_{i\hat{i}}^{T}}{\partial x_{i^{\prime }}}\frac{\partial f}{\partial x_{\hat{i}}}),{\left(a^{T}\nabla \right)}_{k}{f\rangle }_{R^{n}}\\ & & -\sum_{i,k=1}^{n}\sum_{i^{\prime },\hat{i},\hat{k}=1}^{n+m}{\langle a_{k\hat{k}}^{T}a_{ii^{\prime }}^{T}\left(\frac{\partial }{\partial x_{\hat{k}}}\frac{\partial a_{i\hat{i}}^{T}}{\partial x_{i^{\prime }}}\right)\frac{\partial f}{\partial x_{\hat{i}}},{\left(a^{T}\nabla \right)}_{k}f\rangle }_{R^{n}},\\ R_{zb}\left(\nabla f,\nabla f\right) & = & R_{z}\left(\nabla f,\nabla f\right)-\sum_{i=1}^{m}\sum_{\hat{i},\hat{k}=1}^{n+m}{\langle \left(z_{i\hat{i}}^{T}\frac{\partial b_{\hat{k}}}{\partial x_{\hat{i}}}\frac{\partial f}{\partial x_{\hat{k}}}-b_{\hat{k}}\frac{\partial z_{i\hat{i}}^{T}}{\partial x_{\hat{k}}}\frac{\partial f}{\partial x_{\hat{i}}}\right),{\left(z^{T}\nabla f\right)}_{i}\rangle }_{R^{m}},\\ R_{z}\left(\nabla f,\nabla f\right) & = & \sum_{i=1}^{n}\sum_{k=}^{m}\sum_{i^{\prime },\hat{i},\hat{k}=1}^{n+m}{\langle a_{ii^{\prime }}^{T}\left(\frac{\partial a_{i\hat{i}}^{T}}{\partial x_{i^{\prime }}}\frac{\partial z_{k\hat{k}}^{T}}{\partial x_{\hat{i}}}\frac{\partial f}{\partial x_{\hat{k}}}\right),{\left(z^{T}\nabla \right)}_{k}f\rangle }_{R^{m}}\\ & & +\sum_{i=1}^{n}\sum_{k=}^{m}\sum_{i^{\prime },\hat{i},\hat{k}=1}^{n+m}{\langle a_{ii^{\prime }}^{T}a_{i\hat{i}}^{T}\left(\frac{\partial }{\partial x_{i^{\prime }}}\frac{\partial z_{k\hat{k}}^{T}}{\partial x_{\hat{i}}}\right)\left(\frac{\partial f}{\partial x_{\hat{k}}}\right),{\left(z^{T}\nabla \right)}_{k}f\rangle }_{R^{m}}\\ & & -\sum_{i=1}^{n}\sum_{k=1}^{m}\sum_{i^{\prime },\hat{i},\hat{k}=1}^{n+m}\langle z_{k\hat{k}}^{T}\frac{\partial a_{ii^{\prime }}^{T}}{\partial x_{\hat{k}}}\frac{\partial a_{i\hat{i}}^{T}}{\partial x_{i^{\prime }}}\frac{\partial f}{\partial x_{\hat{i}}}),{\left(z^{T}\nabla \right)}_{k}{f\rangle }_{R^{m}}\\ & & -\sum_{i=1}^{n}\sum_{k=1}^{m}\sum_{i^{\prime },\hat{i},\hat{k}=1}^{n+m}{\langle z_{k\hat{k}}^{T}a_{ii^{\prime }}^{T}\left(\frac{\partial }{\partial x_{\hat{k}}}\frac{\partial a_{i\hat{i}}^{T}}{\partial x_{i^{\prime }}}\right)\frac{\partial f}{\partial x_{\hat{i}}},{\left(z^{T}\nabla \right)}_{k}f\rangle }_{R^{m}}, \end{array}$$
*and*
$$\begin{array}{ccc} R_{\pi }\left(\nabla f,\nabla f\right) & = & 2\sum_{k=1}^{m}\sum_{i=1}^{n}\sum_{k^{\prime },\hat{k},\hat{i},i^{\prime }=1}^{n+m}\left[\frac{\partial }{\partial x_{k^{\prime }}}z_{kk^{\prime }}^{T}z_{k\hat{k}}^{T}\frac{\partial }{\partial x_{\hat{k}}}a_{i\hat{i}}^{T}\frac{\partial f}{\partial x_{\hat{i}}}a_{ii^{\prime }}^{T}\frac{\partial f}{\partial x_{i^{\prime }}}\right]\\ & & +2\sum_{k=1}^{m}\sum_{i=1}^{n}\sum_{k^{\prime },\hat{k},\hat{i},i^{\prime }=1}^{n+m}\left[z_{kk^{\prime }}^{T}\frac{\partial }{\partial x_{k^{\prime }}}z_{k\hat{k}}^{T}\frac{\partial }{\partial x_{\hat{k}}}a_{i\hat{i}}^{T}\frac{\partial f}{\partial x_{\hat{i}}}a_{ii^{\prime }}^{T}\frac{\partial f}{\partial x_{i^{\prime }}}\right.\\ & & \;\;\;\;\;\;\;\;\;+z_{kk^{\prime }}^{T}z_{k\hat{k}}^{T}\frac{\partial^{2}}{\partial x_{k^{\prime }}\partial x_{\hat{k}}}a_{i\hat{i}}^{T}\frac{\partial f}{\partial x_{\hat{i}}}a_{ii^{\prime }}^{T}\frac{\partial f}{\partial x_{i^{\prime }}}\\ & & \left.\;\;\;\;\;\;\;\;\;+z_{kk^{\prime }}^{T}z_{k\hat{k}}^{T}\frac{\partial }{\partial x_{\hat{k}}}a_{i\hat{i}}^{T}\frac{\partial f}{\partial x_{\hat{i}}}\frac{\partial }{\partial x_{k^{\prime }}}a_{ii^{\prime }}^{T}\frac{\partial f}{\partial x_{i^{\prime }}}\right]\\ & & +2\sum_{k=1}^{m}\sum_{i=1}^{n}\sum_{\hat{k},\hat{i},i^{\prime }=1}^{n+m}{\left(z^{T}\nabla \log\pi \right)}_{k}\left[z_{k\hat{k}}^{T}\frac{\partial }{\partial x_{\hat{k}}}a_{i\hat{i}}^{T}\frac{\partial f}{\partial x_{\hat{i}}}a_{ii^{\prime }}^{T}\frac{\partial f}{\partial x_{i^{\prime }}}\right]\\ & & -2\sum_{j=1}^{m}\sum_{l=1}^{n}\sum_{l^{\prime },\hat{l},\hat{j},j^{\prime }=1}^{n+m}\left[\frac{\partial }{\partial x_{l^{\prime }}}a_{ll^{\prime }}^{T}a_{l\hat{l}}^{T}\frac{\partial }{\partial x_{\hat{l}}}z_{j\hat{j}}^{T}\frac{\partial f}{\partial x_{\hat{j}}}z_{jj^{\prime }}^{T}\frac{\partial f}{\partial x_{j^{\prime }}}\right]\\ & & -2\sum_{j=1}^{m}\sum_{l=1}^{n}\sum_{l^{\prime },\hat{l},\hat{j},j^{\prime }=1}^{n+m}\left[a_{ll^{\prime }}^{T}\frac{\partial }{\partial x_{l^{\prime }}}a_{l\hat{l}}^{T}\frac{\partial }{\partial x_{\hat{l}}}z_{j\hat{j}}^{T}\frac{\partial f}{\partial x_{\hat{j}}}z_{jj^{\prime }}^{T}\frac{\partial f}{\partial x_{j^{\prime }}}\right.\\ & & \;\;\;\;\;\;\;\;\;+a_{ll^{\prime }}^{T}a_{l\hat{l}}^{T}\frac{\partial^{2}}{\partial x_{l^{\prime }}\partial x_{\hat{l}}}z_{j\hat{j}}^{T}\frac{\partial f}{\partial x_{\hat{j}}}z_{jj^{\prime }}^{T}\frac{\partial f}{\partial x_{j^{\prime }}}\\ & & \left.\;\;\;\;\;\;\;\;\;+a_{ll^{\prime }}^{T}a_{l\hat{l}}^{T}\frac{\partial }{\partial x_{\hat{l}}}z_{j\hat{j}}^{T}\frac{\partial f}{\partial x_{\hat{j}}}\frac{\partial }{\partial x_{l^{\prime }}}z_{jj^{\prime }}^{T}\frac{\partial f}{\partial x_{j^{\prime }}}\right]\\ & & -2\sum_{j=1}^{m}\sum_{l=1}^{n}\sum_{\hat{l},\hat{j},j^{\prime }=1}^{n+m}{\left(a^{T}\nabla \log\pi \right)}_{l}\left[a_{l\hat{l}}^{T}\frac{\partial }{\partial x_{\hat{l}}}z_{j\hat{j}}^{T}\frac{\partial f}{\partial x_{\hat{j}}}z_{jj^{\prime }}^{T}\frac{\partial f}{\partial x_{j^{\prime }}}\right]. \end{array}$$
*Here, we also denote $R=R\left(x\right)\in R^{\left(n+m)\times (n+m\right)}$, such that ${\left(\nabla f\right)}^{T}R\left(x\right)\nabla f=R\left(\nabla f,\nabla f\right)$.*


The main theorem is presented below, and its proof is postponed to Theorem 3 in Section 5.

**Theorem** **1**(Generalized *z* Bochner’s formula)**.**
*If Assumption 1 is satisfied, then the following decomposition holds:*
$$\begin{array}{ccc} \Gamma_{2}\left(f,f\right)+\Gamma_{2}^{z,\pi }\left(f,f\right) & = & ∥{\mathrm{Hess}}_{a,z}{f∥}^{2}+R\left(\nabla f,\nabla f\right), \end{array}$$*where we define*
$$\begin{array}{ccc} ∥{\mathrm{Hess}}_{a,z}{f∥}^{2} & = & {\left[X+\Lambda_{1}\right]}^{T}Q^{T}Q\left[X+\Lambda_{1}\right]+{\left[X+\Lambda_{2}\right]}^{T}P^{T}P\left[X+\Lambda_{2}\right],\\ R(\nabla f,\nabla f) & = & -\Lambda_{1}^{T}Q^{T}Q\Lambda_{1}-\Lambda_{2}^{T}P^{T}P\Lambda_{2}+D^{T}D+E^{T}E\\ & & +R_{ab}\left(\nabla f,\nabla f\right)+R_{zb}\left(\nabla f,\nabla f\right)+R^{\pi }\left(\nabla f,\nabla f\right). \end{array}$$

We are now ready to prove the convergence property of the degenerate drift–diffusion process (Equation 1) and related functional inequalities. Denote the Kullback–Leibler divergence as
$$D_{\mathrm{KL}}\left(\rho ∥\pi \right):=\int_{R^{n+m}}\rho \left(x\right)\log\frac{\rho (x)}{\pi (x)}dx.$$

Denote the a,z-relative Fisher information functional as
$$\begin{array}{c} I_{a,z}\left(\rho \right):=\int_{R^{n+m}}\left(\nabla \log\frac{\rho }{\pi },aa^{T}\nabla \log\frac{\rho }{\pi }\right)\rho dx+\int_{R^{n+m}}\left(\nabla \log\frac{\rho }{\pi },zz^{T}\nabla \log\frac{\rho }{\pi }\right)\rho dx. \end{array}$$

**Theorem** **2**(Exponential convergence in the $L^{1}$ distance)**.**
*Suppose there exists a constant κ>0 such that*
$$R⪰\kappa (aa^{T}+zz^{T}).$$
*Let $\rho_{0}$ be a smooth initial distribution and ρ=ρ(t,x) be the probability density function of (Equation 1). Then, ρ converges to the invariant measure π in the sense of*

$$I_{a,z}\left(\rho \right)\leq e^{-2\kappa t}I_{a,z}\left(\rho_{0}\right).$$


*In addition,*

$$\int_{R^{n+m}}\left|\rho \left(t,x\right)-\pi \left(x\right)\right|dx\leq \sqrt{2D_{\mathrm{KL}}\left(\rho_{0}∥\pi \right)}e^{-\kappa t}.$$



The proof of Theorem 2 is postponed to Proposition (14).

**Remark** **3**(Functional inequalities)**.**
*Suppose $R⪰\kappa (aa^{T}+zz^{T})$ with κ>0, then the z-log-Sobolev inequalities hold:*
$$\int_{R^{n+m}}\rho \log\frac{\rho }{\pi }dx\leq \frac{1}{2\kappa }I_{a,z}\left(\rho \right),$$*for any smooth density function ρ.*

**Remark** **4.**
*In the literature [2], the $\Gamma_{2,z}$ operator is defined by *(Equation 10)*, i.e., $\Gamma_{2}^{z}\left(f,f\right)=\frac{1}{2}L\Gamma_{1}^{z}\left(f,f\right)-\Gamma_{1}^{z}\left(Lf,f\right)$. In fact, this definition is under the assumption of $\Gamma_{1}\left(\Gamma_{1}^{z}\left(f,f\right),f\right)=\Gamma_{1}^{z}\left(\Gamma_{1}\left(f,f\right),f\right)$. This assumption holds true only for the special choice of a and z. In the generalized Gamma z calculus, we introduce a new term *(Equation 11)*, which removes the assumption $\Gamma_{1}\left(\Gamma_{1}^{z}\left(f,f\right),f\right)=\Gamma_{1}^{z}\left(\Gamma_{1}\left(f,f\right),f\right)$. In fact, in the paper, we show that *(Equation 11)* is exactly the new bilinear form behind the assumption in [2] by considering the weak form.*


**Remark** **5.**
*Following [35] (Assumption 1), we know that, for any i∈{1,⋯,n} and*
*k∈{1,⋯,m}, if*

(20)
$$\begin{array}{c} z_{k}^{T}\nabla a_{i}^{T}\in \mathrm{Span}\left\{a_{1}^{T},\cdots ,a_{n}^{T}\right\}, \end{array}$$

*there exist vectors ${\hat{\Lambda }}_{1}$ and ${\hat{\Lambda }}_{2}$, such that the Hessian operator associated with the generator of the SDE and the metric ${\left(aa^{T}\right)}^{†}$ could be represented as*

$${∥\mathrm{Hess}f∥}^{2}={\left[QX+{\hat{\Lambda }}_{1}\right]}^{T}\left[QX+{\hat{\Lambda }}_{1}\right]+{\left[PX+{\hat{\Lambda }}_{2}\right]}^{T}\left[PX+{\hat{\Lambda }}_{2}\right].$$


*Furthermore, we have the following relation:*

$$\begin{array}{cc} \; & {\left[QX+{\hat{\Lambda }}_{1}\right]}^{T}\left[QX+{\hat{\Lambda }}_{1}\right]+{\left[PX+{\hat{\Lambda }}_{2}\right]}^{T}\left[PX+{\hat{\Lambda }}_{2}\right]-{\hat{\Lambda }}_{1}^{T}{\hat{\Lambda }}_{1}-{\hat{\Lambda }}_{2}^{T}{\hat{\Lambda }}_{2}\\ = & {\left[X+\Lambda_{1}\right]}^{T}Q^{T}Q\left[X+\Lambda_{1}\right]+{\left[X+\Lambda_{2}\right]}^{T}P^{T}P\left[X+\Lambda_{2}\right]-\Lambda_{1}^{T}Q^{T}Q\Lambda_{1}-\Lambda_{2}^{T}P^{T}P\Lambda_{2}, \end{array}$$

*if there exist $\Lambda_{1}$ and $\Lambda_{2}$ as in Assumption 1 such that*

(21)
$$\begin{array}{c} {\hat{\Lambda }}_{1}^{T}=\Lambda_{1}^{T}Q^{T}\;\mathrm{and}\;{\hat{\Lambda }}_{2}^{T}=\Lambda_{2}^{T}P^{T}. \end{array}$$


*Assumption 1 is true if Conditions 20 and 21 hold. See the detailed connections in [35] (Remark 11).*


## 3. Examples

In this section, we consider the following degenerate drift–diffusion process:(22)$$dX_{t}=-a\left(X_{t}\right)a{\left(X_{t}\right)}^{T}\nabla V\left(X_{t}\right)dt+\sqrt{2}a\left(X_{t}\right)\circ dB_{t},$$
where $a:R^{n+m}\to R^{(n+m)\times n}$ is a matrix-valued function, for $n,m\in Z_{+}$, and $V\in C^{\infty }\left(R^{n+m}\right)$ is a smooth potential function. We denote the invariant measure of SDE (Equation 22) as π. We further assumed that
$$\begin{array}{c} -aa^{T}\nabla V=a⊗\nabla a+aa^{T}\nabla \log\pi . \end{array}$$

The above assumption holds for the later three examples.

**Remark** **6.**
*For V=0, the invariant measure π in the above assumption exists if $\left\{a_{1},\cdots ,a_{n}\right\}$ forms left-invariant structures on unimodular Lie groups. In this case, the sub-Laplacian is the sum of squares of horizontal vector fields and the invariant measure is also symmetric. Stratonovich SDE *(Equation 22)* defines the horizontal Brownian motion on sub-Riemannian structure $\left(R^{n+m},\tau ,{\left(aa^{T}\right)}^{†}{|}_{\tau }\right)$, and π is the volume form associated with the horizontal Laplacian. In general, if the Lie group structure is not unimodular, the drift b≠0. See the related studies about the diffusion process on general manifolds in [36,37,38,39,40,41,42,43]. See the related studies on log-Sobolev inequality in [44,45].*


**Remark** **7.**
*It is also worth mentioning that many sub-Riemannian manifolds are non-compact. Hence, there may not exist a positive constant κ for both classical $\Gamma_{1}$ and $\Gamma_{1}^{z}$ directions in the non-compact domain. The non-compactness of the domain brings additional difficulties. To prove the associated inequalities in this case, we need to extend the result derived in [46,47]. This is a direction for future work.*


**Remark** **8.**
*It is known that the Heisenberg group is an example of Lie groups in quantum mechanics [48]. In future work, we shall investigate the general convergence analysis of SDEs in Lie groups and their connections with quantum SDEs.*


### 3.1. Heisenberg Group

In this subsection, we apply our general theory to the well-known example in sub-Riemannian geometry, which is the Heisenberg group. A related LSI for the horizontal Wiener measure was studied in [46]. Recall briefly that the Heisenberg group $H^{1}$ admits left-invariant vector fields: $X=\frac{\partial }{\partial x}-\frac{1}{2}y\frac{\partial }{\partial z},\;Y=\frac{\partial }{\partial y}+\frac{1}{2}x\frac{\partial }{\partial z},\;Z=\frac{\partial }{\partial z}$. Here, {X,Y,Z} forms an orthonormal basis for the tangent bundle of $H^{1}$. In this case, $\pi =e^{-V}$. In particular, *X* and *Y* generate the horizontal distribution τ. To fit into our general theory from the previous section, we take matrices *a* and *z* as below:(23)$$\begin{array}{c} a^{T}=\left(\begin{array}{ccc} 1 & 0 & -y/2\\ 0 & 1 & x/2 \end{array}\right),\;z^{T}=\left(0,0,1\right). \end{array}$$

In particular, we have
$$\begin{array}{c} a^{T}\nabla f={\left({\left(a^{T}\nabla \right)}_{1}f,{\left(a^{T}\nabla \right)}_{2}f\right)}^{T},\;{\left(a^{T}\nabla \right)}_{1}f=\left(\frac{\partial f}{\partial x}-\frac{y}{2}\frac{\partial f}{\partial z}\right),\;{\left(a^{T}\nabla \right)}_{2}f=\left(\frac{\partial f}{\partial y}+\frac{x}{2}\frac{\partial f}{\partial z}\right). \end{array}$$

We have the following proposition for Heisenberg group following Theorem 1.

**Proposition** **1.**
*For any smooth function $f\in C^{\infty }\left(H^{1}\right)$, one has*

Γ2(f,f)+Γ2z,π(f,f)=∥Hessa,zf∥2+R(∇f,∇f),

*where*

$$\begin{array}{ccc} \Lambda_{1}^{T} & = & (0,0,0,0,0,0,0,0,0);\\ \Lambda_{2}^{T} & = & (0,0,0,0,0,0,{\left(a^{T}\nabla \right)}_{2}f,-{\left(a^{T}\nabla \right)}_{1}f,0);\\ & & R_{ab}\left(\nabla f,\nabla f\right)-\Lambda_{1}^{T}Q^{T}Q\Lambda_{1}-\Lambda_{2}^{T}P^{T}P\Lambda_{2}+D^{T}D+E^{T}E\\ & = & -\Gamma_{1}\left(f,f\right)+\frac{1}{2}\Gamma_{1}^{z}\left(f,f\right)-{\left(a^{T}\nabla \right)}_{1}V\partial_{z}f{\left(a^{T}\nabla \right)}_{2}f\\ & & +{\left(a^{T}\nabla \right)}_{2}V\partial_{z}f{\left(a^{T}\nabla \right)}_{1}f\\ & & +\left[\frac{\partial^{2}V}{\partial x\partial x}+\frac{y^{2}}{4}\frac{\partial^{2}V}{\partial z\partial z}-y\frac{\partial^{2}V}{\partial x\partial z}\right]{\left|{\left(a^{T}\nabla \right)}_{1}f\right|}^{2}\\ & & +\left[\frac{\partial^{2}V}{\partial y\partial y}+\frac{x^{2}}{4}\frac{\partial^{2}V}{\partial z\partial z}+x\frac{\partial^{2}V}{\partial y\partial z}\right]{\left|{\left(a^{T}\nabla \right)}_{2}f\right|}^{2}\\ & & +2\left[\frac{\partial^{2}V}{\partial x\partial y}+\frac{x}{2}\frac{\partial^{2}V}{\partial x\partial z}-\frac{y}{2}\frac{\partial^{2}V}{\partial y\partial z}-\frac{xy}{4}\frac{\partial^{2}V}{\partial z\partial z}\right]{\left(a^{T}\nabla \right)}_{1}f{\left(a^{T}\nabla \right)}_{2}f;\\ R_{zb}\left(\nabla f,\nabla f\right) & = & \left(\frac{\partial^{2}V}{\partial x\partial z}-\frac{y}{2}\frac{\partial^{2}V}{\partial z\partial z}\right){\left(a^{T}\nabla \right)}_{1}f{\left(z^{T}\nabla \right)}_{1}f\\ & & +\left(\frac{\partial^{2}V}{\partial y\partial z}+\frac{x}{2}\frac{\partial^{2}V}{\partial z\partial z}\right){\left(z^{T}\nabla \right)}_{1}f{\left(a^{T}\nabla \right)}_{2}f;\\ R_{\pi }\left(\nabla f,\nabla f\right) & = & 0. \end{array}$$



The proof of Proposition of 1 follows from the proof of Theorem 1 (i.e., Theorem 3) and Lemmas 1–3. The following convergence result follows directly from Theorem 2.

**Proposition** **2.**
*If there exists κ>0 as shown in Theorem 2, the exponential dissipation result in the $L^{1}$ distance holds:*

∫|ρ(t,x)−π(x)|dx=O(e−κt).



We next formulate the curvature tensor into a matrix format. Denote
(24)$$\begin{array}{c} U={\left({\left(a^{T}\nabla \right)}_{1}f,{\left(a^{T}\nabla \right)}_{2}f,{\left(z^{T}\nabla \right)}_{1}f\right)}_{3\times 1}, \end{array}$$
and denote $I_{3\times 3}$ as the identity matrix. With a little abuse of notation, there exists a symmetric matrix R such that we can represent the tensor as below.
(25)$$\begin{array}{c} R\left(\nabla f,\nabla f\right)={\left(U\right)}^{T}\cdot R\cdot U, \end{array}$$
which implies that
$$\begin{array}{c} R⪰\kappa \left(aa^{T}+zz^{T}\right)\Rightarrow R\left(\nabla f,\nabla f\right)⪰\kappa \left(\Gamma_{1}\left(f,f\right)+\Gamma_{1}^{z}\left(f,f\right)\right). \end{array}$$

In other words, we need to estimate the smallest eigenvalue of matrix R. We next present the formulation of matrix R for the Heisenberg group as follows.

**Corollary** **1.**
*The matrix R associated with the Heisenberg group has the following form:*

$$\begin{array}{ccc} R_{11} & = & \left[\frac{\partial^{2}V}{\partial x\partial x}+\frac{y^{2}}{4}\frac{\partial^{2}V}{\partial z\partial z}-y\frac{\partial^{2}V}{\partial x\partial z}\right]-1;\\ R_{22} & = & \left[\frac{\partial^{2}V}{\partial y\partial y}+\frac{x^{2}}{4}\frac{\partial^{2}V}{\partial z\partial z}+x\frac{\partial^{2}V}{\partial y\partial z}\right]-1;\;R_{33}=\frac{1}{2};\\ R_{12} & = & R_{21}=\left[\frac{\partial^{2}V}{\partial x\partial y}+\frac{x}{2}\frac{\partial^{2}V}{\partial x\partial z}-\frac{y}{2}\frac{\partial^{2}V}{\partial y\partial z}-\frac{xy}{4}\frac{\partial^{2}V}{\partial z\partial z}\right];\\ R_{13} & = & R_{31}=\frac{1}{2}{\left(a^{T}\nabla \right)}_{2}V+\frac{1}{2}\left(\frac{\partial^{2}V}{\partial x\partial z}-\frac{y}{2}\frac{\partial^{2}V}{\partial z\partial z}\right);\\ R_{23} & = & R_{32}=-\frac{1}{2}{\left(a^{T}\nabla \right)}_{1}V+\frac{1}{2}\left(\frac{\partial^{2}V}{\partial y\partial z}+\frac{x}{2}\frac{\partial^{2}V}{\partial z\partial z}\right). \end{array}$$



**Proof.** The explicit form of matrix R follows from the definition in Theorem 1 and the notation in (Equation 24) and (Equation 25). We have
$$\begin{array}{ccc} R(\nabla f,\nabla f) & = & -\Lambda_{1}^{T}Q^{T}Q\Lambda_{1}-\Lambda_{2}^{T}P^{T}P\Lambda_{2}+D^{T}D+E^{T}E\\ & & +R_{ab}\left(\nabla f,\nabla f\right)+R_{zb}\left(\nabla f,\nabla f\right)+R^{\pi }\left(\nabla f,\nabla f\right)\\ & = & {\left(U\right)}^{T}\cdot R\cdot U. \end{array}$$Plugging the explicit representation from Proposition 1 into the above formula and applying matrix symmetrization for the off-diagonal terms, we obtain the desired matrix R. □

Next, we present the three key lemmas.

**Lemma** **1.**
*For the Heisenberg group, we have*

$$\begin{array}{ccc} Q & = & \left(\begin{array}{ccccccccc} 1 & 0 & -\frac{y}{2} & 0 & 0 & 0 & -\frac{y}{2} & 0 & \frac{y^{2}}{4}\\ 0 & 1 & \frac{x}{2} & 0 & 0 & 0 & 0 & -\frac{y}{2} & -\frac{xy}{4}\\ 0 & 0 & 0 & 1 & 0 & -\frac{y}{2} & \frac{x}{2} & 0 & -\frac{xy}{4}\\ 0 & 0 & 0 & 0 & 1 & \frac{x}{2} & 0 & \frac{x}{2} & \frac{x^{2}}{4} \end{array}\right);\\ P & = & \left(\begin{array}{ccccccccc} 0 & 0 & 0 & 0 & 0 & 0 & 1 & 0 & -\frac{y}{2}\\ 0 & 0 & 0 & 0 & 0 & 0 & 0 & 1 & \frac{x}{2} \end{array}\right);\\ D^{T} & = & \left(0,\frac{1}{2}\partial_{z}f,-\frac{1}{2}\partial_{z}f,0\right);\;E^{T}=\left(0,0\right);\\ F^{T} & = & G^{T}=\left(0,0,0,0,0,0,0,0,0\right);\\ C^{T} & = & (0,0,\frac{x}{4}\partial_{z}f+\frac{1}{2}\partial_{y}f,0,0,\frac{y}{4}\partial_{z}f-\frac{1}{2}\partial_{x}f\\ & & ,\frac{x}{4}\partial_{z}f+\frac{1}{2}\partial_{y}f,\frac{y}{4}\partial_{z}f-\frac{1}{2}\partial_{x}f,-\frac{y}{2}\partial_{y}f-\frac{x}{2}\partial_{x}f). \end{array}$$



**Proof.** The proof of this lemma follows from routine computations. Plugging matrices *a* and *z* from (Equation 23) into Notation 1, we obtain the desired vectors and matrices. We skip the detailed computation here. □

**Lemma** **2.**
*On $H^{1}$, vectors F and G are zero vectors, and we have*

[QX+D]T[QX+D]+[PX+E]T[PX+E]+2CTX=∥Hessa,zf∥2−Λ1TQTQΛ1−Λ2TPTPΛ2+DTD+ETE.


*In particular, we have*

$$\begin{array}{ccc} ∥{\mathrm{Hess}}_{a,z}{f∥}^{2} & = & {\left[X+\Lambda_{1}\right]}^{T}Q^{T}Q\left[X+\Lambda_{1}\right]+{\left[X+\Lambda_{2}\right]}^{T}P^{T}P\left[X+\Lambda_{2}\right];\\ \Lambda_{1}^{T} & = & (0,0,0,0,0,0,0,0,0);\\ \Lambda_{2}^{T} & = & (0,0,0,0,0,0,{\left(a^{T}\nabla \right)}_{2}f,-{\left(a^{T}\nabla \right)}_{1}f,0);\\ & & -\Lambda_{1}^{T}Q^{T}Q\Lambda_{1}-\Lambda_{2}^{T}P^{T}P\Lambda_{2}+D^{T}D+E^{T}E)\\ & = & -\Gamma_{1}\left(f,f\right)+\frac{1}{2}\Gamma_{1}^{z}\left(f,f\right). \end{array}$$



**Lemma** **3.**
*By routine computations, we obtain*

$$\begin{array}{ccc} R_{ab}\left(\nabla f,\nabla f\right) & = & -{\left(a^{T}\nabla \right)}_{1}V\partial_{z}f{\left(a^{T}\nabla \right)}_{2}f+{\left(a^{T}\nabla \right)}_{2}V\partial_{z}f{\left(a^{T}\nabla \right)}_{1}f\\ & & +\left[\frac{\partial^{2}V}{\partial x\partial x}+\frac{y^{2}}{4}\frac{\partial^{2}V}{\partial z\partial z}-y\frac{\partial^{2}V}{\partial x\partial z}\right]{\left|{\left(a^{T}\nabla \right)}_{1}f\right|}^{2}\\ & & +\left[\frac{\partial^{2}V}{\partial y\partial y}+\frac{x^{2}}{4}\frac{\partial^{2}V}{\partial z\partial z}+x\frac{\partial^{2}V}{\partial y\partial z}\right]{\left|{\left(a^{T}\nabla \right)}_{2}f\right|}^{2}\\ & & +2\left[\frac{\partial^{2}V}{\partial x\partial y}+\frac{x}{2}\frac{\partial^{2}V}{\partial x\partial z}-\frac{y}{2}\frac{\partial^{2}V}{\partial y\partial z}-\frac{xy}{4}\frac{\partial^{2}V}{\partial z\partial z}\right]{\left(a^{T}\nabla \right)}_{1}f{\left(a^{T}\nabla \right)}_{2}f;\\ R_{zb}\left(\nabla f,\nabla f\right) & = & \left(\frac{\partial^{2}V}{\partial x\partial z}-\frac{y}{2}\frac{\partial^{2}V}{\partial z\partial z}\right){\left(a^{T}\nabla \right)}_{1}f{\left(z^{T}\nabla \right)}_{1}f\\ & & +\left(\frac{\partial^{2}V}{\partial y\partial z}+\frac{x}{2}\frac{\partial^{2}V}{\partial z\partial z}\right){\left(z^{T}\nabla \right)}_{1}f{\left(a^{T}\nabla \right)}_{2}f;\\ R_{\pi }\left(\nabla f,\nabla f\right) & = & 0. \end{array}$$



**Proof** **of** **Lemma** **2.**We first have
$$\begin{array}{ccc} 2C^{T}X & = & \sum_{\hat{i},\hat{k}=1}^{3}2C_{\hat{i}\hat{k}}^{T}X_{\hat{i}\hat{k}}\\ & = & 2\left[\frac{\partial^{2}f}{\partial x\partial z}\frac{{\left(a^{T}\nabla \right)}_{2}f}{2}-\frac{\partial^{2}f}{\partial y\partial z}\frac{{\left(a^{T}\nabla \right)}_{1}f}{2}+\frac{\partial^{2}f}{\partial z\partial x}\frac{{\left(a^{T}\nabla \right)}_{2}f}{2}\right]\\ & & -2\left[\frac{\partial^{2}f}{\partial z\partial y}\frac{{\left(a^{T}\nabla \right)}_{1}f}{2}+\frac{\partial^{2}f}{\partial z\partial z}\left(\frac{y}{2}\partial_{y}f+\frac{x}{2}\partial_{x}f\right)\right]\\ & = & 2\frac{\partial^{2}f}{\partial x\partial z}{\left(a^{T}\nabla \right)}_{2}f-2\frac{\partial^{2}f}{\partial y\partial z}{\left(a^{T}\nabla \right)}_{1}f-2\frac{\partial^{2}f}{\partial z\partial z}\left(\frac{y}{2}\partial_{y}f+\frac{x}{2}\partial_{x}f\right)\\ & = & 2{\left(a^{T}\nabla \right)}_{2}f\left[\frac{\partial^{2}f}{\partial x\partial z}-\frac{y}{2}\frac{\partial^{2}f}{\partial z\partial z}\right]-2{\left(a^{T}\nabla \right)}_{1}f\left[\frac{\partial^{2}f}{\partial y\partial z}+\frac{x}{2}\frac{\partial^{2}f}{\partial z\partial z}\right]. \end{array}$$By direct computations, we have
$$\begin{array}{ccc} & & {\left[QX+D\right]}^{T}\left[QX+D\right]+{\left[PX+E\right]}^{T}\left[PX+E\right]+2C^{T}X\\ & = & {\left[\frac{\partial^{2}f}{\partial x\partial x}-y\frac{\partial^{2}f}{\partial x\partial z}+\frac{y^{2}}{4}\frac{\partial^{2}f}{\partial z\partial z}\right]}^{2}+{\left[\frac{\partial^{2}f}{\partial x\partial y}+\frac{x}{2}\frac{\partial^{2}f}{\partial x\partial z}-\frac{y}{2}\frac{\partial^{2}f}{\partial y\partial z}-\frac{xy}{4}\frac{\partial^{2}f}{\partial z\partial z}+\frac{1}{2}\partial_{z}f\right]}^{2}\\ & & +{\left[\frac{\partial^{2}f}{\partial x\partial y}-\frac{y}{2}\frac{\partial^{2}f}{\partial y\partial z}+\frac{x}{2}\frac{\partial^{2}f}{\partial x\partial z}-\frac{xy}{4}\frac{\partial^{2}f}{\partial z\partial z}-\frac{1}{2}\partial_{z}f\right]}^{2}+{\left[\frac{\partial^{2}f}{\partial y\partial y}+x\frac{\partial^{2}f}{\partial y\partial z}+\frac{x^{2}}{4}\frac{\partial^{2}f}{\partial z\partial z}\right]}^{2}\\ & & +{\left[\frac{\partial^{2}f}{\partial x\partial z}-\frac{y}{2}\frac{\partial^{2}f}{\partial z\partial z}\right]}^{2}+{\left[\frac{\partial^{2}f}{\partial y\partial z}+\frac{x}{2}\frac{\partial^{2}f}{\partial z\partial z}\right]}^{2}\\ & & +2{\left(a^{T}\nabla \right)}_{2}f\left[\frac{\partial^{2}f}{\partial x\partial z}-\frac{y}{2}\frac{\partial^{2}f}{\partial z\partial z}\right]-2{\left(a^{T}\nabla \right)}_{1}f\left[\frac{\partial^{2}f}{\partial y\partial z}+\frac{x}{2}\frac{\partial^{2}f}{\partial z\partial z}\right]. \end{array}$$Completing the squares for the cross terms involving the type of $“\nabla f\nabla^{2}f$” and following the reformulation as below:
$$\begin{array}{ccc} & & {\left[\frac{\partial^{2}f}{\partial x\partial y}+\frac{x}{2}\frac{\partial^{2}f}{\partial x\partial z}-\frac{y}{2}\frac{\partial^{2}f}{\partial y\partial z}-\frac{xy}{4}\frac{\partial^{2}f}{\partial z\partial z}+\frac{1}{2}\partial_{z}f\right]}^{2}\\ & & +{\left[\frac{\partial^{2}f}{\partial x\partial y}-\frac{y}{2}\frac{\partial^{2}f}{\partial y\partial z}+\frac{x}{2}\frac{\partial^{2}f}{\partial x\partial z}-\frac{xy}{4}\frac{\partial^{2}f}{\partial z\partial z}-\frac{1}{2}\partial_{z}f\right]}^{2}\\ & = & 2{\left[\frac{\partial^{2}f}{\partial x\partial y}-\frac{y}{2}\frac{\partial^{2}f}{\partial y\partial z}+\frac{x}{2}\frac{\partial^{2}f}{\partial x\partial z}-\frac{xy}{4}\frac{\partial^{2}f}{\partial z\partial z}\right]}^{2}+\frac{1}{2}{\left|\partial_{z}f\right|}^{2}, \end{array}$$
we have
$$\begin{array}{ccc} & & {\left[QX+D\right]}^{T}\left[QX+D\right]+{\left[PX+E\right]}^{T}\left[PX+E\right]+2C^{T}X\\ & = & {\left[\frac{\partial^{2}f}{\partial x\partial x}-y\frac{\partial^{2}f}{\partial x\partial z}+\frac{y^{2}}{4}\frac{\partial^{2}f}{\partial z\partial z}\right]}^{2}+2{\left[\frac{\partial^{2}f}{\partial x\partial y}-\frac{y}{2}\frac{\partial^{2}f}{\partial y\partial z}+\frac{x}{2}\frac{\partial^{2}f}{\partial x\partial z}-\frac{xy}{4}\frac{\partial^{2}f}{\partial z\partial z}\right]}^{2}\\ & & +{\left[\frac{\partial^{2}f}{\partial y\partial y}+x\frac{\partial^{2}f}{\partial y\partial z}+\frac{x^{2}}{4}\frac{\partial^{2}f}{\partial z\partial z}\right]}^{2}+{\left[\frac{\partial^{2}f}{\partial x\partial z}-\frac{y}{2}\frac{\partial^{2}f}{\partial z\partial z}+{\left(a^{T}\nabla \right)}_{2}f\right]}^{2}\\ & & +{\left[\frac{\partial^{2}f}{\partial y\partial z}+\frac{x}{2}\frac{\partial^{2}f}{\partial z\partial z}-{\left(a^{T}\nabla \right)}_{1}f\right]}^{2}-|{\left(a^{T}\nabla \right)}_{2}{f|}^{2}-|{\left(a^{T}\nabla \right)}_{1}{f|}^{2}+\frac{1}{2}{\left|{\left(z^{T}\nabla \right)}_{1}f\right|}^{2}. \end{array}$$The sum of squares terms give $∥{\mathrm{Hess}}_{a,z}{∥}_{F}^{2}$, hence $\Lambda_{1}$ and $\Lambda_{2}$. The remainders generate $-\Lambda_{1}^{T}Q^{T}Q\Lambda_{1}-\Lambda_{2}^{T}P^{T}P\Lambda_{2}+D^{T}D+E^{T}E$, which equals $-\Gamma_{1}\left(f,f\right)+\frac{1}{2}\Gamma_{1}^{z}\left(f,f\right)$. □

We are now left to compute the tensors.

**Proof** **of** **Lemma** **3.**By direct computation, we have
$$\begin{array}{ccc} R_{a}\left(\nabla f,\nabla f\right) & = & \sum_{i,k=1}^{2}\sum_{i^{\prime },\hat{i},\hat{k}=1}^{3}\langle a_{ii^{\prime }}^{T}\left(\frac{\partial a_{i\hat{i}}^{T}}{\partial x_{i^{\prime }}}\frac{\partial a_{k\hat{k}}^{T}}{\partial x_{\hat{i}}}\frac{\partial f}{\partial x_{\hat{k}}}\right),{\left(a^{T}\nabla \right)}_{k}f\rangle {}_{R^{2}}\\ & & +\sum_{i,k=2}^{2}\sum_{i^{\prime },\hat{i},\hat{k}=1}^{3}\langle a_{ii^{\prime }}^{T}a_{i\hat{i}}^{T}\left(\frac{\partial }{\partial x_{i^{\prime }}}\frac{\partial a_{k\hat{k}}^{T}}{\partial x_{\hat{i}}}\right)\left(\frac{\partial f}{\partial x_{\hat{k}}}\right),{\left(a^{T}\nabla \right)}_{k}f\rangle {}_{R^{2}}\\ & & -\sum_{i,k=1}^{2}\sum_{i^{\prime },\hat{i},\hat{k}=1}^{3}\langle a_{k\hat{k}}^{T}\frac{\partial a_{ii^{\prime }}^{T}}{\partial x_{\hat{k}}}\frac{\partial a_{i\hat{i}}^{T}}{\partial x_{i^{\prime }}}\frac{\partial f}{\partial x_{\hat{i}}}),{\left(a^{T}\nabla \right)}_{k}{f\rangle }_{R^{2}}\\ & & -\sum_{i,k=1}^{2}\sum_{i^{\prime },\hat{i},\hat{k}=1}^{3}\langle a_{k\hat{k}}^{T}a_{ii^{\prime }}^{T}\left(\frac{\partial }{\partial x_{\hat{k}}}\frac{\partial a_{i\hat{i}}^{T}}{\partial x_{i^{\prime }}}\right)\frac{\partial f}{\partial x_{\hat{i}}},{\left(a^{T}\nabla \right)}_{k}f\rangle {}_{R^{2}},\\ & = & I_{1}+I_{2}+I_{3}+I_{4}. \end{array}$$For the four terms above, we have
$$\begin{array}{ccc} I_{1} & = & \sum_{i=1}^{2}\sum_{i^{\prime },\hat{i},\hat{k}=1}^{3}a_{ii^{\prime }}^{T}\left(\frac{\partial a_{i\hat{i}}^{T}}{\partial x_{i^{\prime }}}\frac{\partial a_{1\hat{k}}^{T}}{\partial x_{\hat{i}}}\frac{\partial f}{\partial x_{\hat{k}}}\right){\left(a^{T}\nabla \right)}_{1}f\\ & & +\sum_{i=1}^{2}\sum_{i^{\prime },\hat{i},\hat{k}=1}^{3}a_{ii^{\prime }}^{T}\left(\frac{\partial a_{i\hat{i}}^{T}}{\partial x_{i^{\prime }}}\frac{\partial a_{2\hat{k}}^{T}}{\partial x_{\hat{i}}}\frac{\partial f}{\partial x_{\hat{k}}}\right){\left(a^{T}\nabla \right)}_{2}f=0 \end{array}$$
$$\begin{array}{ccc} I_{2} & = & \sum_{i=2}^{n}\sum_{i^{\prime },\hat{i},\hat{k}=1}^{3}a_{ii^{\prime }}^{T}a_{i\hat{i}}^{T}\left(\frac{\partial }{\partial x_{i^{\prime }}}\frac{\partial a_{1\hat{k}}^{T}}{\partial x_{\hat{i}}}\right)\left(\frac{\partial f}{\partial x_{\hat{k}}}\right){\left(a^{T}\nabla \right)}_{1}f\\ & & +\sum_{i=2}^{n}\sum_{i^{\prime },\hat{i},\hat{k}=1}^{3}a_{ii^{\prime }}^{T}a_{i\hat{i}}^{T}\left(\frac{\partial }{\partial x_{i^{\prime }}}\frac{\partial a_{2\hat{k}}^{T}}{\partial x_{\hat{i}}}\right)\left(\frac{\partial f}{\partial x_{\hat{k}}}\right){\left(a^{T}\nabla \right)}_{2}f=0\\ I_{3} & = & -\sum_{i=1}^{2}\sum_{i^{\prime },\hat{i},\hat{k}=1}^{3}a_{1\hat{k}}^{T}\frac{\partial a_{ii^{\prime }}^{T}}{\partial x_{\hat{k}}}\frac{\partial a_{i\hat{i}}^{T}}{\partial x_{i^{\prime }}}\frac{\partial f}{\partial x_{\hat{i}}}){\left(a^{T}\nabla \right)}_{1}f\\ & & -\sum_{i=1}^{2}\sum_{i^{\prime },\hat{i},\hat{k}=1}^{3}a_{2\hat{k}}^{T}\frac{\partial a_{ii^{\prime }}^{T}}{\partial x_{\hat{k}}}\frac{\partial a_{i\hat{i}}^{T}}{\partial x_{i^{\prime }}}\frac{\partial f}{\partial x_{\hat{i}}}){\left(a^{T}\nabla \right)}_{2}f=0\\ I_{4} & = & -\sum_{i=1}^{2}\sum_{i^{\prime },\hat{i},\hat{k}=1}^{3}a_{1\hat{k}}^{T}a_{ii^{\prime }}^{T}\left(\frac{\partial }{\partial x_{\hat{k}}}\frac{\partial a_{i\hat{i}}^{T}}{\partial x_{i^{\prime }}}\right)\frac{\partial f}{\partial x_{\hat{i}}}{\left(a^{T}\nabla \right)}_{1}f\\ & & -\sum_{i=1}^{2}\sum_{i^{\prime },\hat{i},\hat{k}=1}^{3}a_{2\hat{k}}^{T}a_{ii^{\prime }}^{T}\left(\frac{\partial }{\partial x_{\hat{k}}}\frac{\partial a_{i\hat{i}}^{T}}{\partial x_{i^{\prime }}}\right)\frac{\partial f}{\partial x_{\hat{i}}}{\left(a^{T}\nabla \right)}_{2}f=0. \end{array}$$Similar computation applies to the tensor terms $R_{\pi }$ and $R_{zb}$. Since *z* is a constant matrix, we obtain
$$\begin{array}{ccc} R_{zb}\left(\nabla f,\nabla f\right) & = & -\sum_{\hat{i},\hat{k}=1}^{3}\langle \left(z_{1\hat{i}}^{T}\frac{\partial b_{\hat{k}}}{\partial x_{\hat{i}}}\frac{\partial f}{\partial x_{\hat{k}}}-b_{\hat{k}}\frac{\partial z_{1\hat{i}}^{T}}{\partial x_{\hat{k}}}\frac{\partial f}{\partial x_{\hat{i}}}\right),{\left(z^{T}\nabla f\right)}_{1}\rangle {}_{R},\\ R_{\pi } & = & 0. \end{array}$$We now compute the tensor terms involving the drift *b*. For the drift term in tensor $R_{ab}$, taking $b=-aa^{T}\nabla V$, which means $b=-{\left(a_{\hat{k}k}a_{kk^{\prime }}^{T}\frac{\partial V}{\partial x_{k^{\prime }}}\right)}_{\hat{k}=1,2,3}$ in local coordinates,
$$\begin{array}{ccc} R_{b}^{a} & = & \sum_{i,k=1}^{2}\sum_{\hat{i},\hat{k},k^{\prime }=1}^{3}\left[a_{i\hat{i}}^{T}\frac{\partial a_{k\hat{k}}^{T}}{\partial x_{\hat{i}}}a_{kk^{\prime }}^{T}\frac{\partial V}{\partial x_{k^{\prime }}}\frac{\partial f}{\partial x_{\hat{k}}}{\left(a^{T}\nabla \right)}_{i}f\right]\\ & & +\sum_{i,k=1}^{2}\sum_{\hat{i},\hat{k},k^{\prime }=1}^{3}\left[a_{i\hat{i}}^{T}\frac{\partial a_{kk^{\prime }}^{T}}{\partial x_{\hat{i}}}a_{k\hat{k}}^{T}\frac{\partial V}{\partial x_{k^{\prime }}}\frac{\partial f}{\partial x_{\hat{k}}}{\left(a^{T}\nabla \right)}_{i}f\right]\\ & & +\sum_{i,k=1}^{2}\sum_{\hat{i},\hat{k},k^{\prime }=1}^{3}\left[a_{i\hat{i}}^{T}a_{k\hat{k}}^{T}a_{kk^{\prime }}^{T}\frac{\partial^{2}V}{\partial x_{\hat{i}}\partial x_{k^{\prime }}}\frac{\partial f}{\partial x_{\hat{k}}}{\left(a^{T}\nabla \right)}_{i}f\right]\\ & & -\sum_{i,k=1}^{2}\sum_{\hat{i},\hat{k},k^{\prime }=1}^{3}\left[a_{k\hat{k}}^{T}a_{kk^{\prime }}^{T}\frac{\partial a_{i\hat{i}}^{T}}{\partial x_{\hat{k}}}\frac{\partial V}{\partial x_{k^{\prime }}}\frac{\partial f}{\partial x_{\hat{i}}}{\left(a^{T}\nabla \right)}_{i}f\right]\\ & = & J_{1}+J_{2}+J_{3}+J_{4}. \end{array}$$We now derive the explicit formulas for the above four terms.
$$\begin{array}{ccc} J_{1} & = & \sum_{\hat{i},\hat{k},k^{\prime }=1}^{3}\left[a_{1\hat{i}}^{T}\frac{\partial a_{1\hat{k}}^{T}}{\partial x_{\hat{i}}}a_{1k^{\prime }}^{T}\frac{\partial V}{\partial x_{k^{\prime }}}\frac{\partial f}{\partial x_{\hat{k}}}{\left(a^{T}\nabla \right)}_{1}f+a_{2\hat{i}}^{T}\frac{\partial a_{1\hat{k}}^{T}}{\partial x_{\hat{i}}}a_{1k^{\prime }}^{T}\frac{\partial V}{\partial x_{k^{\prime }}}\frac{\partial f}{\partial x_{\hat{k}}}{\left(a^{T}\nabla \right)}_{2}f\right]\\ & & +\sum_{\hat{i},\hat{k},k^{\prime }=1}^{3}\left[a_{1\hat{i}}^{T}\frac{\partial a_{2\hat{k}}^{T}}{\partial x_{\hat{i}}}a_{2k^{\prime }}^{T}\frac{\partial V}{\partial x_{k^{\prime }}}\frac{\partial f}{\partial x_{\hat{k}}}{\left(a^{T}\nabla \right)}_{1}f+a_{2\hat{i}}^{T}\frac{\partial a_{2\hat{k}}^{T}}{\partial x_{\hat{i}}}a_{2k^{\prime }}^{T}\frac{\partial V}{\partial x_{k^{\prime }}}\frac{\partial f}{\partial x_{\hat{k}}}{\left(a^{T}\nabla \right)}_{2}f\right]\\ & = & -\frac{1}{2}{\left(a^{T}\nabla \right)}_{1}V\partial_{z}f{\left(a^{T}\nabla \right)}_{2}f+\frac{1}{2}{\left(a^{T}\nabla \right)}_{2}V\partial_{z}f{\left(a^{T}\nabla \right)}_{1}f; \end{array}$$
$$\begin{array}{ccc} J_{2} & = & \sum_{\hat{i},\hat{k},k^{\prime }=1}^{3}\left[a_{1\hat{i}}^{T}\frac{\partial a_{1k^{\prime }}^{T}}{\partial x_{\hat{i}}}a_{1\hat{k}}^{T}\frac{\partial V}{\partial x_{k^{\prime }}}\frac{\partial f}{\partial x_{\hat{k}}}{\left(a^{T}\nabla \right)}_{1}f+a_{2\hat{i}}^{T}\frac{\partial a_{1k^{\prime }}^{T}}{\partial x_{\hat{i}}}a_{1\hat{k}}^{T}\frac{\partial V}{\partial x_{k^{\prime }}}\frac{\partial f}{\partial x_{\hat{k}}}{\left(a^{T}\nabla \right)}_{2}f\right]\\ & & \sum_{\hat{i},\hat{k},k^{\prime }=1}^{3}\left[a_{1\hat{i}}^{T}\frac{\partial a_{2k^{\prime }}^{T}}{\partial x_{\hat{i}}}a_{2\hat{k}}^{T}\frac{\partial V}{\partial x_{k^{\prime }}}\frac{\partial f}{\partial x_{\hat{k}}}{\left(a^{T}\nabla \right)}_{1}f+a_{2\hat{i}}^{T}\frac{\partial a_{2k^{\prime }}^{T}}{\partial x_{\hat{i}}}a_{2\hat{k}}^{T}\frac{\partial V}{\partial x_{k^{\prime }}}\frac{\partial f}{\partial x_{\hat{k}}}{\left(a^{T}\nabla \right)}_{2}f\right]\\ & = & -\frac{1}{2}\frac{\partial V}{\partial z}{\left(a^{T}\nabla \right)}_{1}f{\left(a^{T}\nabla \right)}_{2}f+\frac{1}{2}\frac{\partial V}{\partial z}{\left(a^{T}\nabla \right)}_{1}f{\left(a^{T}\nabla \right)}_{2}f=0;\\ J_{3} & = & \sum_{\hat{i},\hat{k},k^{\prime }=1}^{3}\left[a_{1\hat{i}}^{T}a_{1\hat{k}}^{T}a_{1k^{\prime }}^{T}\frac{\partial^{2}V}{\partial x_{\hat{i}}\partial x_{k^{\prime }}}\frac{\partial f}{\partial x_{\hat{k}}}{\left(a^{T}\nabla \right)}_{1}f+a_{2\hat{i}}^{T}a_{1\hat{k}}^{T}a_{1k^{\prime }}^{T}\frac{\partial^{2}V}{\partial x_{\hat{i}}\partial x_{k^{\prime }}}\frac{\partial f}{\partial x_{\hat{k}}}{\left(a^{T}\nabla \right)}_{2}f\right]\\ & & \sum_{\hat{i},\hat{k},k^{\prime }=1}^{3}\left[a_{1\hat{i}}^{T}a_{2\hat{k}}^{T}a_{2k^{\prime }}^{T}\frac{\partial^{2}V}{\partial x_{\hat{i}}\partial x_{k^{\prime }}}\frac{\partial f}{\partial x_{\hat{k}}}{\left(a^{T}\nabla \right)}_{1}f+a_{2\hat{i}}^{T}a_{2\hat{k}}^{T}a_{2k^{\prime }}^{T}\frac{\partial^{2}V}{\partial x_{\hat{i}}\partial x_{k^{\prime }}}\frac{\partial f}{\partial x_{\hat{k}}}{\left(a^{T}\nabla \right)}_{2}f\right]\\ & = & \sum_{\hat{i},k^{\prime }=1}^{3}\left[a_{1\hat{i}}^{T}a_{1k^{\prime }}^{T}\frac{\partial^{2}V}{\partial x_{\hat{i}}\partial x_{k^{\prime }}}{\left|{\left(a^{T}\nabla \right)}_{1}f\right|}^{2}+a_{2\hat{i}}^{T}a_{1k^{\prime }}^{T}\frac{\partial^{2}V}{\partial x_{\hat{i}}\partial x_{k^{\prime }}}{\left(a^{T}\nabla \right)}_{1}f{\left(a^{T}\nabla \right)}_{2}f\right]\\ & & \sum_{\hat{i},k^{\prime }=1}^{3}\left[a_{1\hat{i}}^{T}a_{2k^{\prime }}^{T}\frac{\partial^{2}V}{\partial x_{\hat{i}}\partial x_{k^{\prime }}}{\left(a^{T}\nabla \right)}_{2}f{\left(a^{T}\nabla \right)}_{1}f+a_{2\hat{i}}^{T}a_{2k^{\prime }}^{T}\frac{\partial^{2}V}{\partial x_{\hat{i}}\partial x_{k^{\prime }}}{\left|{\left(a^{T}\nabla \right)}_{2}f\right|}^{2}\right]\\ & = & \left[\frac{\partial^{2}V}{\partial x\partial x}+\frac{y^{2}}{4}\frac{\partial^{2}V}{\partial z\partial z}-y\frac{\partial^{2}V}{\partial x\partial z}\right]{\left|{\left(a^{T}\nabla \right)}_{1}f\right|}^{2}\\ & & +\left[\frac{\partial^{2}V}{\partial y\partial y}+\frac{x^{2}}{4}\frac{\partial^{2}V}{\partial z\partial z}+x\frac{\partial^{2}V}{\partial y\partial z}\right]{\left|{\left(a^{T}\nabla \right)}_{2}f\right|}^{2}\\ & & +2\left[\frac{\partial^{2}V}{\partial x\partial y}+\frac{x}{2}\frac{\partial^{2}V}{\partial x\partial z}-\frac{y}{2}\frac{\partial^{2}V}{\partial y\partial z}-\frac{xy}{4}\frac{\partial^{2}V}{\partial z\partial z}\right]{\left(a^{T}\nabla \right)}_{1}f{\left(a^{T}\nabla \right)}_{2}f;\\ J_{4} & = & -\sum_{\hat{i},\hat{k},k^{\prime }=1}^{3}\left[a_{1\hat{k}}^{T}a_{1k^{\prime }}^{T}\frac{\partial a_{1\hat{i}}^{T}}{\partial x_{\hat{k}}}\frac{\partial V}{\partial x_{k^{\prime }}}\frac{\partial f}{\partial x_{\hat{i}}}{\left(a^{T}\nabla \right)}_{1}f+a_{1\hat{k}}^{T}a_{1k^{\prime }}^{T}\frac{\partial a_{2\hat{i}}^{T}}{\partial x_{\hat{k}}}\frac{\partial V}{\partial x_{k^{\prime }}}\frac{\partial f}{\partial x_{\hat{i}}}{\left(a^{T}\nabla \right)}_{2}f\right]\\ & & -\sum_{\hat{i},\hat{k},k^{\prime }=1}^{3}\left[a_{2\hat{k}}^{T}a_{2k^{\prime }}^{T}\frac{\partial a_{1\hat{i}}^{T}}{\partial x_{\hat{k}}}\frac{\partial V}{\partial x_{k^{\prime }}}\frac{\partial f}{\partial x_{\hat{i}}}{\left(a^{T}\nabla \right)}_{1}f+a_{2\hat{k}}^{T}a_{2k^{\prime }}^{T}\frac{\partial a_{2\hat{i}}^{T}}{\partial x_{\hat{k}}}\frac{\partial V}{\partial x_{k^{\prime }}}\frac{\partial f}{\partial x_{\hat{i}}}{\left(a^{T}\nabla \right)}_{2}f\right]\\ & = & -\frac{1}{2}{\left(a^{T}\nabla \right)}_{1}V\partial_{z}f{\left(a^{T}\nabla \right)}_{2}f+\frac{1}{2}{\left(a^{T}\nabla \right)}_{2}V\partial_{z}f{\left(a^{T}\nabla \right)}_{1}f. \end{array}$$Summing up the above formulas, we obtain $R_{ab}$. We now compute the drift tensor term of $R_{zb}$. By taking $b=-aa^{T}\nabla V$, we have
$$\begin{array}{ccc} R_{b}^{z}\left(\nabla f,\nabla f\right) & = & -\sum_{\hat{i},\hat{k}=1}^{3}\left[z_{1\hat{i}}^{T}\frac{\partial b_{\hat{k}}}{\partial x_{\hat{i}}}\frac{\partial f}{\partial x_{\hat{k}}}{\left(z^{T}\nabla f\right)}_{1}-b_{\hat{k}}\frac{\partial z_{i\hat{i}}^{T}}{\partial x_{\hat{k}}}\frac{\partial f}{\partial x_{\hat{i}}}{\left(z^{T}\nabla f\right)}_{1}\right]\\ & = & \sum_{k=1}^{2}\sum_{\hat{i},\hat{k},k^{\prime }=1}^{3}\left[z_{1\hat{i}}^{T}\frac{\partial a_{k\hat{k}}^{T}}{\partial x_{\hat{i}}}a_{kk^{\prime }}^{T}\frac{\partial V}{\partial x_{k^{\prime }}}\frac{\partial f}{\partial x_{\hat{k}}}{\left(z^{T}\nabla \right)}_{1}f\right]\\ & & +\sum_{k=1}^{2}\sum_{\hat{i},\hat{k},k^{\prime }=1}^{3}\left[z_{1\hat{i}}^{T}\frac{\partial a_{kk^{\prime }}^{T}}{\partial x_{\hat{i}}}a_{k\hat{k}}^{T}\frac{\partial V}{\partial x_{k^{\prime }}}\frac{\partial f}{\partial x_{\hat{k}}}{\left(z^{T}\nabla \right)}_{1}f\right]\\ & & +\sum_{k=1}^{2}\sum_{\hat{i},\hat{k},k^{\prime }=1}^{3}\left[z_{1\hat{i}}^{T}a_{k\hat{k}}^{T}a_{kk^{\prime }}^{T}\frac{\partial^{2}V}{\partial x_{\hat{i}}\partial x_{k^{\prime }}}\frac{\partial f}{\partial x_{\hat{k}}}{\left(z^{T}\nabla \right)}_{1}f\right]\\ & & -\sum_{k=1}^{2}\sum_{\hat{i},\hat{k},k^{\prime }=1}^{3}\left[a_{k\hat{k}}^{T}a_{kk^{\prime }}^{T}\frac{\partial z_{1\hat{i}}^{T}}{\partial x_{\hat{k}}}\frac{\partial V}{\partial x_{k^{\prime }}}\frac{\partial f}{\partial x_{\hat{i}}}{\left(z^{T}\nabla \right)}_{1}f\right]\\ & = & J_{1}^{z}+J_{2}^{z}+J_{3}^{z}+J_{4}^{z}. \end{array}$$We further compute as below by taking advantage of the constant matrix *z*:
$$\begin{array}{ccc} J_{1}^{z} & = & \sum_{k=1}^{2}\sum_{\hat{i},\hat{k},k^{\prime }=1}^{3}\left[z_{1\hat{i}}^{T}\frac{\partial a_{k\hat{k}}^{T}}{\partial x_{\hat{i}}}a_{kk^{\prime }}^{T}\frac{\partial V}{\partial x_{k^{\prime }}}\frac{\partial f}{\partial x_{\hat{k}}}{\left(z^{T}\nabla \right)}_{1}f\right]=0;\\ J_{2}^{z} & = & \sum_{k=1}^{2}\sum_{\hat{i},\hat{k},k^{\prime }=1}^{3}\left[z_{1\hat{i}}^{T}\frac{\partial a_{kk^{\prime }}^{T}}{\partial x_{\hat{i}}}a_{k\hat{k}}^{T}\frac{\partial V}{\partial x_{k^{\prime }}}\frac{\partial f}{\partial x_{\hat{k}}}{\left(z^{T}\nabla \right)}_{1}f\right]=0;\\ J_{4}^{z} & = & -\sum_{k=1}^{2}\sum_{\hat{i},\hat{k},k^{\prime }=1}^{3}\left[a_{k\hat{k}}^{T}a_{kk^{\prime }}^{T}\frac{\partial z_{1\hat{i}}^{T}}{\partial x_{\hat{k}}}\frac{\partial V}{\partial x_{k^{\prime }}}\frac{\partial f}{\partial x_{\hat{i}}}{\left(z^{T}\nabla \right)}_{1}f\right]=0\\ J_{3}^{z} & = & \sum_{k=1}^{2}\sum_{\hat{i},\hat{k},k^{\prime }=1}^{3}\left[z_{1\hat{i}}^{T}a_{k\hat{k}}^{T}a_{kk^{\prime }}^{T}\frac{\partial^{2}V}{\partial x_{\hat{i}}\partial x_{k^{\prime }}}\frac{\partial f}{\partial x_{\hat{k}}}{\left(z^{T}\nabla \right)}_{1}f\right]\\ & = & \left(\frac{\partial^{2}V}{\partial x\partial z}-\frac{y}{2}\frac{\partial^{2}V}{\partial z\partial z}\right){\left(a^{T}\nabla \right)}_{1}f{\left(z^{T}\nabla \right)}_{1}f+\left(\frac{\partial^{2}V}{\partial y\partial z}+\frac{x}{2}\frac{\partial^{2}V}{\partial z\partial z}\right){\left(z^{T}\nabla \right)}_{1}f{\left(a^{T}\nabla \right)}_{2}f. \end{array}$$The proof is thus completed. □

### 3.2. Displacement Group

In this section, we derive the generalized curvature dimension bound for the displacement group, which is one example of three-dimensional solvable Lie groups. We adapted the general setting from [49] below. Denote g as the three-dimensional solvable Lie algebra, and denote H⊂g as the horizontal subspace satisfying Hörmander’s condition, then for a given inner product 〈·,·〉 on *H*, there exists a canonical basis {X,Y,Z} for (g,H,〈·,·〉), such that {X,Y} forms an orthonormal basis for *H* and satisfies the following Lie-bracket-generating condition for parameters α and β≥0:[X,Y]=Z,[X,Z]=αY+βZ,[Y,Z]=0.

When the parameters α=0 and β≠0, the Lie algebra g has a faithful representation. In particular, it was shown in [49] that the elements of g, in local coordinates (θ,x,y), correspond to the following left-invariant differential operators:$$\begin{array}{c} X=\frac{\partial }{\partial \theta },\;Y=e^{\beta \theta }\frac{\partial }{\partial x}+\frac{\partial }{\partial y},\;R=-\beta \frac{\partial }{\partial y}, \end{array}$$
with the following relation:$$\begin{array}{c} {}[X,Y]=\beta Y+R,\;[X,R]=0,\;[Y,R]=0. \end{array}$$

In terms of local coordinates (θ,x,y), we have
$$X=\left(\begin{array}{c} 1\\ 0\\ 0 \end{array}\right),\;Y=\left(\begin{array}{c} 0\\ e^{\beta \theta }\\ 1 \end{array}\right),\;R=\left(\begin{array}{c} 0\\ 0\\ -\beta \end{array}\right).$$

The corresponding Lie group of this special Lie algebra g is called the displacement group, denoted as G. We chose {X,Y} as the horizontal orthonormal basis for subalgebra *H*. To fit into the general framework from the previous section, we take
(26)$$a=\left(X,Y\right)=\left(\begin{array}{cc} 1 & 0\\ 0 & e^{\beta \theta }\\ 0 & 1 \end{array}\right),\;a^{T}=\left(\begin{array}{ccc} 1 & 0 & 0\\ 0 & e^{\beta \theta } & 1 \end{array}\right),\;z^{T}=\left(\begin{array}{ccc} 0 & 0 & -g(\theta ,x,y) \end{array}\right),$$
with g(θ,x,y)≠0. Our focus here is to derive the curvature tensor in terms of $\pi =\frac{1}{Z}e^{-V}$. We then used ${\left(aa^{T}\right)}_{|H}^{†}$ as the horizontal metric on *H*. Thus, the sub-Riemannian structure is given by $\left(G,H,{\left(aa^{T}\right)}_{|H}^{†}\right)$. By direct computations, it is easy to show that, for general smooth function *f*, $\Gamma_{1}\left(f,\Gamma_{1}^{z}\left(f,f\right)\right)\neq \Gamma_{1}^{z}\left(f,\Gamma_{1}\left(f,f\right)\right)$. Hence, the classical Gamma *z* calculus proposed in [2] can not be extended for this case to derive the zLSI. Thus, we need to compute vector *G* and the tensor term $R_{\pi }$. Following Theorem 1, we have the following z-Bochner’s formula for G.

**Proposition** **3.**
*For any smooth function $f\in C^{\infty }\left(G\right)$, one has*

Γ2(f,f)+Γ2z,π(f,f)=∥Hessa,zf∥2+R(∇f,∇f),

*where*

$$\begin{array}{ccc} \Lambda_{1}^{T} & = & (0,\beta \partial_{x}f,\frac{\beta \partial_{y}f}{2},\beta \partial_{x}f,0,0,\frac{\beta \partial_{y}f}{2},0,-\beta \partial_{\theta }f);\\ \Lambda_{2}^{T} & = & (0,0,0,0,0,0,\lambda_{6},0,\lambda_{9});\\ \lambda_{6} & = & \frac{\partial_{\theta }g\partial_{y}f}{g}-\frac{\beta {\left(a^{T}\nabla \right)}_{2}f}{g^{2}}-\frac{\partial_{\theta }g\partial_{y}f}{g};\\ \lambda_{9} & = & \frac{{\left(a^{T}\nabla \right)}_{2}g\partial_{y}f}{g}+\frac{\beta \partial_{\theta }f}{g^{2}}-\frac{{\left(a^{T}\nabla \right)}_{2}g\partial_{y}f}{g}; \end{array}$$

*and*

$$\begin{array}{ccc} & & R_{ab}\left(\nabla f,\nabla f\right)-\Lambda_{1}^{T}Q^{T}Q\Lambda_{1}-\Lambda_{2}^{T}P^{T}P\Lambda_{2}+D^{T}D+E^{T}E)\\ & = & \Gamma_{1}\left(\log g,\log g\right)\Gamma_{1}^{z}\left(f,f\right)-\beta^{2}\left(1+\frac{1}{g^{2}}\right)\Gamma_{1}\left(f,f\right)+\frac{\beta^{2}}{2g^{2}}\Gamma_{1}^{z}\left(f,f\right)\\ & & +\beta^{2}e^{\beta \theta }\frac{\partial f}{\partial x}{\left(a^{T}\nabla \right)}_{2}f+\beta e^{\beta \theta }{\left(a^{T}\nabla \right)}_{2}V\frac{\partial f}{\partial x}{\left(a^{T}\nabla \right)}_{1}f\\ & & +\beta e^{\beta \theta }\frac{\partial V}{\partial x}{\left(a^{T}\nabla \right)}_{2}f{\left(a^{T}\nabla \right)}_{1}f\\ & & +\frac{\partial^{2}V}{\partial \theta \partial \theta }{\left|{\left(a^{T}\nabla \right)}_{1}f\right|}^{2}+2\left(e^{\beta \theta }\frac{\partial^{2}V}{\partial \theta \partial x}+\frac{\partial^{2}V}{\partial \theta \partial y}\right){\left(a^{T}\nabla \right)}_{1}f{\left(a^{T}\nabla \right)}_{2}f\\ & & +\sum_{\hat{i},k^{\prime }=1}^{3}a_{2\hat{i}}^{T}a_{2k^{\prime }}^{T}\frac{\partial^{2}V}{\partial x_{\hat{i}}\partial x_{k^{\prime }}})|{\left(a^{T}\nabla \right)}_{2}{f|}^{2}-\beta e^{\beta \theta }{\left(a^{T}\nabla \right)}_{1}V\frac{\partial f}{\partial x}{\left(a^{T}\nabla \right)}_{2}f;\\ & & R_{zb}\left(\nabla f,\nabla f\right)\\ & = & \sum_{i=1}^{2}\sum_{i^{\prime },\hat{i}=1}^{3}a_{ii^{\prime }}^{T}a_{i\hat{i}}^{T}\frac{\partial^{2}z_{13}^{T}}{\partial x_{i^{\prime }}\partial x_{\hat{i}}}\partial_{y}f{\left(z^{T}\nabla \right)}_{1}f-\sum_{k=1}^{2}{\left(a^{T}\nabla \right)}_{k}z_{13}^{T}{\left(a^{T}\nabla \right)}_{k}V\partial_{y}f{\left(z^{T}\nabla \right)}_{1}f\\ & & -g\frac{\partial^{2}V}{\partial \theta \partial y}{\left(a^{T}\nabla \right)}_{1}f{\left(z^{T}\nabla \right)}_{1}f-g\left(e^{\beta \theta }\frac{\partial^{2}V}{\partial x\partial y}+\frac{\partial^{2}V}{\partial y\partial y}\right){\left(a^{T}\nabla \right)}_{2}f{\left(z^{T}\nabla \right)}_{1}f;\\ & & R_{\pi }\left(\nabla f,\nabla f\right)\\ & = & -2\sum_{l=1}^{2}\sum_{l^{\prime },\hat{l}=1}^{3}a_{ll^{\prime }}^{T}a_{l\hat{l}}^{T}\frac{\partial^{2}z_{13}^{T}}{\partial x_{l^{\prime }}\partial x_{\hat{l}}}\partial_{y}f{\left(z^{T}\nabla \right)}_{1}f\\ & & -2\Gamma_{1}\left(\log\pi ,\log g\right)|{\left(z^{T}\nabla \right)}_{1}{f|}^{2}-2\Gamma_{1}\left(\log g,\log g\right){\left|{\left(z^{T}\nabla \right)}_{1}f\right|}^{2}. \end{array}$$


*In particular, we have*

$$\begin{array}{ccc} & & \sum_{\hat{i},k^{\prime }=1}^{3}a_{2\hat{i}}^{T}a_{2k^{\prime }}^{T}\frac{\partial^{2}V}{\partial x_{\hat{i}}\partial x_{k^{\prime }}}{\left|{\left(a^{T}\nabla \right)}_{2}f\right|}^{2}\\ & = & \left[e^{2\beta \theta }\frac{\partial^{2}V}{\partial x\partial x}+2e^{\beta \theta }\frac{\partial^{2}V}{\partial x\partial y}+\frac{\partial^{2}V}{\partial y\partial y}\right]{\left|{\left(a^{T}\nabla \right)}_{2}f\right|}^{2}; \end{array}$$


$$\begin{array}{ccc} & & \sum_{i=1}^{2}\sum_{i^{\prime },\hat{i}=1}^{3}a_{ii^{\prime }}^{T}a_{i\hat{i}}^{T}\frac{\partial^{2}z_{13}^{T}}{\partial x_{i^{\prime }}\partial x_{\hat{i}}}\partial_{y}f{\left(z^{T}\nabla \right)}_{1}f\\ & = & \left[\frac{\partial^{2}g}{\partial \theta \partial \theta }+e^{2\beta \theta }\frac{\partial^{2}g}{\partial x\partial x}+\frac{\partial^{2}g}{\partial y\partial y}+2e^{\beta \theta }\frac{\partial^{2}g}{\partial x\partial y}\right]\frac{|{\left(z^{T}\nabla \right)}_{1}{f|}^{2}}{g}. \end{array}$$



The proof of Proposition 3 follows from the proof of Theorem 1 (i.e., Theorem 3) and Lemmas 4–6 below. The following convergence result follows directly from Theorem 2.

**Proposition** **4.**
*If there exists κ>0 as shown in Theorem 2, the exponential dissipation result in the $L^{1}$ distance holds:*

∫|ρ(t,x)−π(x)|dx=O(e−κt).



Similarly, we formulated the curvature tensor into a matrix format of R. Using the fact $e^{\beta \theta }\frac{\partial f}{\partial x}={\left(a^{T}\nabla \right)}_{2}f+\frac{1}{g}{\left(z^{T}\nabla f\right)}_{1}f$, we have the following representation.

**Corollary** **2.**
*The matrix R associated with G has the following representation:*

$$\begin{array}{ccc} R_{11} & = & \frac{\partial^{2}V}{\partial \theta \partial \theta }-\beta^{2}\left(1+\frac{1}{g^{2}}\right);\\ R_{22} & = & \left[e^{2\beta \theta }\frac{\partial^{2}V}{\partial x\partial x}+2e^{\beta \theta }\frac{\partial^{2}V}{\partial x\partial y}+\frac{\partial^{2}V}{\partial y\partial y}\right]-\frac{\beta^{2}}{g^{2}}-\beta {\left(a^{T}\nabla \right)}_{1}V;\\ R_{33} & = & \frac{\beta^{2}}{2g^{2}}-\Gamma_{1}\left(\log g,\log g\right)-2\Gamma_{1}\left(\log\pi ,\log g\right)-\Gamma_{1}\left(\log g,V\right)\\ & & -\frac{1}{g}\left[\frac{\partial^{2}g}{\partial \theta \partial \theta }+e^{2\beta \theta }\frac{\partial^{2}g}{\partial x\partial x}+\frac{\partial^{2}g}{\partial y\partial y}+2e^{\beta \theta }\frac{\partial^{2}g}{\partial x\partial y}\right];\\ R_{12} & = & R_{21}=\frac{1}{2}\left(\beta e^{\beta \theta }\frac{\partial V}{\partial x}+2\left(e^{\beta \theta }\frac{\partial^{2}V}{\partial \theta \partial x}+\frac{\partial^{2}V}{\partial \theta \partial y}\right)+\beta {\left(a^{T}\nabla \right)}_{2}V\right);\\ R_{13} & = & R_{31}=\frac{1}{2}\left(\frac{\beta }{g}{\left(a^{T}\nabla \right)}_{2}V-g\frac{\partial^{2}V}{\partial \theta \partial y}\right);\\ R_{23} & = & R_{32}=-\frac{1}{2}\left(\frac{\beta }{g}{\left(a^{T}\nabla \right)}_{1}-\frac{\beta^{2}}{g}\right)-\frac{1}{2}g\left(e^{\beta \theta }\frac{\partial^{2}V}{\partial x\partial y}+\frac{\partial^{2}V}{\partial y\partial y}\right). \end{array}$$



**Proof.** The derivation for the explicit form of matrix R follows from a similar equivalent representation as shown in the proof of Corollary 1 and the explicit bilinear terms derived in Proposition 3. □

**Remark** **9.**
*By taking g(θ,x,y)=β as a constant, Proposition 3 reduces to a simple version; in particular, the tensors reduce to be*

$$\begin{array}{ccc} & & R_{ab}\left(\nabla f,\nabla f\right)-\Lambda_{1}^{T}Q^{T}Q\Lambda_{1}-\Lambda_{2}^{T}P^{T}P\Lambda_{2}+D^{T}D+E^{T}E)\\ & = & -\left(1+\beta^{2}\right)\Gamma_{1}\left(f,f\right)+\frac{1}{2}\Gamma_{1}^{z}\left(f,f\right)\\ & & +\beta^{2}e^{\beta \theta }\frac{\partial f}{\partial x}{\left(a^{T}\nabla \right)}_{2}f+\beta e^{\beta \theta }{\left(a^{T}\nabla \right)}_{2}V\frac{\partial f}{\partial x}{\left(a^{T}\nabla \right)}_{1}f+\beta e^{\beta \theta }\frac{\partial V}{\partial x}{\left(a^{T}\nabla \right)}_{2}f{\left(a^{T}\nabla \right)}_{1}f\\ & & +\frac{\partial^{2}V}{\partial \theta \partial \theta }{\left|{\left(a^{T}\nabla \right)}_{1}f\right|}^{2}+2\left(e^{\beta \theta }\frac{\partial^{2}V}{\partial \theta \partial x}+\frac{\partial^{2}V}{\partial \theta \partial y}\right){\left(a^{T}\nabla \right)}_{1}f{\left(a^{T}\nabla \right)}_{2}f\\ & & +\sum_{\hat{i},k^{\prime }=1}^{3}a_{2\hat{i}}^{T}a_{2k^{\prime }}^{T}\frac{\partial^{2}V}{\partial x_{\hat{i}}\partial x_{k^{\prime }}})|{\left(a^{T}\nabla \right)}_{2}{f|}^{2}-\beta e^{\beta \theta }{\left(a^{T}\nabla \right)}_{1}V\frac{\partial f}{\partial x}{\left(a^{T}\nabla \right)}_{2}f; \end{array}$$


$$\begin{array}{ccc} R_{zb}\left(\nabla f,\nabla f\right) & = & -\beta \frac{\partial^{2}V}{\partial \theta \partial y}{\left(a^{T}\nabla \right)}_{1}f{\left(z^{T}\nabla \right)}_{1}f\\ & & -\beta \left(e^{\beta \theta }\frac{\partial^{2}V}{\partial x\partial y}+\frac{\partial^{2}V}{\partial y\partial y}\right){\left(a^{T}\nabla \right)}_{2}f{\left(z^{T}\nabla \right)}_{1}f;\\ R_{\pi }\left(\nabla f,\nabla f\right) & = & 0. \end{array}$$



Next, we present the following three key lemmas.

**Lemma** **4.**
*For displacement group G, we have*

$$\begin{array}{ccc} Q & = & \left(\begin{array}{ccccccccc} 1 & 0 & 0 & 0 & 0 & 0 & 0 & 0 & 0\\ 0 & e^{\beta \theta } & 1 & 0 & 0 & 0 & 0 & 0 & 0\\ 0 & 0 & 0 & e^{\beta \theta } & 0 & 0 & 1 & 0 & 0\\ 0 & 0 & 0 & 0 & e^{2\beta \theta } & e^{\beta \theta } & 0 & e^{\beta \theta } & 1 \end{array}\right);\\ P & = & \left(\begin{array}{ccccccccc} 0 & 0 & 0 & 0 & 0 & 0 & -g(\theta ,x,y) & 0 & 0\\ 0 & 0 & 0 & 0 & 0 & 0 & 0 & -g\left(\theta ,x,y\right)e^{\beta \theta } & -g(\theta ,x,y) \end{array}\right);\\ D^{T} & = & \left(0,\beta e^{\beta \theta }\partial_{x}f,0,0\right),\;E^{T}=\left(-\partial_{y}f\partial_{\theta }g,-\partial_{y}f\partial_{y}g-e^{\beta \theta }\partial_{y}f\partial_{x}g\right);\\ C^{T} & = & (0,\beta e^{\beta \theta }\partial_{y}f+\beta e^{2\beta \theta }\partial_{x}f,0,0,-\beta e^{2\beta \theta }\partial_{\theta }f,-\beta e^{\beta \theta }\partial_{\theta }f,0,0,0). \end{array}$$

*$F=\left(\begin{array}{c} 0\\ 0\\ g\partial_{\theta }g\partial_{y}f\\ 0\\ 0\\ e^{\beta \theta }g\partial_{y}f\partial_{y}g+e^{2\beta \theta }g\partial_{y}f\partial_{x}g\\ 0\\ 0\\ g\partial_{y}f\partial_{y}g+e^{\beta \theta }g\partial_{y}f\partial_{x}g \end{array}\right),\;G=\left(\begin{array}{c} 0\\ 0\\ -2g\partial_{y}f\partial_{\theta }g\\ 0\\ 0\\ -2e^{\beta \theta }g\partial_{y}f\partial_{y}g-2e^{2\beta \theta }g\partial_{y}f\partial_{x}g\\ 0\\ 0\\ -2g\partial_{y}f\partial_{y}g-2e^{\beta \theta }g\partial_{y}f\partial_{x}g \end{array}\right)$.*


**Proof.** The proof follows by plugging matrices *a* and *z* from (Equation 26) into Notation 1. □

**Lemma** **5.**
*On displacement group G, we have*

$$\begin{array}{ccc} & & {\left[QX+D\right]}^{T}\left[QX+D\right]+{\left[PX+E\right]}^{T}\left[PX+E\right]+2\left[C^{T}+F^{T}+G^{T}\right]X\\ & = & ∥{\mathrm{Hess}}_{a,z}{f∥}^{2}-\Lambda_{1}^{T}Q^{T}Q\Lambda_{1}-\Lambda_{2}^{T}P^{T}P\Lambda_{2}+D^{T}D+E^{T}E. \end{array}$$


*In particular, we have*

$$\begin{array}{ccc} ∥{\mathrm{Hess}}_{a,z}{f∥}^{2} & = & {\left[X+\Lambda_{1}\right]}^{T}Q^{T}Q\left[X+\Lambda_{1}\right]+{\left[X+\Lambda_{2}\right]}^{T}P^{T}P\left[X+\Lambda_{2}\right];\\ \Lambda_{1}^{T} & = & (0,\beta \partial_{x}f,\frac{\beta \partial_{y}f}{2},\beta \partial_{x}f,0,0,\frac{\beta \partial_{y}f}{2},0,-\beta \partial_{\theta }f);\\ \Lambda_{2}^{T} & = & (0,0,0,0,0,0,\lambda_{6},0,\lambda_{9});\\ \lambda_{6} & = & \frac{\partial_{\theta }g\partial_{y}f}{g}-\frac{\beta {\left(a^{T}\nabla \right)}_{2}f}{g^{2}}-\frac{\partial_{\theta }g\partial_{y}f}{g}; \end{array}$$


$$\begin{array}{ccc} \lambda_{9} & = & \frac{{\left(a^{T}\nabla \right)}_{2}g\partial_{y}f}{g}+\frac{\beta \partial_{\theta }f}{g^{2}}-\frac{{\left(a^{T}\nabla \right)}_{2}g\partial_{y}f}{g};\\ & & -\Lambda_{1}^{T}Q^{T}Q\Lambda_{1}-\Lambda_{2}^{T}P^{T}P\Lambda_{2}+D^{T}D+E^{T}E\\ & = & \Gamma_{1}\left(\log g,\log g\right)\Gamma_{1}^{z}\left(f,f\right)-\beta^{2}\left(1+\frac{1}{g^{2}}\right)\Gamma_{1}\left(f,f\right)+\frac{\beta^{2}}{2g^{2}}\Gamma_{1}^{z}\left(f,f\right). \end{array}$$



**Lemma** **6.**
*By routine computations, we obtain*

$$\begin{array}{ccc} & & R_{ab}\left(\nabla f,\nabla f\right)\\ & = & \beta^{2}e^{\beta \theta }\frac{\partial f}{\partial x}{\left(a^{T}\nabla \right)}_{2}f+\beta e^{\beta \theta }{\left(a^{T}\nabla \right)}_{2}V\frac{\partial f}{\partial x}{\left(a^{T}\nabla \right)}_{1}f+\beta e^{\beta \theta }\frac{\partial V}{\partial x}{\left(a^{T}\nabla \right)}_{2}f{\left(a^{T}\nabla \right)}_{1}f\\ & & +\frac{\partial^{2}V}{\partial \theta \partial \theta }{\left|{\left(a^{T}\nabla \right)}_{1}f\right|}^{2}+2\left(e^{\beta \theta }\frac{\partial^{2}V}{\partial \theta \partial x}+\frac{\partial^{2}V}{\partial \theta \partial y}\right){\left(a^{T}\nabla \right)}_{1}f{\left(a^{T}\nabla \right)}_{2}f\\ & & +\sum_{\hat{i},k^{\prime }=1}^{3}a_{2\hat{i}}^{T}a_{2k^{\prime }}^{T}\frac{\partial^{2}V}{\partial x_{\hat{i}}\partial x_{k^{\prime }}}{\left|{\left(a^{T}\nabla \right)}_{2}f\right|}^{2}-\beta e^{\beta \theta }{\left(a^{T}\nabla \right)}_{1}V\frac{\partial f}{\partial x}{\left(a^{T}\nabla \right)}_{2}f;\\ & & R_{zb}\left(\nabla f,\nabla f\right)\\ & = & \sum_{i=1}^{2}\sum_{i^{\prime },\hat{i}=1}^{3}a_{ii^{\prime }}^{T}a_{i\hat{i}}^{T}\frac{\partial^{2}z_{1\hat{k}}^{T}}{\partial x_{i^{\prime }}\partial x_{\hat{i}}}\partial_{y}f{\left(z^{T}\nabla \right)}_{1}f-\sum_{k=1}^{2}{\left(a^{T}\nabla \right)}_{k}z_{13}^{T}{\left(a^{T}\nabla \right)}_{k}V\partial_{y}f{\left(z^{T}\nabla \right)}_{1}f\\ & & -g\frac{\partial^{2}V}{\partial \theta \partial y}{\left(a^{T}\nabla \right)}_{1}f{\left(z^{T}\nabla \right)}_{1}f-g\left(e^{\beta \theta }\frac{\partial^{2}V}{\partial x\partial y}+\frac{\partial^{2}V}{\partial y\partial y}\right){\left(a^{T}\nabla \right)}_{2}f{\left(z^{T}\nabla \right)}_{1}f;\\ & & R_{\pi }\left(\nabla f,\nabla f\right)\\ & = & -2\sum_{l=1}^{2}\sum_{l^{\prime },\hat{l}=1}^{3}a_{ll^{\prime }}^{T}a_{l\hat{l}}^{T}\frac{\partial^{2}z_{13}^{T}}{\partial x_{l^{\prime }}\partial x_{\hat{l}}}\partial_{y}f{\left(z^{T}\nabla \right)}_{1}f-2\sum_{l=1}^{2}\sum_{l^{\prime },\hat{l}=1}^{3}a_{ll^{\prime }}^{T}a_{l\hat{l}}^{T}\frac{\partial z_{13}^{T}}{\partial x_{\hat{l}}}\frac{\partial z_{13}^{T}}{\partial x_{l^{\prime }}}{\left|\partial_{y}f\right|}^{2}\\ & & -2\sum_{l=1}^{2}\sum_{\hat{l}=1}^{3}{\left(a^{T}\nabla \right)}_{l}\log\pi a_{l\hat{l}}^{T}\frac{\partial z_{13}^{T}}{\partial x_{\hat{l}}}\partial_{y}f{\left(z^{T}\nabla \right)}_{1}f. \end{array}$$



**Proof** **of** **Lemma** **5.**According to Lemma 4 and observing the fact that G=−2F and${\left(a^{T}\nabla \right)}_{2}f=e^{\beta \theta }\partial_{x}f+\partial_{y}f$, we first have
$$\begin{array}{ccc} 2C^{T}X & = & 2\left[\beta e^{\beta \theta }\partial_{y}f+\beta e^{2\beta \theta }\partial_{x}f\right]\frac{\partial^{2}f}{\partial \theta \partial x}+2\left[-\beta e^{2\beta \theta }\partial_{\theta }f\right]\frac{\partial^{2}f}{\partial x\partial x}+2\left[-\beta e^{\beta \theta }\partial_{\theta }f\right]\frac{\partial^{2}f}{\partial x\partial y};\\ 2[F^{T}+G^{T}]X & = & -2\left(g\partial_{\theta }g\partial_{y}f\frac{\partial^{2}f}{\partial \theta \partial y}+e^{\beta \theta }g{\left(a^{T}\nabla \right)}_{2}g\partial_{y}f\frac{\partial^{2}f}{\partial x\partial y}+g{\left(a^{T}\nabla \right)}_{2}g\partial_{y}f\frac{\partial^{2}f}{\partial y\partial y}\right). \end{array}$$By direct computations, we have
$$\begin{array}{ccc} & & {\left[QX+D\right]}^{T}\left[QX+D\right]+{\left[PX+E\right]}^{T}\left[PX+E\right]+2C^{T}X+2F^{T}X+2G^{T}X\\ & = & {\left[\frac{\partial^{2}f}{\partial \theta \partial \theta }\right]}^{2}+{\left[e^{2\beta \theta }\frac{\partial^{2}f}{\partial x\partial x}+2e^{\beta \theta }\frac{\partial^{2}f}{\partial x\partial y}+\frac{\partial^{2}f}{\partial y\partial y}\right]}^{2}+{\left[e^{\beta \theta }\frac{\partial^{2}f}{\partial \theta \partial x}+\frac{\partial^{2}f}{\partial \theta \partial y}+\beta e^{\beta \theta }\frac{\partial f}{\partial x}\right]}^{2}\\ & & +{\left[e^{\beta \theta }\frac{\partial^{2}f}{\partial \theta \partial x}+\frac{\partial^{2}f}{\partial \theta \partial y}\right]}^{2}+{\left[-g\frac{\partial^{2}f}{\partial \theta \partial y}-\partial_{y}f\partial_{\theta }g\right]}^{2}\\ & & +{\left[-ge^{\beta \theta }\frac{\partial^{2}f}{\partial x\partial y}-g\frac{\partial^{2}f}{\partial y\partial y}-{\left(a^{T}\nabla \right)}_{2}g\partial_{y}f\right]}^{2}\\ & & +2\left[\beta e^{\beta \theta }\partial_{y}f+\beta e^{2\beta \theta }\partial_{x}f\right]\frac{\partial^{2}f}{\partial \theta \partial x}+2\left[-\beta e^{2\beta \theta }\partial_{\theta }f\right]\frac{\partial^{2}f}{\partial x\partial x}+2\left[-\beta e^{\beta \theta }\partial_{\theta }f\right]\frac{\partial^{2}f}{\partial x\partial y}\\ & & -2\left(g\partial_{\theta }g\partial_{y}f\frac{\partial^{2}f}{\partial \theta \partial y}+e^{\beta \theta }g{\left(a^{T}\nabla \right)}_{2}g\partial_{y}f\frac{\partial^{2}f}{\partial x\partial y}+g{\left(a^{T}\nabla \right)}_{2}g\partial_{y}f\frac{\partial^{2}f}{\partial y\partial y}\right)\\ & = & {\left[\frac{\partial^{2}f}{\partial \theta \partial \theta }\right]}^{2}+{\left[e^{2\beta \theta }\frac{\partial^{2}f}{\partial x\partial x}+2e^{\beta \theta }\frac{\partial^{2}f}{\partial x\partial y}+\frac{\partial^{2}f}{\partial y\partial y}\right]}^{2}+{\left[e^{\beta \theta }\frac{\partial^{2}f}{\partial \theta \partial x}+\frac{\partial^{2}f}{\partial \theta \partial y}+\beta e^{\beta \theta }\frac{\partial f}{\partial x}\right]}^{2} \end{array}$$
$$\begin{array}{ccc} & & +{\left[e^{\beta \theta }\frac{\partial^{2}f}{\partial \theta \partial x}+\frac{\partial^{2}f}{\partial \theta \partial y}\right]}^{2}+{\left[-g\frac{\partial^{2}f}{\partial \theta \partial y}-\partial_{y}f\partial_{\theta }g\right]}^{2}\\ & & +{\left[-ge^{\beta \theta }\frac{\partial^{2}f}{\partial x\partial y}-g\frac{\partial^{2}f}{\partial y\partial y}-{\left(a^{T}\nabla \right)}_{2}g\partial_{y}f\right]}^{2}\\ & & +2\beta {\left(a^{T}\nabla \right)}_{2}f\left[e^{\beta \theta }\frac{\partial^{2}f}{\partial \theta \partial x}+\frac{\partial^{2}f}{\partial \theta \partial y}\right]-2\beta {\left(a^{T}\nabla \right)}_{2}f\frac{\partial^{2}f}{\partial \theta \partial y}-2g\partial_{\theta }g\partial_{y}f\frac{\partial^{2}f}{\partial \theta \partial y}\\ & & -2\beta \partial_{\theta }f\left[2e^{\beta \theta }\frac{\partial^{2}f}{\partial x\partial y}+e^{2\beta \theta }\frac{\partial^{2}f}{\partial x\partial x}+\frac{\partial^{2}f}{\partial y\partial y}\right]+2\beta \partial_{\theta }f\left[e^{\beta \theta }\frac{\partial^{2}f}{\partial x\partial y}+\frac{\partial^{2}f}{\partial y\partial y}\right]\\ & & -2g{\left(a^{T}\nabla \right)}_{2}g\partial_{y}f\left[e^{\beta \theta }\frac{\partial^{2}f}{\partial x\partial y}+\frac{\partial^{2}f}{\partial y\partial y}\right]. \end{array}$$Completing the squares for the above terms, we have
$$\begin{array}{ccc} & & {\left[QX+D\right]}^{T}\left[QX+D\right]+{\left[PX+E\right]}^{T}\left[PX+E\right]+2C^{T}X+2F^{T}X+2G^{T}X\\ & = & {\left[\frac{\partial^{2}f}{\partial \theta \partial \theta }\right]}^{2}+{\left[e^{2\beta \theta }\frac{\partial^{2}f}{\partial x\partial x}+2e^{\beta \theta }\frac{\partial^{2}f}{\partial x\partial y}+\frac{\partial^{2}f}{\partial y\partial y}-\beta \partial_{\theta }f\right]}^{2}-\beta^{2}{\left|\partial_{\theta }f\right|}^{2}\\ & & +{\left[e^{\beta \theta }\frac{\partial^{2}f}{\partial \theta \partial x}+\frac{\partial^{2}f}{\partial \theta \partial y}+\beta e^{\beta \theta }\frac{\partial f}{\partial x}\right]}^{2}+{\left[e^{\beta \theta }\frac{\partial^{2}f}{\partial \theta \partial x}+\frac{\partial^{2}f}{\partial \theta \partial y}+\beta {\left(a^{T}\nabla \right)}_{2}f\right]}^{2}-\beta^{2}{\left|{\left(a^{T}\nabla \right)}_{2}f\right|}^{2}\\ & & +{\left[g\frac{\partial^{2}f}{\partial \theta \partial y}+\partial_{\theta }g\partial_{y}f-\frac{\beta {\left(a^{T}\nabla \right)}_{2}f}{g}-\partial_{\theta }g\partial_{y}f\right]}^{2}-{\left[\frac{\beta {\left(a^{T}\nabla \right)}_{2}f}{g}+\partial_{\theta }g\partial_{y}f\right]}^{2}\\ & & +{\left[ge^{\beta \theta }\frac{\partial^{2}f}{\partial x\partial y}+g\frac{\partial^{2}f}{\partial y\partial y}+{\left(a^{T}\nabla \right)}_{2}g\partial_{y}f+\frac{\beta \partial_{\theta }f}{g}-{\left(a^{T}\nabla \right)}_{2}g\partial_{y}f\right]}^{2}\\ & & -{\left[\frac{\beta \partial_{\theta }f}{g}-{\left(a^{T}\nabla \right)}_{2}g\partial_{y}f\right]}^{2}+2\left[\frac{\beta {\left(a^{T}\nabla \right)}_{2}f}{g}+\partial_{\theta }g\partial_{y}f\right]\partial_{\theta }g\partial_{y}f\\ & & -2\partial_{y}f{\left(a^{T}\nabla \right)}_{2}g\times \left[\frac{\beta \partial_{\theta }f}{g}-{\left(a^{T}\nabla \right)}_{2}g\partial_{y}f\right]. \end{array}$$The first-order terms generate $-\Lambda_{1}^{T}Q^{T}Q\Lambda_{1}-\Lambda_{2}^{T}P^{T}P\Lambda_{2}+D^{T}D+E^{T}E$, and the sum of squares terms generate vectors $\Lambda_{1}$ and $\Lambda_{2}$. We further formulate the above two terms as below:
$$\begin{array}{ccc} & & {\left[e^{\beta \theta }\frac{\partial^{2}f}{\partial \theta \partial x}+\frac{\partial^{2}f}{\partial \theta \partial y}+\beta e^{\beta \theta }\frac{\partial f}{\partial x}\right]}^{2}+{\left[e^{\beta \theta }\frac{\partial^{2}f}{\partial \theta \partial x}+\frac{\partial^{2}f}{\partial \theta \partial y}+\beta {\left(a^{T}\nabla \right)}_{2}f\right]}^{2}\\ & = & 2{\left[e^{\beta \theta }\frac{\partial^{2}f}{\partial \theta \partial x}+\frac{\partial^{2}f}{\partial \theta \partial y}+\beta e^{\beta \theta }\frac{\partial f}{\partial x}+\frac{\beta }{2}\partial_{y}f\right]}^{2}+\frac{\beta^{2}}{2}{\left|\partial_{y}f\right|}^{2}. \end{array}$$Adding $\frac{\beta^{2}}{2}{\left|\partial_{y}f\right|}^{2}$ into the term $-\Lambda_{1}^{T}Q^{T}Q\Lambda_{1}-\Lambda_{2}^{T}P^{T}P\Lambda_{2}+D^{T}D+E^{T}E$ again, we further expand as below:
$$\begin{array}{ccc} & & -\Lambda_{1}^{T}Q^{T}Q\Lambda_{1}-\Lambda_{2}^{T}P^{T}P\Lambda_{2}+D^{T}D+E^{T}E\\ & = & -\beta^{2}\left[\right|\partial_{\theta }{f|}^{2}+|{\left(a^{T}\nabla \right)}_{2}f{|}^{2}]-{\left[\frac{\beta \partial_{\theta }f}{g}-{\left(a^{T}\nabla \right)}_{2}g\partial_{y}f\right]}^{2}\\ & & -{\left[\frac{\beta {\left(a^{T}\nabla \right)}_{2}f}{g}+\partial_{\theta }g\partial_{y}f\right]}^{2}+2[\frac{\beta {\left(a^{T}\nabla \right)}_{2}f}{g}\\ & & +\partial_{\theta }g\partial_{y}f]\partial_{\theta }g\partial_{y}f-2\partial_{y}f{\left(a^{T}\nabla \right)}_{2}g\times \left[\frac{\beta \partial_{\theta }f}{g}-{\left(a^{T}\nabla \right)}_{2}g\partial_{y}f\right]+\frac{\beta^{2}}{2}{\left|\partial_{y}f\right|}^{2} \end{array}$$
$$\begin{array}{ccc} & = & -\beta^{2}\Gamma_{1}\left(f,f\right)-\frac{\beta^{2}}{g^{2}}|{\left(a^{T}\nabla \right)}_{1}{f|}^{2}-|{\left(a^{T}\nabla \right)}_{2}\left(\log g\right){|}^{2}{\left|{\left(z^{T}\nabla \right)}_{1}f\right|}^{2}\\ & & -2\frac{\beta }{g}{\left(a^{T}\nabla \right)}_{2}\log g{\left(a^{T}\nabla \right)}_{1}f{\left(z^{T}\nabla \right)}_{1}f\\ & & -\frac{\beta^{2}}{g^{2}}|{\left(a^{T}\nabla \right)}_{2}{f|}^{2}-|{\left(a^{T}\nabla \right)}_{1}{\log g|}^{2}{\left|{\left(z^{T}\nabla \right)}_{1}f\right|}^{2}\\ & & +2\frac{\beta }{g}{\left(a^{T}\nabla \right)}_{1}\log g{\left(a^{T}\nabla \right)}_{2}f{\left(z^{T}\nabla \right)}_{1}f\\ & & -2\frac{\beta }{g}{\left(a^{T}\nabla \right)}_{1}\log g{\left(a^{T}\nabla \right)}_{2}f{\left(z^{T}\nabla \right)}_{1}f+2|{\left(a^{T}\nabla \right)}_{1}{\log g|}^{2}{\left|{\left(z^{T}\nabla \right)}_{1}f\right|}^{2}\\ & & +2\frac{\beta }{g}{\left(a^{T}\nabla \right)}_{2}\log g{\left(a^{T}\nabla \right)}_{1}f{\left(z^{T}\nabla \right)}_{1}f+2|{\left(a^{T}\nabla \right)}_{2}{\log g|}^{2}{\left|{\left(z^{T}\nabla \right)}_{1}f\right|}^{2}+\frac{\beta^{2}}{2g^{2}}\Gamma_{1}^{z}\left(f,f\right). \end{array}$$By grouping the bilinear terms of ∇f, we obtain
$$\begin{array}{ccc} & & -\Lambda_{1}^{T}Q^{T}Q\Lambda_{1}-\Lambda_{2}^{T}P^{T}P\Lambda_{2}+D^{T}D+E^{T}E\\ & = & \Gamma_{1}\left(\log g,\log g\right)\Gamma_{1}^{z}\left(f,f\right)-\beta^{2}\left(1+\frac{1}{g^{2}}\right)\Gamma_{1}\left(f,f\right)+\frac{\beta^{2}}{2g^{2}}\Gamma_{1}^{z}\left(f,f\right). \end{array}$$□

We are now left to compute the three tensor terms.

**Proof** **of** **Lemma** **6.**For displacement group G, we have n=2 and m=1. Recall Theorem 1; we denote $R_{ab}\left(\nabla f,\nabla f\right)=R_{a}\left(\nabla f,\nabla f\right)+R_{b}\left(\nabla f,\nabla f\right)$, where $R_{b}\left(\nabla f,\nabla f\right)$ represents the tensor term involving drift *b*. We thus have
$$\begin{array}{ccc} R_{a}\left(\nabla f,\nabla f\right) & = & \sum_{i,k=1}^{2}\sum_{i^{\prime },\hat{i},\hat{k}=1}^{3}\langle a_{ii^{\prime }}^{T}\left(\frac{\partial a_{i\hat{i}}^{T}}{\partial x_{i^{\prime }}}\frac{\partial a_{k\hat{k}}^{T}}{\partial x_{\hat{i}}}\frac{\partial f}{\partial x_{\hat{k}}}\right),{\left(a^{T}\nabla \right)}_{k}f\rangle {}_{R^{2}}\\ & & +\sum_{i,k=2}^{n}\sum_{i^{\prime },\hat{i},\hat{k}=1}^{3}\langle a_{ii^{\prime }}^{T}a_{i\hat{i}}^{T}\left(\frac{\partial }{\partial x_{i^{\prime }}}\frac{\partial a_{k\hat{k}}^{T}}{\partial x_{\hat{i}}}\right)\left(\frac{\partial f}{\partial x_{\hat{k}}}\right),{\left(a^{T}\nabla \right)}_{k}f\rangle {}_{R^{2}}\\ & & -\sum_{i,k=1}^{2}\sum_{i^{\prime },\hat{i},\hat{k}=1}^{3}\langle a_{k\hat{k}}^{T}\frac{\partial a_{ii^{\prime }}^{T}}{\partial x_{\hat{k}}}\frac{\partial a_{i\hat{i}}^{T}}{\partial x_{i^{\prime }}}\frac{\partial f}{\partial x_{\hat{i}}}),{\left(a^{T}\nabla \right)}_{k}{f\rangle }_{R^{2}}\\ & & -\sum_{i,k=1}^{2}\sum_{i^{\prime },\hat{i},\hat{k}=1}^{3}\langle a_{k\hat{k}}^{T}a_{ii^{\prime }}^{T}\left(\frac{\partial }{\partial x_{\hat{k}}}\frac{\partial a_{i\hat{i}}^{T}}{\partial x_{i^{\prime }}}\right)\frac{\partial f}{\partial x_{\hat{i}}},{\left(a^{T}\nabla \right)}_{k}f\rangle {}_{R^{2}},\\ & = & I_{1}+I_{2}+I_{3}+I_{4}. \end{array}$$By direct computations, we have
$$\begin{array}{ccc} I_{1} & = & \sum_{i=1}^{2}\sum_{i^{\prime },\hat{i},\hat{k}=1}^{3}[a_{ii^{\prime }}^{T}\left(\frac{\partial a_{i\hat{i}}^{T}}{\partial x_{i^{\prime }}}\frac{\partial a_{1\hat{k}}^{T}}{\partial x_{\hat{i}}}\frac{\partial f}{\partial x_{\hat{k}}}\right){\left(a^{T}\nabla \right)}_{1}f\\ & & \;\;\;+a_{ii^{\prime }}^{T}\left(\frac{\partial a_{i\hat{i}}^{T}}{\partial x_{i^{\prime }}}\frac{\partial a_{2\hat{k}}^{T}}{\partial x_{\hat{i}}}\frac{\partial f}{\partial x_{\hat{k}}}\right){\left(a^{T}\nabla \right)}_{2}f]=0; \end{array}$$
$$\begin{array}{ccc} I_{2} & = & \sum_{i=2}^{n}\sum_{i^{\prime },\hat{i},\hat{k}=1}^{3}[a_{ii^{\prime }}^{T}a_{i\hat{i}}^{T}\left(\frac{\partial }{\partial x_{i^{\prime }}}\frac{\partial a_{1\hat{k}}^{T}}{\partial x_{\hat{i}}}\right)\left(\frac{\partial f}{\partial x_{\hat{k}}}\right){\left(a^{T}\nabla \right)}_{1}f\\ & & \;\;\;+a_{ii^{\prime }}^{T}a_{i\hat{i}}^{T}\left(\frac{\partial }{\partial x_{i^{\prime }}}\frac{\partial a_{2\hat{k}}^{T}}{\partial x_{\hat{i}}}\right)\left(\frac{\partial f}{\partial x_{\hat{k}}}\right){\left(a^{T}\nabla \right)}_{2}f]\\ & = & a_{11}^{T}a_{11}^{T}\frac{\partial^{2}}{\partial \theta \partial \theta }a_{22}^{T}\frac{\partial f}{\partial x}{\left(a^{T}\nabla \right)}_{2}f=\beta^{2}e^{\beta \theta }\frac{\partial f}{\partial x}{\left(a^{T}\nabla \right)}_{2}f;\\ I_{3} & = & -\sum_{i=1}^{2}\sum_{i^{\prime },\hat{i},\hat{k}=1}^{3}[a_{1\hat{k}}^{T}\frac{\partial a_{ii^{\prime }}^{T}}{\partial x_{\hat{k}}}\frac{\partial a_{i\hat{i}}^{T}}{\partial x_{i^{\prime }}}\frac{\partial f}{\partial x_{\hat{i}}}){\left(a^{T}\nabla \right)}_{1}f\\ & & \;\;\;+a_{2\hat{k}}^{T}\frac{\partial a_{ii^{\prime }}^{T}}{\partial x_{\hat{k}}}\frac{\partial a_{i\hat{i}}^{T}}{\partial x_{i^{\prime }}}\frac{\partial f}{\partial x_{\hat{i}}}){\left(a^{T}\nabla \right)}_{2}f]=0;\\ I_{4} & = & -\sum_{i=1}^{2}\sum_{i^{\prime },\hat{i},\hat{k}=1}^{3}[a_{1\hat{k}}^{T}a_{ii^{\prime }}^{T}\left(\frac{\partial }{\partial x_{\hat{k}}}\frac{\partial a_{i\hat{i}}^{T}}{\partial x_{i^{\prime }}}\right)\frac{\partial f}{\partial x_{\hat{i}}}{\left(a^{T}\nabla \right)}_{1}f\\ & & \;\;\;+a_{2\hat{k}}^{T}a_{ii^{\prime }}^{T}\left(\frac{\partial }{\partial x_{\hat{k}}}\frac{\partial a_{i\hat{i}}^{T}}{\partial x_{i^{\prime }}}\right)\frac{\partial f}{\partial x_{\hat{i}}}{\left(a^{T}\nabla \right)}_{2}f]=0. \end{array}$$For the drift term in tensor $R_{ab}$, taking $b=-aa^{T}\nabla V$, we obtain
$$\begin{array}{ccc} R_{b}^{a} & = & \sum_{i,k=1}^{2}\sum_{\hat{i},\hat{k},k^{\prime }=1}^{3}\left[a_{i\hat{i}}^{T}\frac{\partial a_{k\hat{k}}^{T}}{\partial x_{\hat{i}}}a_{kk^{\prime }}^{T}\frac{\partial V}{\partial x_{k^{\prime }}}\frac{\partial f}{\partial x_{\hat{k}}}{\left(a^{T}\nabla \right)}_{i}f\right]\\ & & +\sum_{i,k=1}^{2}\sum_{\hat{i},\hat{k},k^{\prime }=1}^{3}\left[a_{i\hat{i}}^{T}\frac{\partial a_{kk^{\prime }}^{T}}{\partial x_{\hat{i}}}a_{k\hat{k}}^{T}\frac{\partial V}{\partial x_{k^{\prime }}}\frac{\partial f}{\partial x_{\hat{k}}}{\left(a^{T}\nabla \right)}_{i}f\right]\\ & & +\sum_{i,k=1}^{2}\sum_{\hat{i},\hat{k},k^{\prime }=1}^{3}\left[a_{i\hat{i}}^{T}a_{k\hat{k}}^{T}a_{kk^{\prime }}^{T}\frac{\partial^{2}V}{\partial x_{\hat{i}}\partial x_{k^{\prime }}}\frac{\partial f}{\partial x_{\hat{k}}}{\left(a^{T}\nabla \right)}_{i}f\right]\\ & & -\sum_{i,k=1}^{2}\sum_{\hat{i},\hat{k},k^{\prime }=1}^{3}\left[a_{k\hat{k}}^{T}a_{kk^{\prime }}^{T}\frac{\partial a_{i\hat{i}}^{T}}{\partial x_{\hat{k}}}\frac{\partial V}{\partial x_{k^{\prime }}}\frac{\partial f}{\partial x_{\hat{i}}}{\left(a^{T}\nabla \right)}_{i}f\right]\\ & = & J_{1}+J_{2}+J_{3}+J_{4}. \end{array}$$Plugging into the matrix $a^{T}$, we obtain
$$\begin{array}{ccc} J_{1} & = & \sum_{\hat{i},\hat{k},k^{\prime }=1}^{3}\left[a_{1\hat{i}}^{T}\frac{\partial a_{1\hat{k}}^{T}}{\partial x_{\hat{i}}}a_{1k^{\prime }}^{T}\frac{\partial V}{\partial x_{k^{\prime }}}\frac{\partial f}{\partial x_{\hat{k}}}{\left(a^{T}\nabla \right)}_{1}f+a_{2\hat{i}}^{T}\frac{\partial a_{1\hat{k}}^{T}}{\partial x_{\hat{i}}}a_{1k^{\prime }}^{T}\frac{\partial V}{\partial x_{k^{\prime }}}\frac{\partial f}{\partial x_{\hat{k}}}{\left(a^{T}\nabla \right)}_{2}f\right]\\ & & +\sum_{\hat{i},\hat{k},k^{\prime }=1}^{3}\left[a_{1\hat{i}}^{T}\frac{\partial a_{2\hat{k}}^{T}}{\partial x_{\hat{i}}}a_{2k^{\prime }}^{T}\frac{\partial V}{\partial x_{k^{\prime }}}\frac{\partial f}{\partial x_{\hat{k}}}{\left(a^{T}\nabla \right)}_{1}f+a_{2\hat{i}}^{T}\frac{\partial a_{2\hat{k}}^{T}}{\partial x_{\hat{i}}}a_{2k^{\prime }}^{T}\frac{\partial V}{\partial x_{k^{\prime }}}\frac{\partial f}{\partial x_{\hat{k}}}{\left(a^{T}\nabla \right)}_{2}f\right]\\ & = & \beta e^{\beta \theta }{\left(a^{T}\nabla \right)}_{2}V\frac{\partial f}{\partial x}{\left(a^{T}\nabla \right)}_{1}f; \end{array}$$
$$\begin{array}{ccc} J_{2} & = & \sum_{\hat{i},\hat{k},k^{\prime }=1}^{3}\left[a_{1\hat{i}}^{T}\frac{\partial a_{1k^{\prime }}^{T}}{\partial x_{\hat{i}}}a_{1\hat{k}}^{T}\frac{\partial V}{\partial x_{k^{\prime }}}\frac{\partial f}{\partial x_{\hat{k}}}{\left(a^{T}\nabla \right)}_{1}f+a_{2\hat{i}}^{T}\frac{\partial a_{1k^{\prime }}^{T}}{\partial x_{\hat{i}}}a_{1\hat{k}}^{T}\frac{\partial V}{\partial x_{k^{\prime }}}\frac{\partial f}{\partial x_{\hat{k}}}{\left(a^{T}\nabla \right)}_{2}f\right]\\ & & +\sum_{\hat{i},\hat{k},k^{\prime }=1}^{3}\left[a_{1\hat{i}}^{T}\frac{\partial a_{2k^{\prime }}^{T}}{\partial x_{\hat{i}}}a_{2\hat{k}}^{T}\frac{\partial V}{\partial x_{k^{\prime }}}\frac{\partial f}{\partial x_{\hat{k}}}{\left(a^{T}\nabla \right)}_{1}f+a_{2\hat{i}}^{T}\frac{\partial a_{2k^{\prime }}^{T}}{\partial x_{\hat{i}}}a_{2\hat{k}}^{T}\frac{\partial V}{\partial x_{k^{\prime }}}\frac{\partial f}{\partial x_{\hat{k}}}{\left(a^{T}\nabla \right)}_{2}f\right]\\ & = & \beta e^{\beta \theta }\frac{\partial V}{\partial x}{\left(a^{T}\nabla \right)}_{2}f{\left(a^{T}\nabla \right)}_{1}f; \end{array}$$
$$\begin{array}{ccc} J_{3} & = & \sum_{\hat{i},k^{\prime }=1}^{3}\left[a_{1\hat{i}}^{T}a_{1k^{\prime }}^{T}\frac{\partial^{2}V}{\partial x_{\hat{i}}\partial x_{k^{\prime }}}{\left|{\left(a^{T}\nabla \right)}_{1}f\right|}^{2}+a_{2\hat{i}}^{T}a_{1k^{\prime }}^{T}\frac{\partial^{2}V}{\partial x_{\hat{i}}\partial x_{k^{\prime }}}{\left(a^{T}\nabla \right)}_{1}f{\left(a^{T}\nabla \right)}_{2}f\right]\\ & & +\sum_{\hat{i},k^{\prime }=1}^{3}\left[a_{1\hat{i}}^{T}a_{2k^{\prime }}^{T}\frac{\partial^{2}V}{\partial x_{\hat{i}}\partial x_{k^{\prime }}}{\left(a^{T}\nabla \right)}_{2}f{\left(a^{T}\nabla \right)}_{1}f+a_{2\hat{i}}^{T}a_{2k^{\prime }}^{T}\frac{\partial^{2}V}{\partial x_{\hat{i}}\partial x_{k^{\prime }}}{\left|{\left(a^{T}\nabla \right)}_{2}f\right|}^{2}\right]\\ & = & \frac{\partial^{2}V}{\partial \theta \partial \theta }{\left|{\left(a^{T}\nabla \right)}_{1}f\right|}^{2}+2\left(e^{\beta \theta }\frac{\partial^{2}V}{\partial \theta \partial x}+\frac{\partial^{2}V}{\partial \theta \partial y}\right){\left(a^{T}\nabla \right)}_{1}f{\left(a^{T}\nabla \right)}_{2}f\\ & & +\sum_{\hat{i},k^{\prime }=1}^{3}a_{2\hat{i}}^{T}a_{2k^{\prime }}^{T}\frac{pa^{2}V}{\partial x_{\hat{i}}\partial x_{k^{\prime }}})|{\left(a^{T}\nabla \right)}_{2}{f|}^{2};\\ J_{4} & = & -\sum_{\hat{i},\hat{k},k^{\prime }=1}^{3}\left[a_{1\hat{k}}^{T}a_{1k^{\prime }}^{T}\frac{\partial a_{1\hat{i}}^{T}}{\partial x_{\hat{k}}}\frac{\partial V}{\partial x_{k^{\prime }}}\frac{\partial f}{\partial x_{\hat{i}}}{\left(a^{T}\nabla \right)}_{1}f+a_{1\hat{k}}^{T}a_{1k^{\prime }}^{T}\frac{\partial a_{2\hat{i}}^{T}}{\partial x_{\hat{k}}}\frac{\partial V}{\partial x_{k^{\prime }}}\frac{\partial f}{\partial x_{\hat{i}}}{\left(a^{T}\nabla \right)}_{2}f\right]\\ & & -\sum_{\hat{i},\hat{k},k^{\prime }=1}^{3}\left[a_{2\hat{k}}^{T}a_{2k^{\prime }}^{T}\frac{\partial a_{1\hat{i}}^{T}}{\partial x_{\hat{k}}}\frac{\partial V}{\partial x_{k^{\prime }}}\frac{\partial f}{\partial x_{\hat{i}}}{\left(a^{T}\nabla \right)}_{1}f+a_{2\hat{k}}^{T}a_{2k^{\prime }}^{T}\frac{\partial a_{2\hat{i}}^{T}}{\partial x_{\hat{k}}}\frac{\partial V}{\partial x_{k^{\prime }}}\frac{\partial f}{\partial x_{\hat{i}}}{\left(a^{T}\nabla \right)}_{2}f\right]\\ & = & -\beta e^{\beta \theta }{\left(a^{T}\nabla \right)}_{1}V\frac{\partial f}{\partial x}{\left(a^{T}\nabla \right)}_{2}f. \end{array}$$Combining the above computations, we obtain the tensor $R_{ab}$. Now, we turn to the second tensor $R_{zb}$, which has the following form:
$$\begin{array}{ccc} R_{zb}\left(\nabla f,\nabla f\right) & = & \sum_{i=1}^{2}\sum_{i^{\prime },\hat{i},\hat{k}=1}^{3}\langle a_{ii^{\prime }}^{T}\left(\frac{\partial a_{i\hat{i}}^{T}}{\partial x_{i^{\prime }}}\frac{\partial z_{1\hat{k}}^{T}}{\partial x_{\hat{i}}}\frac{\partial f}{\partial x_{\hat{k}}}\right),{\left(z^{T}\nabla \right)}_{1}f\rangle {}_{R}\\ & & +\sum_{i=1}^{2}\sum_{i^{\prime },\hat{i},\hat{k}=1}^{3}\langle a_{ii^{\prime }}^{T}a_{i\hat{i}}^{T}\left(\frac{\partial }{\partial x_{i^{\prime }}}\frac{\partial z_{1\hat{k}}^{T}}{\partial x_{\hat{i}}}\right)\left(\frac{\partial f}{\partial x_{\hat{k}}}\right),{\left(z^{T}\nabla \right)}_{1}f\rangle {}_{R}\\ & & -\sum_{i=1}^{2}\sum_{i^{\prime },\hat{i},\hat{k}=1}^{3}\langle z_{1\hat{k}}^{T}\frac{\partial a_{ii^{\prime }}^{T}}{\partial x_{\hat{k}}}\frac{\partial a_{i\hat{i}}^{T}}{\partial x_{i^{\prime }}}\frac{\partial f}{\partial x_{\hat{i}}}),{\left(z^{T}\nabla \right)}_{1}{f\rangle }_{R}\\ & & -\sum_{i=1}^{2}\sum_{i^{\prime },\hat{i},\hat{k}=1}^{3}\langle z_{1\hat{k}}^{T}a_{ii^{\prime }}^{T}\left(\frac{\partial }{\partial x_{\hat{k}}}\frac{\partial a_{i\hat{i}}^{T}}{\partial x_{i^{\prime }}}\right)\frac{\partial f}{\partial x_{\hat{i}}},{\left(z^{T}\nabla \right)}_{1}f\rangle {}_{R}\\ & & -\sum_{i=1}^{2}\sum_{\hat{i},\hat{k}=1}^{3}\langle \left(z_{1\hat{i}}^{T}\frac{\partial b_{\hat{k}}}{\partial x_{\hat{i}}}\frac{\partial f}{\partial x_{\hat{k}}}-b_{\hat{k}}\frac{\partial z_{1\hat{i}}^{T}}{\partial x_{\hat{k}}}\frac{\partial f}{\partial x_{\hat{i}}}\right),{\left(z^{T}\nabla f\right)}_{1}\rangle {}_{R},\\ & = & I_{1}^{z}+I_{2}^{z}+I_{3}^{z}+I_{4}^{z}+R_{b}^{z}\left(\nabla f,\nabla f\right). \end{array}$$
where we denote further that
$$\begin{array}{c} R_{b}^{z}\left(\nabla f,\nabla f\right)=-\sum_{\hat{i},\hat{k}=1}^{3}\left(z_{1\hat{i}}^{T}\frac{\partial b_{\hat{k}}}{\partial x_{\hat{i}}}\frac{\partial f}{\partial x_{\hat{k}}}-b_{\hat{k}}\frac{\partial z_{i\hat{i}}^{T}}{\partial x_{\hat{k}}}\frac{\partial f}{\partial x_{\hat{i}}}\right){\left(z^{T}\nabla f\right)}_{1}. \end{array}$$By taking $b=-aa^{T}\nabla V$, we further obtain that
$$\begin{array}{ccc} R_{b}^{z}\left(\nabla f,\nabla f\right) & = & -\sum_{\hat{i},\hat{k}=1}^{3}\left[z_{1\hat{i}}^{T}\frac{\partial b_{\hat{k}}}{\partial x_{\hat{i}}}\frac{\partial f}{\partial x_{\hat{k}}}{\left(z^{T}\nabla f\right)}_{1}-b_{\hat{k}}\frac{\partial z_{i\hat{i}}^{T}}{\partial x_{\hat{k}}}\frac{\partial f}{\partial x_{\hat{i}}}{\left(z^{T}\nabla f\right)}_{1}\right]\\ & = & \sum_{k=1}^{2}\sum_{\hat{i},\hat{k},k^{\prime }=1}^{3}\left[z_{1\hat{i}}^{T}\frac{\partial a_{k\hat{k}}^{T}}{\partial x_{\hat{i}}}a_{kk^{\prime }}^{T}\frac{\partial V}{\partial x_{k^{\prime }}}\frac{\partial f}{\partial x_{\hat{k}}}{\left(z^{T}\nabla \right)}_{1}f\right]\\ & & +\sum_{k=1}^{2}\sum_{\hat{i},\hat{k},k^{\prime }=1}^{3}\left[z_{1\hat{i}}^{T}\frac{\partial a_{kk^{\prime }}^{T}}{\partial x_{\hat{i}}}a_{k\hat{k}}^{T}\frac{\partial V}{\partial x_{k^{\prime }}}\frac{\partial f}{\partial x_{\hat{k}}}{\left(z^{T}\nabla \right)}_{1}f\right] \end{array}$$
$$\begin{array}{ccc} & & +\sum_{k=1}^{2}\sum_{\hat{i},\hat{k},k^{\prime }=1}^{3}\left[z_{1\hat{i}}^{T}a_{k\hat{k}}^{T}a_{kk^{\prime }}^{T}\frac{\partial^{2}V}{\partial x_{\hat{i}}\partial x_{k^{\prime }}}\frac{\partial f}{\partial x_{\hat{k}}}{\left(z^{T}\nabla \right)}_{1}f\right]\\ & & -\sum_{k=1}^{2}\sum_{\hat{i},\hat{k},k^{\prime }=1}^{3}\left[a_{k\hat{k}}^{T}a_{kk^{\prime }}^{T}\frac{\partial z_{1\hat{i}}^{T}}{\partial x_{\hat{k}}}\frac{\partial V}{\partial x_{k^{\prime }}}\frac{\partial f}{\partial x_{\hat{i}}}{\left(z^{T}\nabla \right)}_{1}f\right]\\ & = & J_{1}^{z}+J_{2}^{z}+J_{3}^{z}+J_{4}^{z}. \end{array}$$By direct computations, it is not hard to observe that
$$\begin{array}{ccc} I_{1}^{z} & = & \sum_{i=1}^{2}\sum_{i^{\prime },\hat{i},\hat{k}=1}^{3}\langle a_{ii^{\prime }}^{T}\left(\frac{\partial a_{i\hat{i}}^{T}}{\partial x_{i^{\prime }}}\frac{\partial z_{1\hat{k}}^{T}}{\partial x_{\hat{i}}}\frac{\partial f}{\partial x_{\hat{k}}}\right),{\left(z^{T}\nabla \right)}_{1}f\rangle {}_{R}=0;\\ I_{2}^{z} & = & \sum_{i=1}^{2}\sum_{i^{\prime },\hat{i},\hat{k}=1}^{3}\langle a_{ii^{\prime }}^{T}a_{i\hat{i}}^{T}\left(\frac{\partial }{\partial x_{i^{\prime }}}\frac{\partial z_{1\hat{k}}^{T}}{\partial x_{\hat{i}}}\right)\left(\frac{\partial f}{\partial x_{\hat{k}}}\right),{\left(z^{T}\nabla \right)}_{1}f\rangle {}_{R}\\ & = & \sum_{i=1}^{2}\sum_{i^{\prime },\hat{i}=1}^{3}a_{ii^{\prime }}^{T}a_{i\hat{i}}^{T}\frac{\partial^{2}z_{1\hat{k}}^{T}}{\partial x_{i^{\prime }}\partial x_{\hat{i}}}\partial_{y}f{\left(z^{T}\nabla \right)}_{1}f;\\ I_{3}^{z} & = & -\sum_{i=1}^{2}\sum_{i^{\prime },\hat{i},\hat{k}=1}^{3}\langle z_{1\hat{k}}^{T}\frac{\partial a_{ii^{\prime }}^{T}}{\partial x_{\hat{k}}}\frac{\partial a_{i\hat{i}}^{T}}{\partial x_{i^{\prime }}}\frac{\partial f}{\partial x_{\hat{i}}}),{\left(z^{T}\nabla \right)}_{1}{f\rangle }_{R}=0;\\ I_{4}^{z} & = & -\sum_{i=1}^{2}\sum_{i^{\prime },\hat{i},\hat{k}=1}^{3}\langle z_{1\hat{k}}^{T}a_{ii^{\prime }}^{T}\left(\frac{\partial }{\partial x_{\hat{k}}}\frac{\partial a_{i\hat{i}}^{T}}{\partial x_{i^{\prime }}}\right)\frac{\partial f}{\partial x_{\hat{i}}},{\left(z^{T}\nabla \right)}_{1}f\rangle {}_{R}=0, \end{array}$$
and
$$\begin{array}{ccc} J_{1}^{z} & = & \sum_{k=1}^{2}\sum_{\hat{i},\hat{k},k^{\prime }=1}^{3}\left[z_{1\hat{i}}^{T}\frac{\partial a_{k\hat{k}}^{T}}{\partial x_{\hat{i}}}a_{kk^{\prime }}^{T}\frac{\partial V}{\partial x_{k^{\prime }}}\frac{\partial f}{\partial x_{\hat{k}}}{\left(z^{T}\nabla \right)}_{1}f\right]=0;\\ J_{2}^{z} & = & \sum_{k=1}^{2}\sum_{\hat{i},\hat{k},k^{\prime }=1}^{3}\left[z_{1\hat{i}}^{T}\frac{\partial a_{kk^{\prime }}^{T}}{\partial x_{\hat{i}}}a_{k\hat{k}}^{T}\frac{\partial V}{\partial x_{k^{\prime }}}\frac{\partial f}{\partial x_{\hat{k}}}{\left(z^{T}\nabla \right)}_{1}f\right]=0;\\ J_{4}^{z} & = & -\sum_{k=1}^{2}\sum_{\hat{i},\hat{k},k^{\prime }=1}^{3}\left[a_{k\hat{k}}^{T}a_{kk^{\prime }}^{T}\frac{\partial z_{1\hat{i}}^{T}}{\partial x_{\hat{k}}}\frac{\partial V}{\partial x_{k^{\prime }}}\frac{\partial f}{\partial x_{\hat{i}}}{\left(z^{T}\nabla \right)}_{1}f\right]\\ & = & -\sum_{k=1}^{2}{\left(a^{T}\nabla \right)}_{k}z_{13}^{T}{\left(a^{T}\nabla \right)}_{k}V\partial_{y}f{\left(z^{T}\nabla \right)}_{1}f;\\ J_{3}^{z} & = & \sum_{k=1}^{2}\sum_{\hat{i},\hat{k},k^{\prime }=1}^{3}\left[z_{1\hat{i}}^{T}a_{k\hat{k}}^{T}a_{kk^{\prime }}^{T}\frac{\partial^{2}V}{\partial x_{\hat{i}}\partial x_{k^{\prime }}}\frac{\partial f}{\partial x_{\hat{k}}}{\left(z^{T}\nabla \right)}_{1}f\right]\\ & = & \sum_{\hat{i},\hat{k},k^{\prime }=1}^{3}\left[z_{1\hat{i}}^{T}a_{1\hat{k}}^{T}a_{1k^{\prime }}^{T}\frac{\partial^{2}V}{\partial x_{\hat{i}}\partial x_{k^{\prime }}}\frac{\partial f}{\partial x_{\hat{k}}}{\left(z^{T}\nabla \right)}_{1}f+z_{1\hat{i}}^{T}a_{2\hat{k}}^{T}a_{2k^{\prime }}^{T}\frac{\partial^{2}V}{\partial x_{\hat{i}}\partial x_{k^{\prime }}}\frac{\partial f}{\partial x_{\hat{k}}}{\left(z^{T}\nabla \right)}_{1}f\right]\\ & = & \sum_{\hat{i},k^{\prime }=1}^{3}\left[z_{1\hat{i}}^{T}a_{1k^{\prime }}^{T}\frac{\partial^{2}V}{\partial x_{\hat{i}}\partial x_{k^{\prime }}}{\left(a^{T}\nabla \right)}_{1}f{\left(z^{T}\nabla \right)}_{1}f+z_{1\hat{i}}^{T}a_{2k^{\prime }}^{T}\frac{\partial^{2}V}{\partial x_{\hat{i}}\partial x_{k^{\prime }}}{\left(a^{T}\nabla \right)}_{2}f{\left(z^{T}\nabla \right)}_{1}f\right]\\ & = & -g\frac{\partial^{2}V}{\partial \theta \partial y}{\left(a^{T}\nabla \right)}_{1}f{\left(z^{T}\nabla \right)}_{1}f-g\left(e^{\beta \theta }\frac{\partial^{2}V}{\partial x\partial y}+\frac{\partial^{2}V}{\partial y\partial y}\right){\left(a^{T}\nabla \right)}_{2}f{\left(z^{T}\nabla \right)}_{1}f. \end{array}$$Now, we are left to compute the term $R_{\pi }$. Recall that
$$\begin{array}{ccc} & & R_{\pi }\left(\nabla f,\nabla f\right)\\ & = & 2\sum_{k=1}^{1}\sum_{i=1}^{2}\sum_{k^{\prime },\hat{k},\hat{i},i^{\prime }=1}^{3}\left[\frac{\partial }{\partial x_{k^{\prime }}}z_{kk^{\prime }}^{T}z_{k\hat{k}}^{T}\frac{\partial }{\partial x_{\hat{k}}}a_{i\hat{i}}^{T}\frac{\partial f}{\partial x_{\hat{i}}}a_{ii^{\prime }}^{T}\frac{\partial f}{\partial x_{i^{\prime }}}\right]\\ & & +2\sum_{k=1}^{1}\sum_{i=1}^{2}\sum_{k^{\prime },\hat{k},\hat{i},i^{\prime }=1}^{3}\left[z_{kk^{\prime }}^{T}\frac{\partial }{\partial x_{k^{\prime }}}z_{k\hat{k}}^{T}\frac{\partial }{\partial x_{\hat{k}}}a_{i\hat{i}}^{T}\frac{\partial f}{\partial x_{\hat{i}}}a_{ii^{\prime }}^{T}\frac{\partial f}{\partial x_{i^{\prime }}}\right.\\ & & \;\;\;\;\;\;\;\;+z_{kk^{\prime }}^{T}z_{k\hat{k}}^{T}\frac{\partial^{2}}{\partial x_{k^{\prime }}\partial x_{\hat{k}}}a_{i\hat{i}}^{T}\frac{\partial f}{\partial x_{\hat{i}}}a_{ii^{\prime }}^{T}\frac{\partial f}{\partial x_{i^{\prime }}}\\ & & \left.\;\;\;\;\;\;\;\;+z_{kk^{\prime }}^{T}z_{k\hat{k}}^{T}\frac{\partial }{\partial x_{\hat{k}}}a_{i\hat{i}}^{T}\frac{\partial f}{\partial x_{\hat{i}}}\frac{\partial }{\partial x_{k^{\prime }}}a_{ii^{\prime }}^{T}\frac{\partial f}{\partial x_{i^{\prime }}}\right].\\ & & +2\sum_{k=1}^{1}\sum_{i=1}^{2}\sum_{\hat{k},\hat{i},i^{\prime }=1}^{3}{\left(z^{T}\nabla \log\pi \right)}_{k}\left[z_{k\hat{k}}^{T}\frac{\partial }{\partial x_{\hat{k}}}a_{i\hat{i}}^{T}\frac{\partial f}{\partial x_{\hat{i}}}a_{ii^{\prime }}^{T}\frac{\partial f}{\partial x_{i^{\prime }}}\right]\\ & & -2\sum_{j=1}^{1}\sum_{l=1}^{2}\sum_{l^{\prime },\hat{l},\hat{j},j^{\prime }=1}^{3}\left[\frac{\partial }{\partial x_{l^{\prime }}}a_{ll^{\prime }}^{T}a_{l\hat{l}}^{T}\frac{\partial }{\partial x_{\hat{l}}}z_{j\hat{j}}^{T}\frac{\partial f}{\partial x_{\hat{j}}}z_{jj^{\prime }}^{T}\frac{\partial f}{\partial x_{j^{\prime }}}\right]\\ & & -2\sum_{j=1}^{1}\sum_{l=1}^{2}\sum_{l^{\prime },\hat{l},\hat{j},j^{\prime }=1}^{3}\left[a_{ll^{\prime }}^{T}\frac{\partial }{\partial x_{l^{\prime }}}a_{l\hat{l}}^{T}\frac{\partial }{\partial x_{\hat{l}}}z_{j\hat{j}}^{T}\frac{\partial f}{\partial x_{\hat{j}}}z_{jj^{\prime }}^{T}\frac{\partial f}{\partial x_{j^{\prime }}}\right.\\ & & \;\;\;\;\;\;\;\;+a_{ll^{\prime }}^{T}a_{l\hat{l}}^{T}\frac{\partial^{2}}{\partial x_{l^{\prime }}\partial x_{\hat{l}}}z_{j\hat{j}}^{T}\frac{\partial f}{\partial x_{\hat{j}}}z_{jj^{\prime }}^{T}\frac{\partial f}{\partial x_{j^{\prime }}}\\ & & \left.\;\;\;\;\;\;\;\;+a_{ll^{\prime }}^{T}a_{l\hat{l}}^{T}\frac{\partial }{\partial x_{\hat{l}}}z_{j\hat{j}}^{T}\frac{\partial f}{\partial x_{\hat{j}}}\frac{\partial }{\partial x_{l^{\prime }}}z_{jj^{\prime }}^{T}\frac{\partial f}{\partial x_{j^{\prime }}}\right]\\ & & -2\sum_{j=1}^{1}\sum_{l=1}^{2}\sum_{\hat{l},\hat{j},j^{\prime }=1}^{3}{\left(a^{T}\nabla \log\pi \right)}_{l}\left[a_{l\hat{l}}^{T}\frac{\partial }{\partial x_{\hat{l}}}z_{j\hat{j}}^{T}\frac{\partial f}{\partial x_{\hat{j}}}z_{jj^{\prime }}^{T}\frac{\partial f}{\partial x_{j^{\prime }}}\right]\\ & = & \sum_{i=1}^{10}K_{i}. \end{array}$$By direct computation, we obtain
$$\begin{array}{ccc} K_{1} & = & 0,\;K_{2}=0,\;K_{3}=0,\;K_{4}=0,\;K_{5}=0,\;K_{6}=0,\;K_{7}=0;\\ K_{8} & = & -2\sum_{l=1}^{2}\sum_{l^{\prime },\hat{l}=1}^{3}a_{ll^{\prime }}^{T}a_{l\hat{l}}^{T}\frac{\partial^{2}z_{13}^{T}}{\partial x_{l^{\prime }}\partial x_{\hat{l}}}\partial_{y}f{\left(z^{T}\nabla \right)}_{1}f;\\ K_{9} & = & -2\sum_{l=1}^{2}\sum_{l^{\prime },\hat{l}=1}^{3}a_{ll^{\prime }}^{T}a_{l\hat{l}}^{T}\frac{\partial z_{13}^{T}}{\partial x_{\hat{l}}}\frac{\partial z_{13}^{T}}{\partial x_{l^{\prime }}}|\partial_{y}{f|}^{2}=-2\Gamma_{1}\left(\log g,\log g\right){\left|{\left(z^{T}\nabla \right)}_{1}f\right|}^{2};\\ K_{10} & = & -2\sum_{l=1}^{2}\sum_{\hat{l}=1}^{3}{\left(a^{T}\nabla \right)}_{l}\log\pi a_{l\hat{l}}^{T}\frac{\partial z_{13}^{T}}{\partial x_{\hat{l}}}\partial_{y}f{\left(z^{T}\nabla \right)}_{1}f=-2\Gamma_{1}\left(\log\pi ,\log g\right){\left|{\left(z^{T}\nabla \right)}_{1}f\right|}^{2}. \end{array}$$□

### 3.3. Martinet Flat Sub-Riemannian Structure

In this part, we apply our result to the Martinet flat sub-Riemannian structure, which satisfies the bracket-generating condition and has a non-equiregular sub-Riemannian structure (see [37]). The sub-Riemannian structure is defined on $R^{3}$ through the kernel of one-form $\eta :\;=dz-\frac{1}{2}y^{2}dx.$ A global orthonormal basis for the horizontal distribution H adapts the following differential operator representation, in local coordinates (x,y,z):$$X=\frac{\partial }{\partial x}+\frac{y^{2}}{2}\frac{\partial }{\partial z},\;Y=\frac{\partial }{\partial y}.$$

The commutative relation gives
$$\begin{array}{c} \left[X,Y\right]=-yZ,\;\left[Y,\left[X,Y\right]\right]=Z,\;\mathrm{where}\;Z=\frac{\partial }{\partial z}. \end{array}$$

To apply it in our framework, we take
(27)$$\begin{array}{ccc} a & = & \left(\begin{array}{cc} 1 & 0\\ 0 & 1\\ \frac{y^{2}}{2} & 0 \end{array}\right),\;a^{T}=\left(\begin{array}{ccc} 1 & 0 & \frac{y^{2}}{2}\\ 0 & 1 & 0 \end{array}\right),\\ z^{T} & = & \left(0,0,1\right),\;aa^{T}=\left(\begin{array}{ccc} 1 & 0 & \frac{y^{2}}{2}\\ 0 & 1 & 0\\ \frac{y^{2}}{2} & 0 & \frac{y^{4}}{4} \end{array}\right). \end{array}$$

Thus, the sub-Riemannian structure has the form $\left(M,H,{\left(aa^{T}\right)}_{|H}^{†}\right)$.

**Proposition** **5.**
*In this setting,*

$$\pi =e^{-\frac{y^{2}}{2}-V},$$

*then*

$$-aa^{T}\nabla \log\pi =a⊗\nabla a+aa^{T}\nabla V.$$



**Proof.** The poof follows from the observation that
$$a⊗\nabla a={\left(\begin{array}{ccc} 0 & y & 0 \end{array}\right)}^{T},\;aa^{T}\nabla \log e^{-\frac{y^{2}}{2}}={\left(\begin{array}{ccc} 0 & y & 0 \end{array}\right)}^{T}.$$□

Similar to the previous displacement group case, we have the following identity.

**Proposition** **6.**
*For any smooth function $f\in C^{\infty }\left(M\right)$, one has*

Γ2(f,f)+Γ2z,π(f,f)=∥Hessa,zf∥2+R(∇f,∇f),

*where*

$$\begin{array}{ccc} \Lambda_{1}^{T} & = & (0,y\partial_{z}f/2,0,y\partial_{z}f/2,0,0,0,0,0);\\ \Lambda_{2}^{T} & = & (0,0,0,0,0,0,-y\partial_{y}f,\frac{y^{3}}{2}\partial_{z}f+y\partial_{x}f,0);\\ & & R_{ab}\left(\nabla f,\nabla f\right)-\Lambda_{1}^{T}Q^{T}Q\Lambda_{1}-\Lambda_{2}^{T}P^{T}P\Lambda_{2}+D^{T}D+E^{T}E\\ & = & \frac{y^{2}}{2}\Gamma_{1}^{z}\left(f,f\right)-y^{2}\Gamma_{1}\left(f,f\right)\\ & & +\frac{\partial f}{\partial z}{\left(a^{T}\nabla \right)}_{1}f+y{\left(a^{T}\nabla \right)}_{1}V\frac{\partial f}{\partial z}{\left(a^{T}\nabla \right)}_{2}f+y\frac{\partial V}{\partial z}{\left(a^{T}\nabla \right)}_{1}f{\left(a^{T}\nabla \right)}_{2}f\\ & & +\sum_{\hat{i},k^{\prime }=1}^{3}a_{1\hat{i}}^{T}a_{1k^{\prime }}^{T}\frac{\partial^{2}V}{\partial x_{\hat{i}}\partial x_{k^{\prime }}}{\left|{\left(a^{T}\nabla \right)}_{1}f\right|}^{2}+2\left(\frac{\partial^{2}V}{\partial x\partial y}+\frac{y^{2}}{2}\frac{\partial^{2}V}{\partial y\partial z}\right){\left(a^{T}\nabla \right)}_{1}f{\left(a^{T}\nabla \right)}_{2}f \end{array}$$


$$\begin{array}{ccc} & & +\frac{\partial^{2}V}{\partial y\partial y}{\left|{\left(a^{T}\nabla \right)}_{2}f\right|}^{2}-y\frac{\partial V}{\partial y}\frac{\partial f}{\partial z}{\left(a^{T}\nabla \right)}_{1}f;\\ R_{zb}\left(\nabla f,\nabla f\right) & = & \left(\frac{\partial^{2}V}{\partial x\partial z}+\frac{y^{2}}{2}\frac{\partial^{2}V}{\partial z\partial z}\right){\left(a^{T}\nabla \right)}_{1}f{\left(z^{T}\nabla \right)}_{1}f+\frac{\partial^{2}V}{\partial y\partial z}{\left(a^{T}\nabla \right)}_{2}f{\left(z^{T}\nabla \right)}_{1}f;\\ R_{\pi }\left(\nabla f,\nabla f\right) & = & 0. \end{array}$$


*In particular, we have*

$$\begin{array}{c} \sum_{\hat{i},k^{\prime }=1}^{3}a_{1\hat{i}}^{T}a_{1k^{\prime }}^{T}\frac{\partial^{2}V}{\partial x_{\hat{i}}\partial x_{k^{\prime }}}|{\left(a^{T}\nabla \right)}_{1}{f|}^{2}=\left(\frac{\partial^{2}V}{\partial x\partial x}+y^{2}\frac{\partial^{2}V}{\partial x\partial z}+\frac{y^{4}}{4}\frac{\partial^{2}V}{\partial z\partial z}\right){\left|{\left(a^{T}\nabla \right)}_{1}f\right|}^{2}. \end{array}$$



The proof of Proposition 6 follows from the proof of Theorem 1 (i.e., Theorem 3) and Lemmas 7–9 below. The following convergence results are a direct consequence of Theorem 2.

**Proposition** **7.**
*If there exists κ>0 as shown in Theorem 2, the exponential dissipation result in the $L^{1}$ distance holds:*

∫|ρ(t,x)−π(x)|dx=O(e−κt).



Similarly, we summarize the sub-Riemannian Ricci tensor in terms of R as follows.

**Corollary** **3.**
*The matrix R associated with the Martinet sub-Riemannian structure has the following form:*

$$\begin{array}{ccc} R_{11} & = & \left(\frac{\partial^{2}V}{\partial x\partial x}+y^{2}\frac{\partial^{2}V}{\partial x\partial z}+\frac{y^{4}}{4}\frac{\partial^{2}V}{\partial z\partial z}\right)-y^{2};\\ R_{22} & = & \frac{\partial^{2}V}{\partial y\partial y}-y^{2};\;R_{33}=\frac{y^{2}}{2};\\ R_{12} & = & R_{21}=\frac{y}{2}\frac{\partial V}{\partial z}+\left(\frac{\partial^{2}V}{\partial x\partial y}+\frac{y^{2}}{2}\frac{\partial^{2}V}{\partial y\partial z}\right);\\ R_{13} & = & R_{31}=\frac{1}{2}-\frac{y}{2}\frac{\partial V}{\partial y}+\frac{1}{2}\left(\frac{\partial^{2}V}{\partial x\partial z}+\frac{y^{2}}{2}\frac{\partial^{2}V}{\partial z\partial z}\right);\;R_{23}=R_{32}=\frac{1}{2}y{\left(a^{T}\nabla \right)}_{1}V+\frac{1}{2}\frac{\partial^{2}V}{\partial y\partial z}. \end{array}$$



**Proof.** The proof follows from the similar equivalent matrix formulation as shown in the proof of Corollary 1 and the explicit bilinear forms in Proposition 6. □

Next, we prove the following three key lemmas.

**Lemma** **7.**
*For Martinet sub-Riemannian structure $\left(M,H,{\left(aa^{T}\right)}_{|H}^{†}\right)$, we have*

$$\begin{array}{ccc} Q & = & \left(\begin{array}{ccccccccc} 1 & 0 & \frac{y^{2}}{2} & 0 & 0 & 0 & \frac{y^{2}}{2} & 0 & \frac{y^{4}}{4}\\ 0 & 1 & 0 & 0 & 0 & 0 & 0 & \frac{y^{2}}{2} & 0\\ 0 & 0 & 0 & 1 & 0 & \frac{y^{2}}{2} & 0 & 0 & 0\\ 0 & 0 & 0 & 0 & 1 & 0 & 0 & 0 & 0 \end{array}\right);\\ P & = & \left(\begin{array}{ccccccccc} 0 & 0 & 0 & 0 & 0 & 0 & 1 & 0 & y^{2}/2\\ 0 & 0 & 0 & 0 & 0 & 0 & 0 & 1 & 0 \end{array}\right);\\ C^{T} & = & (0,0,0,0,0,\frac{y^{3}}{2}\partial_{z}f+y\partial_{x}f,-y\partial_{y}f,0,-\frac{y^{3}}{2}\partial_{y}f);\\ D^{T} & = & \left(0,0,y\partial_{z}f,0\right),\;E^{T}=\left(0,0\right);\\ F^{T} & = & G^{T}=\left(0,0,0,0,0,0,0,0,0\right). \end{array}$$



**Proof.** Plugging matrices *a* and *z* from (Equation 27) into Notation 1, we complete the proof. □

**Lemma** **8.**
*For the Martinet sub-Riemannian structure, F and G are zero vectors, and we have*

$$\begin{array}{ccc} & & {\left[QX+D\right]}^{T}\left[QX+D\right]+{\left[PX+E\right]}^{T}\left[PX+E\right]+2C^{T}X\\ & = & ∥{\mathrm{Hess}}_{a,z}{f∥}^{2}-\Lambda_{1}^{T}Q^{T}Q\Lambda_{1}-\Lambda_{2}^{T}P^{T}P\Lambda_{2}+D^{T}D+E^{T}E. \end{array}$$


*In particular, we have*

$$\begin{array}{ccc} ∥{\mathrm{Hess}}_{a,z}{f∥}^{2} & = & {\left[X+\Lambda_{1}\right]}^{T}Q^{T}Q\left[X+\Lambda_{1}\right]+{\left[X+\Lambda_{2}\right]}^{T}P^{T}P\left[X+\Lambda_{2}\right];\\ \Lambda_{1}^{T} & = & (0,y\partial_{z}f/2,0,y\partial_{z}f/2,0,0,0,0,0);\\ \Lambda_{2}^{T} & = & (0,0,0,0,0,0,-y\partial_{y}f,\frac{y^{3}}{2}\partial_{z}f+y\partial_{x}f,0);\\ & & -\Lambda_{1}^{T}Q^{T}Q\Lambda_{1}-\Lambda_{2}^{T}P^{T}P\Lambda_{2}+D^{T}D+E^{T}E\\ & = & \frac{y^{2}}{2}\Gamma_{1}^{z}\left(f,f\right)-y^{2}\Gamma_{1}\left(f,f\right). \end{array}$$



**Lemma** **9.**
*By routine computations, we obtain*

$$\begin{array}{ccc} & & R_{ab}\left(\nabla f,\nabla f\right)\\ & = & \frac{\partial f}{\partial z}{\left(a^{T}\nabla \right)}_{1}f+y{\left(a^{T}\nabla \right)}_{1}V\frac{\partial f}{\partial z}{\left(a^{T}\nabla \right)}_{2}f+y\frac{\partial V}{\partial z}{\left(a^{T}\nabla \right)}_{1}f{\left(a^{T}\nabla \right)}_{2}f\\ & & +\sum_{\hat{i},k^{\prime }=1}^{3}a_{1\hat{i}}^{T}a_{1k^{\prime }}^{T}\frac{\partial^{2}V}{\partial x_{\hat{i}}\partial x_{k^{\prime }}}{\left|{\left(a^{T}\nabla \right)}_{1}f\right|}^{2}+2\left(\frac{\partial^{2}V}{\partial x\partial y}+\frac{y^{2}}{2}\frac{\partial^{2}V}{\partial y\partial z}\right){\left(a^{T}\nabla \right)}_{1}f{\left(a^{T}\nabla \right)}_{2}f\\ & & +\frac{\partial^{2}V}{\partial y\partial y}{\left|{\left(a^{T}\nabla \right)}_{2}f\right|}^{2}-y\frac{\partial V}{\partial y}\frac{\partial f}{\partial z}{\left(a^{T}\nabla \right)}_{1}f;\\ & & R_{zb}\left(\nabla f,\nabla f\right)\\ & = & \left(\frac{\partial^{2}V}{\partial x\partial z}+\frac{y^{2}}{2}\frac{\partial^{2}V}{\partial z\partial z}\right){\left(a^{T}\nabla \right)}_{1}f{\left(z^{T}\nabla \right)}_{1}f+\frac{\partial^{2}V}{\partial y\partial z}{\left(a^{T}\nabla \right)}_{2}f{\left(z^{T}\nabla \right)}_{1}f;\\ & & R_{\pi }\left(\nabla f,\nabla f\right)=0. \end{array}$$



**Proof** **of** **Lemma** **8.**Since *F* and *G* are zero vectors, we have
$$\begin{array}{c} 2C^{T}X=2\left[\frac{\partial^{2}f}{\partial y\partial z}\left(\frac{y^{3}}{2}\partial_{z}f+y\partial_{x}f\right)-\frac{\partial^{2}f}{\partial x\partial z}\left(y\partial_{y}f\right)-\frac{\partial^{2}f}{\partial z\partial z}\left(\frac{y^{3}}{2}\partial_{y}f\right)\right]. \end{array}$$By routine computation, we observe that
$$\begin{array}{ccc} & & {\left[QX+D\right]}^{T}\left[QX+D\right]+{\left[PX+E\right]}^{T}\left[PX+E\right]+2C^{T}X\\ & = & {\left[\frac{\partial^{2}f}{\partial x\partial x}+\frac{y^{2}}{2}\frac{\partial^{2}f}{\partial x\partial z}+\frac{y^{2}}{2}\frac{\partial^{2}f}{\partial z\partial x}+\frac{y^{4}}{4}\frac{\partial^{2}f}{\partial z\partial z}\right]}^{2}+{\left[\frac{\partial^{2}f}{\partial y\partial x}+\frac{y^{2}}{2}\frac{\partial^{2}f}{\partial z\partial y}+y\partial_{z}f\right]}^{2}\\ & & +{\left[\frac{\partial^{2}f}{\partial y\partial x}+\frac{y^{2}}{2}\frac{\partial^{2}f}{\partial z\partial y}\right]}^{2}+{\left[\frac{\partial^{2}f}{\partial y\partial y}\right]}^{2}\\ & & +{\left[\frac{\partial^{2}f}{\partial z\partial x}+\frac{y^{2}}{2}\frac{\partial^{2}f}{\partial z\partial z}\right]}^{2}+{\left[\frac{\partial^{2}f}{\partial z\partial y}\right]}^{2}\\ & & +2\frac{\partial^{2}f}{\partial y\partial z}\left(\frac{y^{3}}{2}\partial_{z}f+y\partial_{x}f\right)-2\frac{\partial^{2}f}{\partial x\partial z}\left(y\partial_{y}f\right)-2\frac{\partial^{2}f}{\partial z\partial z}\left(\frac{y^{3}}{2}\partial_{y}f\right) \end{array}$$
$$\begin{array}{ccc} & = & {\left[\frac{\partial^{2}f}{\partial x\partial x}+\frac{y^{2}}{2}\frac{\partial^{2}f}{\partial x\partial z}+\frac{y^{2}}{2}\frac{\partial^{2}f}{\partial z\partial x}+\frac{y^{4}}{4}\frac{\partial^{2}f}{\partial z\partial z}\right]}^{2}+{\left[\frac{\partial^{2}f}{\partial y\partial x}+\frac{y^{2}}{2}\frac{\partial^{2}f}{\partial z\partial y}+y\partial_{z}f\right]}^{2}\\ & & +{\left[\frac{\partial^{2}f}{\partial y\partial x}+\frac{y^{2}}{2}\frac{\partial^{2}f}{\partial z\partial y}\right]}^{2}+{\left[\frac{\partial^{2}f}{\partial y\partial y}\right]}^{2}\\ & & +{\left[\frac{\partial^{2}f}{\partial z\partial x}+\frac{y^{2}}{2}\frac{\partial^{2}f}{\partial z\partial z}-y\partial_{y}f\right]}^{2}+{\left[\frac{\partial^{2}f}{\partial z\partial y}+\left(\frac{y^{3}}{2}\partial_{z}f+y\partial_{x}f\right)\right]}^{2}\\ & & -y^{2}{\left|\partial_{y}f\right|}^{2}-{\left(\frac{y^{3}}{2}\partial_{z}f+y\partial_{x}f\right)}^{2}\\ & = & |{\mathrm{Hess}}_{a,z}{f|}^{2}+\frac{y^{2}}{2}\Gamma_{1}^{z}\left(f,f\right)-y^{2}\Gamma_{1}\left(f,f\right), \end{array}$$
where we use the fact
$$\begin{array}{ccc} & & {\left[\frac{\partial^{2}f}{\partial y\partial x}+\frac{y^{2}}{2}\frac{\partial^{2}f}{\partial z\partial y}+y\partial_{z}f\right]}^{2}+{\left[\frac{\partial^{2}f}{\partial y\partial x}+\frac{y^{2}}{2}\frac{\partial^{2}f}{\partial z\partial y}\right]}^{2}\\ & = & 2{\left[\frac{\partial^{2}f}{\partial y\partial x}+\frac{y^{2}}{2}\frac{\partial^{2}f}{\partial z\partial y}+\frac{1}{2}y\partial_{z}f\right]}^{2}+\frac{y^{2}}{2}{\left|\partial_{z}f\right|}^{2}. \end{array}$$The proof is thus completed. □

We are now left to compute the three tensor terms.

**Proof** **of** **Lemma** **9.**Similar to the proof of Lemma 6, we have
$$\begin{array}{ccc} R_{a}\left(\nabla f,\nabla f\right) & = & \sum_{i,k=1}^{2}\sum_{i^{\prime },\hat{i},\hat{k}=1}^{3}\langle a_{ii^{\prime }}^{T}\left(\frac{\partial a_{i\hat{i}}^{T}}{\partial x_{i^{\prime }}}\frac{\partial a_{k\hat{k}}^{T}}{\partial x_{\hat{i}}}\frac{\partial f}{\partial x_{\hat{k}}}\right),{\left(a^{T}\nabla \right)}_{k}f\rangle {}_{R^{2}}\\ & & +\sum_{i,k=2}^{n}\sum_{i^{\prime },\hat{i},\hat{k}=1}^{3}\langle a_{ii^{\prime }}^{T}a_{i\hat{i}}^{T}\left(\frac{\partial }{\partial x_{i^{\prime }}}\frac{\partial a_{k\hat{k}}^{T}}{\partial x_{\hat{i}}}\right)\left(\frac{\partial f}{\partial x_{\hat{k}}}\right),{\left(a^{T}\nabla \right)}_{k}f\rangle {}_{R^{2}}\\ & & -\sum_{i,k=1}^{2}\sum_{i^{\prime },\hat{i},\hat{k}=1}^{3}\langle a_{k\hat{k}}^{T}\frac{\partial a_{ii^{\prime }}^{T}}{\partial x_{\hat{k}}}\frac{\partial a_{i\hat{i}}^{T}}{\partial x_{i^{\prime }}}\frac{\partial f}{\partial x_{\hat{i}}}),{\left(a^{T}\nabla \right)}_{k}{f\rangle }_{R^{2}}\\ & & -\sum_{i,k=1}^{2}\sum_{i^{\prime },\hat{i},\hat{k}=1}^{3}\langle a_{k\hat{k}}^{T}a_{ii^{\prime }}^{T}\left(\frac{\partial }{\partial x_{\hat{k}}}\frac{\partial a_{i\hat{i}}^{T}}{\partial x_{i^{\prime }}}\right)\frac{\partial f}{\partial x_{\hat{i}}},{\left(a^{T}\nabla \right)}_{k}f\rangle {}_{R^{2}},\\ & = & I_{1}+I_{2}+I_{3}+I_{4}. \end{array}$$By direct computations, we have
$$\begin{array}{ccc} I_{1} & = & \sum_{i=1}^{2}\sum_{i^{\prime },\hat{i},\hat{k}=1}^{3}\left[a_{ii^{\prime }}^{T}\left(\frac{\partial a_{i\hat{i}}^{T}}{\partial x_{i^{\prime }}}\frac{\partial a_{1\hat{k}}^{T}}{\partial x_{\hat{i}}}\frac{\partial f}{\partial x_{\hat{k}}}\right){\left(a^{T}\nabla \right)}_{1}f+a_{ii^{\prime }}^{T}\left(\frac{\partial a_{i\hat{i}}^{T}}{\partial x_{i^{\prime }}}\frac{\partial a_{2\hat{k}}^{T}}{\partial x_{\hat{i}}}\frac{\partial f}{\partial x_{\hat{k}}}\right){\left(a^{T}\nabla \right)}_{2}f\right]=0;\\ I_{2} & = & \sum_{i=2}^{n}\sum_{i^{\prime },\hat{i},\hat{k}=1}^{3}\left[a_{ii^{\prime }}^{T}a_{i\hat{i}}^{T}\left(\frac{\partial }{\partial x_{i^{\prime }}}\frac{\partial a_{1\hat{k}}^{T}}{\partial x_{\hat{i}}}\right)\left(\frac{\partial f}{\partial x_{\hat{k}}}\right){\left(a^{T}\nabla \right)}_{1}f+a_{ii^{\prime }}^{T}a_{i\hat{i}}^{T}\left(\frac{\partial }{\partial x_{i^{\prime }}}\frac{\partial a_{2\hat{k}}^{T}}{\partial x_{\hat{i}}}\right)\left(\frac{\partial f}{\partial x_{\hat{k}}}\right){\left(a^{T}\nabla \right)}_{2}f\right]\\ & = & a_{22}^{T}a_{22}^{T}\frac{\partial^{2}}{\partial y\partial y}a_{13}^{T}\frac{\partial f}{\partial z}{\left(a^{T}\nabla \right)}_{1}f=\frac{\partial f}{\partial z}{\left(a^{T}\nabla \right)}_{1}f;\\ I_{3} & = & -\sum_{i=1}^{2}\sum_{i^{\prime },\hat{i},\hat{k}=1}^{3}\left[a_{1\hat{k}}^{T}\frac{\partial a_{ii^{\prime }}^{T}}{\partial x_{\hat{k}}}\frac{\partial a_{i\hat{i}}^{T}}{\partial x_{i^{\prime }}}\frac{\partial f}{\partial x_{\hat{i}}}){\left(a^{T}\nabla \right)}_{1}f+a_{2\hat{k}}^{T}\frac{\partial a_{ii^{\prime }}^{T}}{\partial x_{\hat{k}}}\frac{\partial a_{i\hat{i}}^{T}}{\partial x_{i^{\prime }}}\frac{\partial f}{\partial x_{\hat{i}}}){\left(a^{T}\nabla \right)}_{2}f\right]=0;\\ I_{4} & = & -\sum_{i=1}^{2}\sum_{i^{\prime },\hat{i},\hat{k}=1}^{3}\left[a_{1\hat{k}}^{T}a_{ii^{\prime }}^{T}\left(\frac{\partial }{\partial x_{\hat{k}}}\frac{\partial a_{i\hat{i}}^{T}}{\partial x_{i^{\prime }}}\right)\frac{\partial f}{\partial x_{\hat{i}}}{\left(a^{T}\nabla \right)}_{1}f+a_{2\hat{k}}^{T}a_{ii^{\prime }}^{T}\left(\frac{\partial }{\partial x_{\hat{k}}}\frac{\partial a_{i\hat{i}}^{T}}{\partial x_{i^{\prime }}}\right)\frac{\partial f}{\partial x_{\hat{i}}}{\left(a^{T}\nabla \right)}_{2}f\right]=0. \end{array}$$For the drift term, we take $b=-aa^{T}\nabla V$
$$\begin{array}{ccc} R_{b}^{a} & = & \sum_{i,k=1}^{2}\sum_{\hat{i},\hat{k},k^{\prime }=1}^{3}\left[a_{i\hat{i}}^{T}\frac{\partial a_{k\hat{k}}^{T}}{\partial x_{\hat{i}}}a_{kk^{\prime }}^{T}\frac{\partial V}{\partial x_{k^{\prime }}}\frac{\partial f}{\partial x_{\hat{k}}}{\left(a^{T}\nabla \right)}_{i}f+a_{i\hat{i}}^{T}\frac{\partial a_{kk^{\prime }}^{T}}{\partial x_{\hat{i}}}a_{k\hat{k}}^{T}\frac{\partial V}{\partial x_{k^{\prime }}}\frac{\partial f}{\partial x_{\hat{k}}}{\left(a^{T}\nabla \right)}_{i}f\right]\\ & & +\sum_{i,k=1}^{2}\sum_{\hat{i},\hat{k},k^{\prime }=1}^{3}\left[a_{i\hat{i}}^{T}a_{k\hat{k}}^{T}a_{kk^{\prime }}^{T}\frac{\partial^{2}V}{\partial x_{\hat{i}}\partial x_{k^{\prime }}}\frac{\partial f}{\partial x_{\hat{k}}}{\left(a^{T}\nabla \right)}_{i}f\right]\\ & & -\sum_{i,k=1}^{2}\sum_{\hat{i},\hat{k},k^{\prime }=1}^{3}\left[a_{k\hat{k}}^{T}a_{kk^{\prime }}^{T}\frac{\partial a_{i\hat{i}}^{T}}{\partial x_{\hat{k}}}\frac{\partial V}{\partial x_{k^{\prime }}}\frac{\partial f}{\partial x_{\hat{i}}}{\left(a^{T}\nabla \right)}_{i}f\right]\\ & = & J_{1}+J_{2}+J_{3}+J_{4}. \end{array}$$Plugging into the matrices of $a^{T}$, we obtain
$$\begin{array}{ccc} J_{1} & = & \sum_{\hat{i},\hat{k},k^{\prime }=1}^{3}\left[a_{1\hat{i}}^{T}\frac{\partial a_{1\hat{k}}^{T}}{\partial x_{\hat{i}}}a_{1k^{\prime }}^{T}\frac{\partial V}{\partial x_{k^{\prime }}}\frac{\partial f}{\partial x_{\hat{k}}}{\left(a^{T}\nabla \right)}_{1}f+a_{2\hat{i}}^{T}\frac{\partial a_{1\hat{k}}^{T}}{\partial x_{\hat{i}}}a_{1k^{\prime }}^{T}\frac{\partial V}{\partial x_{k^{\prime }}}\frac{\partial f}{\partial x_{\hat{k}}}{\left(a^{T}\nabla \right)}_{2}f\right]\\ & & +\sum_{\hat{i},\hat{k},k^{\prime }=1}^{3}\left[a_{1\hat{i}}^{T}\frac{\partial a_{2\hat{k}}^{T}}{\partial x_{\hat{i}}}a_{2k^{\prime }}^{T}\frac{\partial V}{\partial x_{k^{\prime }}}\frac{\partial f}{\partial x_{\hat{k}}}{\left(a^{T}\nabla \right)}_{1}f+a_{2\hat{i}}^{T}\frac{\partial a_{2\hat{k}}^{T}}{\partial x_{\hat{i}}}a_{2k^{\prime }}^{T}\frac{\partial V}{\partial x_{k^{\prime }}}\frac{\partial f}{\partial x_{\hat{k}}}{\left(a^{T}\nabla \right)}_{2}f\right]\\ & = & a_{22}^{T}\frac{\partial a_{13}^{T}}{\partial y}{\left(a^{T}\nabla \right)}_{1}V\frac{\partial f}{\partial z}{\left(a^{T}\nabla \right)}_{2}f=y{\left(a^{T}\nabla \right)}_{1}V\frac{\partial f}{\partial z}{\left(a^{T}\nabla \right)}_{2}f; \end{array}$$
$$\begin{array}{ccc} J_{2} & = & \sum_{\hat{i},\hat{k},k^{\prime }=1}^{3}\left[a_{1\hat{i}}^{T}\frac{\partial a_{1k^{\prime }}^{T}}{\partial x_{\hat{i}}}a_{1\hat{k}}^{T}\frac{\partial V}{\partial x_{k^{\prime }}}\frac{\partial f}{\partial x_{\hat{k}}}{\left(a^{T}\nabla \right)}_{1}f+a_{2\hat{i}}^{T}\frac{\partial a_{1k^{\prime }}^{T}}{\partial x_{\hat{i}}}a_{1\hat{k}}^{T}\frac{\partial V}{\partial x_{k^{\prime }}}\frac{\partial f}{\partial x_{\hat{k}}}{\left(a^{T}\nabla \right)}_{2}f\right]\\ & & +\sum_{\hat{i},\hat{k},k^{\prime }=1}^{3}\left[a_{1\hat{i}}^{T}\frac{\partial a_{2k^{\prime }}^{T}}{\partial x_{\hat{i}}}a_{2\hat{k}}^{T}\frac{\partial V}{\partial x_{k^{\prime }}}\frac{\partial f}{\partial x_{\hat{k}}}{\left(a^{T}\nabla \right)}_{1}f+a_{2\hat{i}}^{T}\frac{\partial a_{2k^{\prime }}^{T}}{\partial x_{\hat{i}}}a_{2\hat{k}}^{T}\frac{\partial V}{\partial x_{k^{\prime }}}\frac{\partial f}{\partial x_{\hat{k}}}{\left(a^{T}\nabla \right)}_{2}f\right]\\ & = & y\frac{\partial V}{\partial z}{\left(a^{T}\nabla \right)}_{1}f{\left(a^{T}\nabla \right)}_{2}f; \end{array}$$
$$\begin{array}{ccc} J_{3} & = & \sum_{\hat{i},k^{\prime }=1}^{3}\left[a_{1\hat{i}}^{T}a_{1k^{\prime }}^{T}\frac{\partial^{2}V}{\partial x_{\hat{i}}\partial x_{k^{\prime }}}{\left|{\left(a^{T}\nabla \right)}_{1}f\right|}^{2}+a_{2\hat{i}}^{T}a_{1k^{\prime }}^{T}\frac{\partial^{2}V}{\partial x_{\hat{i}}\partial x_{k^{\prime }}}{\left(a^{T}\nabla \right)}_{1}f{\left(a^{T}\nabla \right)}_{2}f\right]\\ & & +\sum_{\hat{i},k^{\prime }=1}^{3}\left[a_{1\hat{i}}^{T}a_{2k^{\prime }}^{T}\frac{\partial^{2}V}{\partial x_{\hat{i}}\partial x_{k^{\prime }}}{\left(a^{T}\nabla \right)}_{2}f{\left(a^{T}\nabla \right)}_{1}f+a_{2\hat{i}}^{T}a_{2k^{\prime }}^{T}\frac{\partial^{2}V}{\partial x_{\hat{i}}\partial x_{k^{\prime }}}{\left|{\left(a^{T}\nabla \right)}_{2}f\right|}^{2}\right]\\ & = & \sum_{\hat{i},k^{\prime }=1}^{3}a_{1\hat{i}}^{T}a_{1k^{\prime }}^{T}\frac{\partial^{2}V}{\partial x_{\hat{i}}\partial x_{k^{\prime }}}{\left|{\left(a^{T}\nabla \right)}_{1}f\right|}^{2}+2\left(\frac{\partial^{2}V}{\partial x\partial y}+\frac{y^{2}}{2}\frac{\partial^{2}V}{\partial y\partial z}\right){\left(a^{T}\nabla \right)}_{1}f{\left(a^{T}\nabla \right)}_{2}f\\ & & +\frac{\partial^{2}V}{\partial y\partial y}{\left|{\left(a^{T}\nabla \right)}_{2}f\right|}^{2}; \end{array}$$
$$\begin{array}{ccc} J_{4} & = & -\sum_{\hat{i},\hat{k},k^{\prime }=1}^{3}\left[a_{1\hat{k}}^{T}a_{1k^{\prime }}^{T}\frac{\partial a_{1\hat{i}}^{T}}{\partial x_{\hat{k}}}\frac{\partial V}{\partial x_{k^{\prime }}}\frac{\partial f}{\partial x_{\hat{i}}}{\left(a^{T}\nabla \right)}_{1}f+a_{1\hat{k}}^{T}a_{1k^{\prime }}^{T}\frac{\partial a_{2\hat{i}}^{T}}{\partial x_{\hat{k}}}\frac{\partial V}{\partial x_{k^{\prime }}}\frac{\partial f}{\partial x_{\hat{i}}}{\left(a^{T}\nabla \right)}_{2}f\right] \end{array}$$
$$\begin{array}{ccc} & & -\sum_{\hat{i},\hat{k},k^{\prime }=1}^{3}\left[a_{2\hat{k}}^{T}a_{2k^{\prime }}^{T}\frac{\partial a_{1\hat{i}}^{T}}{\partial x_{\hat{k}}}\frac{\partial V}{\partial x_{k^{\prime }}}\frac{\partial f}{\partial x_{\hat{i}}}{\left(a^{T}\nabla \right)}_{1}f+a_{2\hat{k}}^{T}a_{2k^{\prime }}^{T}\frac{\partial a_{2\hat{i}}^{T}}{\partial x_{\hat{k}}}\frac{\partial V}{\partial x_{k^{\prime }}}\frac{\partial f}{\partial x_{\hat{i}}}{\left(a^{T}\nabla \right)}_{2}f\right]\\ & = & -y\frac{\partial V}{\partial y}\frac{\partial f}{\partial z}{\left(a^{T}\nabla \right)}_{1}f. \end{array}$$Combing the above computations, we obtain the tensor $R_{ab}$. Now, we turn to the second tensor $R_{zb}$. Since $z^{T}=\left(0,0,1\right)$, it is obvious to see that only the drift term of the tensor $R_{zb}$ remains, where we denote
$$\begin{array}{c} R_{b}^{z}\left(\nabla f,\nabla f\right)=-\sum_{\hat{i},\hat{k}=1}^{3}\left(z_{1\hat{i}}^{T}\frac{\partial b_{\hat{k}}}{\partial x_{\hat{i}}}\frac{\partial f}{\partial x_{\hat{k}}}-b_{\hat{k}}\frac{\partial z_{1\hat{i}}^{T}}{\partial x_{\hat{k}}}\frac{\partial f}{\partial x_{\hat{i}}}\right){\left(z^{T}\nabla \right)}_{1}f. \end{array}$$By taking $b=-aa^{T}\nabla V$, we further obtain that
$$\begin{array}{ccc} R_{b}^{z}\left(\nabla f,\nabla f\right) & = & -\sum_{\hat{i},\hat{k}=1}^{3}\left[z_{1\hat{i}}^{T}\frac{\partial b_{\hat{k}}}{\partial x_{\hat{i}}}\frac{\partial f}{\partial x_{\hat{k}}}{\left(z^{T}\nabla f\right)}_{1}-b_{\hat{k}}\frac{\partial z_{1\hat{i}}^{T}}{\partial x_{\hat{k}}}\frac{\partial f}{\partial x_{\hat{i}}}{\left(z^{T}\nabla f\right)}_{1}\right]\\ & = & \sum_{k=1}^{2}\sum_{\hat{i},\hat{k},k^{\prime }=1}^{3}\left[z_{1\hat{i}}^{T}\frac{\partial a_{k\hat{k}}^{T}}{\partial x_{\hat{i}}}a_{kk^{\prime }}^{T}\frac{\partial V}{\partial x_{k^{\prime }}}\frac{\partial f}{\partial x_{\hat{k}}}{\left(z^{T}\nabla \right)}_{1}f\right]\\ & & +\sum_{k=1}^{2}\sum_{\hat{i},\hat{k},k^{\prime }=1}^{3}\left[z_{1\hat{i}}^{T}\frac{\partial a_{kk^{\prime }}^{T}}{\partial x_{\hat{i}}}a_{k\hat{k}}^{T}\frac{\partial V}{\partial x_{k^{\prime }}}\frac{\partial f}{\partial x_{\hat{k}}}{\left(z^{T}\nabla \right)}_{1}f\right]\\ & & +\sum_{k=1}^{2}\sum_{\hat{i},\hat{k},k^{\prime }=1}^{3}\left[z_{1\hat{i}}^{T}a_{k\hat{k}}^{T}a_{kk^{\prime }}^{T}\frac{\partial^{2}V}{\partial x_{\hat{i}}\partial x_{k^{\prime }}}\frac{\partial f}{\partial x_{\hat{k}}}{\left(z^{T}\nabla \right)}_{1}f\right]\\ & & -\sum_{k=1}^{2}\sum_{\hat{i},\hat{k},k^{\prime }=1}^{3}\left[a_{k\hat{k}}^{T}a_{kk^{\prime }}^{T}\frac{\partial z_{1\hat{i}}^{T}}{\partial x_{\hat{k}}}\frac{\partial V}{\partial x_{k^{\prime }}}\frac{\partial f}{\partial x_{\hat{i}}}{\left(z^{T}\nabla \right)}_{1}f\right]\\ & = & J_{1}^{z}+J_{2}^{z}+J_{3}^{z}+J_{4}^{z}. \end{array}$$By direct computations, it is not hard to observe that
$$\begin{array}{ccc} J_{1}^{z} & = & \sum_{k=1}^{2}\sum_{\hat{i},\hat{k},k^{\prime }=1}^{3}\left[z_{1\hat{i}}^{T}\frac{\partial a_{k\hat{k}}^{T}}{\partial x_{\hat{i}}}a_{kk^{\prime }}^{T}\frac{\partial V}{\partial x_{k^{\prime }}}\frac{\partial f}{\partial x_{\hat{k}}}{\left(z^{T}\nabla \right)}_{1}f\right]=0;\\ J_{2}^{z} & = & \sum_{k=1}^{2}\sum_{\hat{i},\hat{k},k^{\prime }=1}^{3}\left[z_{1\hat{i}}^{T}\frac{\partial a_{kk^{\prime }}^{T}}{\partial x_{\hat{i}}}a_{k\hat{k}}^{T}\frac{\partial V}{\partial x_{k^{\prime }}}\frac{\partial f}{\partial x_{\hat{k}}}{\left(z^{T}\nabla \right)}_{1}f\right]=0;\\ J_{4}^{z} & = & -\sum_{k=1}^{2}\sum_{\hat{i},\hat{k},k^{\prime }=1}^{3}\left[a_{k\hat{k}}^{T}a_{kk^{\prime }}^{T}\frac{\partial z_{1\hat{i}}^{T}}{\partial x_{\hat{k}}}\frac{\partial V}{\partial x_{k^{\prime }}}\frac{\partial f}{\partial x_{\hat{i}}}{\left(z^{T}\nabla \right)}_{1}f\right]=0. \end{array}$$The only non-zero term has the following form:
$$\begin{array}{ccc} J_{3}^{z} & = & \sum_{k=1}^{2}\sum_{\hat{i},\hat{k},k^{\prime }=1}^{3}\left[z_{1\hat{i}}^{T}a_{k\hat{k}}^{T}a_{kk^{\prime }}^{T}\frac{\partial^{2}V}{\partial x_{\hat{i}}\partial x_{k^{\prime }}}\frac{\partial f}{\partial x_{\hat{k}}}{\left(z^{T}\nabla \right)}_{1}f\right]\\ & = & \sum_{\hat{i},\hat{k},k^{\prime }=1}^{3}\left[z_{1\hat{i}}^{T}a_{1\hat{k}}^{T}a_{1k^{\prime }}^{T}\frac{\partial^{2}V}{\partial x_{\hat{i}}\partial x_{k^{\prime }}}\frac{\partial f}{\partial x_{\hat{k}}}{\left(z^{T}\nabla \right)}_{1}f+z_{1\hat{i}}^{T}a_{2\hat{k}}^{T}a_{2k^{\prime }}^{T}\frac{\partial^{2}V}{\partial x_{\hat{i}}\partial x_{k^{\prime }}}\frac{\partial f}{\partial x_{\hat{k}}}{\left(z^{T}\nabla \right)}_{1}f\right]\\ & = & \sum_{\hat{i},k^{\prime }=1}^{3}\left[z_{1\hat{i}}^{T}a_{1k^{\prime }}^{T}\frac{\partial^{2}V}{\partial x_{\hat{i}}\partial x_{k^{\prime }}}{\left(a^{T}\nabla \right)}_{1}f{\left(z^{T}\nabla \right)}_{1}f+z_{1\hat{i}}^{T}a_{2k^{\prime }}^{T}\frac{\partial^{2}V}{\partial x_{\hat{i}}\partial x_{k^{\prime }}}{\left(a^{T}\nabla \right)}_{2}f{\left(z^{T}\nabla \right)}_{1}f\right]\\ & = & \left(\frac{\partial^{2}V}{\partial x\partial z}+\frac{y^{2}}{2}\frac{\partial^{2}V}{\partial z\partial z}\right){\left(a^{T}\nabla \right)}_{1}f{\left(z^{T}\nabla \right)}_{1}f+\frac{\partial^{2}V}{\partial y\partial z}{\left(a^{T}\nabla \right)}_{2}f{\left(z^{T}\nabla \right)}_{1}f. \end{array}$$Since matrix $z^{T}$ is a constant matrix and matrix $a^{T}$ contains only variable *y*, it is easy to observe that
$$R_{\pi }\left(\nabla f,\nabla f\right)=0.$$□

## 4. Lyapunov Analysis in Sub-Riemannian Density Manifold

In this section, we illustrate the motivation of this paper, which is to design a matrix condition, whose smallest eigenvalue characterizes the convergence rate of the degenerate SDE.

The outline of this section is given below. Consider a density space over the sub-Riemannian manifold. The finite-dimensional sub-Riemannian structure introduces the density space the *infinite-dimensional sub-Riemannian structure*. We name it the *sub-Riemannian density manifold* (SDM). We provide the geometric calculations in the SDM. We studied the Fokker–Planck equation as the sub-Riemannian gradient flow in the SDM. We derived the equivalence relation between the second-order calculus of the relative entropy in the SDM and the generalized Gamma z calculus.

### 4.1. Sub-Riemannian Density Manifold

Given a finite-dimensional sub-Riemannian manifold $\left(R^{n+m},\tau ,g_{\tau }\right)$ with $g_{\tau }={\left(aa^{T}\right)}^{†}$, consider the probability density space:$$P\left(R^{n+m}\right)=\left\{\rho \left(x\right)\in C^{\infty }\left(R^{n+m}\right):\int \rho \left(x\right)dx=1,\;\rho \left(x\right)\geq 0\right\}.$$

Consider the tangent space at $\rho \in P(R^{n+m})$:$$T_{\rho }P\left(R^{n+m}\right)=\left\{\sigma \left(x\right)\in C^{\infty }\left(R^{n+m}\right):\int \sigma \left(x\right)dx=0\right\}.$$

We introduce the sub-Riemannian structure in probability density space $P(R^{n+m})$.

**Definition** **3**(sub-Riemannian Wasserstein metric tensor)**.**
*The $L^{2}$ sub-Riemannian-Wasserstein metric $g_{\rho }^{W_{a}}:T_{\rho }P\left(R^{n+m}\right)\times T_{\rho }P\left(R^{n+m}\right)\to R$ is defined by*
$$g_{\rho }^{W_{a}}\left(\sigma_{1},\sigma_{2}\right)=\int \left(\sigma_{1}\left(x\right),{\left(-\Delta_{\rho }^{a}\right)}^{†}\sigma_{2}\left(x\right)\right)dx.$$*Here, $\sigma_{1},\sigma_{2}\in T_{\rho }P\left(R^{n+m}\right)$, (·,·) is the metric on $R^{n+m}$, and ${\left(\Delta_{\rho }^{a}\right)}^{†}:T_{\rho }P\left(R^{n+m}\right)\to T_{\rho }P\left(R^{n+m}\right)$ is the pseudo-inverse of the sub-elliptic operator:*
$$\Delta_{\rho }^{a}=\nabla \cdot \left(\rho aa^{T}\nabla \right).$$

For some special choices of *a* as studied in [19] or $aa^{T}$ forming a positive definite matrix, then $\Delta_{\rho }^{a}$ is an elliptic operator. In this case, $\left(P\left(R^{n+m}\right),g^{W_{a}}\right)$ still forms a Riemannian density manifold. In general, given a sub-Riemannian manifold $\left(R^{n+m},{\left(aa^{T}\right)}^{†}\right)$, $\Delta_{\rho }^{a}$ is only a sub-elliptic operator. Thus, $\left(P\left(R^{n+m}\right),g^{W_{a}}\right)$ forms an infinite-dimensional sub-Riemannian manifold.

We next present the sub-Riemannian calculus in $\left(P\left(R^{n+m}\right),g^{W_{a}}\right)$, including both geodesics and the Hessian operator in the tangent bundle. Consider an identification map:$$V:C^{\infty }\left(R^{n+m}\right)\to T_{\rho }P\left(R^{n+m}\right),\;V_{\Phi }=-\Delta_{\rho }^{a}\Phi =-\nabla \cdot \left(\rho aa^{T}\nabla \Phi \right).$$

Here, $\Phi \in T_{\pi }P\left(R^{n+m}\right)=C^{\infty }\left(R^{n+m}\right)/∼$. This $T_{\pi }P\left(R^{n+m}\right)$ is the cotangent space in the SDM, and ∼ represents a constant shift relation. Thus,
$$g_{\rho }^{W_{a}}\left(V_{\Phi_{1}},V_{\Phi_{2}}\right)=\int \Gamma_{1}\left(\Phi_{1},\Phi_{2}\right)\rho \left(x\right)dx.$$

In other words,
(28)$$\begin{array}{cc} g_{\rho }^{W_{a}}\left(V_{\Phi_{1}},V_{\Phi_{2}}\right)= & \int V_{\Phi_{1}}{\left(-\Delta_{\rho }^{a}\right)}^{†}V_{\Phi_{2}}dx\\ = & \int \Phi_{1}\left(-\Delta_{\rho }^{a}\right){\left(-\Delta_{\rho }^{a}\right)}^{†}\left(-\Delta_{\rho }^{a}\right)\Phi_{2}dx\\ = & \int (\Phi_{1},-\Delta_{\rho }^{a}\Phi_{2})dx\\ = & \int \Phi_{1}(-\nabla \cdot \left(\rho aa^{T}\nabla \Phi_{2}\right)dx\\ = & \int (a^{T}\nabla \Phi_{1},a^{T}\nabla \Phi_{2})\rho dx, \end{array}$$
where the second equality holds by $\left(-\Delta_{\rho }^{a}\right){\left(-\Delta_{\rho }^{a}\right)}^{†}\left(-\Delta_{\rho }^{a}\right)=-\Delta_{\rho }^{a}$ and the last equality holds by the integration by parts.

We next derive several basic geometric calculations in the SDM.

**Proposition** **8**(Geodesics in the SDM)**.**
*The sub-Riemannian geodesics in the cotangent bundle forms*
(29)$$\left\{\begin{array}{c} \partial_{t}\rho_{t}+\nabla \cdot \left(\rho_{t}aa^{T}\nabla \Phi_{t}\right)=0,\\ \partial_{t}\Phi_{t}+\frac{1}{2}\left(a^{T}\nabla \Phi_{t},a^{T}\nabla \Phi_{t}\right)=0. \end{array}\right.$$

**Proof.** We considered the Lagrangian formulation of geodesics in density. Here, the minimization of the geometric action functional forms
$$L\left(\rho_{t},\partial_{t}\rho_{t}\right)=\int_{0}^{1}\int \frac{1}{2}\left(\partial_{t}\rho_{t},{\left(-\Delta_{\rho_{t}}^{a}\right)}^{†}\partial_{t}\rho_{t}\right)dxdt,$$
where $\rho_{t}=\rho \left(t,x\right)$ is a density path with fixed boundary points $\rho_{0}$, $\rho_{1}$. Then, the Euler–Lagrange equation in density space forms
$$\frac{\partial }{\partial t}\delta_{\partial_{t}\rho_{t}}L\left(\rho_{t},\partial_{t}\rho_{t}\right)=\delta_{\rho_{t}}L\left(\rho_{t},\partial_{t}\rho_{t}\right),$$
where $\delta_{\partial_{t}\rho_{t}}$ is the $L^{2}$ first variation with respect to $\partial_{t}\rho_{t}$ and $\delta_{\rho_{t}}$ is the $L^{2}$ first variation with respect to $\rho_{t}$. Here,
(30)$$\begin{array}{cc} \partial_{t}\left({\left(-\Delta_{\rho_{t}}^{a}\right)}^{†}\partial_{t}\rho_{t}\right)= & \delta_{\rho }\int \frac{1}{2}\left(\partial_{t}\rho ,{\left(-\Delta_{\rho_{t}}^{a}\right)}^{†}\partial_{t}\rho_{t}\right)dx\\ = & -\frac{1}{2}\left(a^{T}\nabla {\left(-\Delta_{\rho }^{a}\right)}^{†}\partial_{t}\rho_{t},a^{T}\nabla {\left(-\Delta_{\rho }^{a}\right)}^{†}\partial_{t}\rho_{t}\right), \end{array}$$
where the last equality uses the following fact:
$$\partial_{t}{\left(\Delta_{\rho_{t}}^{a}\right)}^{†}=-{\left(\Delta_{\rho_{t}}^{a}\right)}^{†}\cdot \Delta_{\partial_{t}\rho_{t}}^{a}\cdot {\left(\Delta_{\rho_{t}}^{a}\right)}^{†},$$Denote $\partial_{t}\rho_{t}=-\Delta_{\rho_{t}}^{a}\Phi_{t}$, then the Euler–Lagrange Equation (Equation 30) forms the sub-Riemannian geodesics flow (Equation 29). In other words,
$$\partial_{t}\Phi +\frac{1}{2}\left(a^{T}\nabla \Phi ,a^{T}\nabla \Phi \right)=0.$$□

**Proposition** **9**(Gradient and Hessian operators in the SDM)**.**
*Given a functional $F:P(R^{n+m})\to R$, the gradient operator of F in $\left(P,g^{W_{a}}\right)$ satisfies*
$$grad_{W_{a}}F\left(\rho \right)=-\nabla \cdot \left(\rho aa^{T}\nabla \delta F\left(\rho \right)\right).$$
*The Hessian operator of F in $\left(P,g^{W_{a}}\right)$ satisfies*

$$\begin{array}{cc} \; & Hess_{W_{a}}F\left(V_{\Phi },V_{\Phi }\right)\\ = & \int \int \left(a{\left(y\right)}^{T}\nabla_{y}\right)\left(a{\left(x\right)}^{T}\nabla_{x}\right)\delta^{2}F\left(\rho \right)\left(x,y\right)\left(a{\left(x\right)}^{T}\nabla_{x}\Phi \left(x\right),a{\left(y\right)}^{T}\nabla_{y}\Phi \left(y\right)\right)\rho \left(x\right)\rho \left(y\right)dxdy\\ \; & +\int {\mathrm{Hess}}_{a}\delta F\left(\rho \right)\left(\Phi ,\Phi \right)\rho dx, \end{array}$$

*where*

$${\mathrm{Hess}}_{a}\delta F\left(\rho \right)\left(\Phi ,\Phi \right)=\frac{1}{2}\left\{2\Gamma_{1}\left(\Gamma_{1}\left(\delta F,\Phi \right),\Phi \right)-\Gamma_{1}\left(\Gamma_{1}\left(\Phi ,\Phi \right),\delta F\right)\right\}.$$



**Proof.** We first derive the sub-Riemannian gradient operator. We recall the identification map by $-\Delta_{\rho }^{a}\Phi =-\nabla \cdot \left(\rho aa^{T}\nabla \Phi \right)$. Hence, the gradient operator in the SDM satisfies
$$\begin{array}{cc} {\mathrm{grad}}_{W_{a}}F\left(\rho \right)= & {\left({\left(-\Delta_{\rho }^{a}\right)}^{†}\right)}^{†}\frac{\delta }{\delta \rho (x)}F\left(\rho \right)\\ = & -\Delta_{\rho }^{a}\frac{\delta }{\delta \rho (x)}F\left(\rho \right)\\ = & -\nabla \cdot (\rho aa^{T}\nabla \frac{\delta }{\delta \rho (x)}F\left(\rho \right)). \end{array}$$The Hessian operator in the SDM satisfies
$${\mathrm{Hess}}_{W_{a}}F\left(\rho \right)\left(V_{\Phi },V_{\Phi }\right)=\frac{d^{2}}{dt^{2}}F\left(\rho_{t}\right){|}_{t=0},$$
where $\left(\rho_{t},\Phi_{t}\right)$ satisfies the geodesics Equation (Equation 29) with $\rho_{0}=\rho$, $\Phi_{0}=\Phi$. Notice the fact that
$$\begin{array}{cc} \frac{d}{dt}F\left(\rho_{t}\right){|}_{t=0}= & \int \partial_{t}\rho_{t}{\delta F\left(\rho \right)dx|}_{t=0}\\ = & \int \left(-\nabla \cdot \left(\rho aa^{T}\nabla \Phi \right)\right)\delta F\left(\rho \right)dx\\ = & \int (a^{T}\nabla \delta F\left(\rho \right),a^{T}\nabla \Phi )\rho dx. \end{array}$$In addition,
(31)$$\begin{array}{c} \begin{array}{cc} \; & \frac{d^{2}}{dt^{2}}F\left(\rho_{t}\right){{|}_{t=0}=\frac{d}{dt}\int \left(a^{T}\nabla \delta F\left(\rho_{t}\right),a^{T}\nabla \Phi_{t}\right)\rho_{t}dx|}_{t=0}\\ = & \int \int \delta^{2}F\left(\rho \right)\left(x,y\right)\partial_{t}\rho_{t}\left(x\right)\partial_{t}\rho_{t}\left(y\right)dxdy+\int \left(a^{T}\nabla \delta F\left(\rho_{t}\right),a^{T}\nabla \partial_{t}\Phi_{t}\right)\rho_{t}dx\\ \; & +\int \left(a^{T}\nabla \delta F\left(\rho_{t}\right),a^{T}\nabla \partial_{t}\Phi_{t}\right)\partial_{t}\rho_{t}{dx|}_{t=0}\\ = & \int \int \delta^{2}F\left(\rho \right)\left(x,y\right)\nabla \cdot \left(\rho aa^{T}\nabla \Phi \right)\left(x\right)\nabla \cdot \left(\rho aa^{T}\nabla \Phi \right)\left(y\right)dxdy\\ \; & -\frac{1}{2}\int \left(a^{T}\nabla \delta F\left(\rho \right),a^{T}\nabla \Gamma_{1}\left(\Phi ,\Phi \right)\right)\rho dx\\ \; & +\int \Gamma_{1}\left(\Phi ,\delta F\left(\rho \right)\right)\left(-\nabla \cdot (\rho aa^{T}\nabla \Phi )\right)dx\\ = & \int \int \left(a{\left(y\right)}^{T}\nabla_{y}\right)\left(a{\left(x\right)}^{T}\nabla_{x}\right)\delta^{2}F\left(\rho \right)\left(x,y\right)\left(a{\left(x\right)}^{T}\nabla_{x}\Phi \left(x\right),a{\left(y\right)}^{T}\nabla_{y}\Phi \left(y\right)\right)\rho \left(x\right)\rho \left(y\right)dxdy\\ + & \frac{1}{2}\int \left\{2\Gamma_{1}\left(\Gamma_{1}\left(\delta F,\Phi \right),\Phi \right)-\Gamma_{1}\left(\Gamma_{1}\left(\Phi ,\Phi \right),\delta F\right)\right\}\rho dx, \end{array} \end{array}$$
where the last equality holds by the integration by parts formula. □

We next show the equivalence relation between the Hessian of the relative entropy in the SDM and the classical Gamma two operator. We first demonstrate the relation among $L^{*}$, $\Delta_{a}$ and the gradient operator of the entropy. In particular, we show that the Fokker–Planck equation is a sub-Riemannian gradient flow in the SDM. Denote the KL divergence as
(32)$$D\left(\rho \right)=D_{\mathrm{KL}}\left(\rho ∥\pi \right)=\int \rho \left(x\right)\log\frac{\rho (x)}{\pi (x)}dx.$$

**Proposition** **10**(Gradient flow)**.**
*The negative gradient operator in $\left(P,g^{W_{a}}\right)$ forms*
$$-{\mathrm{grad}}_{W_{a}}D\left(\rho \right)=L^{*}\rho =\nabla \cdot \left(\rho aa^{T}\nabla \log\frac{\rho }{\pi }\right).$$
*In addition, the sub-Riemannian gradient flow of D(ρ) in $\left(P,g^{W_{a}}\right)$ forms the Fokker–Planck equation:*

(33)
$$\partial_{t}\rho =\nabla \cdot \left(\rho aa^{T}\nabla \log\frac{\rho }{\pi }\right).$$



**Proof.** We first derive the sub-Riemannian gradient operator of the entropy and relative entropy. Notice that
$$\delta_{\rho (x)}D\left(\rho \right)=\log\rho \left(x\right)+1-\log\pi \left(x\right).$$Thus,
$${\mathrm{grad}}_{W_{a}}D\left(\rho \right)=-\nabla \cdot \left(\rho aa^{T}\nabla \log\rho \right)+\nabla \cdot \left(\rho aa^{T}\nabla \log\pi \right),$$
where $\rho \nabla \log\rho =\rho \frac{\nabla \rho }{\rho }=\nabla \rho$. Following the gradient flow formulation:
$$\frac{\partial \rho_{t}}{\partial t}=-{\mathrm{grad}}_{W_{a}}D\left(\rho_{t}\right)=L^{*}\rho_{t},$$
we finish the derivation of (Equation 33). □

We next demonstrate that the Hessian of the relative entropy (KL divergence) is equivalent to the classical Bakry–Émery calculus.

**Proposition** **11**(Hessian of entropy and Bakry–Émery calculus)**.**
*Given $\Phi_{1}$, $\Phi_{2}\in C^{\infty }\left(R^{n+m}\right)$, then*
$$\begin{array}{cc} Hess_{W_{a}}D\left(\rho \right)\left(V_{\Phi },V_{\Phi }\right)= & \int \Gamma_{2}\left(\Phi ,\Phi \right)\rho \left(x\right)dx. \end{array}$$

**Proof.** We first derive the Hessian of D(ρ) in the SDM. Notice the fact that $\delta^{2}D\left(\rho \right)\left(x,y\right)=\frac{1}{\rho }\delta_{x=y}$. For simplicity, we denote $\delta^{2}D\left(\rho \right)=\frac{1}{\rho (x)}$. By using (Equation 31), we have
(34)$$\begin{array}{ccc} {\mathrm{Hess}}_{W_{a}}D\left(\rho \right)\left(V_{\Phi },V_{\Phi }\right)= & \int \delta^{2}D\left(\rho \right)\left(x\right){\left(\nabla \cdot (\rho aa^{T}\nabla \Phi )\right)}^{2}dx & \left(a\right)\\ \; & -\frac{1}{2}\int \left(a^{T}\nabla \delta D\left(\rho \right),a^{T}\nabla \Gamma_{1}\left(\Phi ,\Phi \right)\right)\rho dx & \left(b\right)\\ \; & +\int \Gamma_{1}\left(\Phi ,\delta D\left(\rho \right)\right)\left(-\nabla \cdot (\rho aa^{T}\nabla \Phi )\right)dx. & \left(c\right) \end{array}$$We next rewrite (Equation 34) into the iterative Gamma calculus. We first show that
$$\begin{array}{cc} (a)+(c)= & \int \left(\delta^{2}D\left(\rho \right)\nabla \cdot \left(\rho aa^{T}\nabla \Phi \right)-\Gamma_{1}\left(\Phi ,\delta D\left(\rho \right)\right)\right)\nabla \cdot \left(\rho aa^{T}\nabla \Phi \right)\rho dx\\ = & \int \left(\frac{1}{\rho }\nabla \cdot \left(\rho aa^{T}\nabla \Phi \right)-\left(aa^{T}\nabla \log\frac{\rho }{\pi },\nabla \Phi \right)\right)\nabla \cdot \left(\rho aa^{T}\nabla \Phi \right)\rho dx\\ = & \int \left(\frac{1}{\rho }\left(\nabla \rho ,aa^{T}\nabla \Phi \right)+\nabla \cdot \left(aa^{T}\nabla \Phi \right)-\left(aa^{T}\nabla \log\frac{\rho }{\pi },\nabla \Phi \right)\right)\nabla \cdot \left(\rho aa^{T}\nabla \Phi \right)\rho dx\\ = & \int (\left(\nabla \log\rho ,aa^{T}\nabla \Phi \right)+\nabla \cdot \left(aa^{T}\nabla \Phi \right)\\ \; & \;-\left(\nabla \log\rho ,aa^{T}\nabla \Phi \right)+\left(aa^{T}\nabla \log\pi ,\nabla \Phi \right))\nabla \cdot \left(\rho aa^{T}\nabla \Phi \right)\rho dx\\ = & \int \left((\nabla \cdot \left(aa^{T}\nabla \Phi \right)+\left(aa^{T}\nabla \log\pi ,\nabla \Phi \right)\right)\nabla \cdot \left(\rho aa^{T}\nabla \Phi \right)\rho dx\\ = & \int L\Phi \nabla \cdot (\rho aa^{T}\nabla \Phi )dx\\ = & -\int \Gamma_{1}\left(L\Phi ,\Phi \right)\rho dx, \end{array}$$
where the fourth equality uses the fact that $\frac{\nabla \rho }{\rho }=\nabla \log\rho$, while the last equality follows the integration by parts.We secondly show that
$$\begin{array}{cc} (b)= & -\frac{1}{2}\int \left(a^{T}\nabla \delta D\left(\rho \right),a^{T}\nabla \Gamma_{1}\left(\Phi ,\Phi \right)\right)\rho dx\\ = & \frac{1}{2}\int \Gamma_{1}\left(\Phi ,\Phi \right))\nabla \cdot \left(\rho aa^{T}\nabla \delta D\left(\rho \right)\right)dx\\ = & \frac{1}{2}\int \Gamma_{1}\left(\Phi ,\Phi \right))L^{*}\rho dx\\ = & \frac{1}{2}\int L\Gamma_{1}\left(\Phi ,\Phi \right))\rho dx, \end{array}$$
where the second equality applies the fact that $L^{*}\rho =\nabla \cdot \left(\rho aa^{T}\nabla \delta D\right)$, while the last inequality uses the dual-relation between Kolmogorov operators *L* and $L^{*}$ in $L^{2}\left(\rho \right)$, i.e.,
$$\int f\left(x\right)L^{*}\rho \left(t,x\right)dx=\int Lf\left(x\right)\rho \left(t,x\right)dx,\;\mathrm{for}\mathrm{any}f\in C_{c}^{\infty }\left(R^{n+m}\right).$$Combining the equality of (a),(b),(c), we prove the result. □

**Remark** **10.**
*We remark that the above formulations in terms of $aa^{T}$ hold for both Riemannian and sub-Riemannian density manifolds. Here, the major difference is whether matrix function a is full rank or degenerate. In this sense, all formulas derived in this subsection recover the classical Bakry–Émery calculus. However, the classical Hessian operator of the entropy is not enough to study the convergence behavior of degenerate diffusion processes. Briefly, we use a modified Lyapunov functional and derive a tensor for the gradient flow in the SDM. It provides the convergence rate of the degenerate diffusion process.*


### 4.2. Gamma z Calculus via Second-Order Calculus of Relative Entropy in SDM

In this subsection, we introduce the motivation of our new Gamma z calculus from the SDM viewpoint. Consider the SDM gradient flow (Equation 33):$$\partial_{t}\rho_{t}=\Delta_{\rho_{t}}^{a}\delta D\left(\rho_{t}\right).$$

When *a* is a degenerate matrix, the classical relative Fisher information $I_{a}$ may not be the Lyapunov functional. In other words, along the gradient flow, it is possible that $\frac{d}{dt}I_{a}\left(\rho_{t}\right)\geq 0$.

To handle this issue, a new Lyapunov function is considered. It is to add a new direction *z* into the relative Fisher information functional. Denote $\Delta_{\rho }^{z}=\nabla \cdot \left(\rho zz^{T}\nabla \right)$ and $I_{z}\left(\rho \right)=\int \left(\delta D,(-\Delta_{\rho }^{z})\delta D\right)dx$. Construct
$$I_{a,z}\left(\rho \right):=I_{a}\left(\rho \right)+I_{z}\left(\rho \right)=\int \left(\delta D,(-\Delta_{\rho }^{a}-\Delta_{\rho }^{z})\delta D\right)dx.$$

We next prove the following proposition.

**Proposition** **12.**

$$\frac{d}{dt}I_{a,z}\left(\rho_{t}\right)=-2\int \left(\Gamma_{2}\left(\delta D,\delta D\right)+{\tilde{\Gamma }}_{2}^{z}\left(\delta D,\delta D\right))\right)\rho_{t}dx,$$

*where*

(35)
$$\begin{array}{cc} {\tilde{\Gamma }}_{2}^{z}\left(\Phi ,\Phi \right):= & \frac{1}{2}L\left(\Gamma_{1}^{z}\left(\Phi ,\Phi \right)\right)-\Gamma_{1}\left(L_{z}\Phi ,\Phi \right), \end{array}$$

*with the notation $\Delta_{z}=\nabla \cdot \left(zz^{T}\nabla \right)$ and $L_{z}=\nabla \cdot \left(zz^{T}\nabla \right)+\left(\nabla \log\pi ,zz^{T}\nabla \right)$.*


**Proof.** For the simplicity of notation, we denote $\rho =\rho_{t}$. Notice the fact that
$$\frac{d}{dt}I_{a,z}\left(\rho \right)=\frac{d}{dt}I_{a}\left(\rho \right)+\frac{d}{dt}I_{z}\left(\rho \right).$$From Proposition 11, we have
$$\begin{array}{cc} \frac{d^{2}}{dt^{2}}I_{a}\left(\rho \right)= & -2{\mathrm{Hess}}_{g_{a}}D\left(V_{\delta D},V_{\delta D}\right)\\ = & -2\int \Gamma_{2}\left(\delta D,\delta D\right)\rho dx. \end{array}$$□

We only need to show the following claim.


**Claim:**

$$\frac{d}{dt}I_{z}\left(\rho \right)=-2\int {\tilde{\Gamma }}_{2}^{z}\left(\delta D,\delta D\right)\rho dx.$$



**Proof** **of** **Claim.**The proof is similar to the ones in Proposition 11. We need to take care of the *z* direction. Notice that
$$\begin{array}{cc} \frac{d}{dt}I_{z}\left(\rho \right)= & 2\int \delta^{2}D\left(\left(-\Delta_{\rho }^{z}\delta D\right),\partial_{t}\rho \right)dx+\int \left(\nabla \delta D,zz^{T}\nabla \delta D\right)\partial_{t}\rho dx\\ = & 2\int \delta^{2}D\left(\left(-\Delta_{\rho }^{z}\delta D\right),\Delta_{\rho }^{a}\delta D\right)dx+\int \left(\nabla \delta D,zz^{T}\nabla \delta D\right)\left(\Delta_{\rho }^{a}\delta D\right)dx\\ = & -2\int \frac{1}{\rho }\nabla \cdot \left(\rho aa^{T}\nabla \delta D\right)\nabla \cdot \left(\rho zz^{T}\nabla \delta D\right)dx\;\left(I\right)\\ \; & +\int \left(\nabla \delta D,zz^{T}\nabla \delta D\right)\nabla \cdot \left(\rho aa^{T}\nabla \delta D\right)dx\;\left(II\right) \end{array}$$We next estimate (I) and (II) separately. For (I), we notice the fact that
$$\begin{array}{cc} \frac{1}{\rho }\nabla \cdot \left(\rho zz^{T}\nabla \delta D\right)= & \left(\nabla \log\rho ,zz^{T}\nabla \delta D\right)+\nabla \cdot \left(zz^{T}\nabla \delta D\right)\\ = & \left(\nabla \log\frac{\rho }{\pi },zz^{T}\nabla \delta D\right)+\left(\nabla \log\pi ,zz^{T}\nabla \delta D\right)+\nabla \cdot \left(zz^{T}\nabla \delta D\right)\\ = & \left(\nabla \delta D,zz^{T}\nabla \delta D\right)+\left(\nabla \log\pi ,zz^{T}\nabla \delta D\right)+\nabla \cdot \left(zz^{T}\nabla \delta D\right). \end{array}$$Thus,
$$\begin{array}{cc} (I)= & -2\int \frac{1}{\rho }\nabla \cdot \left(\rho aa^{T}\nabla \delta D\right)\nabla \cdot \left(\rho zz^{T}\nabla \delta D\right)dx\\ = & -2\int \nabla \cdot \left(\rho aa^{T}\nabla \delta D\right)(\left(\nabla \delta D,zz^{T}\nabla \delta D\right)\\ \; & \;\;\;+\left(\nabla \log\pi ,zz^{T}\nabla \delta D\right)+\nabla \cdot \left(zz^{T}\nabla \delta D\right))dx\\ = & -2\int \left(\nabla \delta D,zz^{T}\nabla \delta D\right)L^{*}\rho \\ \; & \;\;+\nabla \cdot \left(\rho aa^{T}\nabla \delta D\right)\left(\left(\nabla \log\pi ,zz^{T}\nabla \delta D\right)+\nabla \cdot \left(zz^{T}\nabla \delta D\right)\right)dx\\ = & -2\int L(\nabla \delta D,zz^{T}\nabla \delta D)\rho dx\\ \; & +2\int \left\{\nabla \left(\left(\nabla \log\pi ,zz^{T}\nabla \delta D\right)+\nabla \cdot \left(zz^{T}\nabla \delta D\right)\right),aa^{T}\nabla \delta D\right\}\rho dx, \end{array}$$
where the last equality holds by integration by parts.For (II), we have
$$\begin{array}{cc} (II)= & \int \left(\nabla \delta D,zz^{T}\nabla \delta D\right)\nabla \cdot \left(\rho aa^{T}\nabla \delta D\right)dx\\ = & \int \left(\nabla \delta D,zz^{T}\nabla \delta D\right)L^{*}\rho dx\\ = & \int L(\nabla \delta D,zz^{T}\nabla \delta D)\rho dx. \end{array}$$Combining (I) and (II), we have
$${\tilde{\Gamma }}_{2}^{z}\left(\Phi ,\Phi \right)=\frac{1}{2}L\left(\Gamma_{1}^{z}\left(\Phi ,\Phi \right)\right)-\Gamma_{1}\left(\Delta_{z}\Phi ,\Phi \right)-\Gamma_{1}\left(\Gamma_{1}^{z}\left(\log\pi ,\Phi \right),\Phi \right).$$Using the notation $L_{z}=\Delta_{z}+\left(\nabla \log\pi ,zz^{T}\nabla \right)$, we finish the proof. □

We next prove that ${\tilde{\Gamma }}_{2}^{z}$ and $\Gamma_{2}^{z,\pi }$ in Definition 1 agree with each other in the weak form along the gradient flow.

**Proposition** **13.**
*Denote Φ=δD(ρ), then*

$$\int {\tilde{\Gamma }}_{2}^{z}\left(\Phi ,\Phi \right)\rho dx=\int \Gamma_{2}^{z,\pi }\left(\Phi ,\Phi \right)\rho dx.$$



**Proof.** To prove the proposition, we rewrite ${\tilde{\Gamma }}_{2}^{z}$ as follows.
$$\begin{array}{cc} {\tilde{\Gamma }}_{2}^{z}\left(\Phi ,\Phi \right)= & \frac{1}{2}L\left(\Gamma_{1}^{z}\left(\Phi ,\Phi \right)\right)-\Gamma_{1}\left(L_{z}\Phi ,\Phi \right)\\ = & \frac{1}{2}L\left(\Gamma_{1}^{z}\left(\Phi ,\Phi \right)\right)-\Gamma_{1}^{z}\left(L\Phi ,\Phi \right)\\ \; & +\Gamma_{1}^{z}\left(L\Phi ,\Phi \right)-\Gamma_{1}\left(L_{z}\Phi ,\Phi \right). \end{array}$$□

Here, we need to prove the following equality.


**Claim:**

$$\begin{array}{cc} \; & \int \left\{\Gamma_{1}^{z}\left(L\Phi ,\Phi \right)-\Gamma_{1}\left(L_{z}\Phi ,\Phi \right)\right\}\rho dx\\ = & \int \rho \left\{\frac{1}{\pi }\nabla \cdot \left(zz^{T}\pi \left(\nabla \Phi ,\nabla (aa^{T})\nabla \Phi \right)\right)-\frac{1}{\pi }\nabla \cdot \left(aa^{T}\pi \left(\nabla \Phi ,\nabla (zz^{T})\nabla \Phi \right)\right)\right\}dx. \end{array}$$



**Proof** **of** **Claim.**For the simplicity of notation, let
$$L^{*}\rho =\nabla \cdot \left(aa^{T}\pi \nabla \frac{\rho }{\pi }\right)=\nabla \cdot \left(\rho aa^{T}\nabla \log\frac{\rho }{\pi }\right)$$
and
$$L_{z}^{*}\rho =\nabla \cdot \left(zz^{T}\pi \nabla \frac{\rho }{\pi }\right)=\nabla \cdot \left(\rho zz^{T}\nabla \log\frac{\rho }{\pi }\right).$$The following property is also used in the proof. For any smooth test function *f* and $\Phi =\log\frac{\rho }{\pi }$, then
$$\int L_{z}^{*}\rho fdx=-\int \Gamma_{1}^{z}\left(f,\Phi \right)\rho dx,\;\int L^{*}\rho fdx=-\int \Gamma_{1}\left(f,\Phi \right)\rho dx.$$Notice that $\Phi =\log\frac{\rho }{\pi }$, then
$$\begin{array}{cc} \; & \int \Gamma_{1}^{z}\left(L\Phi ,\Phi \right)\rho dx\\ = & \int (\nabla \left(\nabla \cdot \left(aa^{T}\nabla \Phi \right)-\left(A,\nabla \Phi \right)\right),zz^{T}\nabla \Phi )\rho dx\\ = & \int \left(\nabla \left(\nabla \cdot \left(aa^{T}\nabla \Phi \right)\right),zz^{T}\nabla \Phi \right)\rho dx-\int \left(\nabla \left(A,\nabla \Phi \right),zz^{T}\nabla \Phi \right)\rho dx.\\ \; & \;(a1)\;(a2) \end{array}$$Here,
$$\begin{array}{cc} (a1)= & \int \left(\nabla \left(\nabla \cdot \left(aa^{T}\nabla \Phi \right)\right),zz^{T}\nabla \Phi \right)\rho dx\\ = & -\int \nabla \cdot \left(aa^{T}\nabla \Phi \right)\nabla \cdot \left(\rho zz^{T}\nabla \Phi \right)dx\\ = & -\int \nabla \cdot \left(aa^{T}\nabla \log\frac{\rho }{\pi }\right)\nabla \cdot \left(\rho zz^{T}\nabla \log\frac{\rho }{\pi }\right)dx\\ = & -\int \nabla \cdot \left(\frac{1}{\rho }aa^{T}\pi \nabla \frac{\rho }{\pi }\right)\nabla \cdot \left(\rho zz^{T}\nabla \log\frac{\rho }{\pi }\right)dx\\ = & -\int \left\{\left(\nabla \frac{1}{\rho },aa^{T}\pi \nabla \frac{\rho }{\pi }\right)+\frac{1}{\rho }\nabla \cdot \left(aa^{T}\pi \nabla \frac{\rho }{\pi }\right)\right\}\nabla \cdot \left(\rho zz^{T}\nabla \log\frac{\rho }{\pi }\right)dx\\ = & -\int \left(\nabla \frac{1}{\rho },aa^{T}\pi \nabla \frac{\rho }{\pi }\right)L_{z}^{*}\rho dx-\int \frac{1}{\rho }L^{*}\rho L_{z}^{*}\rho dx\\ = & \int \frac{1}{\rho^{2}}\left(\nabla \rho ,aa^{T}\pi \nabla \frac{\rho }{\pi }\right)L_{z}^{*}\rho dx-\int \frac{1}{\rho }L^{*}\rho L_{z}^{*}\rho dx\\ = & \int \left(\nabla \log\rho ,aa^{T}\nabla \log\frac{\rho }{\pi }\right)L_{z}^{*}\rho dx-\int \frac{1}{\rho }L^{*}\rho L_{z}^{*}\rho dx\\ = & \int \left(\nabla \log\frac{\rho }{\pi },aa^{T}\nabla \log\frac{\rho }{\pi }\right)L_{z}^{*}\rho dx\\ \; & +\int \left(\nabla \log\pi ,aa^{T}\nabla \log\frac{\rho }{\pi }\right)L_{z}^{*}\rho dx-\int \frac{1}{\rho }L^{*}\rho L_{z}^{*}\rho dx\\ = & -\int (\nabla (\nabla \log\frac{\rho }{\pi },aa^{T}\nabla \log\frac{\rho }{\pi }),zz^{T}\nabla \log\frac{\rho }{\pi })\rho dx\\ \; & -\int \Gamma_{1}^{z}\left(\left(\nabla \log\pi ,aa^{T}\nabla \log\frac{\rho }{\pi }\right),\log\frac{\rho }{\pi }\right)\rho dx-\int \frac{1}{\rho }L^{*}\rho L_{z}^{*}\rho dx\\ = & -\int \left((\nabla \log\frac{\rho }{\pi },\nabla \left(aa^{T}\right)\nabla \log\frac{\rho }{\pi }\right),zz^{T}\nabla \frac{\rho }{\pi })\pi dx\\ \; & -\int 2\nabla^{2}\log\frac{\rho }{\pi }\left(aa^{T}\nabla \log\frac{\rho }{\pi },zz^{T}\nabla \log\frac{\rho }{\pi }\right)\rho dx\\ \; & -\int \Gamma_{1}^{z}\left(\left(\nabla \log\pi ,aa^{T}\nabla \log\frac{\rho }{\pi }\right),\log\frac{\rho }{\pi }\right)\rho dx-\int \frac{1}{\rho }L^{*}\rho L_{z}^{*}\rho dx \end{array}$$
$$\begin{array}{cc} = & \int \nabla \cdot (zz^{T}\pi \left((\nabla \log\frac{\rho }{\pi },\nabla \left(aa^{T}\right)\nabla \log\frac{\rho }{\pi }\right))\frac{1}{\pi }\rho dx\\ \; & -\int 2\nabla^{2}\log\frac{\rho }{\pi }\left(aa^{T}\nabla \log\frac{\rho }{\pi },zz^{T}\nabla \log\frac{\rho }{\pi }\right)\rho dx\\ \; & -\int \Gamma_{1}^{z}\left(\left(\nabla \log\pi ,aa^{T}\nabla \log\frac{\rho }{\pi }\right),\log\frac{\rho }{\pi }\right)\rho dx-\int \frac{1}{\rho }L^{*}\rho L_{z}^{*}\rho dx. \end{array}$$Notice the fact that
$$\begin{array}{cc} (a2)= & -\int (\nabla \left(A,\nabla \Phi \right),zz^{T}\nabla \Phi )\rho dx\\ = & \int \left(\nabla \left(\nabla \log\pi ,aa^{T}\nabla \Phi \right),zz^{T}\nabla \Phi \right)\rho dx\\ = & \int \Gamma_{1}\left(\Gamma_{1}^{z}\left(\Phi ,\Phi \right),\Phi \right)\rho dx. \end{array}$$Hence,
$$\begin{array}{cc} \int \Gamma_{1}^{z}\left(L\Phi ,\Phi \right)\rho dx= & \left(a1)+(a2\right)\\ = & \int \nabla \cdot (zz^{T}\pi \left((\nabla \log\frac{\rho }{\pi },\nabla \left(aa^{T}\right)\nabla \log\frac{\rho }{\pi }\right))\frac{1}{\pi }\rho dx\\ \; & -\int 2\nabla^{2}\log\frac{\rho }{\pi }\left(aa^{T}\nabla \log\frac{\rho }{\pi },zz^{T}\nabla \log\frac{\rho }{\pi }\right)\rho dx\\ \; & -\int \frac{1}{\rho }L^{*}\rho L_{z}^{*}\rho dx. \end{array}$$Similarly, by switching *a* and *z*, we have
$$\begin{array}{cc} \int \Gamma_{1}\left(L_{z}\Phi ,\Phi \right)\rho dx= & \int \nabla \cdot (aa^{T}\pi \left((\nabla \log\frac{\rho }{\pi },\nabla \left(zz^{T}\right)\nabla \log\frac{\rho }{\pi }\right))\frac{1}{\pi }\rho dx\\ \; & -\int 2\nabla^{2}\log\frac{\rho }{\pi }\left(aa^{T}\nabla \log\frac{\rho }{\pi },zz^{T}\nabla \log\frac{\rho }{\pi }\right)\rho dx\\ \; & -\int \frac{1}{\rho }L^{*}\rho L_{z}^{*}\rho dx. \end{array}$$Combining the above derivation, we finish the proof. □

**Remark** **11.**
*From the proof, we can show the following identity: denote Φ=δD, then*

$$\begin{array}{cc} \; & \int \left(\Gamma_{1}^{z}\left(L\Phi ,\Phi \right)-\Gamma_{1}\left(L_{z}\Phi ,\Phi \right)\right)\rho dx\\ = & \int \left(\Gamma_{1}\left(\Gamma_{1}^{z}\left(\Phi ,\Phi \right),\Phi \right)-\Gamma_{1}^{z}\left(\Gamma_{1}\left(\Phi ,\Phi \right),\Phi \right)\right)\rho dx. \end{array}$$


*Therefore, it is clear that, if the commutative assumption $\Gamma_{1}\left(\Gamma_{1}^{z}\left(\Phi ,\Phi \right),\Phi \right)=\Gamma_{1}^{z}\left(\Gamma \left(\Phi ,\Phi \right),\Phi \right)$ holds, the above quantity equals zero. In this case,*

$$\int \Gamma_{2}^{z,\pi }\left(\Phi ,\Phi \right)\rho dx=\int \Gamma_{2}^{z}\left(\Phi ,\Phi \right)\rho dx.$$


*This means that, under the commutative assumption, the generalized Gamma z calculus agrees with the classical one [2] in the weak sense.*


With the generalized Gamma z calculus, we are ready to prove the convergence properties and functional inequalities for degenerate drift–diffusion processes.

**Proposition** **14.**
*Suppose $\Gamma_{2}+\Gamma_{2}^{z,\pi }⪰\kappa \left(\Gamma_{1}+\Gamma_{1}^{z}\right)$ with κ>0. Denote $\rho_{t}$ as the solution of the sub-Riemannian gradient flow *(Equation 33)*, then*

$$\frac{d}{dt}\left(I_{a}\left(\rho_{t}\right)+I_{z}\left(\rho_{t}\right)\right)\leq -2\kappa \left(I_{a}\left(\rho_{t}\right)+I_{z}\left(\rho_{t}\right)\right).$$


*In addition, the z-log-Sobolev inequalities holds:*

$$\int_{R^{n+m}}\rho \log\frac{\rho }{\pi }dx\leq \frac{1}{2\kappa }I_{a,z}\left(\rho \right),$$

*for any smooth density function ρ.*

*Finally,*

$$\int_{R^{n+m}}\left|\rho \left(t,x\right)-\pi \left(x\right)\right|dx\leq \sqrt{2D(\rho_{0})}e^{-\kappa t}.$$



**Proof.** Here, the proof is very similar to the one in the previous section. Again, consider the sub-Riemannian gradient flow in the SDM.
$$\partial_{t}\rho_{t}=-{\mathrm{grad}}_{W_{a}}D\left(\rho_{t}\right).$$We know that the log-Sobolev inequality relates to the ratio of $\frac{d}{dt}D\left(\rho_{t}\right)$ and $\frac{d^{2}}{dt^{2}}D\left(\rho_{t}\right)$. If we cannot estimate a ratio κ>0, then
$$\frac{d}{dt}I_{a}\left(\rho_{t}\right)\leq -2\kappa I_{a}\left(\rho_{t}\right).$$We construct the other Lyapunov function:
$$I_{a,z}\left(\rho \right)=I_{a}\left(\rho \right)+I_{z}\left(\rho \right).$$Thus, along the SDM gradient flow (Equation 33), we have
$$\frac{d}{dt}I_{a,z}\left(\rho_{t}\right)=-2\int \left(\Gamma_{2}\left(\delta D,\delta D\right)+\Gamma_{2}^{z,\pi }\left(\delta D,\delta D\right)\right)\rho_{t}dx.$$If $\Gamma_{2}+\Gamma_{2}^{z,\pi }⪰\kappa \left(\Gamma_{1}+\Gamma_{1}^{z}\right)$, then
(36)$$\frac{d}{dt}I_{a,z}\left(\rho_{t}\right)\leq -2\kappa I_{a,z}\left(\rho_{t}\right).$$The convergence result follows directly from Gronwall’s equality.We next prove the *z*-log-Sobolev inequality. Since
$$-\frac{d}{dt}D\left(\rho_{t}\right)=I_{a}\left(\rho_{t}\right)\leq I_{a,z}\left(\rho_{t}\right),$$
then (Equation 36) implies the fact that, denoting $\rho_{0}=\rho$, then
$$\begin{array}{cc} -I_{a,z}\left(\rho \right)= & \int_{0}^{\infty }\frac{d}{dt}I_{a,z}\left(\rho_{t}\right)dt\\ \leq & -2\kappa \int_{0}^{\infty }I_{a,z}\left(\rho_{t}\right)dt=-2\kappa \int_{0}^{\infty }\left(I_{a}\left(\rho_{t}\right)+I_{z}\left(\rho_{t}\right)\right)dt\\ \leq & -2\kappa \int_{0}^{\infty }I_{a}\left(\rho_{t}\right)dt\\ = & -2\kappa \int_{0}^{\infty }\left(-\frac{d}{dt}D\left(\rho_{t}\right)\right)dt\\ = & -2\kappa D(\rho ). \end{array}$$Thus, $I_{a,z}\left(\rho \right)\geq 2\kappa D\left(\rho \right)$. Hence, we prove all the results by the fact that $R⪰\kappa (\Gamma_{1}+\Gamma_{1}^{z})$ implies $\Gamma_{2}+\Gamma_{2}^{z,\pi }⪰\kappa \left(\Gamma_{1}+\Gamma_{1}^{z}\right)$. In other words, the generalized Gamma z calculus implies the z-log-Sobolev equality (zLSI):
$$R⪰\kappa \left(\Gamma_{1}+\Gamma_{1}^{z}\right)\Rightarrow \frac{d}{dt}I_{a,z}\left(\rho_{t}\right)\leq -2\kappa I_{a,z}\left(\rho_{t}\right)\Rightarrow \mathrm{zLSI}.$$We last prove the exponential convergence in the $L^{1}$ distance. Notice that
$$D_{\mathrm{KL}}\left(\rho_{t}∥\pi \right)\leq \frac{1}{2\lambda }I_{a,z}\left(\rho_{t}∥\pi \right)\leq \frac{1}{2\lambda }e^{-2\lambda t}I_{a,z}\left(\rho_{0}∥\pi \right).$$We apply an inequality between the KL divergence and $L_{1}$ distance. In other words,
$$\int_{R^{n+m}}\left|\rho \left(t,x\right)-\pi \left(x\right)\right|dx\leq \sqrt{2D_{\mathrm{KL}}\left(\rho ∥\pi \right)}.$$This finishes the proof. □

**Remark** **12.**
*It is worth mentioning that our derivation of the Gamma z calculus is not a direct Hessian operator of the entropy in the SDM. In fact, it combines both the second-order calculus in the SDM and the property of the $L^{2}$ Hessian operator of the entropy. See similar relations in the mean-field Bakry–Émery calculus [50].*


## 5. Generalized Gamma *z* Calculus

In this section, we introduce the generalized Gamma *z* calculus. For any smooth functions $f,g:R^{n+m}\to R$, the diffusion operator associated with SDE (Equation 2) is denoted as
$$\begin{array}{c} Lf=\Delta_{a}f-A\nabla f+b\nabla f, \end{array}$$
where we denote A=a⊗∇a and
$$\begin{array}{c} \Delta_{a}f=\nabla \cdot \left(aa^{T}\nabla f\right). \end{array}$$

When b=0, we denote the diffusion operator as
$$\begin{array}{ccc} \tilde{L}f & = & \Delta_{a}f-a⊗\nabla a\nabla f. \end{array}$$

We first define the Carré de Champ operator $\Gamma_{1}$ associated with the above second-order diffusion operators. It is easy to check that $\Delta_{a}$, $\tilde{L}$, and *L* share the same $\Gamma_{1}$:(37)$$\begin{array}{c} \Gamma_{1}\left(f,g\right)={\langle a^{T}\nabla f,a^{T}\nabla f\rangle }_{R^{n}}. \end{array}$$

Similarly, we introduce the $\Gamma_{1}^{z}$ operator in the direction of $z=(z_{1},\cdots ,z_{m})$ below:(38)$$\begin{array}{c} \Gamma_{1}^{z}={\langle z^{T}\nabla f,z^{T}\nabla f\rangle }_{R^{m}}, \end{array}$$

Next, we define the iterative $\Gamma_{2}$ and $\Gamma_{2}^{z}$ for operator *L* ($\tilde{L}$, respectively) below:(39)$$\begin{array}{c} \Gamma_{2,L}\left(f,f\right)=\frac{1}{2}L\Gamma_{1}^{z}\left(f,f\right)-\Gamma_{1}^{z}\left(Lf,f\right). \end{array}$$
(40)$$\begin{array}{c} \Gamma_{2,L}^{z}\left(f,f\right)=\frac{1}{2}L\Gamma_{1}^{z}\left(f,f\right)-\Gamma_{1}^{z}\left(Lf,f\right). \end{array}$$

**Definition** **4.**
*We define the generalized Gamma z for operator L below:*

(41)
$$\begin{array}{ccc} \Gamma_{2,L}^{z,\pi }\left(f,f\right) & = & \Gamma_{2,L}^{z}\left(f,f\right)+{\mathrm{div}}_{z}^{\pi }\left(\Gamma_{\nabla (aa^{T})}f,f\right)-{\mathrm{div}}_{a}^{\pi }\left(\Gamma_{\nabla (zz^{T})}f,f\right), \end{array}$$


*For matrices $a\in R^{n\times (n+m)}$ and $z\in R^{m\times (n+m)}$, we denote the divergence operator as*

$$\begin{array}{ccc} {\mathrm{div}}_{z}^{\pi }\left(\Gamma_{\nabla (aa^{T})}f,f\right) & = & \frac{\nabla \cdot (zz^{T}\pi \Gamma_{\nabla (aa^{T})}\left(f,f\right))}{\pi },\\ {\mathrm{div}}_{a}^{\pi }\left(\Gamma_{\nabla (zz^{T})}f,f\right) & = & \frac{\nabla \cdot (aa^{T}\pi \Gamma_{\nabla (zz^{T})}\left(f,f\right))}{\pi }, \end{array}$$

*and*

$$\begin{array}{c} \Gamma_{\nabla (aa^{T)}}\left(f,f\right)=\langle \nabla f,\nabla \left(aa^{T}\right)\nabla f\rangle ,\;\mathrm{and}\;\Gamma_{\nabla (zz^{T)}}\left(f,f\right)=\langle \nabla f,\nabla \left(zz^{T}\right)\nabla f\rangle . \end{array}$$


*Here, we denote π as the invariant distribution associated with the operator L.*


**Remark** **13.**
*In particular, we have the following local coordinates representation.*

(42)
$$\begin{array}{ccc} \langle \nabla f,\nabla \left(aa^{T}\right)\nabla f\rangle & = & {\langle \nabla f,\frac{\partial }{\partial x_{\hat{k}}}\left(aa^{T}\right)\nabla f\rangle }_{\hat{k}=1}^{n+m}=2{\langle a^{T}\nabla f,\frac{\partial }{\partial x_{\hat{k}}}a^{T}\nabla f\rangle }_{\hat{k}=1}^{n+m}\\ & = & {\left(2\sum_{i=1}^{n}\sum_{\hat{i},i^{\prime }=1}^{n+m}\frac{\partial }{\partial x_{\hat{k}}}a_{i\hat{i}}^{T}\frac{\partial f}{\partial x_{\hat{i}}}a_{ii^{\prime }}^{T}\frac{\partial f}{\partial x_{i^{\prime }}}\right)}_{\hat{k}=1}^{n+m},\\ \langle \nabla f,\nabla \left(zz^{T}\right)\nabla f\rangle & = & {\left(2\sum_{j=1}^{n}\sum_{\hat{j},j^{\prime }=1}^{n+m}\frac{\partial }{\partial x_{\hat{k}}}z_{j\hat{j}}^{T}\frac{\partial f}{\partial x_{\hat{j}}}z_{jj^{\prime }}^{T}\frac{\partial f}{\partial x_{j^{\prime }}}\right)}_{\hat{k}=1}^{n+m}. \end{array}$$



We first present the following key lemmas.

**Lemma** **10.**

$$\begin{array}{ccc} {\mathrm{div}}_{z}^{\pi }\left(\Gamma_{\nabla (aa^{T})}f,f\right)-{\mathrm{div}}_{a}^{\pi }\left(\Gamma_{\nabla (zz^{T})}f,f\right) & = & R^{\pi }\left(f,f\right)+2G^{T}X, \end{array}$$

*where X, G are defined in Notation 1 and $R^{\pi }$ is defined in Definition 2.*


**Lemma** **11.**

$$\begin{array}{ccc} & & \Gamma_{2,\tilde{L}}\left(f,f\right)=X^{T}Q^{T}QX+2D^{T}QX+2C^{T}X+D^{T}D+R_{a}\left(\nabla f,\nabla f\right), \end{array}$$

*where Q,X,C,D are introduced in Notation 1 and $R_{a}$ is defined in Definition 2.*


**Lemma** **12.**

$$\begin{array}{c} \Gamma_{2,\tilde{L}}^{z}\left(f,f\right)=X^{T}P^{T}PX+2E^{T}PX+2F^{T}X+E^{T}E+R_{z}\left(\nabla f,\nabla f\right). \end{array}$$

*where P,X,F,E are introduced in Notation 1 and $R_{z}$ is defined in Definition 2.*


We then have the following main theorem. In order to distinguish the operators *L* and $\tilde{L}$, we rewrite Theorem 1 as below, and with some abuse of notation, we denote $\Gamma_{2}\left(f,f\right)=\Gamma_{2,L}\left(f,f\right)$ and $\Gamma_{2}^{z,\pi }\left(f,f\right)=\Gamma_{2,L}^{z,\pi }\left(f,f\right)$.

**Theorem** **3**(z-Bochner’s formula)**.**
*For smooth function $f:R^{n+m}\to R$, assume that Assumption 1 holds, then*
$$\begin{array}{ccc} \Gamma_{2,L}\left(f,f\right)+\Gamma_{2,L}^{z,\pi }\left(f,f\right) & = & ∥{\mathrm{Hess}}_{a,z}{f∥}^{2}+R\left(\nabla f,\nabla f\right), \end{array}$$
*where*
$$\begin{array}{ccc} ∥{\mathrm{Hess}}_{a,z}{f∥}^{2} & = & {\left[X+\Lambda_{1}\right]}^{T}Q^{T}Q\left[X+\Lambda_{1}\right]+{\left[X+\Lambda_{2}\right]}^{T}P^{T}P\left[X+\Lambda_{2}\right]\\ R(\nabla f,\nabla f) & = & -\Lambda_{1}^{T}Q^{T}Q\Lambda_{1}-\Lambda_{2}^{T}P^{T}P\Lambda_{2}+D^{T}D+E^{T}E\\ & & +R_{ab}\left(\nabla f,\nabla f\right)+R_{zb}\left(\nabla f,\nabla f\right)+R_{\pi }\left(\nabla f,\nabla f\right). \end{array}$$
*All the terms are defined in Notation 1 and Definition 2.*


**Proof.** By Definition 4 and Formulae (Equation 39) and (Equation 40), we have
$$\begin{array}{ccc} & & \Gamma_{2,L}\left(f,f\right)+{\tilde{\Gamma }}_{2,L}^{z,\pi }\left(f,f\right)\\ & = & \Gamma_{2,L}\left(f,f\right)+\Gamma_{2,L}^{z}\left(f,f\right)+{\mathrm{div}}_{z}^{\pi }\left(\Gamma_{\nabla (aa^{T})}f,f\right)-{\mathrm{div}}_{a}^{\pi }\left(\Gamma_{\nabla (zz^{T})}f,f\right). \end{array}$$We compute the above terms explicitly in the following four steps.
**Step 1:**

$$\begin{array}{ccc} \Gamma_{2,L}\left(f,f\right) & = & \frac{1}{2}\left(L\Gamma_{1,L}\left(f,f\right)-2\Gamma_{1,L}\left(Lf,f\right)\right)\\ & = & \frac{1}{2}\Delta_{a}\Gamma_{1}\left(f,f\right)-\frac{1}{2}A\nabla \Gamma_{1}\left(f,f\right)+\frac{1}{2}b\nabla \Gamma_{1}\left(f,f\right)\\ & & -\Gamma_{1}\left(\left(\Delta_{a}-A\nabla +b\nabla \right)f,f\right)\\ & = & \Gamma_{2,\tilde{L}}\left(f,f\right)+\left[\frac{1}{2}b\nabla \Gamma_{1}\left(f,f\right)-\Gamma_{1}\left(b\nabla f,f\right)\right]. \end{array}$$

The term $\Gamma_{2,\tilde{L}}\left(f,f\right)$ follows from Lemma 11. We are left with the other two terms:
$$\begin{array}{ccc} \frac{1}{2}b\nabla \Gamma_{1}\left(f,f\right) & = & \frac{1}{2}\sum_{\hat{k}=1}^{n+m}b_{\hat{k}}\frac{\partial }{\partial x_{\hat{k}}}\left({\langle a^{T}\nabla f,a^{T}\nabla f\rangle }_{R^{n}}\right)\\ & = & \sum_{\hat{k},\hat{i}=1}^{n+m}\sum_{i=1}^{n}\left(b_{\hat{k}}\frac{\partial a_{i\hat{i}}^{T}}{\partial x_{\hat{k}}}\frac{\partial f}{\partial x_{\hat{i}}}+b_{\hat{k}}a_{i\hat{i}}^{T}\frac{\partial^{2}f}{\partial x_{\hat{k}}\partial x_{\hat{i}}}\right){\left(a^{T}\nabla f\right)}_{i}, \end{array}$$
and
$$\begin{array}{ccc} -\Gamma_{1}\left(b\nabla f,f\right) & = & -{\langle a^{T}\nabla \left(b\nabla f\right),a^{T}\nabla f\rangle }_{R^{n}}\\ & = & -\sum_{i=1}^{n}\sum_{\hat{k},\hat{i}=1}^{n+m}\left(a_{i\hat{i}}^{T}\frac{\partial b_{\hat{k}}}{\partial x_{\hat{i}}}\frac{\partial f}{\partial x_{\hat{k}}}+a_{i\hat{i}}^{T}b_{\hat{k}}\frac{\partial^{2}f}{\partial x_{\hat{i}}\partial x_{\hat{k}}}\right){\left(a^{T}\nabla f\right)}_{i}. \end{array}$$
**Step 2:**

$$\begin{array}{ccc} \Gamma_{2,L}^{z}\left(f,f\right) & = & \frac{1}{2}\left(L\Gamma_{1}^{z}\left(f,f\right)-2\Gamma_{1}^{z}\left(Lf,f\right)\right)\\ & = & \frac{1}{2}\Delta_{a}\Gamma_{1}^{z}\left(f,f\right)-\frac{1}{2}A\nabla \Gamma_{1}^{z}\left(f,f\right)+\frac{1}{2}b\nabla \Gamma_{1}^{z}\left(f,f\right)\\ & & -\Gamma_{1}^{z}\left(\left(\Delta_{a}-A\nabla +b\nabla \right)f,f\right)\\ & = & \Gamma_{2,\tilde{L}}^{z}\left(f,f\right)+\left[\frac{1}{2}b\nabla \Gamma_{1}^{z}\left(f,f\right)-\Gamma_{1}^{z}\left(b\nabla f,f\right)\right]. \end{array}$$

The term $\Gamma_{2,\tilde{L}}^{z}\left(f,f\right)$ follows from Lemma 12. We are left to compute the last two terms:
$$\begin{array}{ccc} \frac{1}{2}b\nabla \Gamma_{1}^{z}\left(f,f\right) & = & -\frac{1}{2}\sum_{\hat{k}=1}^{n+m}b_{\hat{k}}\frac{\partial }{\partial x_{\hat{k}}}\left({\langle z^{T}\nabla f,z^{T}\nabla f\rangle }_{R^{m}}\right)\\ & = & \sum_{\hat{k},\hat{i}=1}^{n+m}\sum_{i=1}^{m}\left(b_{\hat{k}}\frac{\partial z_{i\hat{i}}^{T}}{\partial x_{\hat{k}}}\frac{\partial f}{\partial x_{\hat{i}}}+b_{\hat{k}}z_{i\hat{i}}^{T}\frac{\partial^{2}f}{\partial x_{\hat{k}}\partial x_{\hat{i}}}\right){\left(z^{T}\nabla f\right)}_{i}, \end{array}$$
and
$$\begin{array}{ccc} -\Gamma_{1}^{z}\left(b\nabla f,f\right) & = & -{\langle z^{T}\nabla \left(b\nabla f\right),z^{T}\nabla f\rangle }_{R^{n}}\\ & = & -\sum_{i=1}^{m}\sum_{\hat{k},\hat{i}=1}^{n+m}\left(z_{i\hat{i}}^{T}\frac{\partial b_{\hat{k}}}{\partial x_{\hat{i}}}\frac{\partial f}{\partial x_{\hat{k}}}+z_{i\hat{i}}^{T}b_{\hat{k}}\frac{\partial^{2}f}{\partial x_{\hat{i}}\partial x_{\hat{k}}}\right){\left(z^{T}\nabla f\right)}_{i}. \end{array}$$**Step 3:** Following Lemma 10, which will be proven shortly in the next section, we have
$$\begin{array}{ccc} {\mathrm{div}}_{z}^{\pi }\left(\Gamma_{\nabla (aa^{T})}f,f\right)-{\mathrm{div}}_{a}^{\pi }\left(\Gamma_{\nabla (zz^{T})}f,f\right) & = & R^{\pi }\left(f,f\right)+2G^{T}X, \end{array}$$
where *X*, *G* are defined in Notation 1 and $R^{\pi }$ is defined in Definition 2.**Step 4:** Combining the above terms $\Gamma_{2,\tilde{L}}\left(f,f\right)$ in Lemma 11, $\Gamma_{2,\tilde{L}}^{z}\left(f,f\right)$ in Lemma 12, and $R^{\pi }\left(f,f\right)+2G^{T}X$, we have
$$\begin{array}{ccc} & & \Gamma_{2,\tilde{L}}\left(f,f\right)+\Gamma_{2,\tilde{L}}^{z}\left(f,f\right)+R^{\pi }\left(f,f\right)+2G^{T}X\\ & = & X^{T}P^{T}PX+2E^{T}PX+2F^{T}X+E^{T}E\\ & & +X^{T}Q^{T}QX+2D^{T}QX+2C^{T}X+D^{T}D+2G^{T}X\\ & & +R_{a}\left(\nabla f,\nabla f\right)+R_{z}\left(\nabla f,\nabla f\right)+R_{\pi }\left(\nabla f,\nabla f\right)\\ & = & X^{T}\left[P^{T}P+Q^{T}Q\right]X+2\left[G^{T}+F^{T}+C^{T}\right]X+2\left[E^{T}P+D^{T}Q\right]X+D^{T}D+E^{T}E\\ & & +R_{a}\left(\nabla f,\nabla f\right)+R_{z}\left(\nabla f,\nabla f\right)+R_{\pi }\left(\nabla f,\nabla f\right). \end{array}$$Assuming that Assumption 1 is satisfied, we obtain
$$\begin{array}{ccc} & & \Gamma_{2,\tilde{L}}\left(f,f\right)+\Gamma_{2,\tilde{L}}^{z}\left(f,f\right)+R^{\pi }\left(f,f\right)+2G^{T}X\\ & = & {\left[X+\Lambda_{1}\right]}^{T}Q^{T}Q\left[X+\Lambda_{1}\right]+{\left[X+\Lambda_{2}\right]}^{T}P^{T}P\left[X+\Lambda_{2}\right]\\ & & +R_{a}\left(\nabla f,\nabla f\right)+R_{z}\left(\nabla f,\nabla f\right)+R^{\pi }\left(\nabla f,\nabla f\right)\\ & & -\Lambda_{1}^{T}Q^{T}Q\Lambda_{1}-\Lambda_{2}^{T}P^{T}P\Lambda_{2}+D^{T}D+E^{T}E. \end{array}$$Adding the drift terms from **Step 1** and **Step 2**, we obtain $R_{ab}$ and $R_{zb}$, which finishes the proof. □

### 5.1. Proof of Lemma 10

**Lemma** **13.**

(43)
$$\begin{array}{ccc} {\mathrm{div}}_{z}^{\pi }\left(\Gamma_{\nabla (aa^{T})}f,f\right)-{\mathrm{div}}_{a}^{\pi }\left(\Gamma_{\nabla (zz^{T})}f,f\right) & = & R^{\pi }\left(f,f\right)+2G^{T}X, \end{array}$$

*where X, G are defined in Notation 1 and $R^{\pi }$ is defined in Definition 2.*


**Proof.** For the first term in the above lemma, we have
$$\begin{array}{ccc} {\mathrm{div}}_{z}^{\pi }\left(\Gamma_{\nabla (aa^{T})}f,f\right) & = & \frac{\nabla \cdot (zz^{T}\pi \Gamma_{\nabla (aa^{T)}}\left(f,f\right))}{\pi }\\ & = & \sum_{k^{\prime }=1}^{n+m}\frac{1}{\pi }\frac{\partial }{\partial x_{k^{\prime }}}\left[\sum_{k=1}^{m}z_{k^{\prime }k}\left(\pi \sum_{\hat{k}=1}^{n+m}z_{k\hat{k}}^{T}{\left(\Gamma_{\nabla (aa^{T)}}\left(f,f\right))\right)}_{\hat{k}}\right)\right]\\ & = & \sum_{k^{\prime }=1}^{n+m}\sum_{k=1}^{m}\left[\frac{\partial }{\partial x_{k^{\prime }}}z_{k^{\prime }k}\left(\sum_{\hat{k}=1}^{n+m}z_{k\hat{k}}^{T}{\left(\Gamma_{\nabla (aa^{T)}}\left(f,f\right))\right)}_{\hat{k}}\right)\right.\\ & & \left.\;\;+z_{k^{\prime }k}\frac{\partial }{\partial x_{k^{\prime }}}\left(\sum_{\hat{k}=1}^{n+m}z_{k\hat{k}}^{T}{\left(\Gamma_{\nabla (aa^{T)}}\left(f,f\right))\right)}_{\hat{k}}\right)\right] \end{array}$$
$$\begin{array}{ccc} & & +\sum_{k^{\prime }=1}^{n+m}\sum_{k=1}^{m}\frac{\partial }{\partial x_{k^{\prime }}}\log\pi \left[z_{k^{\prime }k}\sum_{\hat{k}=1}^{n+m}z_{k\hat{k}}^{T}{\left(\Gamma_{\nabla (aa^{T)}}\left(f,f\right))\right)}_{\hat{k}}\right]\\ & = & \sum_{k^{\prime }=1}^{n+m}\sum_{k=1}^{m}\left[\frac{\partial }{\partial x_{k^{\prime }}}z_{kk^{\prime }}^{T}\left(\sum_{\hat{k}=1}^{n+m}z_{k\hat{k}}^{T}{\left(\Gamma_{\nabla (aa^{T)}}\left(f,f\right))\right)}_{\hat{k}}\right)\right.\\ & & \left.\;\;+z_{kk^{\prime }}^{T}\frac{\partial }{\partial x_{k^{\prime }}}\left(\sum_{\hat{k}=1}^{n+m}z_{k\hat{k}}^{T}{\left(\Gamma_{\nabla (aa^{T)}}\left(f,f\right))\right)}_{\hat{k}}\right)\right]\\ & & +\sum_{k=1}^{m}{\left(z^{T}\nabla \log\pi \right)}_{k}\left[\sum_{\hat{k}=1}^{n+m}z_{k\hat{k}}^{T}{\left(\Gamma_{\nabla (aa^{T)}}\left(f,f\right))\right)}_{\hat{k}}\right], \end{array}$$
where $\Gamma_{\nabla (aa^{T)}}{\left(f,f\right))}_{\hat{k}}$ is defined in (Equation 42). Plugging in (Equation 42), we further obtain
(44)$$\begin{array}{ccc} & & {\mathrm{div}}_{z}^{\pi }\left(\Gamma_{\nabla (aa^{T})}f,f\right)\\ & = & \sum_{k^{\prime }=1}^{n+m}\sum_{k=1}^{m}\left[\frac{\partial }{\partial x_{k^{\prime }}}z_{kk^{\prime }}^{T}\left(\sum_{\hat{k}=1}^{n+m}z_{k\hat{k}}^{T}\left(2\sum_{i=1}^{n}\sum_{\hat{i},i^{\prime }=1}^{n+m}\frac{\partial }{\partial x_{\hat{k}}}a_{i\hat{i}}^{T}\frac{\partial f}{\partial x_{\hat{i}}}a_{ii^{\prime }}^{T}\frac{\partial f}{\partial x_{i^{\prime }}}\right)\right)\right]\\ & & +\sum_{k^{\prime }=1}^{n+m}\sum_{k=1}^{m}\left[z_{kk^{\prime }}^{T}\frac{\partial }{\partial x_{k^{\prime }}}\left(\sum_{\hat{k}=1}^{n+m}z_{k\hat{k}}^{T}\left(2\sum_{i=1}^{n}\sum_{\hat{i},i^{\prime }=1}^{n+m}\frac{\partial }{\partial x_{\hat{k}}}a_{i\hat{i}}^{T}\frac{\partial f}{\partial x_{\hat{i}}}a_{ii^{\prime }}^{T}\frac{\partial f}{\partial x_{i^{\prime }}}\right)\right)\right]\\ & & +\sum_{k=1}^{m}{\left(z^{T}\nabla \log\pi \right)}_{k}\left[\sum_{\hat{k}=1}^{n+m}z_{k\hat{k}}^{T}\left(2\sum_{i=1}^{n}\sum_{\hat{i},i^{\prime }=1}^{n+m}\frac{\partial }{\partial x_{\hat{k}}}a_{i\hat{i}}^{T}\frac{\partial f}{\partial x_{\hat{i}}}a_{ii^{\prime }}^{T}\frac{\partial f}{\partial x_{i^{\prime }}}\right)\right]\\ & = & 2\sum_{k=1}^{m}\sum_{i=1}^{n}\sum_{k^{\prime },\hat{k},\hat{i},i^{\prime }=1}^{n+m}\left[\frac{\partial }{\partial x_{k^{\prime }}}z_{kk^{\prime }}^{T}z_{k\hat{k}}^{T}\frac{\partial }{\partial x_{\hat{k}}}a_{i\hat{i}}^{T}\frac{\partial f}{\partial x_{\hat{i}}}a_{ii^{\prime }}^{T}\frac{\partial f}{\partial x_{i^{\prime }}}\right]\cdots S_{1}^{z}\\ & & +2\sum_{k=1}^{m}\sum_{i=1}^{n}\sum_{k^{\prime },\hat{k},\hat{i},i^{\prime }=1}^{n+m}\left[z_{kk^{\prime }}^{T}\frac{\partial }{\partial x_{k^{\prime }}}\left(z_{k\hat{k}}^{T}\frac{\partial }{\partial x_{\hat{k}}}a_{i\hat{i}}^{T}\frac{\partial f}{\partial x_{\hat{i}}}a_{ii^{\prime }}^{T}\frac{\partial f}{\partial x_{i^{\prime }}}\right)\right]\cdots S_{2}^{z}\\ & & +2\sum_{k=1}^{m}\sum_{i=1}^{n}\sum_{\hat{k},\hat{i},i^{\prime }=1}^{n+m}{\left(z^{T}\nabla \log\pi \right)}_{k}\left[z_{k\hat{k}}^{T}\frac{\partial }{\partial x_{\hat{k}}}a_{i\hat{i}}^{T}\frac{\partial f}{\partial x_{\hat{i}}}a_{ii^{\prime }}^{T}\frac{\partial f}{\partial x_{i^{\prime }}}\right]\cdots S_{3}^{z}\\ & = & S_{1}^{z}+S_{2}^{z}+S_{3}^{z}. \end{array}$$By further expanding $S_{2}^{z}$, we obtain
$$\begin{array}{ccc} S_{2}^{z} & = & 2\sum_{k=1}^{m}\sum_{i=1}^{n}\sum_{k^{\prime },\hat{k},\hat{i},i^{\prime }=1}^{n+m}\left[z_{kk^{\prime }}^{T}\frac{\partial }{\partial x_{k^{\prime }}}\left(z_{k\hat{k}}^{T}\frac{\partial }{\partial x_{\hat{k}}}a_{i\hat{i}}^{T}\frac{\partial f}{\partial x_{\hat{i}}}a_{ii^{\prime }}^{T}\frac{\partial f}{\partial x_{i^{\prime }}}\right)\right]\\ & = & 2\sum_{k=1}^{m}\sum_{i=1}^{n}\sum_{k^{\prime },\hat{k},\hat{i},i^{\prime }=1}^{n+m}\left[z_{kk^{\prime }}^{T}\frac{\partial }{\partial x_{k^{\prime }}}z_{k\hat{k}}^{T}\frac{\partial }{\partial x_{\hat{k}}}a_{i\hat{i}}^{T}\frac{\partial f}{\partial x_{\hat{i}}}a_{ii^{\prime }}^{T}\frac{\partial f}{\partial x_{i^{\prime }}}\right.\\ & & \;\;\;\;\;\;\;\;+z_{kk^{\prime }}^{T}z_{k\hat{k}}^{T}\frac{\partial^{2}}{\partial x_{k^{\prime }}\partial x_{\hat{k}}}a_{i\hat{i}}^{T}\frac{\partial f}{\partial x_{\hat{i}}}a_{ii^{\prime }}^{T}\frac{\partial f}{\partial x_{i^{\prime }}}\\ & & \;\;\;\;\;\;\;\;+z_{kk^{\prime }}^{T}z_{k\hat{k}}^{T}\frac{\partial }{\partial x_{\hat{k}}}a_{i\hat{i}}^{T}\frac{\partial^{2}f}{\partial x_{k^{\prime }}\partial x_{\hat{i}}}a_{ii^{\prime }}^{T}\frac{\partial f}{\partial x_{i^{\prime }}}\\ & & \;\;\;\;\;\;\;\;+z_{kk^{\prime }}^{T}z_{k\hat{k}}^{T}\frac{\partial }{\partial x_{\hat{k}}}a_{i\hat{i}}^{T}\frac{\partial f}{\partial x_{\hat{i}}}a_{ii^{\prime }}^{T}\frac{\partial^{2}f}{\partial x_{k^{\prime }}\partial x_{i^{\prime }}}\\ & & \left.\;\;\;\;\;\;\;\;+z_{kk^{\prime }}^{T}z_{k\hat{k}}^{T}\frac{\partial }{\partial x_{\hat{k}}}a_{i\hat{i}}^{T}\frac{\partial f}{\partial x_{\hat{i}}}\frac{\partial }{\partial x_{k^{\prime }}}a_{ii^{\prime }}^{T}\frac{\partial f}{\partial x_{i^{\prime }}}\right]. \end{array}$$Similarly, we obtain
(45)$$\begin{array}{ccc} & & {\mathrm{div}}_{a}^{\pi }\left(\Gamma_{\nabla (zz^{T})}f,f\right)\\ & = & \sum_{l^{\prime }=1}^{n+m}\sum_{l=1}^{n}\left[\frac{\partial }{\partial x_{l^{\prime }}}a_{ll^{\prime }}^{T}\left(\sum_{\hat{l}=1}^{n+m}a_{l\hat{l}}^{T}{\left(\Gamma_{\nabla (zz^{T)}}\left(f,f\right))\right)}_{\hat{l}}\right)\right.\\ & & \left.\;\;+a_{ll^{\prime }}^{T}\frac{\partial }{\partial x_{l^{\prime }}}\left(\sum_{\hat{l}=1}^{n+m}a_{l\hat{l}}^{T}{\left(\Gamma_{\nabla (zz^{T)}}\left(f,f\right))\right)}_{\hat{l}}\right)\right]\\ & & +\sum_{l=1}^{n}{\left(a^{T}\nabla \log\pi \right)}_{l}\left[\sum_{\hat{l}=1}^{n+m}{\left(a_{l\hat{l}}^{T}\Gamma_{\nabla (zz^{T)}}\left(f,f\right))\right)}_{\hat{l}}\right]\\ & = & 2\sum_{j=1}^{m}\sum_{l=1}^{n}\sum_{l^{\prime },\hat{l},\hat{j},j^{\prime }=1}^{n+m}\left[\frac{\partial }{\partial x_{l^{\prime }}}a_{ll^{\prime }}^{T}a_{l\hat{l}}^{T}\frac{\partial }{\partial x_{\hat{l}}}z_{j\hat{j}}^{T}\frac{\partial f}{\partial x_{\hat{j}}}z_{jj^{\prime }}^{T}\frac{\partial f}{\partial x_{j^{\prime }}}\right]\cdots S_{1}^{a}\\ & & \;\;\;\;+2\sum_{j=1}^{m}\sum_{l=1}^{n}\sum_{l^{\prime },\hat{l},\hat{j},j^{\prime }=1}^{n+m}\left[a_{ll^{\prime }}^{T}\frac{\partial }{\partial x_{l^{\prime }}}\left(a_{l\hat{l}}^{T}\frac{\partial }{\partial x_{\hat{l}}}z_{j\hat{j}}^{T}\frac{\partial f}{\partial x_{\hat{j}}}z_{jj^{\prime }}^{T}\frac{\partial f}{\partial x_{j^{\prime }}}\right)\right]\cdots S_{2}^{a}\\ & & +2\sum_{j=1}^{m}\sum_{l=1}^{n}\sum_{\hat{l},\hat{j},j^{\prime }=1}^{n+m}{\left(a^{T}\nabla \log\pi \right)}_{l}\left[a_{l\hat{l}}^{T}\frac{\partial }{\partial x_{\hat{l}}}z_{j\hat{j}}^{T}\frac{\partial f}{\partial x_{\hat{j}}}z_{jj^{\prime }}^{T}\frac{\partial f}{\partial x_{j^{\prime }}}\right]\cdots S_{3}^{a} \end{array}$$
(46)$$\begin{array}{cc} = & S_{1}^{a}+S_{2}^{a}+S_{3}^{a}, \end{array}$$
where we also obtain
$$\begin{array}{ccc} S_{2}^{a} & = & 2\sum_{j=1}^{m}\sum_{l=1}^{n}\sum_{l^{\prime },\hat{l},\hat{j},j^{\prime }=1}^{n+m}\left[a_{ll^{\prime }}^{T}\frac{\partial }{\partial x_{l^{\prime }}}\left(a_{l\hat{l}}^{T}\frac{\partial }{\partial x_{\hat{l}}}z_{j\hat{j}}^{T}\frac{\partial f}{\partial x_{\hat{j}}}z_{jj^{\prime }}^{T}\frac{\partial f}{\partial x_{j^{\prime }}}\right)\right]\\ & = & 2\sum_{j=1}^{m}\sum_{l=1}^{n}\sum_{l^{\prime },\hat{l},\hat{j},j^{\prime }=1}^{n+m}\left[a_{ll^{\prime }}^{T}\frac{\partial }{\partial x_{l^{\prime }}}a_{l\hat{l}}^{T}\frac{\partial }{\partial x_{\hat{l}}}z_{j\hat{j}}^{T}\frac{\partial f}{\partial x_{\hat{j}}}z_{jj^{\prime }}^{T}\frac{\partial f}{\partial x_{j^{\prime }}}\right.\\ & & \;\;\;\;\;\;\;\;+a_{ll^{\prime }}^{T}a_{l\hat{l}}^{T}\frac{\partial^{2}}{\partial x_{l^{\prime }}\partial x_{\hat{l}}}z_{j\hat{j}}^{T}\frac{\partial f}{\partial x_{\hat{j}}}z_{jj^{\prime }}^{T}\frac{\partial f}{\partial x_{j^{\prime }}}\\ & & \;\;\;\;\;\;\;\;+a_{ll^{\prime }}^{T}a_{l\hat{l}}^{T}\frac{\partial }{\partial x_{\hat{l}}}z_{j\hat{j}}^{T}\frac{\partial^{2}f}{\partial x_{l^{\prime }}\partial x_{\hat{j}}}z_{jj^{\prime }}^{T}\frac{\partial f}{\partial x_{j^{\prime }}}\\ & & \;\;\;\;\;\;\;\;+a_{ll^{\prime }}^{T}a_{l\hat{l}}^{T}\frac{\partial }{\partial x_{\hat{l}}}z_{j\hat{j}}^{T}\frac{\partial f}{\partial x_{\hat{j}}}z_{jj^{\prime }}^{T}\frac{\partial^{2}f}{\partial x_{l^{\prime }}\partial x_{j^{\prime }}}\\ & & \left.\;\;\;\;\;\;\;\;+a_{ll^{\prime }}^{T}a_{l\hat{l}}^{T}\frac{\partial }{\partial x_{\hat{l}}}z_{j\hat{j}}^{T}\frac{\partial f}{\partial x_{\hat{j}}}\frac{\partial }{\partial x_{l^{\prime }}}z_{jj^{\prime }}^{T}\frac{\partial f}{\partial x_{j^{\prime }}}\right]. \end{array}$$Combining all the terms above, we have
$$\begin{array}{c} {\mathrm{div}}_{z}^{\pi }\left(\Gamma_{\nabla (aa^{T})}f,f\right)-{\mathrm{div}}_{a}^{\pi }\left(\Gamma_{\nabla (zz^{T})}f,f\right)=S_{1}^{z}+S_{2}^{z}+S_{3}^{z}-\left(S_{1}^{a}+S_{2}^{a}+S_{3}^{a}\right). \end{array}$$By direct computations, we separate the above terms into two groups based on “∂f∂f” and “$\partial^{2}f\partial f$”. We denote $R^{\pi }\left(f,f\right)$ as the sum of all “∂f∂f” terms and denote $2G^{T}X$ as the sum of all “$\partial^{2}f\partial f$” terms. Switching indices for the terms in $2G^{T}X$ to match $\frac{\partial^{2}f}{\partial x_{\hat{i}}\partial x_{\hat{j}}}$, we obtain the following:
$$\begin{array}{ccc} & & 2G^{T}X\\ & = & 2\sum_{k=1}^{m}\sum_{i=1}^{n}\sum_{k^{\prime },\hat{k},\hat{i},i^{\prime }=1}^{n+m}\left[z_{kk^{\prime }}^{T}z_{k\hat{k}}^{T}\frac{\partial }{\partial x_{\hat{k}}}a_{i\hat{i}}^{T}\frac{\partial^{2}f}{\partial x_{k^{\prime }}\partial x_{\hat{i}}}a_{ii^{\prime }}^{T}\frac{\partial f}{\partial x_{i^{\prime }}}+z_{kk^{\prime }}^{T}z_{k\hat{k}}^{T}\frac{\partial }{\partial x_{\hat{k}}}a_{i\hat{i}}^{T}\frac{\partial f}{\partial x_{\hat{i}}}a_{ii^{\prime }}^{T}\frac{\partial^{2}f}{\partial x_{k^{\prime }}\partial x_{i^{\prime }}}\right]\\ & & -2\sum_{j=1}^{m}\sum_{l=1}^{n}\sum_{l^{\prime },\hat{l},\hat{j},j^{\prime }=1}^{n+m}\left[a_{ll^{\prime }}^{T}a_{l\hat{l}}^{T}\frac{\partial }{\partial x_{\hat{l}}}z_{j\hat{j}}^{T}\frac{\partial^{2}f}{\partial x_{l^{\prime }}\partial x_{\hat{j}}}z_{jj^{\prime }}^{T}\frac{\partial f}{\partial x_{j^{\prime }}}+a_{ll^{\prime }}^{T}a_{l\hat{l}}^{T}\frac{\partial }{\partial x_{\hat{l}}}z_{j\hat{j}}^{T}\frac{\partial f}{\partial x_{\hat{j}}}z_{jj^{\prime }}^{T}\frac{\partial^{2}f}{\partial x_{l^{\prime }}\partial x_{j^{\prime }}}\right]\\ & = & 2\sum_{j=1}^{m}\sum_{i=1}^{n}\sum_{j^{\prime },\hat{j},\hat{i},i^{\prime }=1}^{n+m}\left[z_{jj^{\prime }}^{T}z_{j\hat{j}}^{T}\frac{\partial }{\partial x_{\hat{j}}}a_{i\hat{i}}^{T}\frac{\partial^{2}f}{\partial x_{j^{\prime }}\partial x_{\hat{i}}}a_{ii^{\prime }}^{T}\frac{\partial f}{\partial x_{i^{\prime }}}+z_{jj^{\prime }}^{T}z_{j\hat{j}}^{T}\frac{\partial }{\partial x_{\hat{j}}}a_{i\hat{i}}^{T}\frac{\partial f}{\partial x_{\hat{i}}}a_{ii^{\prime }}^{T}\frac{\partial^{2}f}{\partial x_{j^{\prime }}\partial x_{i^{\prime }}}\right]\\ & & -2\sum_{j=1}^{m}\sum_{i=1}^{n}\sum_{i^{\prime },\hat{i},\hat{j},j^{\prime }=1}^{n+m}\left[a_{ii^{\prime }}^{T}a_{i\hat{i}}^{T}\frac{\partial }{\partial x_{\hat{i}}}z_{j\hat{j}}^{T}\frac{\partial^{2}f}{\partial x_{i^{\prime }}\partial x_{\hat{j}}}z_{jj^{\prime }}^{T}\frac{\partial f}{\partial x_{j^{\prime }}}+a_{ii^{\prime }}^{T}a_{i\hat{i}}^{T}\frac{\partial }{\partial x_{\hat{i}}}z_{j\hat{j}}^{T}\frac{\partial f}{\partial x_{\hat{j}}}z_{jj^{\prime }}^{T}\frac{\partial^{2}f}{\partial x_{i^{\prime }}\partial x_{j^{\prime }}}\right]\\ & = & 2\sum_{j=1}^{m}\sum_{i=1}^{n}\sum_{j^{\prime },\hat{j},\hat{i},i^{\prime }=1}^{n+m}\left[z_{j\hat{j}}^{T}z_{jj^{\prime }}^{T}\frac{\partial }{\partial x_{j^{\prime }}}a_{i\hat{i}}^{T}\frac{\partial^{2}f}{\partial x_{\hat{j}}\partial x_{\hat{i}}}a_{ii^{\prime }}^{T}\frac{\partial f}{\partial x_{i^{\prime }}}+z_{j\hat{j}}^{T}z_{jj^{\prime }}^{T}\frac{\partial }{\partial x_{j^{\prime }}}a_{ii^{\prime }}^{T}\frac{\partial f}{\partial x_{i^{\prime }}}a_{i\hat{i}}^{T}\frac{\partial^{2}f}{\partial x_{\hat{j}}\partial x_{\hat{i}}}\right]\\ & & -2\sum_{j=1}^{m}\sum_{i=1}^{n}\sum_{i^{\prime },\hat{i},\hat{j},j^{\prime }=1}^{n+m}\left[a_{i\hat{i}}^{T}a_{ii^{\prime }}^{T}\frac{\partial }{\partial x_{i^{\prime }}}z_{j\hat{j}}^{T}\frac{\partial^{2}f}{\partial x_{\hat{i}}\partial x_{\hat{j}}}z_{jj^{\prime }}^{T}\frac{\partial f}{\partial x_{j^{\prime }}}+a_{i\hat{i}}^{T}a_{ii^{\prime }}^{T}\frac{\partial }{\partial x_{i^{\prime }}}z_{jj^{\prime }}^{T}\frac{\partial f}{\partial x_{j^{\prime }}}z_{j\hat{j}}^{T}\frac{\partial^{2}f}{\partial x_{\hat{i}}\partial x_{\hat{j}}}\right]\\ & = & 2\sum_{\hat{i},\hat{j}=1}^{n+m}\frac{\partial^{2}f}{\partial x_{\hat{i}}\partial x_{\hat{j}}}\left[\sum_{i=1}^{n}\sum_{j=1}^{m}\sum_{j^{\prime },\hat{j},i^{\prime },\hat{i}=1}^{n+m}\left[\left(z_{j\hat{j}}^{T}z_{jj^{\prime }}^{T}\frac{\partial }{\partial x_{j^{\prime }}}a_{i\hat{i}}^{T}a_{ii^{\prime }}^{T}\frac{\partial f}{\partial x_{i^{\prime }}}+z_{j\hat{j}}^{T}z_{jj^{\prime }}^{T}\frac{\partial }{\partial x_{j^{\prime }}}a_{ii^{\prime }}^{T}\frac{\partial f}{\partial x_{i^{\prime }}}a_{i\hat{i}}^{T}\right)\right.\right.\\ & & \left.\left.\;\;\;\;\;\;\;\;\;\;\;-\left(a_{i\hat{i}}^{T}a_{ii^{\prime }}^{T}\frac{\partial }{\partial x_{i^{\prime }}}z_{j\hat{j}}^{T}z_{jj^{\prime }}^{T}\frac{\partial f}{\partial x_{j^{\prime }}}+a_{i\hat{i}}^{T}a_{ii^{\prime }}^{T}\frac{\partial }{\partial x_{i^{\prime }}}z_{jj^{\prime }}^{T}\frac{\partial f}{\partial x_{j^{\prime }}}z_{j\hat{j}}^{T}\right)\right]\right]. \end{array}$$The first equality follows from the quantities we obtained previously, the second equality from switching “k” to “j” and “l” to “i”, and the third equality from switching between $“i^{\prime }$” and $“\hat{i}$”, $“j^{\prime }$” and $“\hat{j}$”. Thus, the proof is completed. □

### 5.2. Proof of Lemma 11

From now on, we keep the following notation: $a^{T}\nabla f=\sum_{i=1}^{n}\sum_{\hat{i}=1}^{n+m}a_{i\hat{i}}^{T}\frac{\partial }{\partial x_{\hat{i}}}f$. Furthermore, we fixed the notation for $a,a^{T}$ with relation $a_{\hat{i}i}=a_{i\hat{i}}^{T}$ for i=1,⋯,n and $\hat{i}=1,\cdots ,n+m.$ Here, we denote $a_{i\hat{i}}^{T}:={\left(a^{T}\right)}_{i\hat{i}}.$ Recall that we define
$$\begin{array}{c} \Gamma_{2,\tilde{L}}\left(f,f\right)=\frac{1}{2}\left(\tilde{L}\Gamma_{1}\left(f,f\right)-2\Gamma_{1}\left(\tilde{L}f,f\right)\right). \end{array}$$

Next, we are ready to prove the following lemma.

**Lemma** **14.**

$$\begin{array}{ccc} & & \Gamma_{2,\tilde{L}}\left(f,f\right)=X^{T}Q^{T}QX+2D^{T}QX+2C^{T}X+D^{T}D+R_{a}\left(\nabla f,\nabla f\right), \end{array}$$

*where Q,X,C,D are introduced in Notation 1 and Ra is defined in Definition 2.*


**Proof.** We plug in the operator $\tilde{L}$ into our definition for $\Gamma_{2}$:
$$\begin{array}{ccc} \Gamma_{2,\tilde{L}}\left(f,f\right) & = & \frac{1}{2}\Delta_{a}\Gamma_{1}\left(f,f\right)-\frac{1}{2}A\nabla \Gamma_{1}\left(f,f\right)-\Gamma_{1}\left(\left(\Delta_{a}-A\nabla \right)f,f\right)\\ & = & \frac{1}{2}\Delta_{a}\Gamma_{1}\left(f,f\right)-\Gamma_{1}\left(\Delta_{a}f,f\right)-\frac{1}{2}A\nabla \Gamma_{1}\left(f,f\right)+\Gamma_{1}\left(A\nabla f,f\right). \end{array}$$Now, we compute the last two terms of the above equation. With A=a⊗∇a, we obtain
$$\begin{array}{ccc} -\frac{1}{2}A\nabla \Gamma_{1}\left(f,f\right) & = & -\frac{1}{2}\sum_{\hat{k}=1}^{n+m}A_{\hat{k}}\nabla_{\frac{\partial }{\partial x_{\hat{k}}}}{\langle a^{T}\nabla f,a^{T}\nabla f\rangle }_{R^{n}}\\ & = & -\sum_{\hat{k}=1}^{n+m}\langle A_{\hat{k}}\left(\nabla_{\frac{\partial }{\partial x_{\hat{k}}}}a^{T}\right)\nabla f,a^{T}\nabla f\rangle {}_{R^{n}}-\sum_{\hat{k}=1}^{n+m}\langle A_{\hat{k}}a^{T}\left(\nabla_{\frac{\partial }{\partial x_{\hat{k}}}}\nabla f\right),a^{T}\nabla f\rangle {}_{R^{n}}\\ & = & J_{1}+J_{2}, \end{array}$$
and
$$\begin{array}{ccc} \Gamma_{1}\left(A\nabla f,f\right) & = & \langle a^{T}\nabla \left(\sum_{\hat{k}=1}^{n+m}A_{\hat{k}}\nabla_{\frac{\partial }{\partial x_{\hat{k}}}}f\right),a^{T}\nabla f\rangle {}_{R^{n}}\\ & = & {\langle a^{T}\left(\sum_{\hat{k}=1}^{n+m}A_{\hat{k}}\nabla \nabla_{\frac{\partial }{\partial x_{\hat{k}}}}f\right),a^{T}\nabla f\rangle }_{R^{n}}+\langle a^{T}\left(\sum_{\hat{k}=1}^{n+m}\nabla A_{\hat{k}}\nabla_{\frac{\partial }{\partial x_{\hat{k}}}}f\right),a^{T}\nabla f\rangle {}_{R^{n}}\\ & = & J_{3}+J_{4}. \end{array}$$It is easy to see
$$\begin{array}{c} J_{2}+J_{3}=0.\; \end{array}$$We now expand $J_{1}$ and $J_{4}$ into local coordinates:
(47)$$\begin{array}{c} J_{1}=-\sum_{l=1}^{n}{\left(a^{T}\nabla f\right)}_{l}\left(\sum_{l^{\prime },\hat{k}=1}^{n+m}\sum_{k=1}^{n}\sum_{k^{\prime }=1}^{n+m}a_{\hat{k}k}\nabla_{\frac{\partial }{\partial x_{k^{\prime }}}}a_{k^{\prime }k}\nabla_{\frac{\partial }{\partial x_{\hat{k}}}}a_{ll^{\prime }}\nabla_{\frac{\partial }{\partial x_{l^{\prime }}}}f\right), \end{array}$$
and
(48)$$\begin{array}{ccc} J_{4} & = & \sum_{l=1}^{n}{\left(a^{T}\nabla f\right)}_{l}\left(\sum_{l^{\prime }=1}^{n+m}a_{ll^{\prime }}^{T}\left(\sum_{\hat{k}=1}^{n+m}\nabla_{\frac{\partial }{\partial x_{l^{\prime }}}}\left(\sum_{k=1}^{n}\sum_{k^{\prime }=1}^{n+m}a_{\hat{k}k}\nabla_{\frac{\partial }{\partial x_{k^{\prime }}}}a_{k^{\prime }k}\right)\nabla_{\frac{\partial }{\partial x_{\hat{k}}}}f\right)\right)\\ & = & \sum_{l=1}^{n}{\left(a^{T}\nabla f\right)}_{l}\left(\sum_{k=1}^{n}\sum_{l^{\prime }=1}^{n+m}\sum_{\hat{k},k^{\prime }=1}^{n+m}a_{ll^{\prime }}^{T}\nabla_{\frac{\partial }{\partial x_{l^{\prime }}}}a_{\hat{k}k}\nabla_{\frac{\partial }{\partial x_{k^{\prime }}}}a_{k^{\prime }k}\nabla_{\frac{\partial }{\partial x_{\hat{k}}}}f\right)\\ & & +\sum_{l=1}^{n}{\left(a^{T}\nabla f\right)}_{l}\left(\sum_{k=1}^{n}\sum_{l^{\prime }=1}^{n+m}\sum_{\hat{k},k^{\prime }=1}^{n+m}a_{ll^{\prime }}^{T}a_{\hat{k}k}\left(\nabla_{\frac{\partial }{\partial x_{l^{\prime }}}}\nabla_{\frac{\partial }{\partial x_{k^{\prime }}}}a_{k^{\prime }k}\right)\nabla_{\frac{\partial }{\partial x_{\hat{k}}}}f\right). \end{array}$$Applying Lemma 15, which will be proven shortly below, we have
$$\begin{array}{ccc} & & \frac{1}{2}\Delta_{a}\Gamma_{1}\left(f,f\right)-\Gamma_{1}\left(\Delta_{a}f,f\right)\\ & = & \frac{1}{2}(a^{T}\nabla \circ (a^{T}\nabla |a^{T}{\nabla f|}^{2}))-{\langle a^{T}\nabla \left(\left(a^{T}\nabla \right)\circ \left(a^{T}\nabla f\right)\right),a^{T}\nabla f\rangle }_{R^{n}}\\ & & +\sum_{l=1}^{n}{\left(a^{T}\nabla f\right)}_{l}\left(\sum_{\hat{i},\hat{k},l^{\prime }=1}^{n+m}\sum_{k=1}^{n}\left(\frac{\partial }{\partial x_{\hat{i}}}a_{\hat{i}k}a_{k\hat{k}}^{T}\left(\frac{\partial }{\partial x_{\hat{k}}}a_{ll^{\prime }}^{T}\frac{\partial }{\partial x_{l^{\prime }}}f\right)-a_{ll^{\prime }}^{T}\frac{\partial }{\partial x_{\hat{i}}}a_{\hat{i}k}\left(\frac{\partial }{\partial x_{l^{\prime }}}a_{k\hat{k}}^{T}\frac{\partial }{\partial x_{\hat{k}}}f\right)\right)\right)\\ & & -{\langle B_{n\times n}a^{T}\nabla f,a^{T}\nabla f\rangle }_{R^{n}}, \end{array}$$
where
$$\begin{array}{c} {\langle B_{n\times n}a^{T}\nabla f,a^{T}\nabla f\rangle }_{R^{n}}=\sum_{l=1}^{n}{\left(a^{T}\nabla f\right)}_{l}\left(\sum_{k=1}^{n}\sum_{l=1}^{n}\sum_{i=1}^{n+m}\sum_{k^{\prime },j^{\prime }=1}^{n+m}a_{lj^{\prime }}^{T}\frac{\partial^{2}}{\partial x_{i}x_{j}^{\prime }}a_{ik}\left(a_{kk^{\prime }}^{T}\frac{\partial }{\partial x_{k^{\prime }}}f\right)\right). \end{array}$$Thus, combining with (Equation 47) and (Equation 48), we have
$$\begin{array}{ccc} \Gamma_{2,\tilde{L}}\left(f,f\right) & = & \frac{1}{2}\Delta_{a}\Gamma_{1}\left(f,f\right)-\Gamma_{1}\left(\Delta_{a}f,f\right)+J_{1}+J_{4}\\ & = & \frac{1}{2}(a^{T}\nabla \circ (a^{T}\nabla |a^{T}{\nabla f|}^{2}))-{\langle a^{T}\nabla \left[\left(a^{T}\nabla \right)\circ \left(a^{T}\nabla f\right)\right],a^{T}\nabla f\rangle }_{R^{n}}. \end{array}$$
where the last term follows from Lemma 16 below. The proof is thus completed. □

**Lemma** **15.**

(49)
$$\begin{array}{ccc} & & \frac{1}{2}\Delta_{a}\Gamma_{1}\left(f,f\right)-\Gamma_{1}\left(\Delta_{a}f,f\right)\\ & = & \frac{1}{2}(a^{T}\nabla \circ (a^{T}\nabla |a^{T}{\nabla f|}^{2}))-{\langle a^{T}\nabla \left(\left[\left(a^{T}\nabla \right)\circ \left(a^{T}\nabla f\right)\right]\right),a^{T}\nabla f\rangle }_{R^{n}}\\ & & -{\langle B_{n\times n}a^{T}\nabla f,a^{T}\nabla f\rangle }_{R^{n}}+B_{0}. \end{array}$$


*Here, the local representations for $B_{n\times n}$ and $B_{0}$ are given as follows. For l,k=1,⋯,n, we denote*

(50)
$$\begin{array}{ccc} B_{lk} & = & \sum_{j^{\prime }=1}^{n+m}a_{lj^{\prime }}^{T}\sum_{i=1}^{n+m}\frac{\partial^{2}}{\partial x_{i}\partial x_{j^{\prime }}}a_{ik}=\sum_{j^{\prime }=1}^{n+m}a_{lj^{\prime }}^{T}\sum_{i=1}^{n+m}\frac{\partial^{2}}{\partial x_{i}\partial x_{j^{\prime }}}a_{ki}^{T},\\ B_{0} & = & \sum_{l=1}^{n}{\left(a^{T}\nabla f\right)}_{l}\left(\sum_{\hat{i},\hat{k},l^{\prime }=1}^{n+m}\sum_{k=1}^{n}\left(\frac{\partial }{\partial x_{\hat{i}}}a_{\hat{i}k}a_{k\hat{k}}^{T}\left(\frac{\partial }{\partial x_{\hat{k}}}a_{ll^{\prime }}^{T}\frac{\partial }{\partial x_{l^{\prime }}}f\right)\right.\right.\\ & & \;\;\;\left.\left.-a_{ll^{\prime }}^{T}\frac{\partial }{\partial x_{\hat{i}}}a_{\hat{i}k}\left(\frac{\partial }{\partial x_{l^{\prime }}}a_{k\hat{k}}^{T}\frac{\partial }{\partial x_{\hat{k}}}f\right)\right)\right). \end{array}$$


*We introduce the following notation (convention) that, for any function F,*

(51)
$$\begin{array}{c} \left(a^{T}\nabla \right)\circ \left(a^{T}\nabla F\right)=\sum_{i=1}^{n}{\left(a^{T}\nabla \right)}_{i}{\left(a^{T}\nabla F\right)}_{i}=\sum_{i=1}^{n}\sum_{\hat{i},i^{\prime }=1}^{n+m}\left(a_{i\hat{i}}^{T}\frac{\partial }{\partial x_{\hat{i}}}\right)\left(a_{ii^{\prime }}^{T}\frac{\partial F}{\partial x_{i^{\prime }}}\right). \end{array}$$



**Proof** **of** **Lemma** **15.**By our definition above, we have
$$\begin{array}{ccc} \Delta_{a}\Gamma_{1}\left(f,f\right) & = & \nabla \cdot (aa^{T}\nabla {\langle a^{T}\nabla f,a^{T}\nabla f\rangle }_{R^{n}})\\ & = & \nabla \cdot (aF)\\ & = & \sum_{\hat{i}=1}^{n+m}\frac{\partial }{\partial x_{\hat{i}}}\left(\sum_{k=1}^{n}a_{\hat{i}k}F_{k}\right)\\ & = & \sum_{\hat{i}=1}^{n+m}\sum_{k=1}^{n}\left(\frac{\partial }{\partial x_{\hat{i}}}a_{\hat{i}k}F_{k}+a_{\hat{i}k}\frac{\partial }{\partial x_{\hat{i}}}F_{k}\right)\\ & = & \sum_{\hat{i}=1}^{n+m}\sum_{k=1}^{n}\left(\frac{\partial }{\partial x_{\hat{i}}}a_{\hat{i}k}F_{k}\right)+a^{T}\nabla \circ \left(a^{T}\nabla {\left(a^{T}\nabla f\right)}^{2}\right), \end{array}$$
where we denote
$$\begin{array}{ccc} F & = & a^{T}\nabla {\langle a^{T}\nabla f,a^{T}\nabla f\rangle }_{R^{n}}\\ & = & a^{T}\nabla \sum_{l=1}^{n}{\left(\sum_{\hat{l}=1}^{n+m}a_{l\hat{l}}^{T}\frac{\partial }{\partial x_{\hat{l}}}f\right)}^{2}\\ & = & {\left(\sum_{\hat{k}=1}^{n+m}a_{k\hat{k}}^{T}\frac{\partial }{\partial x_{\hat{j}}}\sum_{l=1}^{n}{\left(\sum_{\hat{l}=1}^{n+m}a_{l\hat{l}}^{T}\frac{\partial }{\partial x_{\hat{l}}}f\right)}^{2}\right)}_{k=1,\cdots ,n}={\left(F_{1},F_{2},\cdots ,F_{n}\right)}^{T}. \end{array}$$Therefore, we have
(52)$$\begin{array}{ccc} \Delta_{a}\Gamma_{1}\left(f,f\right) & = & \sum_{\hat{i}=1}^{n+m}\sum_{k=1}^{n}\left(\frac{\partial }{\partial x_{\hat{i}}}a_{\hat{i}k}\left(\sum_{\hat{k}=1}^{n+m}a_{k\hat{k}}^{T}\frac{\partial }{\partial x_{\hat{j}}}\sum_{l=1}^{n}{\left(\sum_{\hat{l}=1}^{n+m}a_{l\hat{l}}^{T}\frac{\partial }{\partial x_{\hat{l}}}f\right)}^{2}\right)\right)\\ & & +a^{T}\nabla \circ \left(a^{T}\nabla {\left(a^{T}\nabla f\right)}^{2}\right)\\ & = & \sum_{k=1}^{n}\sum_{\hat{i}=1}^{n+m}\left(\frac{\partial }{\partial x_{\hat{i}}}a_{\hat{i}k}\left(\sum_{\hat{k}=1}^{n+m}a_{k\hat{k}}^{T}\frac{\partial }{\partial x_{\hat{k}}}{\left(a^{T}\nabla f\right)}^{2}\right)\right)\\ & & +\left(a^{T}\nabla \right)\circ \left(a^{T}\nabla {\left(a^{T}\nabla f\right)}^{2}\right)\\ & = & \nabla a\circ \left(a^{T}\nabla {\left(a^{T}\nabla f\right)}^{2}\right)+\left(a^{T}\nabla \right)\circ \left(a^{T}\nabla {\left(a^{T}\nabla f\right)}^{2}\right). \end{array}$$Next, we compute the following quantity.
$$\begin{array}{ccc} \Gamma_{1}\left(\Delta_{a}f,f\right) & = & {\langle a^{T}\nabla \left(\nabla \cdot \left(aa^{T}\nabla f\right)\right),a^{T}\nabla f\rangle }_{R^{n}}, \end{array}$$
where we have
$$\begin{array}{c} \nabla \cdot \left(aa^{T}\nabla f\right)=\nabla \cdot \left(\sum_{k=1}^{n}\sum_{\hat{k}=1}^{n+m}a_{\hat{i}k}a_{k\hat{k}}^{T}\frac{\partial }{\partial x_{\hat{k}}}f\right)=\sum_{\hat{i}=1}^{n+m}\frac{\partial }{\partial x_{\hat{i}}}\left(\sum_{k=1}^{n}\sum_{\hat{k}=1}^{n+m}a_{\hat{i}k}a_{k\hat{k}}^{T}\frac{\partial }{\partial x_{\hat{k}}}f\right) \end{array}$$
$$\begin{array}{ccc} & = & \sum_{\hat{i},\hat{k}=1}^{n+m}\sum_{k=1}^{n}\frac{\partial }{\partial x_{\hat{i}}}a_{\hat{i}k}\left(a_{k\hat{k}}^{T}\frac{\partial }{\partial x_{\hat{k}}}f\right)+\sum_{\hat{i},\hat{k}=1}^{n+m}\sum_{k=1}^{n}a_{\hat{i}k}\frac{\partial }{\partial x_{\hat{i}}}\left(a_{k\hat{k}}^{T}\frac{\partial }{\partial x_{j}}f\right)\\ & = & \sum_{\hat{i},\hat{k}=1}^{n+m}\sum_{k=1}^{n}\frac{\partial }{\partial x_{\hat{i}}}a_{\hat{i}k}\left(a_{k\hat{k}}^{T}\frac{\partial }{\partial x_{\hat{k}}}f\right)+\left(a^{T}\nabla \right)\circ \left(a^{T}\nabla f\right)\\ & = & \nabla a\circ \left(a^{T}\nabla f\right)+\left(a^{T}\nabla \right)\circ \left(a^{T}\nabla f\right). \end{array}$$We continue with our computation as below:
(53)$$\begin{array}{ccc} & & \Gamma_{1}\left(\Delta_{a}f,f\right)\\ & = & {\langle a^{T}\nabla \left[\sum_{\hat{i},\hat{k}=1}^{n+m}\sum_{k=1}^{n}\frac{\partial }{\partial x_{\hat{i}}}a_{\hat{i}k}\left(a_{k\hat{k}}^{T}\frac{\partial }{\partial x_{\hat{k}}}f\right)+\left(a^{T}\nabla \right)\circ \left(a^{T}\nabla f\right)\right],a^{T}\nabla f\rangle }_{R^{n}}\\ & = & {\langle a^{T}\nabla \left[\sum_{\hat{i},\hat{k}=1}^{n+m}\sum_{k=1}^{n}\frac{\partial }{\partial x_{\hat{i}}}a_{\hat{i}k}\left(a_{k\hat{k}}^{T}\frac{\partial }{\partial x_{\hat{k}}}f\right)\right],a^{T}\nabla f\rangle }_{R^{n}}\\ & & +{\langle a^{T}\nabla \left(\left(a^{T}\nabla \right)\circ \left(a^{T}\nabla f\right)\right),a^{T}\nabla f\rangle }_{R^{n}}\\ & = & {\langle a^{T}\nabla \left(\nabla a\cdot (a^{T}\nabla f)\right),a^{T}\nabla f\rangle }_{R^{n}}+{\langle a^{T}\nabla \left(\left(a^{T}\nabla \right)\circ \left(a^{T}\nabla f\right)\right),a^{T}\nabla f\rangle }_{R^{n}}\\ & = & {\langle \left(a^{T}\nabla \nabla a\cdot \left(a^{T}\nabla f\right)\right),a^{T}\nabla f\rangle }_{R^{n}}+{\langle \left(\nabla a\cdot (a^{T}\nabla \left(a^{T}\nabla f\right))\right),a^{T}\nabla f\rangle }_{R^{n}}\\ & & +{\langle a^{T}\nabla \left(\left(a^{T}\nabla \right)\circ \left(a^{T}\nabla f\right)\right),a^{T}\nabla f\rangle }_{R^{n}}. \end{array}$$From the above, combining (Equation 52) and (Equation 53), we further obtain
$$\begin{array}{ccc} & & \frac{1}{2}\Delta_{a}\Gamma_{1}\left(f,f\right)-\Gamma_{1}\left(\Delta_{a}f,f\right)\\ & = & \frac{1}{2}(a^{T}\nabla \circ (a^{T}\nabla |a^{T}{\nabla f|}^{2}))-{\langle a^{T}\nabla \left(\left(a^{T}\nabla \right)\circ \left(a^{T}\nabla f\right)\right),a^{T}\nabla f\rangle }_{R^{n}} \end{array}$$
$$\begin{array}{ccc} & & +\frac{1}{2}\sum_{\hat{i}=1}^{n+m}\sum_{k=1}^{n}\left(\frac{\partial }{\partial x_{\hat{i}}}a_{\hat{i}k}\left(\sum_{\hat{k}=1}^{n+m}a_{k\hat{k}}^{T}\frac{\partial }{\partial x_{\hat{j}}}\sum_{l=1}^{n}{\left(\sum_{\hat{l}=1}^{n+m}a_{l\hat{l}}^{T}\frac{\partial }{\partial x_{\hat{l}}}f\right)}^{2}\right)\right)\\ & & -{\langle a^{T}\nabla \left[\sum_{\hat{i},\hat{k}=1}^{n+m}\sum_{k=1}^{n}\frac{\partial }{\partial x_{\hat{i}}}a_{\hat{i}k}\left(a_{k\hat{k}}^{T}\frac{\partial }{\partial x_{\hat{k}}}f\right)\right],a^{T}\nabla f\rangle }_{R^{n}}\\ & = & \frac{1}{2}(a^{T}\nabla \circ (a^{T}\nabla |a^{T}{\nabla f|}^{2}))-{\langle a^{T}\nabla \left(\left(a^{T}\nabla \right)\circ \left(a^{T}\nabla f\right)\right),a^{T}\nabla f\rangle }_{R^{n}}\\ & & +\sum_{\hat{i}=1}^{n+m}\sum_{k=1}^{n}\left(\frac{\partial }{\partial x_{\hat{i}}}a_{\hat{i}k}\sum_{\hat{k}=1}^{n+m}a_{k\hat{k}}^{T}\sum_{l=1}^{n}\left(\sum_{\hat{l}=1}^{n+m}a_{l\hat{l}}^{T}\frac{\partial }{\partial x_{\hat{l}}}f\right)\frac{\partial }{\partial x_{\hat{k}}}\left(\sum_{\hat{l}=1}^{n+m}a_{l\hat{l}}^{T}\frac{\partial }{\partial x_{\hat{l}}}f\right)\right)\cdots I\\ & & -\sum_{l=1}^{n}\left(\left(\sum_{\hat{l}=1}^{n+m}a_{l\hat{l}}^{T}\frac{\partial }{\partial x_{\hat{l}}}f\right)\left(\sum_{l^{\prime }=1}^{n+m}a_{ll^{\prime }}^{T}\frac{\partial }{\partial x_{l^{\prime }}}\left[\sum_{\hat{i},\hat{k}=1}^{n+m}\sum_{k=1}^{n}\frac{\partial }{\partial x_{\hat{i}}}a_{\hat{i}k}\left(a_{k\hat{k}}^{T}\frac{\partial }{\partial x_{\hat{k}}}f\right)\right]\right)\right)\cdots \mathrm{II}. \end{array}$$Recall that we denote $a^{T}$ to emphasize the transpose of the matrix *a* and $a_{i\hat{i}}^{T}=a_{\hat{i}i}$:
$$\begin{array}{ccc} I & = & \sum_{\hat{i}=1}^{n+m}\sum_{k=1}^{n}\left(\frac{\partial }{\partial x_{\hat{i}}}a_{\hat{i}k}\sum_{\hat{k}=1}^{n+m}a_{k\hat{k}}^{T}\sum_{l=1}^{n}\left(\sum_{\hat{l}=1}^{n+m}a_{l\hat{l}}^{T}\frac{\partial }{\partial x_{\hat{l}}}f\right)\frac{\partial }{\partial x_{\hat{k}}}\left(\sum_{\hat{l}=1}^{n+m}a_{l\hat{l}}^{T}\frac{\partial }{\partial x_{\hat{l}}}f\right)\right)\\ & = & \sum_{\hat{i}=1}^{n+m}\sum_{k=1}^{n}\left(\frac{\partial }{\partial x_{\hat{i}}}a_{\hat{i}k}\sum_{\hat{k}=1}^{n+m}a_{k\hat{k}}^{T}\sum_{l=1}^{n}\left(\sum_{\hat{l}=1}^{n+m}a_{l\hat{l}}^{T}\frac{\partial }{\partial x_{\hat{l}}}f\right)\left(\sum_{\hat{l}=1}^{n+m}\frac{\partial }{\partial x_{\hat{k}}}a_{l\hat{l}}^{T}\frac{\partial }{\partial x_{\hat{l}}}f\right)\right)\\ & & +\sum_{\hat{i}=1}^{n+m}\sum_{k=1}^{n}\left(\frac{\partial }{\partial x_{\hat{i}}}a_{\hat{i}k}\sum_{\hat{k}=1}^{n+m}a_{k\hat{k}}^{T}\sum_{l=1}^{n}\left(\sum_{\hat{l}=1}^{n+m}a_{l\hat{l}}^{T}\frac{\partial }{\partial x_{\hat{l}}}f\right)\left(\sum_{\hat{l}=1}^{n+m}a_{l\hat{l}}^{T}\frac{\partial }{\partial x_{\hat{k}}}\frac{\partial }{\partial x_{\hat{l}}}f\right)\right)\\ & = & \sum_{l=1}^{n}\left(\sum_{\hat{l}=1}^{n+m}a_{l\hat{l}}^{T}\frac{\partial }{\partial x_{\hat{l}}}f\right)\sum_{\hat{i}=1}^{n+m}\sum_{k=1}^{n}\left(\frac{\partial }{\partial x_{\hat{i}}}a_{\hat{i}k}\sum_{\hat{k}=1}^{n+m}a_{k\hat{k}}^{T}\left(\sum_{l^{\prime }=1}^{n+m}\frac{\partial }{\partial x_{\hat{k}}}a_{ll^{\prime }}^{T}\frac{\partial }{\partial x_{l^{\prime }}}f\right)\right)\\ & & +\sum_{l=1}^{n}\left(\sum_{\hat{l}=1}^{n+m}a_{l\hat{l}}^{T}\frac{\partial }{\partial x_{\hat{l}}}f\right)\sum_{\hat{i}=1}^{n+m}\sum_{k=1}^{n}\left(\frac{\partial }{\partial x_{\hat{i}}}a_{\hat{i}k}\sum_{\hat{k}=1}^{n+m}a_{k\hat{k}}^{T}\left(\sum_{\hat{l}=1}^{n+m}a_{ll^{\prime }}^{T}\frac{\partial }{\partial x_{\hat{k}}}\frac{\partial }{\partial x_{l^{\prime }}}f\right)\right), \end{array}$$
and
$$\begin{array}{ccc} \mathrm{II} & = & \sum_{l=1}^{n}\left(\left(\sum_{\hat{l}=1}^{n+m}a_{l\hat{l}}^{T}\frac{\partial }{\partial x_{\hat{l}}}f\right)\left(\sum_{l^{\prime }=1}^{n+m}a_{ll^{\prime }}^{T}\frac{\partial }{\partial x_{l^{\prime }}}\left[\sum_{\hat{i},\hat{k}=1}^{n+m}\sum_{k=1}^{n}\frac{\partial }{\partial x_{\hat{i}}}a_{\hat{i}k}\left(a_{k\hat{k}}^{T}\frac{\partial }{\partial x_{\hat{k}}}f\right)\right]\right)\right)\\ & = & \sum_{l=1}^{n}\left(\left(\sum_{\hat{l}=1}^{n+m}a_{l\hat{l}}^{T}\frac{\partial }{\partial x_{\hat{l}}}f\right)\left(\sum_{l^{\prime }=1}^{n+m}a_{ll^{\prime }}^{T}\left[\sum_{\hat{i},\hat{k}=1}^{n+m}\sum_{k=1}^{n}\frac{\partial }{\partial x_{\hat{i}}}a_{\hat{i}k}\frac{\partial }{\partial x_{l^{\prime }}}\left(a_{k\hat{k}}^{T}\frac{\partial }{\partial x_{\hat{k}}}f\right)\right]\right)\right)\\ & & +\sum_{l=1}^{n}\left(\left(\sum_{\hat{l}=1}^{n+m}a_{l\hat{l}}^{T}\frac{\partial }{\partial x_{\hat{l}}}f\right)\left(\sum_{l^{\prime }=1}^{n+m}a_{ll^{\prime }}^{T}\left[\sum_{\hat{i},\hat{k}=1}^{n+m}\sum_{k=1}^{n}\frac{\partial^{2}}{\partial x_{\hat{i}}x_{l^{\prime }}}a_{\hat{i}k}\left(a_{k\hat{k}}^{T}\frac{\partial }{\partial x_{\hat{k}}}f\right)\right]\right)\right)\\ & = & \sum_{l=1}^{n}\left(\left(\sum_{\hat{l}=1}^{n+m}a_{l\hat{l}}^{T}\frac{\partial }{\partial x_{\hat{l}}}f\right)\left(\sum_{l^{\prime }=1}^{n+m}a_{ll^{\prime }}^{T}\left[\sum_{\hat{i},\hat{k}=1}^{n+m}\sum_{k=1}^{n}\frac{\partial }{\partial x_{\hat{i}}}a_{\hat{i}k}\left(\frac{\partial }{\partial x_{l^{\prime }}}a_{k\hat{k}}^{T}\frac{\partial }{\partial x_{\hat{k}}}f\right)\right]\right)\right)\\ & & +\sum_{l=1}^{n}\left(\left(\sum_{\hat{l}=1}^{n+m}a_{l\hat{l}}^{T}\frac{\partial }{\partial x_{\hat{l}}}f\right)\left(\sum_{l^{\prime }=1}^{n+m}a_{ll^{\prime }}^{T}\left[\sum_{\hat{i},\hat{k}=1}^{n+m}\sum_{k=1}^{n}\frac{\partial }{\partial x_{\hat{i}}}a_{\hat{i}k}\left(a_{k\hat{k}}^{T}\frac{\partial }{\partial x_{l^{\prime }}}\frac{\partial }{\partial x_{\hat{k}}}f\right)\right]\right)\right)\\ & & +\sum_{l=1}^{n}\left(\left(\sum_{\hat{l}=1}^{n+m}a_{l\hat{l}}^{T}\frac{\partial }{\partial x_{\hat{l}}}f\right)\left(\sum_{l^{\prime }=1}^{n+m}a_{ll^{\prime }}^{T}\left[\sum_{\hat{i},\hat{k}=1}^{n+m}\sum_{k=1}^{n}\frac{\partial^{2}}{\partial x_{\hat{i}}x_{l^{\prime }}}a_{\hat{i}k}\left(a_{k\hat{k}}^{T}\frac{\partial }{\partial x_{\hat{k}}}f\right)\right]\right)\right). \end{array}$$Subtracting the above two terms, we have
$$\begin{array}{ccc} & & I-\mathrm{II}\\ & = & -\sum_{l=1}^{n}\left(\left(\sum_{\hat{l}=1}^{n+m}a_{l\hat{l}}^{T}\frac{\partial }{\partial x_{\hat{l}}}f\right)\left(\sum_{l^{\prime }=1}^{n+m}a_{ll^{\prime }}^{T}\left[\sum_{\hat{i},\hat{k}=1}^{n+m}\sum_{k=1}^{n}\frac{\partial^{2}}{\partial x_{\hat{i}}x_{l^{\prime }}}a_{\hat{i}k}\left(a_{k\hat{k}}^{T}\frac{\partial }{\partial x_{\hat{k}}}f\right)\right]\right)\right)\\ & & +\sum_{l=1}^{n}\left(\sum_{\hat{l}=1}^{n+m}a_{l\hat{l}}^{T}\frac{\partial }{\partial x_{\hat{l}}}f\right)\sum_{\hat{i}=1}^{n+m}\sum_{k=1}^{n}\left(\frac{\partial }{\partial x_{\hat{i}}}a_{\hat{i}k}\sum_{\hat{k}=1}^{n+m}a_{k\hat{k}}^{T}\left(\sum_{l^{\prime }=1}^{n+m}\frac{\partial }{\partial x_{\hat{k}}}a_{ll^{\prime }}^{T}\frac{\partial }{\partial x_{l^{\prime }}}f\right)\right)\\ & & -\sum_{l=1}^{n}\left(\left(\sum_{\hat{l}=1}^{n+m}a_{l\hat{l}}^{T}\frac{\partial }{\partial x_{\hat{l}}}f\right)\left(\sum_{l^{\prime }=1}^{n+m}a_{ll^{\prime }}^{T}\left[\sum_{\hat{i},\hat{k}=1}^{n+m}\sum_{k=1}^{n}\frac{\partial }{\partial x_{\hat{i}}}a_{\hat{i}k}\left(\frac{\partial }{\partial x_{l^{\prime }}}a_{k\hat{k}}^{T}\frac{\partial }{\partial x_{\hat{k}}}f\right)\right]\right)\right)\\ & = & -\sum_{l=1}^{n}\left(\left(\sum_{\hat{l}=1}^{n+m}a_{l\hat{l}}^{T}\frac{\partial }{\partial x_{\hat{l}}}f\right)\left(\sum_{l^{\prime }=1}^{n+m}a_{ll^{\prime }}^{T}\left[\sum_{\hat{i},\hat{k}=1}^{n+m}\sum_{k=1}^{n}\frac{\partial^{2}}{\partial x_{\hat{i}}x_{l^{\prime }}}a_{\hat{i}k}\left(a_{k\hat{k}}^{T}\frac{\partial }{\partial x_{\hat{k}}}f\right)\right]\right)\right)\\ & & +\sum_{l=1}^{n}\left(\sum_{\hat{l}=1}^{n+m}a_{l\hat{l}}^{T}\frac{\partial }{\partial x_{\hat{l}}}f\right)\left(\sum_{\hat{i},\hat{k},l^{\prime }=1}^{n+m}\sum_{k=1}^{n}\left(\frac{\partial }{\partial x_{\hat{i}}}a_{\hat{i}k}a_{k\hat{k}}^{T}\left(\frac{\partial }{\partial x_{\hat{k}}}a_{ll^{\prime }}^{T}\frac{\partial }{\partial x_{l^{\prime }}}f\right)\right.\right.\\ & & \left.\left.\;\;\;\;\;-a_{ll^{\prime }}^{T}\frac{\partial }{\partial x_{\hat{i}}}a_{\hat{i}k}\left(\frac{\partial }{\partial x_{l^{\prime }}}a_{k\hat{k}}^{T}\frac{\partial }{\partial x_{\hat{k}}}f\right)\right)\right)\\ & = & -\sum_{l=1}^{n}\left(\left(\sum_{\hat{l}=1}^{n+m}a_{l\hat{l}}^{T}\frac{\partial }{\partial x_{\hat{l}}}f\right)\left(\sum_{l^{\prime }=1}^{n+m}a_{ll^{\prime }}^{T}\left[\sum_{\hat{i},\hat{k}=1}^{n+m}\sum_{k=1}^{n}\frac{\partial^{2}}{\partial x_{\hat{i}}x_{l^{\prime }}}a_{\hat{i}k}\left(a_{k\hat{k}}^{T}\frac{\partial }{\partial x_{\hat{k}}}f\right)\right]\right)\right)\\ & & +\sum_{l=1}^{n}{\left(a^{T}\nabla f\right)}_{l}\left(\sum_{\hat{i},\hat{k},l^{\prime }=1}^{n+m}\sum_{k=1}^{n}\left(\frac{\partial }{\partial x_{\hat{i}}}a_{\hat{i}k}a_{k\hat{k}}^{T}\left(\frac{\partial }{\partial x_{\hat{k}}}a_{ll^{\prime }}^{T}\frac{\partial }{\partial x_{l^{\prime }}}f\right)\right.\right.\\ & & \left.\left.\;\;\;\;\;-a_{ll^{\prime }}^{T}\frac{\partial }{\partial x_{\hat{i}}}a_{\hat{i}k}\left(\frac{\partial }{\partial x_{l^{\prime }}}a_{k\hat{k}}^{T}\frac{\partial }{\partial x_{\hat{k}}}f\right)\right)\right). \end{array}$$Now, we eventually obtain the the following step:
$$\begin{array}{ccc} & & \frac{1}{2}\Delta_{a}\Gamma_{1}\left(f,f\right)-\Gamma_{1}\left(\Delta_{a}f,f\right)\\ & = & \frac{1}{2}(a^{T}\nabla \circ (a^{T}\nabla |a^{T}{\nabla f|}^{2}))-{\langle a^{T}\nabla \left(\left(a^{T}\nabla \right)\circ \left(a^{T}\nabla f\right)\right),a^{T}\nabla f\rangle }_{R^{n}}\\ & & -\langle \sum_{k=1}^{n}\sum_{l=1}^{n}\sum_{i=1}^{n+m}\sum_{j^{\prime }=1}^{n+m}a_{lj^{\prime }}^{T}\frac{\partial^{2}}{\partial x_{i}x_{j}^{\prime }}a_{ik}{\left(a^{T}\nabla f\right)}_{k},a^{T}\nabla f\rangle \\ & & +\sum_{l=1}^{n}{\left(a^{T}\nabla f\right)}_{l}\left(\sum_{\hat{i},\hat{k},l^{\prime }=1}^{n+m}\sum_{k=1}^{n}\left(\frac{\partial }{\partial x_{\hat{i}}}a_{\hat{i}k}a_{k\hat{k}}^{T}\left(\frac{\partial }{\partial x_{\hat{k}}}a_{ll^{\prime }}^{T}\frac{\partial }{\partial x_{l^{\prime }}}f\right)-a_{ll^{\prime }}^{T}\frac{\partial }{\partial x_{\hat{i}}}a_{\hat{i}k}\left(\frac{\partial }{\partial x_{l^{\prime }}}a_{k\hat{k}}^{T}\frac{\partial }{\partial x_{\hat{k}}}f\right)\right)\right)\\ & = & \frac{1}{2}(a^{T}\nabla \circ (a^{T}\nabla |a^{T}{\nabla f|}^{2}))\\ & & -{\langle a^{T}\nabla \left(\left(a^{T}\nabla \right)\circ \left(a^{T}\nabla f\right)\right),a^{T}\nabla f\rangle }_{R^{n}}-{\langle B_{n\times n}a^{T}\nabla f,a^{T}\nabla f\rangle }_{R^{n}}\\ & & +\sum_{l=1}^{n}{\left(a^{T}\nabla f\right)}_{l}\left(\sum_{\hat{i},\hat{k},l^{\prime }=1}^{n+m}\sum_{k=1}^{n}\left(\frac{\partial }{\partial x_{\hat{i}}}a_{\hat{i}k}a_{k\hat{k}}^{T}\left(\frac{\partial }{\partial x_{\hat{k}}}a_{ll^{\prime }}^{T}\frac{\partial }{\partial x_{l^{\prime }}}f\right)-a_{ll^{\prime }}^{T}\frac{\partial }{\partial x_{\hat{i}}}a_{\hat{i}k}\left(\frac{\partial }{\partial x_{l^{\prime }}}a_{k\hat{k}}^{T}\frac{\partial }{\partial x_{\hat{k}}}f\right)\right)\right). \end{array}$$Thus, the proof is completed. □

Below, we further investigate the extra term explicitly in the above Lemma 15.

**Lemma** **16.**

$$\begin{array}{ccc} & & \frac{1}{2}(a^{T}\nabla \circ (a^{T}\nabla |a^{T}{\nabla f|}^{2}))-\langle a^{T}\nabla \left(\left(a^{T}\nabla \right)\circ \left(a^{T}\nabla f\right)\right),a^{T}\nabla f\rangle {}_{R^{n}}\\ & = & X^{T}Q^{T}QX+2D^{T}QX+2C^{T}X+D^{T}D\\ & & +\sum_{i,k=1}^{n}\sum_{i^{\prime },\hat{i},\hat{k}=1}^{n+m}\langle a_{ii^{\prime }}^{T}\left(\frac{\partial a_{i\hat{i}}^{T}}{\partial x_{i^{\prime }}}\frac{\partial a_{k\hat{k}}^{T}}{\partial x_{\hat{i}}}\frac{\partial f}{\partial x_{\hat{k}}}\right),{\left(a^{T}\nabla \right)}_{k}f\rangle {}_{R^{n}}\\ & & +\sum_{i,k=1}^{n}\sum_{i^{\prime },\hat{i},\hat{k}=1}^{n+m}\langle a_{ii^{\prime }}^{T}a_{i\hat{i}}^{T}\left(\frac{\partial }{\partial x_{i^{\prime }}}\frac{\partial a_{k\hat{k}}^{T}}{\partial x_{\hat{i}}}\right)\left(\frac{\partial f}{\partial x_{\hat{k}}}\right),{\left(a^{T}\nabla \right)}_{k}f\rangle {}_{R^{n}} \end{array}$$


(54)
$$\begin{array}{ccc} & & -\sum_{i,k=1}^{n}\sum_{i^{\prime },\hat{i},\hat{k}=1}^{n+m}\langle \left(a_{k\hat{k}}^{T}\frac{\partial a_{ii^{\prime }}^{T}}{\partial x_{\hat{k}}}\frac{\partial a_{i\hat{i}}^{T}}{\partial x_{i^{\prime }}}\frac{\partial f}{\partial x_{\hat{i}}}\right),{\left(a^{T}\nabla \right)}_{k}f\rangle {}_{R^{n}}\\ & & -\sum_{i,k=1}^{n}\sum_{i^{\prime },\hat{i},\hat{k}=1}^{n+m}\langle a_{k\hat{k}}^{T}a_{ii^{\prime }}^{T}\left(\frac{\partial }{\partial x_{\hat{k}}}\frac{\partial a_{i\hat{i}}^{T}}{\partial x_{i^{\prime }}}\right)\frac{\partial f}{\partial x_{\hat{i}}},{\left(a^{T}\nabla \right)}_{k}f\rangle {}_{R^{n}}. \end{array}$$


*Recall that matrix Q and vectors X, C, and D are defined in Notation 1.*


**Proof.** We expand the two terms in Lemma 16. The first term reads as
$$\begin{array}{ccc} & & \frac{1}{2}(a^{T}\nabla \circ (a^{T}\nabla |a^{T}{\nabla f|}^{2}))\\ & = & \frac{1}{2}\sum_{i=1}^{n}\sum_{k=1}^{n}{\left(a^{T}\nabla \right)}_{i}{\left(a^{T}\nabla \right)}_{i}{\left|{\left(a^{T}\nabla \right)}_{k}f\right|}^{2}\\ & = & \sum_{i=1}^{n}\sum_{k=1}^{n}{\left(a^{T}\nabla \right)}_{i}{\langle {\left(a^{T}\nabla \right)}_{i}{\left(a^{T}\nabla \right)}_{k}f,{\left(a^{T}\nabla \right)}_{k}f\rangle }_{R^{n}}\\ & = & \sum_{i=1}^{n}\sum_{k=1}^{n}{\langle {\left(a^{T}\nabla \right)}_{i}{\left(a^{T}\nabla \right)}_{k}f,{\left(a^{T}\nabla \right)}_{i}{\left(a^{T}\nabla \right)}_{k}f\rangle }_{R^{n}}\cdots T_{1}\\ & & +\sum_{i=1}^{n}\sum_{k=1}^{n}\sum_{i^{\prime },\hat{i},\hat{k}=1}^{n+m}{\langle \left(a_{ii^{\prime }}^{T}\frac{\partial }{\partial x_{i^{\prime }}}\right)\left(a_{i\hat{i}}^{T}\frac{\partial }{\partial x_{\hat{i}}}\right)\left(a_{k\hat{k}}^{T}\frac{\partial }{\partial x_{\hat{k}}}\right)f,{\left(a^{T}\nabla \right)}_{k}f\rangle }_{R^{n}}\cdots R_{1}. \end{array}$$The second term reads as
$$\begin{array}{ccc} & & {\langle a^{T}\nabla \left(\left[\left(a^{T}\nabla \right)\circ \left(a^{T}\nabla f\right)\right]\right),a^{T}\nabla f\rangle }_{R^{n}}\\ & = & \sum_{i,k=1}^{n}\langle {\left(a^{T}\nabla \right)}_{k}\left[{\left(a^{T}\nabla \right)}_{i}{\left(a^{T}\nabla \right)}_{i}f\right],{\left(a^{T}\nabla \right)}_{k}f\rangle \\ & = & \sum_{i,k=1}^{n}\sum_{i^{\prime },\hat{i},\hat{k}=1}^{n+m}\langle \left(a_{k\hat{k}}^{T}\frac{\partial }{\partial x_{\hat{k}}}\right)\left[\left(a_{ii^{\prime }}^{T}\frac{\partial }{\partial x_{i^{\prime }}}\right)\left(a_{i\hat{i}}^{T}\frac{\partial }{\partial x_{\hat{i}}}\right)f\right],{\left(a^{T}\nabla \right)}_{k}f\rangle \cdots R_{2}. \end{array}$$Next, we expand $R_{1}$ and $R_{2}$ completely and obtain the following:
$$\begin{array}{ccc} R_{1} & = & \sum_{i,k=1}^{n}\sum_{i^{\prime },\hat{i},\hat{k}=1}^{n+m}\langle \left(a_{ii^{\prime }}^{T}\frac{\partial }{\partial x_{i^{\prime }}}\right)\left(a_{i\hat{i}}^{T}\frac{\partial }{\partial x_{\hat{i}}}\right)\left(a_{k\hat{k}}^{T}\frac{\partial f}{\partial x_{\hat{k}}}\right),{\left(a^{T}\nabla \right)}_{k}f\rangle {}_{R^{n}}\\ & = & \sum_{i,k=1}^{n}\sum_{i^{\prime },\hat{i},\hat{k}=1}^{n+m}\langle a_{ii^{\prime }}^{T}\left(\frac{\partial a_{i\hat{i}}^{T}}{\partial x_{i^{\prime }}}\frac{\partial a_{k\hat{k}}^{T}}{\partial x_{\hat{i}}}\frac{\partial f}{\partial x_{\hat{k}}}\right),{\left(a^{T}\nabla \right)}_{k}f\rangle {}_{R^{n}}\cdots R_{1}^{1}\\ & & +\sum_{i,k=1}^{n}\sum_{i^{\prime },\hat{i},\hat{k}=1}^{n+m}\langle a_{ii^{\prime }}^{T}a_{i\hat{i}}^{T}\left(\frac{\partial }{\partial x_{i^{\prime }}}\frac{\partial a_{k\hat{k}}^{T}}{\partial x_{\hat{i}}}\right)\left(\frac{\partial f}{\partial x_{\hat{k}}}\right),{\left(a^{T}\nabla \right)}_{k}f\rangle {}_{R^{n}}\cdots R_{1}^{2} \end{array}$$
$$\begin{array}{ccc} & & +\sum_{i,k=1}^{n}\sum_{i^{\prime },\hat{i},\hat{k}=1}^{n+m}\langle a_{ii^{\prime }}^{T}a_{i\hat{i}}^{T}\left(\frac{\partial a_{k\hat{k}}^{T}}{\partial x_{\hat{i}}}\right)\left(\frac{\partial }{\partial x_{i^{\prime }}}\frac{\partial f}{\partial x_{\hat{k}}}\right),{\left(a^{T}\nabla \right)}_{k}f\rangle {}_{R^{n}}\cdots R_{1}^{3}\\ & & +\sum_{i,k=1}^{n}\sum_{i^{\prime },\hat{i},\hat{k}=1}^{n+m}\langle \left(a_{ii^{\prime }}^{T}\right)\left(\left(\frac{\partial }{\partial x_{i^{\prime }}}a_{i\hat{i}}^{T}\right)a_{k\hat{k}}^{T}\frac{\partial }{\partial x_{\hat{i}}}\frac{\partial f}{\partial x_{\hat{k}}}\right),{\left(a^{T}\nabla \right)}_{k}f\rangle {}_{R^{n}}\cdots R_{1}^{4}\\ & & +\sum_{i,k=1}^{n}\sum_{i^{\prime },\hat{i},\hat{k}=1}^{n+m}\langle a_{ii^{\prime }}^{T}a_{i\hat{i}}^{T}\left(\frac{\partial }{\partial x_{i^{\prime }}}a_{k\hat{k}}^{T}\right)\frac{\partial }{\partial x_{\hat{i}}}\frac{\partial f}{\partial x_{\hat{k}}}),{\left(a^{T}\nabla \right)}_{k}{f\rangle }_{R^{n}}\cdots R_{1}^{5}\\ & & +\sum_{i,k=1}^{n}\sum_{i^{\prime },\hat{i},\hat{k}=1}^{n+m}\langle a_{ii^{\prime }}^{T}a_{i\hat{i}}^{T}a_{k\hat{k}}^{T}\left(\frac{\partial }{\partial x_{i^{\prime }}}\frac{\partial }{\partial x_{\hat{i}}}\frac{\partial f}{\partial x_{\hat{k}}}\right),{\left(a^{T}\nabla \right)}_{k}f\rangle {}_{R^{n}}\cdots R_{1}^{6}. \end{array}$$Additionally,
$$\begin{array}{ccc} R_{2} & = & \sum_{i,k=1}^{n}\sum_{i^{\prime },\hat{i},\hat{k}=1}^{n+m}\langle \left(a_{k\hat{k}}^{T}\frac{\partial }{\partial x_{\hat{k}}}\right)\left[\left(a_{ii^{\prime }}^{T}\frac{\partial }{\partial x_{i^{\prime }}}\right)\left(a_{i\hat{i}}^{T}\frac{\partial f}{\partial x_{\hat{i}}}\right)\right],{\left(a^{T}\nabla \right)}_{k}f\rangle \\ & = & \sum_{i,k=1}^{n}\sum_{i^{\prime },\hat{i},\hat{k}=1}^{n+m}\langle a_{k\hat{k}}^{T}\frac{\partial a_{ii^{\prime }}^{T}}{\partial x_{\hat{k}}}\frac{\partial a_{i\hat{i}}^{T}}{\partial x_{i^{\prime }}}\frac{\partial f}{\partial x_{\hat{i}}}),{\left(a^{T}\nabla \right)}_{k}f\rangle \cdots R_{2}^{1}\\ & & +\sum_{i,k=1}^{n}\sum_{i^{\prime },\hat{i},\hat{k}=1}^{n+m}\langle a_{k\hat{k}}^{T}a_{ii^{\prime }}^{T}\left(\frac{\partial }{\partial x_{\hat{k}}}\frac{\partial a_{i\hat{i}}^{T}}{\partial x_{i^{\prime }}}\right)\frac{\partial f}{\partial x_{\hat{i}}},{\left(a^{T}\nabla \right)}_{k}f\rangle \cdots R_{2}^{2}\\ & & +\sum_{i,k=1}^{n}\sum_{i^{\prime },\hat{i},\hat{k}=1}^{n+m}\langle a_{k\hat{k}}^{T}a_{ii^{\prime }}^{T}\frac{\partial a_{i\hat{i}}^{T}}{\partial x_{i^{\prime }}}\left(\frac{\partial }{\partial x_{\hat{k}}}\frac{\partial f}{\partial x_{\hat{i}}}\right),{\left(a^{T}\nabla \right)}_{k}f\rangle \cdots R_{2}^{3}=R_{1}^{4}\\ & & +\sum_{i,k=1}^{n}\sum_{i^{\prime },\hat{i},\hat{k}=1}^{n+m}\langle a_{k\hat{k}}^{T}\frac{\partial a_{ii^{\prime }}^{T}}{\partial x_{\hat{k}}}a_{i\hat{i}}^{T}\left(\frac{\partial }{\partial x_{i^{\prime }}}\frac{\partial f}{\partial x_{\hat{i}}}\right),{\left(a^{T}\nabla \right)}_{k}f\rangle \cdots R_{2}^{4}\\ & & +\sum_{i,k=1}^{n}\sum_{i^{\prime },\hat{i},\hat{k}=1}^{n+m}\langle a_{k\hat{k}}^{T}a_{ii^{\prime }}^{T}\frac{\partial a_{i\hat{i}}^{T}}{\partial x_{\hat{k}}}\left(\frac{\partial }{\partial x_{i^{\prime }}}\frac{\partial f}{\partial x_{\hat{i}}}\right),{\left(a^{T}\nabla \right)}_{k}f\rangle \cdots R_{2}^{5}\\ & & +\sum_{i,k=1}^{n}\sum_{i^{\prime },\hat{i},\hat{k}=1}^{n+m}\langle a_{k\hat{k}}^{T}a_{ii^{\prime }}^{T}a_{i\hat{i}}^{T}\left(\frac{\partial }{\partial x_{\hat{k}}}\frac{\partial }{\partial x_{i^{\prime }}}\frac{\partial f}{\partial x_{\hat{i}}}\right),{\left(a^{T}\nabla \right)}_{k}f\rangle \cdots R_{2}^{6}=R_{1}^{6}. \end{array}$$Our next step is to complete the squares for all the above terms. Look at the term $T_{1}$ first.
$$\begin{array}{ccc} T_{1} & = & \sum_{i,k=1}^{n}\langle \sum_{\hat{i},\hat{k}=1}^{n+m}a_{i\hat{i}}^{T}a_{k\hat{k}}^{T}\frac{\partial^{2}f}{\partial x_{\hat{i}}\partial x_{\hat{k}}}+\sum_{\hat{i},\hat{k}=1}^{n+m}a_{i\hat{i}}^{T}\frac{\partial a_{k\hat{k}}^{T}}{\partial x_{\hat{i}}}\frac{\partial f}{\partial x_{\hat{k}}},\\ & & \;\;\;\sum_{i^{\prime },k^{\prime }=1}^{n+m}a_{ii^{\prime }}^{T}a_{kk^{\prime }}^{T}\frac{\partial^{2}f}{\partial x_{i^{\prime }}\partial x_{k^{\prime }}}+\sum_{i^{\prime },k^{\prime }=1}^{n+m}a_{ii^{\prime }}^{T}\frac{\partial a_{kk^{\prime }}^{T}}{\partial x_{i^{\prime }}}\frac{\partial f}{\partial x_{k^{\prime }}}\rangle \\ & = & \sum_{i,k=1}^{n}\langle \sum_{\hat{i},\hat{k}=1}^{n+m}a_{i\hat{i}}^{T}a_{k\hat{k}}^{T}\frac{\partial^{2}f}{\partial x_{\hat{i}}\partial x_{\hat{k}}},\sum_{i^{\prime },k^{\prime }=1}^{n+m}a_{ii^{\prime }}^{T}a_{kk^{\prime }}^{T}\frac{\partial^{2}f}{\partial x_{i^{\prime }}\partial x_{k^{\prime }}}\rangle \cdots T_{1a}\\ & & +\sum_{i,k=1}^{n}\langle \sum_{\hat{i},\hat{k}=1}^{n+m}a_{i\hat{i}}^{T}a_{k\hat{k}}^{T}\frac{\partial^{2}f}{\partial x_{\hat{i}}\partial x_{\hat{k}}},\sum_{i^{\prime },k^{\prime }=1}^{n+m}a_{ii^{\prime }}^{T}\frac{\partial a_{kk^{\prime }}^{T}}{\partial x_{i^{\prime }}}\frac{\partial f}{\partial x_{k^{\prime }}}\rangle \cdots T_{1b}\\ & & +\sum_{i,k=1}^{n}\langle \sum_{\hat{i},\hat{k}=1}^{n+m}a_{i\hat{i}}^{T}\frac{\partial a_{k\hat{k}}^{T}}{\partial x_{\hat{i}}}\frac{\partial f}{\partial x_{\hat{k}}},\sum_{i^{\prime },k^{\prime }=1}^{n+m}a_{ii^{\prime }}^{T}a_{kk^{\prime }}^{T}\frac{\partial^{2}f}{\partial x_{i^{\prime }}\partial x_{k^{\prime }}}\rangle \cdots T_{1c}\\ & & +\sum_{i,k=1}^{n}\langle \sum_{\hat{i},\hat{k}=1}^{n+m}a_{i\hat{i}}^{T}\frac{\partial a_{k\hat{k}}^{T}}{\partial x_{\hat{i}}}\frac{\partial f}{\partial x_{\hat{k}}},\sum_{i^{\prime },k^{\prime }=1}^{n+m}a_{ii^{\prime }}^{T}\frac{\partial a_{kk^{\prime }}^{T}}{\partial x_{i^{\prime }}}\frac{\partial f}{\partial x_{k^{\prime }}}\rangle \cdots T_{1d}. \end{array}$$The terms $T_{1b}=T_{1c}$, $R_{1}^{3}=R_{1}^{5}$, and $R_{2}^{5}=R_{2}^{4}$ play the role of crossing terms inside the complete squares. In particular, for convenience, we change the index inside the sum of $R_{1}^{3}$ and $R_{2}^{5}$, switching $i^{\prime },\hat{i}$ for $R_{1}^{3}$ and switching $i^{\prime },\hat{k}$ for $R_{2}^{5}$. Then, we obtain the following.
$$\begin{array}{ccc} 2R_{1}^{3} & = & 2\sum_{i,k=1}^{n}\sum_{i^{\prime },\hat{i},\hat{k}=1}^{n+m}\langle a_{i\hat{i}}^{T}a_{ii^{\prime }}^{T}\left(\frac{\partial a_{k\hat{k}}^{T}}{\partial x_{i^{\prime }}}\right)\left(\frac{\partial }{\partial x_{\hat{i}}}\frac{\partial f}{\partial x_{\hat{k}}}\right),{\left(a^{T}\nabla \right)}_{k}f\rangle \\ & = & 2\sum_{i,k=1}^{n}\sum_{\hat{i},\hat{k}=1}^{n+m}\sum_{i^{\prime },l=1}^{n+m}\left(a_{i\hat{i}}^{T}a_{ii^{\prime }}^{T}\left(\frac{\partial a_{k\hat{k}}^{T}}{\partial x_{i^{\prime }}}\right)\left(\frac{\partial }{\partial x_{\hat{i}}}\frac{\partial f}{\partial x_{\hat{k}}}\right)a_{kl}^{T}\frac{\partial f}{\partial x_{l}}\right)\\ -2R_{2}^{5} & = & -2\sum_{i,k=1}^{n}\sum_{i^{\prime },\hat{i},\hat{k}=1}^{n+m}\langle a_{ki^{\prime }}^{T}a_{i\hat{k}}^{T}\frac{\partial a_{i\hat{i}}^{T}}{\partial x_{i^{\prime }}}\left(\frac{\partial }{\partial x_{\hat{k}}}\frac{\partial f}{\partial x_{\hat{i}}}\right),{\left(a^{T}\nabla \right)}_{k}f\rangle \\ & = & -2\sum_{i,k=1}^{n}\sum_{\hat{i},\hat{k}=1}^{n+m}\sum_{i^{\prime },l=1}^{n+m}\left(a_{ki^{\prime }}^{T}a_{i\hat{k}}^{T}\frac{\partial a_{i\hat{i}}^{T}}{\partial x_{i^{\prime }}}\left(\frac{\partial }{\partial x_{\hat{k}}}\frac{\partial f}{\partial x_{\hat{i}}}\right)a_{kl}^{T}\frac{\partial f}{\partial x_{l}}\right). \end{array}$$We denote
(55)$$\begin{array}{ccc} \sum_{\hat{i},\hat{k}=1}^{n+m}a_{i\hat{i}}^{T}a_{k\hat{k}}^{T}\frac{\partial^{2}f}{\partial x_{\hat{i}}\partial x_{\hat{k}}} & = & \gamma_{ik}. \end{array}$$The above Equality (Equation 55) can be represented in the following matrix form:
$$\begin{array}{c} Q_{n^{2}\times {\left(n+m\right)}^{2}}X_{{\left(n+m\right)}^{2}\times 1}={\left(\gamma_{11},\cdots ,\gamma_{ik},\cdots ,\gamma_{nn}\right)}_{n^{2}\times 1}^{T}, \end{array}$$
where *Q* and *X* are defined in (Equation 12) and (Equation 19). Now, we can represent term $T_{1a}$ as $\sum_{i,k=1}^{n}\gamma_{ik}^{2}=\gamma^{T}\gamma ={\left(QX\right)}^{T}QX=X^{T}Q^{T}QX$. Next, we want to represent $R_{1}^{3}$ and $R_{2}^{5}$ in the following form in terms of vector *X*:
$$\begin{array}{ccc} & & 2R_{1}^{3}-2R_{2}^{5}\\ & = & 2\sum_{i,k=1}^{n}\sum_{i^{\prime },\hat{i},\hat{k}=1}^{n+m}\left({\langle a_{i\hat{i}}^{T}a_{ii^{\prime }}^{T}\left(\frac{\partial a_{k\hat{k}}^{T}}{\partial x_{i^{\prime }}}\right)\left(\frac{\partial }{\partial x_{\hat{i}}}\frac{\partial f}{\partial x_{\hat{k}}}\right),{\left(a^{T}\nabla \right)}_{k}f\rangle }_{R^{n}}\right.\\ & & \left.\;\;\;\;\;\;-2\sum_{i,k=1}^{n}\sum_{i^{\prime },\hat{i},\hat{k}=1}^{n+m}\langle a_{ki^{\prime }}^{T}a_{i\hat{k}}^{T}\frac{\partial a_{i\hat{i}}^{T}}{\partial x_{i^{\prime }}}\left(\frac{\partial }{\partial x_{\hat{k}}}\frac{\partial f}{\partial x_{\hat{i}}}\right),{\left(a^{T}\nabla \right)}_{k}f\rangle \right)\\ & = & 2\sum_{\hat{i},\hat{k}=1}^{n+m}\left[\sum_{i,k=1}^{n}\sum_{i^{\prime }=1}^{n+m}\left(\langle a_{i\hat{i}}^{T}a_{ii^{\prime }}^{T}\left(\frac{\partial a_{k\hat{k}}^{T}}{\partial x_{i^{\prime }}}\right),{\left(a^{T}\nabla \right)}_{k}f\rangle -\langle a_{ki^{\prime }}^{T}a_{i\hat{k}}^{T}\frac{\partial a_{i\hat{i}}^{T}}{\partial x_{i^{\prime }}},{\left(a^{T}\nabla \right)}_{k}f\rangle \right)\right]\left(\frac{\partial }{\partial x_{\hat{i}}}\frac{\partial f}{\partial x_{\hat{k}}}\right)\\ & = & 2C^{T}X, \end{array}$$
where *C* is defined in (Equation 14). Similarly, we can represent $T_{1b}=T_{1c}$ by *X*:
$$\begin{array}{ccc} T_{1b}=T_{1c} & = & \sum_{i,k=1}^{n}\langle \sum_{\hat{i},\hat{k}=1}^{n+m}a_{i\hat{i}}^{T}\frac{\partial a_{k\hat{k}}^{T}}{\partial x_{\hat{i}}}\frac{\partial f}{\partial x_{\hat{k}}},\sum_{i^{\prime },k^{\prime }=1}^{n+m}a_{ii^{\prime }}^{T}a_{kk^{\prime }}^{T}\frac{\partial^{2}f}{\partial x_{i^{\prime }}\partial x_{k^{\prime }}}\rangle \\ & = & D^{T}QX, \end{array}$$
where *D* is defined in (1). Summingover the above terms, we have the following quadratic form:
(56)$$\begin{array}{c} T_{1}+2R_{1}^{3}-2R_{2}^{5}=X^{T}Q^{T}QX+2D^{T}QX+2C^{T}X+D^{T}D. \end{array}$$Taking into account the fact that $R_{1}^{6}-R_{2}^{6}=0$ and $R_{1}^{4}-R_{2}^{3}=0$, we have
$$\begin{array}{c} T_{1}+R_{1}-R_{2}=T_{1}+2R_{1}^{3}-2R_{2}^{5}+R_{1}^{1}+R_{1}^{2}-R_{2}^{1}-R_{2}^{2}, \end{array}$$
which completes the proof. □

### 5.3. Proof of Lemma 12

**Lemma** **17.**

(57)
$$\begin{array}{c} \Gamma_{2,\tilde{L}}^{z}\left(f,f\right)=X^{T}P^{T}PX+2E^{T}PX+2F^{T}X+E^{T}E+R_{z}\left(\nabla f,\nabla f\right). \end{array}$$

*where $R_{z}$ is defined in Definition 2.*


**Proof.** The proof follows directly from Lemmas 18 and 19. □

**Lemma** **18.**

$$\begin{array}{ccc} & & \frac{1}{2}\tilde{L}\Gamma_{1}^{z}\left(f,f\right)-\Gamma_{1}^{z}\left(\tilde{L}f,f\right)\\ & = & \frac{1}{2}(a^{T}\nabla \circ (a^{T}\nabla |z^{T}{\nabla f|}^{2}))-{\langle z^{T}\nabla \left(\left(a^{T}\nabla \right)\circ \left(a^{T}\nabla f\right)\right),z^{T}\nabla f\rangle }_{R^{m}}. \end{array}$$



**Proof.** **Step 1:** We first define $\Gamma_{1}^{z}={\langle z^{T}\nabla f,z^{T}\nabla f\rangle }_{R^{m}}$, then we have
$$\begin{array}{c} \tilde{L}\Gamma_{1}^{z}\left(f,f\right)=\Delta_{p}\Gamma_{1}^{z}\left(f,f\right)-A\nabla \Gamma_{1}^{z}\left(f,f\right),\;\Gamma_{1}^{z}\left(\tilde{L}f,f\right)=\Gamma_{1}^{z}\left(\Delta_{p}f,f\right)-\Gamma_{1}^{z}\left(A\nabla f,f\right). \end{array}$$By our definition above, we directly obtain
$$\begin{array}{ccc} \Delta_{a}\Gamma_{1}^{z}\left(f,f\right) & = & \nabla \cdot \left(aa^{T}\nabla {\langle z^{T}\nabla f,z^{T}\nabla f\rangle }_{R^{m}}\right)=\nabla \cdot \left(aF^{z}\right)\\ & = & \sum_{\hat{i}=1}^{n+m}\frac{\partial }{\partial x_{\hat{i}}}\left(\sum_{k=1}^{n}a_{\hat{i}k}F_{k}^{z}\right)\\ & = & \sum_{\hat{i}=1}^{n+m}\sum_{k=1}^{n}\left(\frac{\partial }{\partial x_{\hat{i}}}a_{\hat{i}k}F_{k}^{z}+a_{\hat{i}k}\frac{\partial }{\partial x_{\hat{i}}}F_{k}^{z}\right)\\ & = & \sum_{\hat{i}=1}^{n+m}\sum_{k=1}^{n}\left(\frac{\partial }{\partial x_{\hat{i}}}a_{\hat{i}k}F_{k}^{z}\right)+a^{T}\nabla \circ \left(a^{T}\nabla {\left(z^{T}\nabla f\right)}^{2}\right), \end{array}$$
where we denote
$$\begin{array}{ccc} F^{z} & = & a^{T}\nabla {\langle z^{T}\nabla f,z^{T}\nabla f\rangle }_{R^{m}}=a^{T}\nabla \sum_{l=1}^{m}{\left(\sum_{\hat{l}=1}^{n+m}z_{l\hat{l}}^{T}\frac{\partial }{\partial x_{\hat{l}}}f\right)}^{2}\\ & = & {\left(\sum_{\hat{k}=1}^{n+m}a_{k\hat{k}}^{T}\frac{\partial }{\partial x_{\hat{k}}}\sum_{l=1}^{m}{\left(\sum_{\hat{l}=1}^{n+m}z_{l\hat{l}}^{T}\frac{\partial }{\partial x_{\hat{l}}}f\right)}^{2}\right)}_{k=1,\cdots ,n}={\left(F_{1}^{z},F_{2}^{z},\cdots ,F_{n}^{z}\right)}^{T}. \end{array}$$We have
(58)$$\begin{array}{ccc} & & \Delta_{a}\Gamma_{1}^{z}\left(f,f\right)\\ & = & \sum_{\hat{i}=1}^{n+m}\sum_{k=1}^{n}\left(\frac{\partial }{\partial x_{\hat{i}}}a_{\hat{i}k}\left(\sum_{\hat{k}=1}^{n+m}a_{k\hat{k}}^{T}\frac{\partial }{\partial x_{\hat{k}}}\sum_{l=1}^{m}{\left(\sum_{\hat{l}=1}^{n+m}z_{l\hat{l}}^{T}\frac{\partial }{\partial x_{\hat{l}}}f\right)}^{2}\right)\right)+a^{T}\nabla \circ \left(a^{T}\nabla {\left(z^{T}\nabla f\right)}^{2}\right))\\ & = & \sum_{k=1}^{n}\sum_{\hat{i}=1}^{n+m}\left(\frac{\partial }{\partial x_{\hat{i}}}a_{\hat{i}k}\left(\sum_{\hat{k}=1}^{n+m}a_{k\hat{k}}^{T}\frac{\partial }{\partial x_{\hat{k}}}{\left(z^{T}\nabla f\right)}^{2}\right)\right)+\left(a^{T}\nabla \right)\circ \left(a^{T}\nabla {\left(z^{T}\nabla f\right)}^{2}\right)\\ & = & \nabla a\circ \left(a^{T}\nabla {\left(z^{T}\nabla f\right)}^{2}\right)+\left(a^{T}\nabla \right)\circ \left(a^{T}\nabla {\left(z^{T}\nabla f\right)}^{2}\right). \end{array}$$Next, we compute the following quantity.
$$\begin{array}{ccc} \Gamma_{1}^{z}\left(\Delta_{a}f,f\right) & = & {\langle z^{T}\nabla \left(\nabla \cdot \left(aa^{T}\nabla f\right)\right),z^{T}\nabla f\rangle }_{R^{m}}. \end{array}$$From Lemma 15, we have
$$\begin{array}{c} \nabla \cdot \left(aa^{T}\nabla f\right)=\nabla a\circ \left(a^{T}\nabla f\right)+\left(a^{T}\nabla \right)\circ \left(a^{T}\nabla f\right). \end{array}$$We continue with our computation as below:
(59)$$\begin{array}{ccc} & & \Gamma_{1}^{z}\left(\Delta_{a}f,f\right)\\ & = & {\langle z^{T}\nabla \left[\sum_{\hat{i},\hat{k}=1}^{n+m}\sum_{k=1}^{n}\frac{\partial }{\partial x_{\hat{i}}}a_{\hat{i}k}\left(a_{k\hat{k}}^{T}\frac{\partial }{\partial x_{\hat{k}}}f\right)+\left(a^{T}\nabla \right)\circ \left(a^{T}\nabla f\right)\right],z^{T}\nabla f\rangle }_{R^{m}}\\ & = & {\langle z^{T}\nabla \left[\sum_{\hat{i},\hat{k}=1}^{n+m}\sum_{k=1}^{n}\frac{\partial }{\partial x_{\hat{i}}}a_{\hat{i}k}\left(a_{k\hat{k}}^{T}\frac{\partial }{\partial x_{\hat{k}}}f\right)\right],z^{T}\nabla f\rangle }_{R^{m}}+{\langle z^{T}\nabla \left(\left(a^{T}\nabla \right)\circ \left(a^{T}\nabla f\right)\right),z^{T}\nabla f\rangle }_{R^{m}} \end{array}$$
(60)$$\begin{array}{ccc} & = & {\langle z^{T}\nabla \left(\nabla a\cdot (a^{T}\nabla f)\right),z^{T}\nabla f\rangle }_{R^{m}}\\ & & +{\langle z^{T}\nabla \left(\left(a^{T}\nabla \right)\circ \left(a^{T}\nabla f\right)\right),z^{T}\nabla f\rangle }_{R^{m}}\\ & = & {\langle \left(z^{T}\nabla \nabla a\cdot \left(a^{T}\nabla f\right)\right),z^{T}\nabla f\rangle }_{R^{m}}+{\langle \left(\nabla a\cdot (z^{T}\nabla \left(a^{T}\nabla f\right))\right),z^{T}\nabla f\rangle }_{R^{m}}\\ & & +{\langle z^{T}\nabla \left(\left(a^{T}\nabla \right)\circ \left(a^{T}\nabla f\right)\right),z^{T}\nabla f\rangle }_{R^{m}}. \end{array}$$From the above, combining (Equation 58) and (Equation 59), we further obtain
$$\begin{array}{ccc} & & \frac{1}{2}\Delta_{a}\Gamma_{1}^{z}\left(f,f\right)-\Gamma_{1}^{z}\left(\Delta_{a}f,f\right)\\ & = & \frac{1}{2}(a^{T}\nabla \circ (a^{T}\nabla |z^{T}{\nabla f|}^{2}))-{\langle z^{T}\nabla \left(\left(a^{T}\nabla \right)\circ \left(a^{T}\nabla f\right)\right),z^{T}\nabla f\rangle }_{R^{m}}\\ & & +\frac{1}{2}\sum_{\hat{i}=1}^{n+m}\sum_{k=1}^{n}\left(\frac{\partial }{\partial x_{\hat{i}}}a_{\hat{i}k}\left(\sum_{\hat{k}=1}^{n+m}a_{k\hat{k}}^{T}\frac{\partial }{\partial x_{\hat{j}}}\sum_{l=1}^{n}{\left(\sum_{\hat{l}=1}^{n+m}z_{l\hat{l}}^{T}\frac{\partial }{\partial x_{\hat{l}}}f\right)}^{2}\right)\right) \end{array}$$
$$\begin{array}{ccc} & & -{\langle z^{T}\nabla \left[\sum_{\hat{i},\hat{k}=1}^{n+m}\sum_{k=1}^{n}\frac{\partial }{\partial x_{\hat{i}}}a_{\hat{i}k}\left(a_{k\hat{k}}^{T}\frac{\partial }{\partial x_{\hat{k}}}f\right)\right],z^{T}\nabla f\rangle }_{R^{m}}\\ & = & \frac{1}{2}(a^{T}\nabla \circ (a^{T}\nabla |z^{T}{\nabla f|}^{2}))-{\langle z^{T}\nabla \left(\left(a^{T}\nabla \right)\circ \left(a^{T}\nabla f\right)\right),z^{T}\nabla f\rangle }_{R^{m}}\\ & & +\sum_{\hat{i}=1}^{n+m}\sum_{k=1}^{n}\left(\frac{\partial }{\partial x_{\hat{i}}}a_{\hat{i}k}\sum_{\hat{k}=1}^{n+m}a_{k\hat{k}}^{T}\sum_{l=1}^{m}\left(\sum_{\hat{l}=1}^{n+m}z_{l\hat{l}}^{T}\frac{\partial }{\partial x_{\hat{l}}}f\right)\frac{\partial }{\partial x_{\hat{k}}}\left(\sum_{\hat{l}=1}^{n+m}z_{l\hat{l}}^{T}\frac{\partial }{\partial x_{\hat{l}}}f\right)\right)\cdots I\\ & & -\sum_{l=1}^{m}\left(\left(\sum_{\hat{l}=1}^{n+m}z_{l\hat{l}}^{T}\frac{\partial }{\partial x_{\hat{l}}}f\right)\left(\sum_{l^{\prime }=1}^{n+m}z_{ll^{\prime }}^{T}\frac{\partial }{\partial x_{l^{\prime }}}\left[\sum_{\hat{i},\hat{k}=1}^{n+m}\sum_{k=1}^{n}\frac{\partial }{\partial x_{\hat{i}}}a_{\hat{i}k}\left(a_{k\hat{k}}^{T}\frac{\partial }{\partial x_{\hat{k}}}f\right)\right]\right)\right)\cdots \mathrm{II}. \end{array}$$Recall here that we denote $a^{T}$ to emphasize the transpose of the matrix *a* and $a_{i\hat{i}}^{T}=a_{\hat{i}i}$:
$$\begin{array}{ccc} I & = & \sum_{\hat{i}=1}^{n+m}\sum_{k=1}^{n}\left(\frac{\partial }{\partial x_{\hat{i}}}a_{\hat{i}k}\sum_{\hat{k}=1}^{n+m}a_{k\hat{k}}^{T}\sum_{l=1}^{m}\left(\sum_{\hat{l}=1}^{n+m}z_{l\hat{l}}^{T}\frac{\partial }{\partial x_{\hat{l}}}f\right)\frac{\partial }{\partial x_{\hat{k}}}\left(\sum_{\hat{l}=1}^{n+m}z_{l\hat{l}}^{T}\frac{\partial }{\partial x_{\hat{l}}}f\right)\right)\\ & = & \sum_{\hat{i}=1}^{n+m}\sum_{k=1}^{n}\left(\frac{\partial }{\partial x_{\hat{i}}}a_{\hat{i}k}\sum_{\hat{k}=1}^{n+m}a_{k\hat{k}}^{T}\sum_{l=1}^{m}\left(\sum_{\hat{l}=1}^{n+m}z_{l\hat{l}}^{T}\frac{\partial }{\partial x_{\hat{l}}}f\right)\left(\sum_{\hat{l}=1}^{n+m}\frac{\partial }{\partial x_{\hat{k}}}z_{l\hat{l}}^{T}\frac{\partial }{\partial x_{\hat{l}}}f\right)\right)\\ & & +\sum_{\hat{i}=1}^{n+m}\sum_{k=1}^{n}\left(\frac{\partial }{\partial x_{\hat{i}}}a_{\hat{i}k}\sum_{\hat{k}=1}^{n+m}a_{k\hat{k}}^{T}\sum_{l=1}^{m}\left(\sum_{\hat{l}=1}^{n+m}z_{l\hat{l}}^{T}\frac{\partial }{\partial x_{\hat{l}}}f\right)\left(\sum_{\hat{l}=1}^{n+m}z_{l\hat{l}}^{T}\frac{\partial }{\partial x_{\hat{k}}}\frac{\partial }{\partial x_{\hat{l}}}f\right)\right)\\ & = & \sum_{l=1}^{m}\left(\sum_{\hat{l}=1}^{n+m}z_{l\hat{l}}^{T}\frac{\partial }{\partial x_{\hat{l}}}f\right)\sum_{\hat{i}=1}^{n+m}\sum_{k=1}^{n}\left(\frac{\partial }{\partial x_{\hat{i}}}a_{\hat{i}k}\sum_{\hat{k}=1}^{n+m}a_{k\hat{k}}^{T}\left(\sum_{l^{\prime }=1}^{n+m}\frac{\partial }{\partial x_{\hat{k}}}z_{ll^{\prime }}^{T}\frac{\partial }{\partial x_{l^{\prime }}}f\right)\right)\\ & & +\sum_{l=1}^{m}\left(\sum_{\hat{l}=1}^{n+m}z_{l\hat{l}}^{T}\frac{\partial }{\partial x_{\hat{l}}}f\right)\sum_{\hat{i}=1}^{n+m}\sum_{k=1}^{n}\left(\frac{\partial }{\partial x_{\hat{i}}}a_{\hat{i}k}\sum_{\hat{k}=1}^{n+m}a_{k\hat{k}}^{T}\left(\sum_{\hat{l}=1}^{n+m}z_{ll^{\prime }}^{T}\frac{\partial }{\partial x_{\hat{k}}}\frac{\partial }{\partial x_{l^{\prime }}}f\right)\right);\\ \mathrm{II} & = & \sum_{l=1}^{m}\left(\left(\sum_{\hat{l}=1}^{n+m}z_{l\hat{l}}^{T}\frac{\partial }{\partial x_{\hat{l}}}f\right)\left(\sum_{l^{\prime }=1}^{n+m}z_{ll^{\prime }}^{T}\frac{\partial }{\partial x_{l^{\prime }}}\left[\sum_{\hat{i},\hat{k}=1}^{n+m}\sum_{k=1}^{n}\frac{\partial }{\partial x_{\hat{i}}}a_{\hat{i}k}\left(a_{k\hat{k}}^{T}\frac{\partial }{\partial x_{\hat{k}}}f\right)\right]\right)\right)\\ & = & \sum_{l=1}^{m}\left(\left(\sum_{\hat{l}=1}^{n+m}z_{l\hat{l}}^{T}\frac{\partial }{\partial x_{\hat{l}}}f\right)\left(\sum_{l^{\prime }=1}^{n+m}z_{ll^{\prime }}^{T}\left[\sum_{\hat{i},\hat{k}=1}^{n+m}\sum_{k=1}^{n}\frac{\partial }{\partial x_{\hat{i}}}a_{\hat{i}k}\frac{\partial }{\partial x_{l^{\prime }}}\left(a_{k\hat{k}}^{T}\frac{\partial }{\partial x_{\hat{k}}}f\right)\right]\right)\right)\\ & & +\sum_{l=1}^{m}\left(\left(\sum_{\hat{l}=1}^{n+m}z_{l\hat{l}}^{T}\frac{\partial }{\partial x_{\hat{l}}}f\right)\left(\sum_{l^{\prime }=1}^{n+m}z_{ll^{\prime }}^{T}\left[\sum_{\hat{i},\hat{k}=1}^{n+m}\sum_{k=1}^{n}\frac{\partial^{2}}{\partial x_{\hat{i}}x_{l^{\prime }}}a_{\hat{i}k}\left(a_{k\hat{k}}^{T}\frac{\partial }{\partial x_{\hat{k}}}f\right)\right]\right)\right)\\ & = & \sum_{l=1}^{m}\left(\left(\sum_{\hat{l}=1}^{n+m}z_{l\hat{l}}^{T}\frac{\partial }{\partial x_{\hat{l}}}f\right)\left(\sum_{l^{\prime }=1}^{n+m}z_{ll^{\prime }}^{T}\left[\sum_{\hat{i},\hat{k}=1}^{n+m}\sum_{k=1}^{n}\frac{\partial }{\partial x_{\hat{i}}}a_{\hat{i}k}\left(\frac{\partial }{\partial x_{l^{\prime }}}a_{k\hat{k}}^{T}\frac{\partial }{\partial x_{\hat{k}}}f\right)\right]\right)\right)\\ & & +\sum_{l=1}^{m}\left(\left(\sum_{\hat{l}=1}^{n+m}z_{l\hat{l}}^{T}\frac{\partial }{\partial x_{\hat{l}}}f\right)\left(\sum_{l^{\prime }=1}^{n+m}z_{ll^{\prime }}^{T}\left[\sum_{\hat{i},\hat{k}=1}^{n+m}\sum_{k=1}^{n}\frac{\partial }{\partial x_{\hat{i}}}a_{\hat{i}k}\left(a_{k\hat{k}}^{T}\frac{\partial }{\partial x_{l^{\prime }}}\frac{\partial }{\partial x_{\hat{k}}}f\right)\right]\right)\right)\\ & & +\sum_{l=1}^{n}\left(\left(\sum_{\hat{l}=1}^{n+m}z_{l\hat{l}}^{T}\frac{\partial }{\partial x_{\hat{l}}}f\right)\left(\sum_{l^{\prime }=1}^{n+m}z_{ll^{\prime }}^{T}\left[\sum_{\hat{i},\hat{k}=1}^{n+m}\sum_{k=1}^{n}\frac{\partial^{2}}{\partial x_{\hat{i}}x_{l^{\prime }}}a_{\hat{i}k}\left(a_{k\hat{k}}^{T}\frac{\partial }{\partial x_{\hat{k}}}f\right)\right]\right)\right). \end{array}$$Subtracting the above two terms, we obtain the following:
$$\begin{array}{ccc} I-\mathrm{II} & = & -\sum_{l=1}^{m}\left(\left(\sum_{\hat{l}=1}^{n+m}z_{l\hat{l}}^{T}\frac{\partial }{\partial x_{\hat{l}}}f\right)\left(\sum_{l^{\prime }=1}^{n+m}z_{ll^{\prime }}^{T}\left[\sum_{\hat{i},\hat{k}=1}^{n+m}\sum_{k=1}^{n}\frac{\partial }{\partial x_{\hat{i}}}a_{\hat{i}k}\left(\frac{\partial }{\partial x_{l^{\prime }}}a_{k\hat{k}}^{T}\frac{\partial }{\partial x_{\hat{k}}}f\right)\right]\right)\right)\\ & & -\sum_{l=1}^{m}\left(\left(\sum_{\hat{l}=1}^{n+m}z_{l\hat{l}}^{T}\frac{\partial }{\partial x_{\hat{l}}}f\right)\left(\sum_{l^{\prime }=1}^{n+m}z_{ll^{\prime }}^{T}\left[\sum_{\hat{i},\hat{k}=1}^{n+m}\sum_{k=1}^{n}\frac{\partial^{2}}{\partial x_{\hat{i}}x_{l^{\prime }}}a_{\hat{i}k}\left(a_{k\hat{k}}^{T}\frac{\partial }{\partial x_{\hat{k}}}f\right)\right]\right)\right)\\ & & +\sum_{l=1}^{m}\left(\sum_{\hat{l}=1}^{n+m}z_{l\hat{l}}^{T}\frac{\partial }{\partial x_{\hat{l}}}f\right)\sum_{\hat{i}=1}^{n+m}\sum_{k=1}^{n}\left(\frac{\partial }{\partial x_{\hat{i}}}a_{\hat{i}k}\sum_{\hat{k}=1}^{n+m}a_{k\hat{k}}^{T}\left(\sum_{l^{\prime }=1}^{n+m}\frac{\partial }{\partial x_{\hat{k}}}z_{ll^{\prime }}^{T}\frac{\partial }{\partial x_{l^{\prime }}}f\right)\right). \end{array}$$Now, we eventually end up with the following formula:
$$\begin{array}{ccc} & & \frac{1}{2}\Delta_{a}\Gamma_{1}^{z}\left(f,f\right)-\Gamma_{1}^{z}\left(\Delta_{a}f,f\right)\\ & = & \frac{1}{2}(a^{T}\nabla \circ (a^{T}\nabla |z^{T}{\nabla f|}^{2}))-{\langle z^{T}\nabla \left(\left(a^{T}\nabla \right)\circ \left(a^{T}\nabla f\right)\right),z^{T}\nabla f\rangle }_{R^{m}}\\ & & -\sum_{l=1}^{m}\left(\left(\sum_{\hat{l}=1}^{n+m}z_{l\hat{l}}^{T}\frac{\partial }{\partial x_{\hat{l}}}f\right)\left(\sum_{l^{\prime }=1}^{n+m}z_{ll^{\prime }}^{T}\left[\sum_{\hat{i},\hat{k}=1}^{n+m}\sum_{k=1}^{n}\frac{\partial }{\partial x_{\hat{i}}}a_{\hat{i}k}\left(\frac{\partial }{\partial x_{l^{\prime }}}a_{k\hat{k}}^{T}\frac{\partial }{\partial x_{\hat{k}}}f\right)\right]\right)\right)\\ & & -\sum_{l=1}^{m}\left(\left(\sum_{\hat{l}=1}^{n+m}z_{l\hat{l}}^{T}\frac{\partial }{\partial x_{\hat{l}}}f\right)\left(\sum_{l^{\prime }=1}^{n+m}z_{ll^{\prime }}^{T}\left[\sum_{\hat{i},\hat{k}=1}^{n+m}\sum_{k=1}^{n}\frac{\partial^{2}}{\partial x_{\hat{i}}x_{l^{\prime }}}a_{\hat{i}k}\left(a_{k\hat{k}}^{T}\frac{\partial }{\partial x_{\hat{k}}}f\right)\right]\right)\right)\\ & & +\sum_{l=1}^{m}\left(\sum_{\hat{l}=1}^{n+m}z_{l\hat{l}}^{T}\frac{\partial }{\partial x_{\hat{l}}}f\right)\sum_{\hat{i}=1}^{n+m}\sum_{k=1}^{n}\left(\frac{\partial }{\partial x_{\hat{i}}}a_{\hat{i}k}\sum_{\hat{k}=1}^{n+m}a_{k\hat{k}}^{T}\left(\sum_{l^{\prime }=1}^{n+m}\frac{\partial }{\partial x_{\hat{k}}}z_{ll^{\prime }}^{T}\frac{\partial }{\partial x_{l^{\prime }}}f\right)\right). \end{array}$$**Step 2:** Computation of $-\frac{1}{2}A\nabla \Gamma_{1}^{z}\left(f,f\right)+\Gamma_{1}^{z}\left(A\nabla f,f\right)$. Now, we compute the last two terms of the above equation, with A=a⊗∇a:
$$\begin{array}{ccc} -\frac{1}{2}A\nabla \Gamma_{1}^{z}\left(f,f\right) & = & -\frac{1}{2}\sum_{\hat{k}=1}^{n+m}A_{\hat{k}}\nabla_{\frac{\partial }{\partial x_{\hat{k}}}}{\langle z^{T}\nabla f,z^{T}\nabla f\rangle }_{R^{m}}\\ & = & -\sum_{\hat{k}=1}^{n+m}{\langle A_{\hat{k}}\left(\nabla_{\frac{\partial }{\partial x_{\hat{k}}}}z^{T}\right)\nabla f,z^{T}\nabla f\rangle }_{R^{m}}-\sum_{\hat{k}=1}^{n+m}{\langle A_{\hat{k}}z^{T}\left(\nabla_{\frac{\partial }{\partial x_{\hat{k}}}}\nabla f\right),z^{T}\nabla f\rangle }_{R^{m}}\\ & = & {\tilde{J}}_{1}+{\tilde{J}}_{2}, \end{array}$$
$$\begin{array}{ccc} \Gamma_{1}^{z}\left(A\nabla f,f\right) & = & {\langle z^{T}\nabla \left(\sum_{\hat{k}=1}^{n+m}A_{\hat{k}}\nabla_{\frac{\partial }{\partial x_{\hat{k}}}}f\right),z^{T}\nabla f\rangle }_{R^{m}}\\ & = & {\langle z^{T}\left(\sum_{\hat{k}=1}^{n+m}A_{\hat{k}}\nabla \nabla_{\frac{\partial }{\partial x_{\hat{k}}}}f\right),z^{T}\nabla f\rangle }_{R^{m}}+{\langle z^{T}\left(\sum_{\hat{k}=1}^{n+m}\nabla A_{\hat{k}}\nabla_{\frac{\partial }{\partial x_{\hat{k}}}}f\right),z^{T}\nabla f\rangle }_{R^{m}}\\ & = & {\tilde{J}}_{3}+{\tilde{J}}_{4}. \end{array}$$It is easy to see that ${\tilde{J}}_{2}+{\tilde{J}}_{3}=0.$ We now expand ${\tilde{J}}_{1}$ and ${\tilde{J}}_{4}$ into local coordinates:
(61)$$\begin{array}{ccc} {\tilde{J}}_{1} & = & -\sum_{l=1}^{m}{\left(z^{T}\nabla f\right)}_{l}\left(\sum_{l^{\prime },\hat{k}=1}^{n+m}\sum_{k=1}^{n}\sum_{k^{\prime }=1}^{n+m}a_{\hat{k}k}\nabla_{\frac{\partial }{\partial x_{k^{\prime }}}}a_{k^{\prime }k}\nabla_{\frac{\partial }{\partial x_{\hat{k}}}}z_{ll^{\prime }}\nabla_{\frac{\partial }{\partial x_{l^{\prime }}}}f\right), \end{array}$$
(62)$$\begin{array}{ccc} {\tilde{J}}_{4} & = & \sum_{l=1}^{m}{\left(z^{T}\nabla f\right)}_{l}\left(\sum_{l^{\prime }=1}^{n+m}z_{ll^{\prime }}^{T}\left(\sum_{\hat{k}=1}^{n+m}\nabla_{\frac{\partial }{\partial x_{l^{\prime }}}}\left(\sum_{k=1}^{n}\sum_{k^{\prime }=1}^{n+m}a_{\hat{k}k}\nabla_{\frac{\partial }{\partial x_{k^{\prime }}}}a_{k^{\prime }k}\right)\nabla_{\frac{\partial }{\partial x_{\hat{k}}}}f\right)\right)\\ & = & \sum_{l=1}^{m}{\left(z^{T}\nabla f\right)}_{l}\left(\sum_{k=1}^{n}\sum_{l^{\prime }=1}^{n+m}\sum_{\hat{k},k^{\prime }=1}^{n+m}z_{ll^{\prime }}^{T}\nabla_{\frac{\partial }{\partial x_{l^{\prime }}}}a_{\hat{k}k}\nabla_{\frac{\partial }{\partial x_{k^{\prime }}}}a_{k^{\prime }k}\nabla_{\frac{\partial }{\partial x_{\hat{k}}}}f\right)\\ & & +\sum_{l=1}^{m}{\left(z^{T}\nabla f\right)}_{l}\left(\sum_{k=1}^{n}\sum_{l^{\prime }=1}^{n+m}\sum_{\hat{k},k^{\prime }=1}^{n+m}z_{ll^{\prime }}^{T}a_{\hat{k}k}\left(\nabla_{\frac{\partial }{\partial x_{l^{\prime }}}}\nabla_{\frac{\partial }{\partial x_{k^{\prime }}}}a_{k^{\prime }k}\right)\nabla_{\frac{\partial }{\partial x_{\hat{k}}}}f\right). \end{array}$$Combining the above two steps, we thus obtain
$$\begin{array}{c} \frac{1}{2}\tilde{L}\Gamma_{1}^{z}\left(f,f\right)-\Gamma_{1}^{z}\left(\tilde{L}f,f\right)=\frac{1}{2}(a^{T}\nabla \circ (a^{T}\nabla |z^{T}{\nabla f|}^{2}))-{\langle z^{T}\nabla \left(\left(a^{T}\nabla \right)\circ \left(a^{T}\nabla f\right)\right),z^{T}\nabla f\rangle }_{R^{m}}. \end{array}$$□

**Lemma** **19.**

$$\begin{array}{ccc} & & \frac{1}{2}(a^{T}\nabla \circ (a^{T}\nabla |z^{T}{\nabla f|}^{2}))-{\langle z^{T}\nabla \left(\left(a^{T}\nabla \right)\circ \left(a^{T}\nabla f\right)\right),z^{T}\nabla f\rangle }_{R^{m}}\\ & = & X^{T}P^{T}PX+2E^{T}PX+2F^{T}X+E^{T}E+R_{z}\left(\nabla f,\nabla f\right). \end{array}$$



**Proof.** We expand the two terms in Lemma 19.
$$\begin{array}{ccc} & & \frac{1}{2}(a^{T}\nabla \circ (a^{T}\nabla |z^{T}{\nabla f|}^{2}))\\ & = & \frac{1}{2}\sum_{i=1}^{n}\sum_{k=1}^{m}{\left(a^{T}\nabla \right)}_{i}{\left(a^{T}\nabla \right)}_{i}{\left|{\left(z^{T}\nabla \right)}_{k}f\right|}^{2}\\ & = & \sum_{i=1}^{n}\sum_{k=1}^{m}{\left(a^{T}\nabla \right)}_{i}{\langle {\left(a^{T}\nabla \right)}_{i}{\left(z^{T}\nabla \right)}_{k}f,{\left(z^{T}\nabla \right)}_{k}f\rangle }_{R^{m}}\\ & = & \sum_{i=1}^{n}\sum_{k=1}^{m}{\langle {\left(a^{T}\nabla \right)}_{i}{\left(z^{T}\nabla \right)}_{k}f,{\left(a^{T}\nabla \right)}_{i}{\left(z^{T}\nabla \right)}_{k}f\rangle }_{R^{m}}\cdots {\tilde{T}}_{1}\\ & & +\sum_{i=1}^{n}\sum_{k=1}^{m}{\langle \left(a_{ii^{\prime }}^{T}\frac{\partial }{\partial x_{i^{\prime }}}\right)\left(a_{i\hat{i}}^{T}\frac{\partial }{\partial x_{\hat{i}}}\right)\left(z_{k\hat{k}}^{T}\frac{\partial }{\partial x_{\hat{k}}}\right)f,{\left(z^{T}\nabla \right)}_{k}f\rangle }_{R^{m}}\cdots {\tilde{R}}_{1}. \end{array}$$
$$\begin{array}{ccc} & & {\langle z^{T}\nabla \left(\left[\left(a^{T}\nabla \right)\circ \left(a^{T}\nabla f\right)\right]\right),z^{T}\nabla f\rangle }_{R^{m}}\\ & = & \sum_{i=1}^{n}\sum_{k=1}^{m}{\langle {\left(z^{T}\nabla \right)}_{k}\left[{\left(a^{T}\nabla \right)}_{i}{\left(a^{T}\nabla \right)}_{i}f\right],{\left(z^{T}\nabla \right)}_{k}f\rangle }_{R^{m}}\\ & = & \sum_{i=1}^{n}\sum_{k=1}^{m}{\langle \left(z_{k\hat{k}}^{T}\frac{\partial }{\partial x_{\hat{k}}}\right)\left[\left(a_{ii^{\prime }}^{T}\frac{\partial }{\partial x_{i^{\prime }}}\right)\left(a_{i\hat{i}}^{T}\frac{\partial }{\partial x_{\hat{i}}}\right)f\right],{\left(z^{T}\nabla \right)}_{k}f\rangle }_{R^{m}}\cdots {\tilde{R}}_{2}. \end{array}$$Next, we expand ${\tilde{R}}_{1}$ and ${\tilde{R}}_{2}$ completely and obtain the following:
$$\begin{array}{ccc} {\tilde{R}}_{1} & = & \sum_{i=1}^{n}\sum_{k=1}^{m}\sum_{i^{\prime },\hat{i},\hat{k}=1}^{n+m}{\langle \left(a_{ii^{\prime }}^{T}\frac{\partial }{\partial x_{i^{\prime }}}\right)\left(a_{i\hat{i}}^{T}\frac{\partial }{\partial x_{\hat{i}}}\right)\left(z_{k\hat{k}}^{T}\frac{\partial f}{\partial x_{\hat{k}}}\right),{\left(z^{T}\nabla \right)}_{k}f\rangle }_{R^{m}}\\ & = & \sum_{i=1}^{n}\sum_{k=1}^{m}\sum_{i^{\prime },\hat{i},\hat{k}=1}^{n+m}{\langle a_{ii^{\prime }}^{T}\left(\frac{\partial a_{i\hat{i}}^{T}}{\partial x_{i^{\prime }}}\frac{\partial z_{k\hat{k}}^{T}}{\partial x_{\hat{i}}}\frac{\partial f}{\partial x_{\hat{k}}}\right),{\left(z^{T}\nabla \right)}_{k}f\rangle }_{R^{m}}\cdots {\tilde{R}}_{1}^{1}\\ & & +\sum_{i=1}^{n}\sum_{k=1}^{m}\sum_{i^{\prime },\hat{i},\hat{k}=1}^{n+m}{\langle a_{ii^{\prime }}^{T}a_{i\hat{i}}^{T}\left(\frac{\partial }{\partial x_{i^{\prime }}}\frac{\partial z_{k\hat{k}}^{T}}{\partial x_{\hat{i}}}\right)\left(\frac{\partial f}{\partial x_{\hat{k}}}\right),{\left(z^{T}\nabla \right)}_{k}f\rangle }_{R^{m}}\cdots {\tilde{R}}_{1}^{2}\\ & & +\sum_{i=1}^{n}\sum_{k=1}^{m}\sum_{i^{\prime },\hat{i},\hat{k}=1}^{n+m}{\langle a_{ii^{\prime }}^{T}a_{i\hat{i}}^{T}\left(\frac{\partial z_{k\hat{k}}^{T}}{\partial x_{\hat{i}}}\right)\left(\frac{\partial }{\partial x_{i^{\prime }}}\frac{\partial f}{\partial x_{\hat{k}}}\right),{\left(z^{T}\nabla \right)}_{k}f\rangle }_{R^{m}}\cdots {\tilde{R}}_{1}^{3}\\ & & +\sum_{i=1}^{n}\sum_{k=1}^{m}\sum_{i^{\prime },\hat{i},\hat{k}=1}^{n+m}{\langle \left(a_{ii^{\prime }}^{T}\right)\left(\left(\frac{\partial }{\partial x_{i^{\prime }}}a_{i\hat{i}}^{T}\right)z_{k\hat{k}}^{T}\frac{\partial }{\partial x_{\hat{i}}}\frac{\partial f}{\partial x_{\hat{k}}}\right),{\left(z^{T}\nabla \right)}_{k}f\rangle }_{R^{m}}\cdots {\tilde{R}}_{1}^{4}\\ & & +\sum_{i=1}^{n}\sum_{k=1}^{m}\sum_{i^{\prime },\hat{i},\hat{k}=1}^{n+m}\langle a_{ii^{\prime }}^{T}a_{i\hat{i}}^{T}\left(\frac{\partial }{\partial x_{i^{\prime }}}z_{k\hat{k}}^{T}\right)\frac{\partial }{\partial x_{\hat{i}}}\frac{\partial f}{\partial x_{\hat{k}}}),{\left(z^{T}\nabla \right)}_{k}{f\rangle }_{R^{m}}\cdots {\tilde{R}}_{1}^{5}\\ & & +\sum_{i=1}^{n}\sum_{k=1}^{m}\sum_{i^{\prime },\hat{i},\hat{k}=1}^{n+m}{\langle a_{ii^{\prime }}^{T}a_{i\hat{i}}^{T}z_{k\hat{k}}^{T}\left(\frac{\partial }{\partial x_{i^{\prime }}}\frac{\partial }{\partial x_{\hat{i}}}\frac{\partial f}{\partial x_{\hat{k}}}\right),{\left(z^{T}\nabla \right)}_{k}f\rangle }_{R^{m}}\cdots {\tilde{R}}_{1}^{6} \end{array}$$
$$\begin{array}{ccc} {\tilde{R}}_{2} & = & \sum_{i=1}^{n}\sum_{k=1}^{m}\sum_{i^{\prime },\hat{i},\hat{k}=1}^{n+m}{\langle \left(z_{k\hat{k}}^{T}\frac{\partial }{\partial x_{\hat{k}}}\right)\left[\left(a_{ii^{\prime }}^{T}\frac{\partial }{\partial x_{i^{\prime }}}\right)\left(a_{i\hat{i}}^{T}\frac{\partial f}{\partial x_{\hat{i}}}\right)\right],{\left(z^{T}\nabla \right)}_{k}f\rangle }_{R^{m}}\\ & = & \sum_{i=1}^{n}\sum_{k=1}^{m}\sum_{i^{\prime },\hat{i},\hat{k}=1}^{n+m}\langle z_{k\hat{k}}^{T}\frac{\partial a_{ii^{\prime }}^{T}}{\partial x_{\hat{k}}}\frac{\partial a_{i\hat{i}}^{T}}{\partial x_{i^{\prime }}}\frac{\partial f}{\partial x_{\hat{i}}}),{\left(z^{T}\nabla \right)}_{k}{f\rangle }_{R^{m}}\cdots {\tilde{R}}_{2}^{1}\\ & & +\sum_{i=1}^{n}\sum_{k=1}^{m}\sum_{i^{\prime },\hat{i},\hat{k}=1}^{n+m}{\langle z_{k\hat{k}}^{T}a_{ii^{\prime }}^{T}\left(\frac{\partial }{\partial x_{\hat{k}}}\frac{\partial a_{i\hat{i}}^{T}}{\partial x_{i^{\prime }}}\right)\frac{\partial f}{\partial x_{\hat{i}}},{\left(z^{T}\nabla \right)}_{k}f\rangle }_{R^{m}}\cdots {\tilde{R}}_{2}^{2}\\ & & +\sum_{i=1}^{n}\sum_{k=1}^{m}\sum_{i^{\prime },\hat{i},\hat{k}=1}^{n+m}{\langle z_{k\hat{k}}^{T}a_{ii^{\prime }}^{T}\frac{\partial a_{i\hat{i}}^{T}}{\partial x_{i^{\prime }}}\left(\frac{\partial }{\partial x_{\hat{k}}}\frac{\partial f}{\partial x_{\hat{i}}}\right),{\left(z^{T}\nabla \right)}_{k}f\rangle }_{R^{m}}\cdots {\tilde{R}}_{2}^{3}={\tilde{R}}_{1}^{4}\\ & & +\sum_{i=1}^{n}\sum_{k=1}^{m}\sum_{i^{\prime },\hat{i},\hat{k}=1}^{n+m}{\langle z_{k\hat{k}}^{T}\frac{\partial a_{ii^{\prime }}^{T}}{\partial x_{\hat{k}}}a_{i\hat{i}}^{T}\left(\frac{\partial }{\partial x_{i^{\prime }}}\frac{\partial f}{\partial x_{\hat{i}}}\right),{\left(z^{T}\nabla \right)}_{k}f\rangle }_{R^{m}}\cdots {\tilde{R}}_{2}^{4}\\ & & +\sum_{i=1}^{n}\sum_{k=1}^{m}\sum_{i^{\prime },\hat{i},\hat{k}=1}^{n+m}{\langle z_{k\hat{k}}^{T}a_{ii^{\prime }}^{T}\frac{\partial a_{i\hat{i}}^{T}}{\partial x_{\hat{k}}}\left(\frac{\partial }{\partial x_{i^{\prime }}}\frac{\partial f}{\partial x_{\hat{i}}}\right),{\left(z^{T}\nabla \right)}_{k}f\rangle }_{R^{m}}\cdots {\tilde{R}}_{2}^{5}\\ & & +\sum_{i=1}^{n}\sum_{k=1}^{m}\sum_{i^{\prime },\hat{i},\hat{k}=1}^{n+m}{\langle z_{k\hat{k}}^{T}a_{ii^{\prime }}^{T}a_{i\hat{i}}^{T}\left(\frac{\partial }{\partial x_{\hat{k}}}\frac{\partial }{\partial x_{i^{\prime }}}\frac{\partial f}{\partial x_{\hat{i}}}\right),{\left(z^{T}\nabla \right)}_{k}f\rangle }_{R^{m}}\cdots {\tilde{R}}_{2}^{6}={\tilde{R}}_{1}^{6} \end{array}$$Our next step is to complete the squares for all the above terms. We look at term ${\tilde{T}}_{1}$ first.
$$\begin{array}{ccc} {\tilde{T}}_{1} & = & \sum_{i=1}^{n}\sum_{k=1}^{m}\langle \sum_{\hat{i},\hat{k}=1}^{n+m}a_{i\hat{i}}^{T}z_{k\hat{k}}^{T}\frac{\partial^{2}f}{\partial x_{\hat{i}}\partial x_{\hat{k}}}+\sum_{\hat{i},\hat{k}=1}^{n+m}a_{i\hat{i}}^{T}\frac{\partial z_{k\hat{k}}^{T}}{\partial x_{\hat{i}}}\frac{\partial f}{\partial x_{\hat{k}}},\\ & & \;\;\;\;\sum_{i^{\prime },k^{\prime }=1}^{n+m}a_{ii^{\prime }}^{T}z_{kk^{\prime }}^{T}\frac{\partial^{2}f}{\partial x_{i^{\prime }}\partial x_{k^{\prime }}}+\sum_{i^{\prime },k^{\prime }=1}^{n+m}a_{ii^{\prime }}^{T}\frac{\partial z_{kk^{\prime }}^{T}}{\partial x_{i^{\prime }}}\frac{\partial f}{\partial x_{k^{\prime }}}\rangle \\ & = & \sum_{i=1}^{n}\sum_{k=1}^{m}\sum_{k=1}^{m}\langle \sum_{\hat{i},\hat{k}=1}^{n+m}a_{i\hat{i}}^{T}z_{k\hat{k}}^{T}\frac{\partial^{2}f}{\partial x_{\hat{i}}\partial x_{\hat{k}}},\sum_{i^{\prime },k^{\prime }=1}^{n+m}a_{ii^{\prime }}^{T}z_{kk^{\prime }}^{T}\frac{\partial^{2}f}{\partial x_{i^{\prime }}\partial x_{k^{\prime }}}\rangle \cdots {\tilde{T}}_{1a} \end{array}$$
$$\begin{array}{ccc} & & +\sum_{i=1}^{n}\sum_{k=1}^{m}\langle \sum_{\hat{i},\hat{k}=1}^{n+m}a_{i\hat{i}}^{T}z_{k\hat{k}}^{T}\frac{\partial^{2}f}{\partial x_{\hat{i}}\partial x_{\hat{k}}},\sum_{i^{\prime },k^{\prime }=1}^{n+m}a_{ii^{\prime }}^{T}\frac{\partial z_{kk^{\prime }}^{T}}{\partial x_{i^{\prime }}}\frac{\partial f}{\partial x_{k^{\prime }}}\rangle \cdots {\tilde{T}}_{1b}\\ & & +\sum_{i=1}^{n}\sum_{k=1}^{m}\langle \sum_{\hat{i},\hat{k}=1}^{n+m}a_{i\hat{i}}^{T}\frac{\partial z_{k\hat{k}}^{T}}{\partial x_{\hat{i}}}\frac{\partial f}{\partial x_{\hat{k}}},\sum_{i^{\prime },k^{\prime }=1}^{n+m}a_{ii^{\prime }}^{T}z_{kk^{\prime }}^{T}\frac{\partial^{2}f}{\partial x_{i^{\prime }}\partial x_{k^{\prime }}}\rangle \cdots {\tilde{T}}_{1c}\\ & & +\sum_{i=1}^{n}\sum_{k=1}^{m}\langle \sum_{\hat{i},\hat{k}=1}^{n+m}a_{i\hat{i}}^{T}\frac{\partial z_{k\hat{k}}^{T}}{\partial x_{\hat{i}}}\frac{\partial f}{\partial x_{\hat{k}}},\sum_{i^{\prime },k^{\prime }=1}^{n+m}a_{ii^{\prime }}^{T}\frac{\partial z_{kk^{\prime }}^{T}}{\partial x_{i^{\prime }}}\frac{\partial f}{\partial x_{k^{\prime }}}\rangle \cdots {\tilde{T}}_{1d}. \end{array}$$The terms ${\tilde{T}}_{1b}={\tilde{T}}_{1c}$, ${\tilde{R}}_{1}^{3}={\tilde{R}}_{1}^{5}$, and ${\tilde{R}}_{2}^{5}={\tilde{R}}_{2}^{4}$ play the role of crossing terms inside the complete squares. In particular, for convenience, we changed the index inside the sum of ${\tilde{R}}_{1}^{3}$ and ${\tilde{R}}_{2}^{5}$, switched $i^{\prime },\hat{i}$ for ${\tilde{R}}_{1}^{3}$, and switched $i^{\prime },\hat{k}$ for ${\tilde{R}}_{2}^{5}$, then we obtain the following.
$$\begin{array}{ccc} 2{\tilde{R}}_{1}^{3} & = & 2\sum_{i=1}^{n}\sum_{k=1}^{m}\sum_{i^{\prime },\hat{i},\hat{k}=1}^{n+m}{\langle a_{i\hat{i}}^{T}a_{ii^{\prime }}^{T}\left(\frac{\partial z_{k\hat{k}}^{T}}{\partial x_{i^{\prime }}}\right)\left(\frac{\partial }{\partial x_{\hat{i}}}\frac{\partial f}{\partial x_{\hat{k}}}\right),{\left(z^{T}\nabla \right)}_{k}f\rangle }_{R^{n}}\\ & = & 2\sum_{i=1}^{n}\sum_{k=1}^{m}\sum_{\hat{i},\hat{k}=1}^{n+m}\sum_{i^{\prime },l=1}^{n+m}\left(a_{i\hat{i}}^{T}a_{ii^{\prime }}^{T}\left(\frac{\partial z_{k\hat{k}}^{T}}{\partial x_{i^{\prime }}}\right)\left(\frac{\partial }{\partial x_{\hat{i}}}\frac{\partial f}{\partial x_{\hat{k}}}\right)z_{kl}^{T}\frac{\partial f}{\partial x_{l}}\right)\\ -2{\tilde{R}}_{2}^{5} & = & -2\sum_{i=1}^{n}\sum_{k=1}^{m}\sum_{i^{\prime },\hat{i},\hat{k}=1}^{n+m}\langle z_{ki^{\prime }}^{T}a_{i\hat{k}}^{T}\frac{\partial a_{i\hat{i}}^{T}}{\partial x_{i^{\prime }}}\left(\frac{\partial }{\partial x_{\hat{k}}}\frac{\partial f}{\partial x_{\hat{i}}}\right),{\left(z^{T}\nabla \right)}_{k}f\rangle \\ & = & -2\sum_{i=1}^{n}\sum_{k=1}^{m}\sum_{\hat{i},\hat{k}=1}^{n+m}\sum_{i^{\prime },l=1}^{n+m}\left(z_{ki^{\prime }}^{T}a_{i\hat{k}}^{T}\frac{\partial a_{i\hat{i}}^{T}}{\partial x_{i^{\prime }}}\left(\frac{\partial }{\partial x_{\hat{k}}}\frac{\partial f}{\partial x_{\hat{i}}}\right)z_{kl}^{T}\frac{\partial f}{\partial x_{l}}\right) \end{array}$$We denote
(63)$$\begin{array}{ccc} \sum_{\hat{i},\hat{k}=1}^{n+m}a_{i\hat{i}}^{T}z_{k\hat{k}}^{T}\frac{\partial^{2}f}{\partial x_{\hat{i}}\partial x_{\hat{k}}} & = & \omega_{ik}. \end{array}$$The above Equality (Equation 63) can be represented in the following matrix form:
$$\begin{array}{c} P_{\left(n*m\right)\times {\left(n+m\right)}^{2}}X_{{\left(n+m\right)}^{2}\times 1}={\left(\omega_{11},\cdots ,\omega_{ik},\cdots ,\omega_{nm}\right)}_{(n*m)\times 1}^{T} \end{array}$$
where *P* and *X* are defined in (Equation 13) and (Equation 19). Now, we can represent term ${\tilde{T}}_{1a}$ as $\sum_{i=1}^{n}\sum_{k=1}^{m}\omega_{ik}^{2}=\omega^{T}\omega ={\left(PX\right)}^{T}PX=X^{T}P^{T}PX$. Next, we want to represent ${\tilde{R}}_{1}^{3}$ and ${\tilde{R}}_{2}^{5}$ in the following form in terms of vector *X*:
$$\begin{array}{ccc} & & 2{\tilde{R}}_{1}^{3}-2{\tilde{R}}_{2}^{5}\\ & = & 2\sum_{i,k=1}^{n}\sum_{i^{\prime },\hat{i},\hat{k}=1}^{n+m}{\langle a_{i\hat{i}}^{T}a_{ii^{\prime }}^{T}\left(\frac{\partial z_{k\hat{k}}^{T}}{\partial x_{i^{\prime }}}\right)\left(\frac{\partial }{\partial x_{\hat{i}}}\frac{\partial f}{\partial x_{\hat{k}}}\right),{\left(z^{T}\nabla \right)}_{k}f\rangle }_{R^{n}}\\ & & -2\sum_{i,k=1}^{n}\sum_{i^{\prime },\hat{i},\hat{k}=1}^{n+m}\langle z_{ki^{\prime }}^{T}a_{i\hat{k}}^{T}\frac{\partial a_{i\hat{i}}^{T}}{\partial x_{i^{\prime }}}\left(\frac{\partial }{\partial x_{\hat{k}}}\frac{\partial f}{\partial x_{\hat{i}}}\right),{\left(z^{T}\nabla \right)}_{k}f\rangle \\ & = & 2\sum_{\hat{i},\hat{k}=1}^{n+m}\left[\sum_{i,k=1}^{n}\sum_{i^{\prime }=1}^{n+m}\left(\langle a_{i\hat{i}}^{T}a_{ii^{\prime }}^{T}\left(\frac{\partial z_{k\hat{k}}^{T}}{\partial x_{i^{\prime }}}\right),{\left(z^{T}\nabla \right)}_{k}f\rangle -\langle z_{ki^{\prime }}^{T}a_{i\hat{k}}^{T}\frac{\partial a_{i\hat{i}}^{T}}{\partial x_{i^{\prime }}},{\left(z^{T}\nabla \right)}_{k}f\rangle \right)\right]\left(\frac{\partial }{\partial x_{\hat{i}}}\frac{\partial f}{\partial x_{\hat{k}}}\right)\\ & = & 2F^{T}X, \end{array}$$
where *F* is defined in (Equation 16). Similarly, we can represent ${\tilde{T}}_{1b}={\tilde{T}}_{1c}$ by *X*:
$$\begin{array}{ccc} {\tilde{T}}_{1b}={\tilde{T}}_{1c} & = & \sum_{i,k=1}^{n}\langle \sum_{\hat{i},\hat{k}=1}^{n+m}a_{i\hat{i}}^{T}\frac{\partial z_{k\hat{k}}^{T}}{\partial x_{\hat{i}}}\frac{\partial f}{\partial x_{\hat{k}}},\sum_{i^{\prime },k^{\prime }=1}^{n+m}a_{ii^{\prime }}^{T}z_{kk^{\prime }}^{T}\frac{\partial^{2}f}{\partial x_{i^{\prime }}\partial x_{k^{\prime }}}\rangle =E^{T}PX \end{array}$$
where *E* is defined in (Equation 17). We thus have the following form:
$$\begin{array}{c} {\tilde{T}}_{1}+2{\tilde{R}}_{1}^{3}-2{\tilde{R}}_{2}^{5}=X^{T}P^{T}PX+2E^{T}PX+2F^{T}X+E^{T}E \end{array}$$Taking into account the fact that $R_{1}^{6}-R_{2}^{6}=0$ and $R_{1}^{4}-R_{2}^{3}=0$, we have
$$\begin{array}{c} {\tilde{T}}_{1}+{\tilde{R}}_{1}-{\tilde{R}}_{2}={\tilde{T}}_{1}+2{\tilde{R}}_{1}^{3}-2{\tilde{R}}_{2}^{5}+{\tilde{R}}_{1}^{1}+{\tilde{R}}_{1}^{2}-{\tilde{R}}_{2}^{1}-{\tilde{R}}_{2}^{2}, \end{array}$$
which completes the proof. □

## 6. Further Discussions on Other Inequalities

In this section, we apply the generalized Gamma calculus to study the entropic inequality for the semi-group $P_{t}$ associated with the drift–diffusion process. With a little abuse of notation, we denote the generator of the semi-group $P_{t}$ as $\frac{1}{2}L$ instead of *L*, and we denote $X_{t}$ as the corresponding diffusion process.

**Definition** **5.**
*We define the semigroup $P_{t}=e^{\frac{1}{2}tL}$, where L is invariant with respect to the invariant measure dμ=π(x)dx. We denote $P_{t}f\left(x\right)=E\left(f\left(X_{t}\right)\right)$, and*

$$\begin{array}{ccc} E(f\left(X_{t}\right)) & = & \int_{R^{n+m}}f\left(y\right)p\left(t,x,y\right)d\mu \left(y\right)=\int_{R^{n+m}}f\left(y\right)\rho \left(t,x,y\right)dy, \end{array}$$

*where the infinitesimal generator of this process $X_{t}$ is $\frac{1}{2}L$, and we denote ρ(t,·,·) as the product of the transition kernel p(t,·,·) and the volume measure π.*


**Remark** **14.**
*Following the standard treatment as in [2] (Section 5), whenever we consider the differentiating operation on $P_{t}f$, we shall always consider $P_{t}f_{\varepsilon }$ first with $f_{\varepsilon }=f+\varepsilon$, for ∀ε>0. Then, we take the limit as ε→0. Throughout this section, we directly use $P_{t}f$ instead of $P_{t}f_{\varepsilon }$ for convenience.*


**Remark** **15.**
*In the standard sub-Riemannian setting, the semi-groups are in general defined with respect to the invariant measure dμ(y). In this paper, we formulate the semi-group and the transition kernel with respect to the Lebesgue measure dy.*


Following the framework in [2], we also need the following assumption, which is necessary to rigorously justify the computations on functionals of the heat semigroup.

**Assumption** **2.**
*The semigroup $P_{t}$ is stochastically complete, that is, for t≥0, $P_{t}1=1$ and for any T>0 and $f\in C^{\infty }\left(R^{n+m}\right)$ with compact support, we assume that*

(64)
$$\begin{array}{c} \underset{t\in [0,T]}{\sup}∥\Gamma \left(P_{t}f\right){∥}_{\infty }+{∥\Gamma_{1}^{z}\left(P_{t}f\right)∥}_{\infty }<+\infty . \end{array}$$



We believe that the above Assumption 2 should follow from the the assumption $R\geq \kappa (\Gamma_{1}+\Gamma_{1}^{z})$ if we assume the appropriate lower bound κ. We leave this for further studies. Related gradient estimates are presented in order below. For the infinitesimal generator $\frac{1}{2}L$ associated with linear semi-group $P_{t}$, we have the following property.

**Proposition** **15.**
*For all smooth function f, we have:*


*$P_{0}=Id$;*

*For all functions $f\in C_{b}\left(R^{n+m}\right)$, the map $t\mapsto P_{t}f$ is continuous from $R^{+}$ to $L^{2}\left(d\mu \right)$;*

*For all s,t≥0, one has $P_{t}\circ P_{s}=P_{t+s}$;*

*$\forall x\in R^{n+m}$, ∀t≥0,$\frac{\partial }{\partial_{t}}P_{t}f\left(x\right)=\frac{1}{2}L\left(P_{t}f\right)\left(x\right)=\frac{1}{2}P_{t}\left(Lf\right)\left(x\right)$.*



Next, we present the entropic inequality under Assumption 1. We follow closely the framework introduced in [2] and define the following two functionals:$$\begin{array}{ccc} \phi_{a}\left(x,t\right) & =P_{T-t}f\Gamma_{1}\left(\log P_{T-t}f\right)\left(x\right),\;\mathrm{and}\;\phi_{z}\left(x,t\right) & =P_{T-t}f\Gamma_{1}^{z}\left(\log P_{T-t}f\right)\left(x\right). \end{array}$$

**Lemma** **20.**
*We have the following relation:*

(65)
$$\begin{array}{ccc} \frac{1}{2}L\phi_{a}+\frac{\partial }{\partial t}\phi_{a} & = & \left(P_{T-t}f\right)\left(x\right)\Gamma_{2}\left(\log P_{T-t}f,\log P_{T-t}f\right)\left(x\right),\\ \frac{1}{2}L\phi_{z}+\frac{\partial }{\partial t}\phi_{z} & = & \left(P_{T-t}f\right)\left(x\right)\Gamma_{2}^{z}\left(\log P_{T-t}f,\log P_{T-t}f\right)\left(x\right)\\ & & +\left(P_{T-t}f\right)\left(x\right)\Gamma_{1}\left(\log P_{T-t}f,\Gamma_{1}^{z}\left(P_{T-t}f,P_{T-t}f\right)\right)\left(x\right) \end{array}$$


(66)
$$\begin{array}{ccc} & & -\left(P_{T-t}f\right)\left(x\right)\Gamma_{1}^{z}\left(\log P_{T-t}f,\Gamma_{1}\left(P_{T-t}f,P_{T-t}f\right)\right)\left(x\right). \end{array}$$



**Proof.** Denote $g\left(t,x\right)=P_{T-t}f\left(x\right)=\int \rho \left(t,x,\tilde{x}\right)f\left(\tilde{x}\right)d\tilde{x}$, and we have the following relation:
$$\begin{array}{c} L\left(\log g\right)=-\frac{\Gamma_{1}\left(g,g\right)}{{\left(g\right)}^{2}}-2\frac{\partial_{t}g}{g}. \end{array}$$By direct computation, one obtains
$$\begin{array}{ccc} \partial_{t}\phi_{a} & = & \partial_{t}g\Gamma_{1}\left(\log g,\log g\right)+2g{\langle a^{T}\nabla \log g,a^{T}\nabla \left(\frac{\partial_{t}g}{g}\right)\rangle }_{R^{n}}\\ & = & -\frac{1}{2}Lg\Gamma_{1}\left(\log g,\log g\right)-g\Gamma_{1}\left(\log g,L\log g\right)-g\Gamma_{1}\left(\log g,\Gamma_{1}\left(\log g,\log g\right)\right),\\ \frac{1}{2}L\phi_{a} & = & \frac{1}{2}Lg\Gamma_{1}\left(\log g,\log g\right)+\frac{1}{2}gL\Gamma_{1}\left(\log g,\log g\right)+\Gamma_{1}\left(g,\Gamma_{1}\left(\log g,\log g\right)\right), \end{array}$$
where we have $\Gamma_{1}\left(g,\Gamma_{1}\left(\log g,\log g\right)\right)=g\Gamma_{1}\left(\log g,\Gamma_{1}\left(\log g,\log g\right)\right)$; thus, (Equation 66) is proven. Similarly, we obtain the following for $\phi_{z}$:
$$\begin{array}{ccc} \partial_{t}\phi_{z} & = & \partial_{t}g\Gamma_{1}^{z}\left(\log g,\log g\right)+2g\langle z^{T}\nabla \log g,z^{T}\nabla \left(\frac{\partial_{t}g}{g}\right)\rangle {}_{R^{m}}\\ & = & -\frac{1}{2}Lg\Gamma_{1}^{z}\left(\log g,\log g\right)-g\Gamma_{1}^{z}\left(\log g,L\log g\right)-g\Gamma_{1}^{z}\left(\log g,\Gamma_{1}\left(\log g,\log g\right)\right),\\ \frac{1}{2}L\phi_{z} & = & \frac{1}{2}Lg\Gamma_{1}^{z}\left(\log g,\log g\right)+\frac{1}{2}gL\Gamma_{1}^{z}\left(\log g,\log g\right)+\Gamma_{1}\left(g,\Gamma_{1}^{z}\left(\log g,\log g\right)\right). \end{array}$$The proof then follows. □

Now, we are ready to present the following important lemma, which prepares us to prove the new entropy inequality without the assumption:$$\Gamma_{1}\left(\log P_{T-t}f,\Gamma_{1}^{z}\left(P_{T-t}f,P_{T-t}f\right)\right)\left(x\right)=\Gamma_{1}^{z}\left(\log P_{T-t}f,\Gamma_{1}\left(P_{T-t}f,P_{T-t}f\right)\right)\left(x\right).$$

**Lemma** **21.**
*For any 0<s<T, we denote ρ(s,x,y)=p(s,x,y)π(y) as the transition kernel of diffusion process $X_{s}^{x}$ starting at x defined in Definition 5, and the following equality is satisfied:*

$$\begin{array}{ccc} & & E[g\Gamma_{1}\left(\log g,\Gamma_{1}^{z}\left(\log g,\log g\right)\right)-g\Gamma_{1}^{z}\left(\log g,\Gamma_{1}\left(\log g,\log g\right)\right)]\\ & = & \int \frac{\nabla \cdot (\rho \left(s,x,y\right)zz^{T}\Gamma_{\nabla (aa^{T})}\left(\log g\left(s,y\right),\log g\left(s,y\right)\right))}{\rho (s,x,y)}g\left(s,y\right)\rho \left(s,x,y\right)dy\\ & & -\int \frac{\nabla \cdot (\rho \left(s,x,y\right)aa^{T}\Gamma_{\nabla (zz^{T})}\left(\log g\left(s,y\right),\log g\left(s,y\right)\right))}{\rho (s,x,y)}g\left(s,y\right)\rho \left(s,x,y\right)dy. \end{array}$$


*Here, we denote $g\left(s,y\right)=P_{T-s}f\left(y\right)=\int \rho \left(s,y,\tilde{y}\right)f\left(\tilde{y}\right)d\tilde{y}$ and*

$$\begin{array}{ccc} & & E[g\Gamma_{1}\left(\log g,\Gamma_{1}^{z}\left(\log g,\log g\right)\right)]\\ & = & E[g\left(s,X_{s}\right)\Gamma_{1}\left(\log g\left(s,X_{s}^{x}\right),\Gamma_{1}^{z}\left(\log g\left(s,X_{s}^{x}\right),\log g\left(s,X_{s}^{x}\right)\right)\right)]\\ & = & \int g\left(s,y\right)\Gamma_{1}\left(\log g\left(s,y\right),\Gamma_{1}^{z}\left(\log g\left(s,y\right),\log g\left(s,y\right)\right)\right)\rho \left(s,x,y\right)dy. \end{array}$$



**Proof.** We first expand in the following integral form.
$$\begin{array}{ccc} & & E[g\Gamma_{1}\left(\log g,\Gamma_{1}^{z}\left(\log g,\log g\right)\right)-g\Gamma_{1}^{z}\left(\log g,\Gamma_{1}\left(\log g,\log g\right)\right)]\\ & = & \int g\left(s,y\right)\Gamma_{1}\left(\log g\left(s,y\right),\Gamma_{1}^{z}\left(\log g\left(s,y\right),\log g\left(s,y\right)\right)\right)\rho \left(s,x,y\right)dy\\ & & -\int g\left(s,y\right)\Gamma_{1}^{z}\left(\log g\left(s,y\right),\Gamma_{1}\left(\log g\left(s,y\right),\log g\left(s,y\right)\right)\right)\rho \left(s,x,y\right)dy. \end{array}$$We skip x,y,s for simplicity. Take logg=h.**Claim 1**:
$$\begin{array}{ccc} & & \int \Gamma_{1}\left(h,\Gamma_{1}^{z}\left(h,h\right)\right)\rho gdy-\int \Gamma_{1}^{z}\left(h,\Gamma_{1}\left(h,h\right)\right)\rho gdy\\ & = & \int \Gamma_{1}^{z}\left(h,\Delta_{a}h\right)\rho gdy-\int \Gamma_{1}^{z}\left(h,\frac{\Delta_{a}g}{g}\right)\rho gdy-\int \Gamma_{1}\left(h,\Delta_{z}h\right)\rho gdy+\int \Gamma_{1}\left(h,\frac{\Delta_{z}g}{g}\right)\rho gdy. \end{array}$$Recall that we denote $\Delta_{a}=\nabla \cdot \left(aa^{T}\nabla \right)$ and $\Delta_{z}=\nabla \cdot \left(zz^{T}\nabla \right)$. Use the following identity:
$$\begin{array}{c} \Delta_{a}h=\frac{\Delta_{a}g}{g}-\frac{\Gamma_{1}\left(g,g\right)}{g^{2}},\;\mathrm{and}\;\Delta_{z}h=\frac{\Delta_{z}g}{g}-\frac{\Gamma_{1}^{z}\left(g,g\right)}{g^{2}}. \end{array}$$We then obtain
$$\begin{array}{ccc} \int \Gamma_{1}^{z}\left(h,\Delta_{a}h\right)\rho gdy & = & \int \Gamma_{1}^{z}\left(h,\frac{\Delta_{a}g}{g}-\frac{\Gamma_{1}\left(g,g\right)}{g^{2}}\right)\rho gdy\\ & = & -\int \Gamma_{1}^{z}\left(h,\Gamma_{1}\left(h,h\right)\right)\rho gdy+\int \Gamma_{1}^{z}\left(h,\frac{\Delta_{a}g}{g}\right)\rho gdy. \end{array}$$Similarly, the other equality is satisfied.
**Claim 2:**

$$\begin{array}{ccc} & & \int \Gamma_{1}^{z}\left(h,\Delta_{a}h\right)\rho gdy-\int \Gamma_{1}^{z}\left(h,\frac{\Delta_{a}g}{g}\right)\rho gdy-\int \Gamma_{1}\left(h,\Delta_{z}h\right)\rho gdy+\int \Gamma_{1}\left(h,\frac{\Delta_{z}g}{g}\right)\rho gdy\\ & = & \int \frac{\nabla \cdot (\rho zz^{T}\Gamma_{\nabla (aa^{T})}\left(h,h\right))}{\rho }g\rho dy-\int \frac{\nabla \cdot (\rho aa^{T}\Gamma_{\nabla (zz^{T})}\left(h,h\right))}{\rho }g\rho dy. \end{array}$$

First, observe that
$$\begin{array}{ccc} \int \Gamma_{1}^{z}\left(h,\frac{\Delta_{a}g}{g}\right)\rho gdy & = & \int \langle zz^{T}\nabla h,\nabla \left(\frac{\Delta_{a}g}{g}\right)\rangle \rho gdy=-\int \nabla \cdot \left(\rho zz^{T}\nabla g\right)\frac{\Delta_{a}g}{g}dy\\ & = & -\int \frac{\rho }{g}\Delta_{a}g\Delta_{z}gdy-\int \langle \nabla \rho ,zz^{T}\nabla g\rangle \frac{\Delta_{a}g}{g}dy. \end{array}$$Similarly, one obtains
$$\begin{array}{ccc} \int \Gamma_{1}\left(h,\frac{\Delta_{z}g}{g}\right)\rho gdy & = & -\int \frac{\rho }{g}\Delta_{a}g\Delta_{z}gdy-\int \langle \nabla \rho ,aa^{T}\nabla g\rangle \frac{\Delta_{z}g}{g}dy. \end{array}$$For the next term, one obtains
$$\begin{array}{ccc} & & \int \Gamma_{1}^{z}\left(h,\Delta_{a}h\right)\rho gdy\\ & = & \int \langle \nabla \left(\nabla \cdot \left(aa^{T}\nabla h\right)\right),zz^{T}\nabla h\rangle \rho gdy\\ & = & -\int \left[\nabla \cdot \left(aa^{T}\nabla h\right)\right]\left[\nabla \cdot \left(\rho gzz^{T}\nabla h\right)\right]dy\\ & = & -\int \left[\nabla \cdot \left(aa^{T}\frac{1}{g}\nabla g\right)\right]\nabla \cdot \left(\rho zz^{T}\nabla g\right)dy\\ & = & -\int \left(\langle \nabla \frac{1}{g},aa^{T}\nabla g\rangle +\frac{1}{g}\Delta_{a}g\right)\nabla \cdot \left(\rho zz^{T}\nabla g\right)dy\\ & = & \int \langle \frac{1}{g^{2}}\nabla g,aa^{T}\nabla g\rangle \left(\nabla \cdot \left(\rho zz^{T}\nabla g\right)\right)dy-\int \frac{1}{g}\Delta_{a}g\left(\nabla \cdot \left(\rho zz^{T}\nabla g\right)\right)dy\\ & = & -2\int \nabla^{2}h\left(aa^{T}\nabla h,zz^{T}\nabla h\right)\rho gdy-\int \langle \langle \nabla h,\nabla \left(aa^{T}\right)\nabla h\rangle ,zz^{T}\nabla h\rangle \rho gdy\\ & & -\int \frac{1}{g}\Delta_{a}g\langle \nabla \rho ,zz^{T}\nabla g\rangle dy-\int \frac{\rho }{g}\Delta_{a}g\Delta_{z}gdy, \end{array}$$
where the last equality follows from the integration by parts for the first term and the direct expansion of the divergence for the second term. Similarly, we obtain
$$\begin{array}{ccc} & & \int \Gamma_{1}\left(h,\Delta_{z}h\right)\rho gdy\\ & = & -2\int \nabla^{2}h\left(zz^{T}\nabla h,aa^{T}\nabla h\right)\rho gdy-\int \langle \langle \nabla h,\nabla \left(zz^{T}\right)\nabla h\rangle ,aa^{T}\nabla h\rangle \rho gdy\\ & & -\int \frac{1}{g}\Delta_{z}g\langle \nabla \rho ,aa^{T}\nabla g\rangle dy-\int \frac{\rho }{g}\Delta_{a}g\Delta_{z}gdy. \end{array}$$Observing, by integration by parts, we obtain
$$\begin{array}{ccc} & & -\int \langle \langle \nabla h,\nabla \left(aa^{T}\right)\nabla h\rangle ,zz^{T}\nabla h\rangle \rho gdy+\int \langle \langle \nabla h,\nabla \left(zz^{T}\right)\nabla h\rangle ,aa^{T}\nabla h\rangle \rho gdy\\ & = & \int \frac{\nabla \cdot (\rho zz^{T}\Gamma_{\nabla (aa^{T})}\left(h,h\right))}{\rho }g\rho dy-\int \frac{\nabla \cdot (\rho aa^{T}\Gamma_{\nabla (zz^{T})}\left(h,h\right))}{\rho }g\rho dy. \end{array}$$Combining the above formulas, the proof is completed. □

With the above lemma in hand, we are ready to prove the following entropic inequality. We first define the following energy form:$$\begin{array}{ccc} \Phi_{a}\left(x,t\right) & =P_{t}\left(P_{T-t}f\Gamma_{1}\left(\log P_{T-t}f\right)\right)\left(x\right),\;\Phi_{z}\left(x,t\right) & =P_{t}\left(P_{T-t}f\Gamma_{1}^{z}\left(\log P_{T-t}f\right)\right)\left(x\right). \end{array}$$

Recall that we define
$$\begin{array}{ccc} \phi_{a}\left(x,t\right) & =P_{T-t}f\Gamma_{1}\left(\log P_{T-t}f\right)\left(x\right),\;\mathrm{and}\;\phi_{z}\left(x,t\right) & =P_{T-t}f\Gamma_{1}^{z}\left(\log P_{T-t}f\right)\left(x\right). \end{array}$$

**Theorem** **4.**
*Denote $\phi =\phi_{a}+\phi_{z}$; if the following condition is satisfied:*

$$R⪰\kappa (\Gamma_{1}+\Gamma_{1}^{z}),$$

*we then conclude*

(67)
$$\begin{array}{ccc} P_{T}\left(\phi \left(\cdot ,T\right)\right)\left(x\right) & \geq & \phi \left(x,0\right)+\int_{0}^{T}\kappa_{s}\left(\Phi_{a}\left(x,s\right)+\Phi_{z}\left(x,s\right)\right)ds, \end{array}$$

*where $\kappa_{s}$ depends on the estimate of the transition kernel ∇logρ(s,·,·) associated with semi-group $P_{s}$ (see Definition 5).*


**Remark** **16.**
*Based on Theorem 3, we can also prove the above theorem for operator $\tilde{L}$ with the drift term involved. Since the proof is similar, we skip the proof here.*


**Proof.** Take $\phi =\phi_{a}+\phi_{z}$. Let ${\left(X_{t}^{x}\right)}_{t\geq 0}$ be the diffusion Markov process with semigroup $P_{t}$. (Similar proofs can be found in [2] (Proposition 4.5).) Let smooth function $u:R^{n+m}\to R$ be such that, for every T>0, $\sup_{t\in [0,T]}{∥u\left(t,\cdot \right)∥}_{\infty }<\infty$ and $\sup_{t\in [0,T]}{∥\frac{1}{2}Lu\left(t,\cdot \right)+\partial_{t}u\left(t,\cdot \right)∥}_{\infty }<\infty$. We have for every t>0
$$\begin{array}{ccc} u(t,X_{t}^{x}) & = & u\left(0,x\right)+\int_{0}^{T}\left(\frac{1}{2}Lu+\partial_{s}u\right)\left(s,X_{s}^{x}\right)ds+M_{t}, \end{array}$$
where ${\left(M_{t}\right)}_{t\geq 0}$ is a local martingale. Let $T_{n},n\in N$ be an increasing sequence of stopping times such that, almost surely, $T_{n}\to \infty$ and ${\left(M_{t\wedge T_{n}}\right)}_{t\geq 0}$ is a martingale. We obtain
$$\begin{array}{ccc} E[u\left(t\wedge T_{n},X_{t\wedge T_{n}}^{x}\right)] & = & u\left(0,x\right)+E[\int_{0}^{t\wedge T_{n}}\left(\frac{1}{2}Lu+\partial_{s}u\right)\left(s,X_{s}^{x}\right)ds]. \end{array}$$By using the dominated convergence theorem, we obtain
$$\begin{array}{ccc} E[u\left(t,X_{t}^{x}\right)] & = & u\left(0,x\right)+E[\int_{0}^{t}\left(\frac{1}{2}Lu+\partial_{s}u\right)\left(s,X_{s}^{x}\right)ds]. \end{array}$$Applying the above equality to $\phi (t,X_{t}^{x})$, we obtain
$$\begin{array}{ccc} E[\phi \left(t,X_{t}^{x}\right)] & = & \phi \left(0,x\right)+E[\int_{0}^{T}\left(\frac{1}{2}L\phi +\partial_{s}\phi \right)\left(s,X_{s}^{x}\right)ds]\\ & = & \phi \left(0,x\right)+\int_{0}^{T}E\left[\left(\frac{1}{2}L\phi +\partial_{s}\phi \right)\left(s,X_{s}^{x}\right)\right]ds. \end{array}$$We now look at the term $E[\left(\frac{1}{2}L\phi +\partial_{s}\phi \right)\left(s,X_{s}^{x}\right)]$ with $g\left(s,x\right)=\left(P_{T-s}f\right)\left(x\right)=E\left[f\left(X_{t}^{x}\right)\right]$=∫ρ(x,y,s)f(y)dy:
$$\begin{array}{ccc} E[\left(\frac{1}{2}L\phi +\partial_{s}\phi \right)\left(s,X_{s}^{x}\right)] & = & E[g\Gamma_{2}\left(\log g,\log g\right)+g\Gamma_{2}^{z}\left(\log g,\log g\right)]\\ & & +E[g\Gamma_{1}\left(\log g,\Gamma_{1}^{z}\left(\log g,\log g\right)\right)-g\Gamma_{1}^{z}\left(\log g,\Gamma_{1}\left(\log g,\log g\right)\right)]. \end{array}$$By using the above Lemma 21, let h=logg, and we obtain
$$\begin{array}{ccc} & & E[\left(\frac{1}{2}L\phi +\partial_{s}\phi \right)\left(s,X_{s}^{x}\right)]\\ & = & \int g\rho \left(\Gamma_{2}\left(h,h\right)+\Gamma_{2}^{z}\left(h,h\right)+\frac{\nabla \cdot (\rho zz^{T}\Gamma_{\nabla (aa^{T})}\left(h,h\right))}{\rho }-\frac{\nabla \cdot (\rho aa^{T}\Gamma_{\nabla (zz^{T})}\left(h,h\right))}{\rho }\right)dy\\ & = & \int g\rho \left(\Gamma_{2}\left(h,h\right)+{\tilde{\Gamma }}_{2}^{z,\rho }\left(h,h\right)\right)dy. \end{array}$$Applying Theorem 3 here with π=ρ(s,·,·) as the transition kernel function, we obtain a time-dependent version of Theorem 3. Assume that the following bound is satisfied where the bound $\kappa_{s}$ depends on kernel ρ(s,·,·):
$$\begin{array}{c} R\left(\nabla f,\nabla f\right)\geq \kappa_{s}\left(\Gamma_{1}\left(f,f\right)+\Gamma_{1}^{z}\left(f,f\right)\right). \end{array}$$We then conclude with the following bound:
$$\begin{array}{ccc} E[\left(\frac{1}{2}L\phi +\partial_{s}\phi \right)\left(s,X_{s}^{x}\right)] & \geq & \int \rho \left(s,x,y\right)g\kappa_{s}\left(\Gamma_{1}\left(h,h\right)\left(y\right)+\Gamma_{1}^{z}\left(h,h\right)\left(y\right)\right)dy\\ & = & \int p\left(s,x,y\right)g\kappa_{s}\left(\Gamma_{1}\left(h,h\right)\left(y\right)+\Gamma_{1}^{z}\left(h,h\right)\left(y\right)\right)\pi \left(y\right)dy\\ & \geq & P_{s}\left(\kappa_{s}g\left(\Gamma_{1}\left(\log g,\log g\right)+\Gamma_{1}^{z}\left(\log g,\log g\right)\right)\right). \end{array}$$Plugging into the time integral $\int_{0}^{T}E\left[\left(\frac{1}{2}L\phi +\partial_{s}\phi \right)\left(s,X_{s}^{x}\right)\right]ds$, the proof follows. □

**Remark** **17.**
*We prove the entropic inequality Theorem 4 in this section without the the assumption: $\Gamma_{1}\left(f,\Gamma_{1}^{z}\left(f,f\right)\right)=\Gamma_{1}^{z}\left(f,\Gamma_{1}\left(f,f\right)\right)$. A similar entropic inequality under the assumption $\Gamma_{1}\left(f,\Gamma_{1}^{z}\left(f,f\right)\right)=\Gamma_{1}^{z}\left(f,\Gamma_{1}\left(f,f\right)\right)$ was first proven in [2] (Proposition 4.5 and Theorem 5.2). With this new inequality Theorem 4 in hand, similar gradient estimates and other inequalities from [2] follow. We leave them for future studies. Proposition 4.5 in [2] is based on a pointwise estimate given the commutative assumption of $\Gamma_{1}$ and $\Gamma_{1}^{z}$. We removed the commutative assumption, and our estimate is in a weak form, which is presented in the above Lemma 21.*

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
