# Peer review of "Entropy Dissipation for Degenerate Stochastic Differential Equations via Sub-Riemannian Density Manifold"

_entropy, 2023, doi:10.3390/e25050786_

Round 1
Reviewer 1 Report
See the attached pdf file.

Author Response
We would like to thank the Editor and all Referees for their very careful reading of
our paper and for many constructive comments. We address all the referees’ comments and highlight the major changes in the revision with blue texts. We corrected typos and grammar issues in the new version. All the labels mentioned below are referred to the revision of the paper. We also move the bibliography after the Appendix following Editor’s suggestions. We believe that the paper is improved a lot after the revision. The detailed response to reviewer letter is attached below.

Reviewer 2 Report
The authors of this manuscript performed a convergence analysis of stochastic differential equation (1.1) to its invariant distribution.
Classical studies are limited to the non-degenerate diffusion coefficient matrix “a” and make use of Gamma calculus. Here it is presented a Lyapunov convergence analysis for the degenerate diffusion
process. To this end a class of z-Fisher information (with “z” a matrix function different from matrix “a”) was selected as the Lyapunov functional.
Then, generalized Bochner's formula and exponential convergence condition were obtained.
Several concrete examples are also discussed.
The presented approach extends the classical optimal transport geometry and has potentially broad application. For this reasons the manuscript warrants publication.
However, its presentation must be improved before publication.
In particular, it is not clear where some of the results are proved.
Here is a partial list: Where Theorems 1 and 2 are proved?
Lemma 3.6 is just an observation….
Where Proposition 3.4 is proved?
Where Corollary 3.5 is proved?
Where Propositions 3.9, 3.10 and Corollary 3.11 are proved?
What is the purpose of Lemmas 3.13, 3.14, 3.15?
Where Proposition 3.18 and Corollary 3.19 are proved?
Additionally, given the applications to Heisenberg group and to the displacement group,
the authors should also comment about the extension of their results to the quantum SDEs.
Author Response

(The authors gave the same response as above.)

Reviewer 3 Report
The aim of the submitted paper is to study the dynamical behavior of degenerate stochastic differential equations. The main tools to conduct such study are an auxiliary Fisher information functional as a Lyapunov functional and a sort of generalized Gamma calculus.
By the help of generalized Fisher information, the authors were able to conduct a Lyapunov-exponential convergence analysis of degenerate stochastic differential equations of Stratonivich type.
The authors then discuss examples of a generalized Bochner's formula valid for the Heisenberg group, the displacement group, and the Martinet sub-Riemannian structure. In this context, the authors were able to show that the discussed generalized Bochner's formula follows a generalized second-order calculus of Kullback-Leibler divergence in density space endowed with a sub-Riemannian optimal-transport metric.
The paper is certainly worth publishing provided some small typos will get fixed before entering the publication stage.
Author Response

(The authors gave the same response as above.)

Round 2
Reviewer 1 Report
No further comments
Reviewer 2 Report
The authors have revised the manuscript taking into account (among others also) my comments. I am satisfied with the present version.